# THE JOINT EFFECT OF TASK SIMILARITY AND OVER-PARAMETERIZATION ON CATASTROPHIC FORGETTING — AN ANALYTICAL MODEL

**Daniel Goldfarb**[*1]**, Itay Evron**[*2]**, Nir Weinberger**[2]**, Daniel Soudry**[2]**, Paul Hand**[1]

[1] Northeastern University [2] Department of Electrical and Computer Engineering, Technion

## ABSTRACT

In continual learning, catastrophic forgetting is affected by multiple aspects of the tasks. Previous works have analyzed separately how forgetting is affected by either task similarity or overparameterization. In contrast, our paper examines how task similarity and overparameterization *jointly* affect forgetting in an analyzable model. Specifically, we focus on two-task continual linear regression, where the second task is a random orthogonal transformation of an arbitrary first task (an abstraction of random permutation tasks). We derive an exact analytical expression for the expected forgetting — and uncover a nuanced pattern. In highly overparameterized models, intermediate task similarity causes the most forgetting. However, near the interpolation threshold, forgetting decreases monotonically with the expected task similarity. We validate our findings with linear regression on synthetic data, and with neural networks on established permutation task benchmarks.

## 1 INTRODUCTION

Modern neural networks achieve state-of-the-art performance in a wide range of applications, but when trained on multiple tasks in sequence, they typically suffer from a drop in performance on earlier tasks, known as the *catastrophic forgetting problem* (Goodfellow et al., 2013). Continual learning research is mostly dedicated to designing neural architectures and optimization methods that better suit learning sequentially (*e.g.,* Zenke et al. (2017); De Lange et al. (2021)). Despite these efforts, it is still unclear in which regimes forgetting is most pronounced, even for elementary models.

A number of works have explored the relationship between task similarity and catastrophic forgetting (Bennani et al., 2020; Ramasesh et al., 2021; Doan et al., 2021; Lee et al., 2021; Evron et al., 2022; Lin et al., 2023). While earlier works struggled to have a consensus on whether similar or different tasks are most prone to forgetting, recent works suggested that continual learning is the easiest when tasks have either high similarity or low similarity, and that it is most difficult for tasks that have an intermediate level of similarity (Evron et al., 2022; Lin et al., 2023). The main experimental evidence of this claim so far focused on the similarity of learned feature representations of a neural network (Ramasesh et al., 2021). Theoretically, Lin et al. (2023) quantified task similarity using Euclidean distances between underlying "teacher" models. However, this notion of similarity cannot capture the task similarity in standard benchmarks (e.g., permuted MNIST), where typically the input features are changing between tasks (and not the teacher). Others (Doan et al., 2021; Evron et al., 2022) interpreted task similarity as the principal angles between data matrices.

In practice, neural networks are typically extremely overparameterized. Several works have studied, empirically and analytically, the beneficial effect of overparameterization in continual learning, particularly on the commonly used permutation and rotation benchmarks (Goldfarb & Hand, 2023; Mirzadeh et al., 2022). In this paper, we propose a more refined analysis, which is able to show the *combined* effect of overparameterization and task similarity for a general data model. We show that task similarity alone cannot explain the difficulty of a continual learning problem, rather it depends also on the model's overparameterization level.

---

[*]Equal contribution. Correspondence to <goldfarb.d@northeastern.edu> or <itay@evron.me>.

Our main result is an exact analytical expression for the worst-case forgetting under a two-task linear regression model trained using the simplest (S)GD scheme. Data for each task is related by a random orthogonal transformation over a randomly chosen subspace, and the **D**imensionality **o**f the **T**ransformed **S**ubspace (DOTS) controls the task similarity — the higher the DOTS, the lower the similarity between tasks. This similarity notion provides a natural knob that controls how similar tasks are after a random transformation. This notion also closely characterizes popular permutation benchmarks, for which it was first suggested by the seminal work of Kirkpatrick et al. (2017).

Figure 1 informally illustrates the essence of our result. When the model is suitably overparameterized, the relationship between task similarity and expected forgetting is non-monotonic, where the most forgetting occurs for intermediately similar tasks. However, if the model is critically parameterized, then the behavior is monotonic and the continual learning problem is most difficult for the highest dissimilarity level. This behavior illustrates how hard it is to estimate the difficulty of continual learning in different regimes.

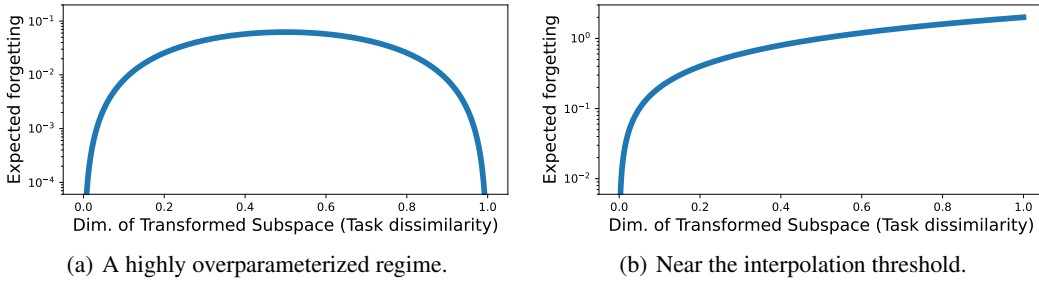

(a) A highly overparameterized regime.      (b) Near the interpolation threshold.

Figure 1: Informal illustration of our theoretical result. Formal details are shared in Section 2.

The contributions of this paper are:

- We present a linear regression data model motivated by empirically-studied permutation tasks that exhibits a joint effect of task similarity and overparameterization on catastrophic forgetting.
- We derive an exact non-asymptotic expression for the worst-case expected forgetting under our model. We reveal a non-monotonic behavior in task similarity when the model is suitably overparameterized, and a monotonic behavior when it is critically overparameterized. We demonstrate that contrary to common belief, overparameterization alone cannot always prevent forgetting.
- We replicate this theoretically-observed interaction of task similarity and overparameterization using a fully connected neural network in a permuted image setting.

## 2 ANALYSIS

We study a linear regression model trained continually on a sequence of two tasks under varying overparameterization and task-similarity levels.

### 2.1 DATA MODEL AND ASSUMPTIONS

Formally, we consider two regression tasks given by two $p$-dimensional data matrices $\mathbf{X}_1, \mathbf{X}_2 \in \mathbb{R}^{n \times p}$ and a single label vector $\mathbf{y} \in \mathbb{R}^n$. The first data matrix $\mathbf{X}_1$ can be *arbitrary*. For ease of notation, we often denote $\mathbf{X} \triangleq \mathbf{X}_1$.

The second task's data matrix $\mathbf{X}_2$ is given by rotating the first task's $p$-dimensional data in a *random* $m$-dimensional subspace for some $m \in [p]$. Specifically, we set $\mathbf{X}_2 = \mathbf{X}_1 \mathbf{O} = \mathbf{X} \mathbf{O}$, where $\mathbf{O} \in \mathbb{R}^{p \times p}$ is a random orthogonal operator defined as,

$$\mathbf{O} = \mathbf{Q}_p \begin{bmatrix} \mathbf{Q}_m & \\ & \mathbf{I}_{p-m} \end{bmatrix} \mathbf{Q}_p^\top , \tag{1}$$

and $\mathbf{Q}_m \sim O(m), \mathbf{Q}_p \sim O(p)$ are orthogonal operators, sampled "uniformly", *i.e.,* using the Haar measure of the orthogonal group. Our definition of the operator $\mathbf{O}$ results in a more mathematically-tractable orthogonal transformation version of popular permutation task datasets (like the ones we use in Section 3), where the labels stay fixed while features permute. This definition also reduces (when $m = p$) to the random rotations studied in Goldfarb & Hand (2023).

To facilitate our analysis, we assume that the first task is realizable by a linear model (implying that the second one is realizable as well). A similar assumption has been made in a previous theoretical paper (Evron et al., 2022) that analyzed, as we do, arbitrary data matrices (rather than assuming random isotropic data). This assumption is especially reasonable in overparameterized regimes, such as in wide neural networks under the NTK regime.

**Assumption 1** (Realizability). *There exists a solution* $\mathbf{w}^\star \in \mathbb{R}^p$ *such that* $\mathbf{X}_1 \mathbf{w}^\star = \mathbf{X} \mathbf{w}^\star = \mathbf{y}$.

Note that this assumption implies that the second task is also realizable (since $\mathbf{X}_2(\mathbf{O}^\top \mathbf{w}^\star) = \mathbf{X}\mathbf{O}\mathbf{O}^\top \mathbf{w}^\star = \mathbf{X}\mathbf{w}^\star = \mathbf{y}$). When $p \geq 2\,\mathrm{rank}(\mathbf{X})$, it is readily seen that the tasks are also *jointly*-realizable w.h.p.

## 2.2 THE ANALYZED LEARNING SCHEME AND ITS LEARNING DYNAMICS

We analyze the most natural continual learning scheme. That is, given two tasks $(\mathbf{X}_1, \mathbf{y}), (\mathbf{X}_2, \mathbf{y})$, we learn them sequentially with a gradient algorithm, without explicitly trying to prevent forgetting.

---

**Scheme 1** Continual learning of two tasks

Initialize $\mathbf{w}_0 = \mathbf{0}_p$
Start from $\mathbf{w}_0$ and obtain $\mathbf{w}_1$ by minimizing $\mathcal{L}_1(\mathbf{w}) \triangleq \|\mathbf{X}_1\mathbf{w} - \mathbf{y}\|^2$ with (S)GD (to convergence)
Start from $\mathbf{w}_1$ and obtain $\mathbf{w}_2$ by minimizing $\mathcal{L}_2(\mathbf{w}) \triangleq \|\mathbf{X}_2\mathbf{w} - \mathbf{y}\|^2$ with (S)GD (to convergence)
Output $\mathbf{w}_2$

---

The learning scheme above is known to mathematically converge[1] to the following iterates,

$$\mathbf{w}_1 = \left( \arg\min_{\mathbf{w}\in\mathbb{R}^p} \|\mathbf{w} - \mathbf{w}_0\|^2 \text{ s.t. } \mathbf{y} = \mathbf{X}_1\mathbf{w} \right) = \mathbf{X}_1^+ \mathbf{y}, \tag{2}$$

$$\mathbf{w}_2 = \left( \arg\min_{\mathbf{w}\in\mathbb{R}^p} \|\mathbf{w} - \mathbf{w}_1\|^2 \text{ s.t. } \mathbf{y} = \mathbf{X}_2\mathbf{w} \right) = \mathbf{X}_2^+ \mathbf{y} + \left( \mathbf{I}_p - \mathbf{X}_2^+ \mathbf{X}_2 \right) \mathbf{w}_1, \tag{3}$$

where $\mathbf{X}^+$ is the pseudoinverse of $\mathbf{X}$ (we use $\mathbf{X}_1^+, \mathbf{X}_2^+$ for analysis only; we do not compute them).

We now define our main quantity of interest.

**Definition 2** (Forgetting). *The forgetting after learning the two tasks (parameterized by* $\mathbf{X}, \mathbf{w}^\star, \mathbf{O}$), *is defined as the degradation in the loss of the first task. More formally,*

$$F(\mathbf{O}; \mathbf{X}, \mathbf{w}^\star) \triangleq \mathcal{L}_1(\mathbf{w}_2) - \mathcal{L}_1(\mathbf{w}_1) = \|\mathbf{X}_1\mathbf{w}_2 - \mathbf{y}\|^2 - \underbrace{\|\mathbf{X}_1\mathbf{w}_1 - \mathbf{y}\|^2}_{=0} = \|\mathbf{X}\mathbf{w}_2 - \mathbf{y}\|^2.$$

Our forgetting definition is natural and relates to definitions in previous papers (*e.g.,* Doan et al. (2021)). Notice that under our Assumption 1, the forgetting is the training loss of the first task (exactly as in Evron et al. (2022)). Alternatively, one can study the degradation in the *generalization* loss instead, but this often requires additional data-distribution assumptions (*e.g.,* random isotropic data as in Goldfarb & Hand (2023); Lin et al. (2023)), while our analysis is valid for any arbitrary data matrix. Our analysis thus gives a better insight into the problem's expected *worst-case* error.

To analyze the forgetting, we utilize our data model from Section 2.1 to show that,

$$F(\mathbf{O}; \mathbf{X}, \mathbf{w}^\star) = \|\mathbf{X}\mathbf{w}_2 - \mathbf{y}\|^2 = \left\| \mathbf{X} \left( \mathbf{X}_2^+ \mathbf{y} + \left( \mathbf{I}_p - \mathbf{X}_2^+ \mathbf{X}_2 \right) \mathbf{w}_1 \right) - \mathbf{y} \right\|^2$$

$$[\mathbf{X}_2 = \mathbf{X}\mathbf{O}] = \left\| \mathbf{X}(\mathbf{X}\mathbf{O})^+ \mathbf{y} + \mathbf{X}\mathbf{w}_1 - \mathbf{X}(\mathbf{X}\mathbf{O})^+ (\mathbf{X}\mathbf{O}) \mathbf{w}_1 - \mathbf{y} \right\|^2$$

$$\left[ \begin{array}{c} \mathbf{y} = \mathbf{X}\mathbf{w}_1, \\ \mathbf{w}_1 = \mathbf{X}^+\mathbf{y} = \mathbf{X}^+\mathbf{X}\mathbf{w}^\star \end{array} \right] = \left\| \mathbf{X}(\mathbf{X}\mathbf{O})^+ \mathbf{X} \left( \mathbf{I} - \mathbf{O} \right) \mathbf{w}_1 \right\|^2 = \left\| \mathbf{X}(\mathbf{X}\mathbf{O})^+ \mathbf{X} \left( \mathbf{I} - \mathbf{O} \right) \mathbf{X}^+ \mathbf{X} \mathbf{w}^\star \right\|^2 \tag{4}$$

$$[\text{pseudoinverse properties}] = \left\| \left( \mathbf{X}\mathbf{X}^+\mathbf{X} \right) \left( \mathbf{O}^\top \mathbf{X}^+ \right) \mathbf{X} \left( \mathbf{I} - \mathbf{O} \right) \mathbf{X}^+ \mathbf{X} \mathbf{w}^\star \right\|^2$$

$$[\forall \mathbf{X}, \mathbf{v}: \|\mathbf{X}\mathbf{v}\|_2 \leq \|\mathbf{X}\|_2 \|\mathbf{v}\|_2] \leq \left\| \mathbf{X} \right\|_2^2 \left\| \mathbf{X}^+ \mathbf{X}\mathbf{O}^\top \mathbf{X}^+ \mathbf{X} \left( \mathbf{I} - \mathbf{O} \right) \mathbf{X}^+ \mathbf{X} \mathbf{w}^\star \right\|^2,$$

where $\|\mathbf{X}\|_2 = \sigma_{\max}(\mathbf{X})$. We used two pseudoinverse properties (any matrix $\mathbf{X}$ and orthogonal matrix $\mathbf{O}$ hold $\mathbf{X} = \mathbf{X}\mathbf{X}^+\mathbf{X}$ and $(\mathbf{X}\mathbf{O})^+ = \mathbf{O}^\top \mathbf{X}^+$). Our upper bound is *sharp*, *i.e.*, it saturates when all nonzero singular values of $\mathbf{X}$ are identical. This allows for *exact* worst-case forgetting analysis.

---

[1] In the realizable case, minimizing the squared loss of a linear model using (stochastic) gradient descent, is known to converge to the projection of the initialization onto the solution space (see Sec. 2.1 in Gunasekar et al. (2018)). This happens regardless of the batch size (as long as the learning rate is small enough). Finally, the projections are given mathematically using the pseudoinverses (*e.g.,* see Sec. 1.3 in Needell & Tropp (2014)).

## 2.3 Key result: Interplay between Task-Similarity and Overparameterization

We now present our main theorem and illustrate it in Figure 2 on random synthetic data.

**Theorem 3.** *Let $p \geq 4, d \in \{1, \ldots, p\}, m \geq 2$. Define $\mathcal{X}_{p,d} \triangleq \{\mathbf{X} \in \mathbb{R}^{n \times p} \mid n \geq \text{rank}(\mathbf{X}) = d\}$. Define the **D**imensionality of **T**ransformed **S**ubspace $\alpha \triangleq \frac{m}{p}$ as our proxy for task dissimilarity and $\beta \triangleq 1 - \frac{d}{p}$ as our proxy for overparameterization. Then, for any solution $\mathbf{w}^\star \in \mathbb{R}^p$ (Assumption 1), the (normalized) worst-case expected forgetting per Def. 2 (obtained by Scheme 1) is*

$$\max_{\mathbf{X} \in \mathcal{X}_{p,d}} \frac{\mathbb{E}_{\mathbf{O}} F(\mathbf{O}; \mathbf{X}, \mathbf{w}^\star)}{\|\mathbf{X}\|_2^2 \|\mathbf{X}^+ \mathbf{X} \mathbf{w}^\star\|^2} = \alpha \Big(2 + \beta \left(\alpha^3 - 6\alpha^2 + 11\alpha - 8\right) + \beta^2 \left(-5\alpha^3 + 22\alpha^2 - 30\alpha + 12\right) + $$

$$\beta^3 \left(5\alpha^3 - 18\alpha^2 + 20\alpha - 6\right)\Big) + \mathcal{O}\left(\frac{1}{p}\right),$$

*where $\mathbf{X}^+ \mathbf{X} \mathbf{w}^\star$ projects $\mathbf{w}^\star$ onto the column space of $\mathbf{X}$. Notice that $\|\mathbf{X}\|_2^2 \|\mathbf{X}^+ \mathbf{X} \mathbf{w}^\star\|^2$ is a necessary scaling factor, since the forgetting $\|\mathbf{X}\mathbf{w}_2 - \mathbf{y}\|^2 = \|\mathbf{X}\mathbf{w}_2 - \mathbf{X}\mathbf{w}^\star\|^2$ naturally scales with $\|\mathbf{X}\|_2^2$ and $\|\mathbf{X}^+ \mathbf{X} \mathbf{w}^\star\|^2$. The exact expression (without the $\mathcal{O}$ notation) appears in Eq. (8) in Appendix C.*

The full proof is given in Appendix C. Below we outline an informal sketch of the proof.

**Proof sketch.** In our proof, we show that the expected forgetting is controlled by two important terms, namely, $\mathbb{E}_{\mathbf{O}} \left(\mathbf{e}_i^\top \mathbf{O}^\top \mathbf{\Sigma}^+ \mathbf{\Sigma} (\mathbf{I} - \mathbf{O}) \mathbf{e}_i\right)^2$ and $\mathbb{E}_{\mathbf{O}} \left(\mathbf{e}_i^\top \mathbf{O}^\top \mathbf{\Sigma}^+ \mathbf{\Sigma} (\mathbf{I} - \mathbf{O}) \mathbf{e}_j\right)^2$ for $i \neq j$, where $\mathbf{e}_i$ is the $i^{\text{th}}$ standard unit vector. Each of these expectations is essentially a polynomial of the entries of our random $\mathbf{Q}_p, \mathbf{Q}_m$ from Eq. (1). To compute these expectations (in Lemmas 6 and 8), we employ *exact* formulas for the integrals of monomials over the orthogonal groups in $p$ and $m$ dimensions (Gorin, 2002). A more detailed proof outline is given in Appendix C.

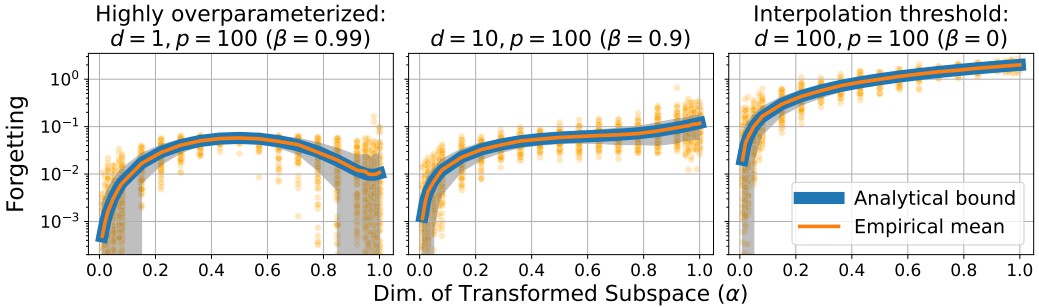

Figure 2: Empirically illustrating the worst-case forgetting under different overparameterization levels. Points indicate the forgetting under 1000 sampled random transformations applied on a (single) random data matrix $\mathbf{X}$. Their mean is shown in the thin orange line, with the standard deviation represented by a gray band. The thick blue line depicts the analytical expression of Theorem 3. Here, we restrict the nonzero singular values of $\mathbf{X}$ to be identical, saturating the inequality in Eq. (4). Indeed, the analytical bound matches the empirical mean, thus exemplifying the tightness of our analysis. For completeness, in Appendix C.3, we repeat this experiment with $p = 10$ and $p = 1000$.

### 2.3.1 Interesting extremal cases

To help interpret our result, we exemplify it in several interesting regimes, taking either the task similarity proxy $\alpha$ or the overparameterization proxy $\beta$ to their extremes.

**Highly overparameterized regime** ($\beta = 1 - \frac{d}{p} \to 1$). Plugging in $\beta = 1$ into Theorem 3, we get

$$\max_{\mathbf{X} \in \mathcal{X}_{p,d}} \frac{\mathbb{E}_{\mathbf{O}} F(\mathbf{O}; \mathbf{X}, \mathbf{w}^\star)}{\|\mathbf{X}\|_2^2 \|\mathbf{X}^+ \mathbf{X} \mathbf{w}^\star\|^2} = \alpha^2 (1 - \alpha)^2 = \left(\frac{m}{p}\right)^2 \left(1 - \frac{m}{p}\right)^2. \tag{5}$$

This behavior is illustrated in Figure 1(a) and in the top part of Figure 3. The non-monotonic nature of this behavior and its peak at $\alpha = 0.5$ corresponding to *intermediate* similarity, seem to agree with Figure 3(a) in Evron et al. (2022) (especially when no repetitions are made, *i.e.*, their $k = 2$ curve).

**At the interpolation threshold** ($d = p \implies \beta = 1 - \frac{d}{p} = 0$). In this extreme, the theorem asserts,

$$\max_{\mathbf{X} \in \mathcal{X}_{p,d}} \frac{\mathbb{E}_{\mathbf{O}} F(\mathbf{O}; \mathbf{X}, \mathbf{w}^\star)}{\|\mathbf{X}\|_2^2 \|\mathbf{X}^+ \mathbf{X} \mathbf{w}^\star\|^2} = 2\alpha = \frac{2m}{p} \,.$$

This behavior is illustrated in Figure 1(b) and in the rightmost plot of Figure 2. Notably, we get a monotonic decrease in forgetting as tasks become more similar. This seems to contradict the conclusions of Evron et al. (2022), according to which intermediate task similarity should be the worst. We settle this alleged disagreement in our discussion in Section 4.

**Minimal task similarity** ($m = p \implies \alpha = \frac{m}{p} = 1$). Plugging in $\alpha = 1$ into Theorem 3, we get

$$\max_{\mathbf{X} \in \mathcal{X}_{p,d}} \frac{\mathbb{E}_{\mathbf{O}} F(\mathbf{O}; \mathbf{X}, \mathbf{w}^\star)}{\|\mathbf{X}\|_2^2 \|\mathbf{X}^+ \mathbf{X} \mathbf{w}^\star\|^2} = 2 - 2\beta - \beta^2 + \beta^3 = \frac{d}{p} + 2\left(\frac{d}{p}\right)^2 - \left(\frac{d}{p}\right)^3 \,.$$

This minimal task similarity regime matches the (noiseless) model of Goldfarb & Hand (2023), where a (generalization) risk bound with a scaling of $\sqrt{\frac{n}{p}}$ was proven under a particular data model.

### 2.3.2   THE ENTIRE INTERPLAY BETWEEN TASK SIMILARITY AND OVERPARAMETERIZATION

The figure below illustrates our main result (Theorem 3). While high task similarity consistently reduces forgetting, this effect becomes more evident with sufficient overparameterization. Noticeably, non-monotonic effects of task similarity (our DOTS), only occur when the model turns highly overparameterized. Importantly, we observe once more that even with extremely high overparameterization levels ($\beta = 1 - \frac{d}{p} \to 1$), forgetting does not vanish entirely. Instead, this outcome is contingent on task similarity, as captured by our DOTS measure. For instance, Eq. (5) shows that when $\alpha = \frac{m}{p} = 0.5$, the worst-case forgetting becomes $(0.5)^4 = 0.0625$ when $\beta = 1 - \frac{d}{p} \to 1$.

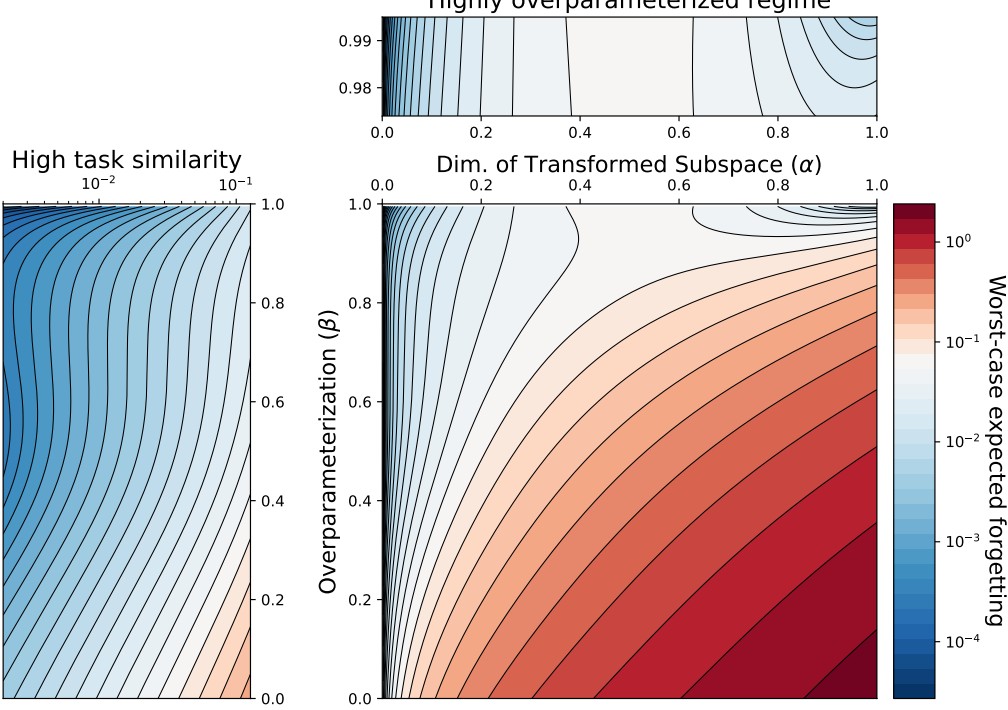

Figure 3:   Levelsets depicting our main result from Theorem 3. The entire space (combinations of $\alpha, \beta$) appears on the lower-right subplot. We zoom into more interesting regimes, *i.e.,* high task similarity and high overparameterization, on the lower-left and upper-right subplots (respectively).

## 2.4 Empirical Forgetting under an Average-Case Data Model

We aim to simulate the (linear) model from Section 2 and show that the average-case behavior matches the joint effect that task similarity and overparameterization have on the worst-case forgetting as analyzed in our Theorem 3. We choose the data model of Goldfarb & Hand (2023) for its clear analogies to learning with neural networks. Under this model, $n$ samples of effective dimensionality $d$ are sampled independently. The latent dimensionality $p$ of the samples controls the overparameterization level. These parameters are precisely defined in Section 2.1 therein. We also utilize the same hyperparameters and noise levels as defined in their numerical simulations in Section 2.3. We compute the statistical risk and training error (MSE) of $\mathbf{w}_1$, $\mathbf{w}_2$, and the null estimator on task 1 as a function of $\alpha \in [0, 1]$ for $p \in [500, 1000, 3000]$ averaged over 100 runs.

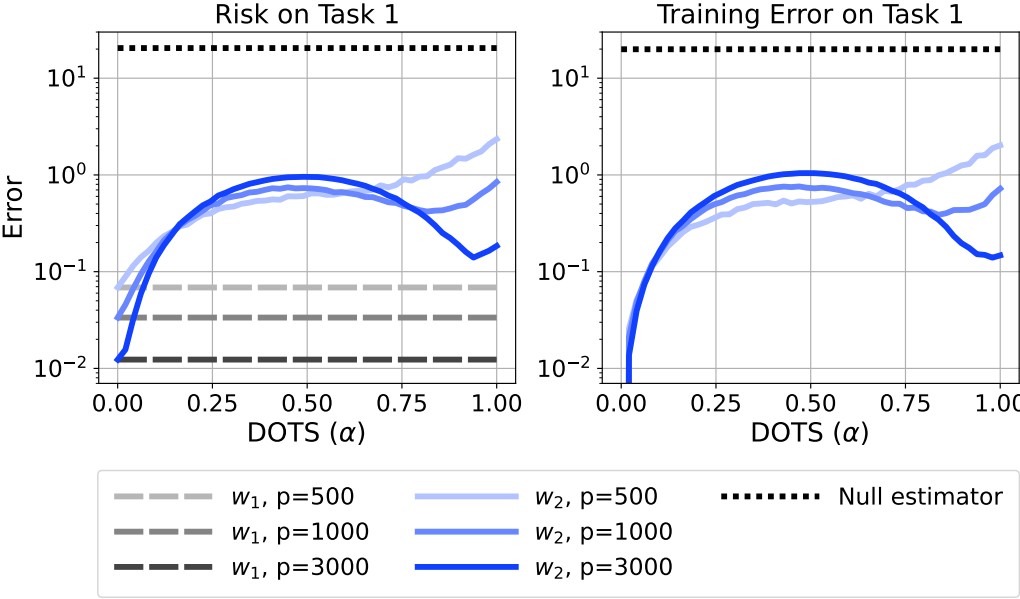

Figure 4: Results of the numerical simulation. Risk and training error on task 1 are plotted as a function of $\alpha$ for various levels of $p$. The solid (blue) curves denote performance on task 1 of an estimator that is trained on task 1 and then on task 2. The dashed dark lines denote performance on task 1 of an estimator trained on task 1 only. The dotted black line denotes the performance of the null estimator. Training error curves for $w_1$ are omitted as these values are 0 for $p > d$.

The results of the experiment are shown in Figure 4. The dotted black line denotes performance of the null estimator (the parameter vector $\mathbf{0}_p$) providing a level for which any non-trivial estimator should beat. The dashed horizontal lines denote the risk of the single task estimator on task 1, providing a hypothetical best-case bound for $\mathbf{w}_2$. The forgetting of the model is then defined by the difference between the performance of $\mathbf{w}_1$ (grey curves) and the performance of $\mathbf{w}_2$ (blue curves). The grey curves are omitted for the training error plot as the single-task training error is 0 for $p > d$. Thus, forgetting in training error is controlled solely by the performance of $\mathbf{w}_2$.

Comparing Figure 2 (for the worst-case forgetting) and Figure 4 here (for the average-case data model) reveals that the interplay between task similarity and overparameterization in both cases agrees with our analytical result in Theorem 3. Here, for large $p$ (3000), we observe a $\cap$-shaped curve for the forgetting of $\mathbf{w}_2$, where the highest amount of forgetting occurs in the intermediate similarity regime. The model under small $p$ (500) has the greatest forgetting when tasks are most dissimilar. Forgetting risk at the extreme of $\alpha = 1$ reduces to the model of Goldfarb & Hand (2023), whose main result explains why overparameterization is beneficial in this setting. Additionally, it is not surprising that higher levels of overparameterization benefit the single task risk setting of $\mathbf{w}_1$, as this reduces to the model of Hastie et al. (2022) where the double descent behavior was observed.

## 3 NEURAL NETWORK EXPERIMENTS

In this section, we examine whether our analytical results apply to a continual learning benchmark using neural networks. The permuted MNIST (LeCun, 1998) benchmark is a popular continual learning problem that has been used to measure the performance of many state-of-the-art continual learning algorithms (Kirkpatrick et al., 2017; Zenke et al., 2017; Li et al., 2019). One performs a number of uniformly chosen permutations on the $28 \times 28$ images of the MNIST dataset. This results in equally difficult tasks (for an MLP): each task is a different pixel-shuffled version of MNIST. One then trains in sequence on these tasks and measures catastrophic forgetting. We consider a variant of the permuted MNIST benchmark, first suggested by Kirkpatrick et al. (2017), where instead of permuting the entire image, we only permute a square grid of pixels in the center of the image. Define the width/length of the permuted square to be the "permutation size" ($PS$). When $PS = 0$, each task is identical and thus extremely similar. When $PS = 28$, each task is fully permuted and thus extremely different. Any value of $PS \in (0, 28)$ can be deemed to have some level of intermediate similarity. Figure 5 shows examples of high, intermediate, and low similarities in this setup. We are interested in the relationship between $PS$ and catastrophic forgetting. Based on prior work (Ramasesh et al., 2021; Evron et al., 2022), it seems we should expect the most forgetting to occur in the regime of intermediate similarity in a two-task scenario.

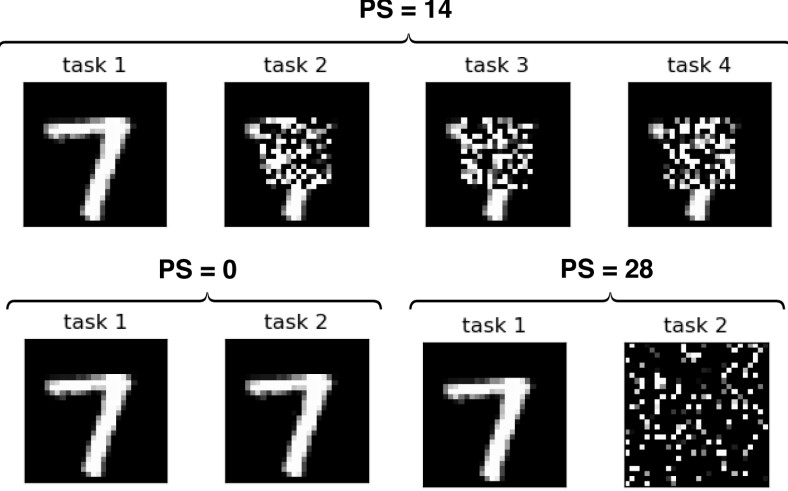

Figure 5: Three versions of permuted MNIST for $PS = 0$ (high similarity), $PS = 14$ (intermediate similarity), $PS = 28$ (low similarity).

We use vanilla SGD to train a 2-layer MLP of width 400 on sequences of up to 4 tasks of MNIST and EMNIST (a more difficult 26-class version of MNIST for handwritten English letters; see Cohen et al. (2017)). After each task is trained, we report the test error on all seen tasks. We compare forgetting across varying permutation sizes. See Appendix A for complete implementational details. The results are shown in Figure 6. The leftmost plots best illustrate the relationship between $PS$ and forgetting. The remaining plots are intended to ensure that each new task is being sufficiently fit. We observe a ∩-shape behavior where the most forgetting occurs around $PS \in [16, 24]$, agreeing with our hypothesis that forgetting is most severe for intermediately similar tasks.

Now consider the experiment in Figure 6 but using a 2-layer MLP with a lower overparameterization level. For each dataset, we find an overparameterization level that is significantly lower than before but still saturates to the training data and has comparable single-task test error as the highly overparameterized version. For MNIST we choose width 20 and for EMNIST we choose width 40. The results of this experiment are shown in Figure 7. We can observe that the ∩-shape behavior is now less pronounced and it appears that the continual learning problem has relatively equal difficulty for intermediately similar tasks as for extremely dissimilar tasks. This general behavior agrees with our previous results on the relationship of overparameterization and forgetting in Section 2, both in the theoretical results and numerical simulations. See Appendix B for evidence of the connection between the notion of similarity in permuted MNIST and the NTK feature regime.

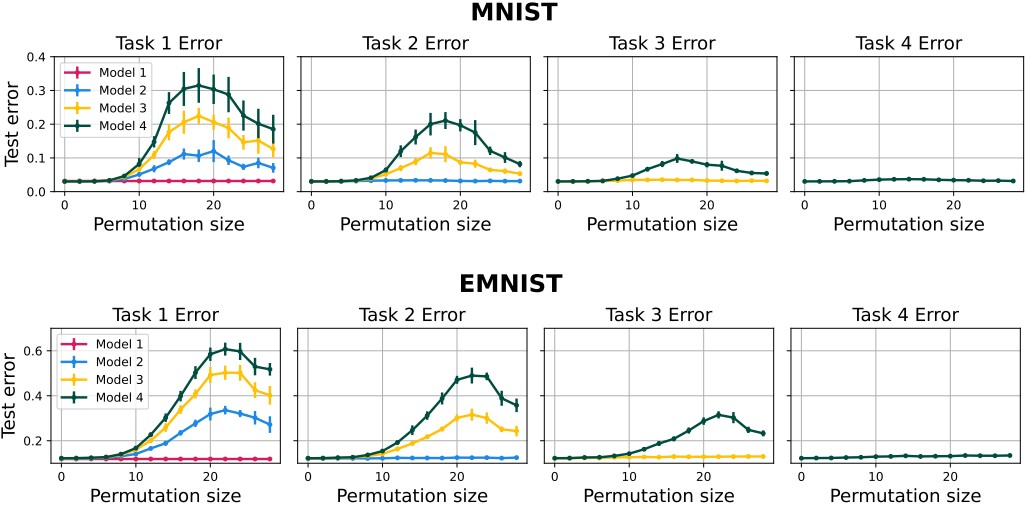

Figure 6: Results of the permutation experiment for varying levels of permutation size. The architecture is a 2-layer MLP of width 400. Model 1 corresponds to the net that is trained on task 1, Model 2 corresponds to the net that is trained on task 1 then task 2, and so on. Plotted curves have been averaged over 10 runs and error bars denote standard deviation of test error over the runs.

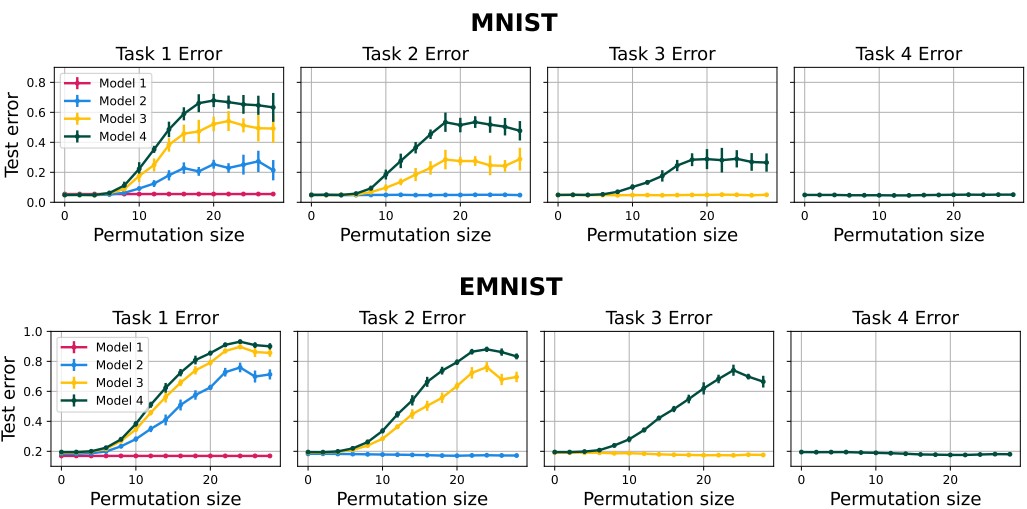

Figure 7: Replication of the experiment in Figure 6 using less overparameterized 2-layer MLP models (width 20 for MNIST and width 40 for EMNIST).

## 4 DISCUSSION

While several related works study continual learning theoretically (Kim et al., 2022; Heckel, 2022), only some of them study the relationship between task similarity and catastrophic forgetting. Bennani et al. (2020) prove generalization bounds which suggest that forgetting is more severe when tasks are dissimilar. Lee et al. (2021) analyze a student-teacher setup, showing that intermediate tasks forget the most. Li et al. (2023) show regimes for which dissimilar tasks may be difficult and where performance on intermediately similar tasks can benefit from regularization. Experimentally, Ramasesh et al. (2021) provide evidence that intermediate task similarity is most difficult by studying learned feature representations during training on a number of modern neural networks.

**Geometric interpretation and Comparison to Evron et al. (2022).** Previous studies utilize principal angles between the solution subspaces of the two data matrices $(\mathbf{X}_1, \mathbf{X}_2)$ to quantify task similarity (Doan et al., 2021; Evron et al., 2022). Evron et al. (2022) show analytically that intermediate task similarity (an angle of $45°$) is most difficult in two-task linear regression. Their analysis applies to *any* two arbitrary tasks, and thus seemingly contradicts the behavior observed, *e.g.,* in our Figure 1(b), where maximal dissimilarity is most difficult. The key to settling this apparent disagreement is the *randomness* of our transformations (Eq. (1)). Their analysis focuses on any two *deterministic* tasks, while our second task is given by a *random* transformation of the first, as done in many popular continual learning benchmarks (*e.g.,* permutation and rotation tasks).

To gain a geometric intuition, consider two tasks of rank $d = 1$ $(\mathbf{x}_1, \mathbf{x}_2)$.[2] Consider also a *maximal* task dissimilarity (DOTS of $\alpha = \frac{m}{p} = 1$). Then, $\mathbf{x}_2 = \mathbf{O}\mathbf{x}_1$ is a completely random rotation of $\mathbf{x}_1$ in $p$ dimensions. It is known that $\mathbb{E}\left|\left\langle \frac{\mathbf{x}_1}{\|\mathbf{x}_1\|}, \frac{\mathbf{x}_2}{\|\mathbf{x}_2\|} \right\rangle\right| \approx \frac{1}{\sqrt{p}}$ (Remark 3.2.5 in Vershynin (2018)). Near the interpolation threshold, *e.g.,* when $p = 2$ (recall that $d = 1$), we get $\mathbb{E}\left|\left\langle \frac{\mathbf{x}_1}{\|\mathbf{x}_1\|}, \frac{\mathbf{x}_2}{\|\mathbf{x}_2\|} \right\rangle\right| \approx \frac{1}{\sqrt{2}} \implies \mathbb{E}\angle(\mathbf{x}_1, \mathbf{x}_2) \approx 45°$, corresponding to the *intermediate* task dissimilarity in Evron et al. (2022), where forgetting is *maximal*. Conversely, given high overparameterization levels ($p \to \infty$), we get $\mathbb{E}\left|\left\langle \frac{\mathbf{x}_1}{\|\mathbf{x}_1\|}, \frac{\mathbf{x}_2}{\|\mathbf{x}_2\|} \right\rangle\right| \approx \frac{1}{\sqrt{p}} \to 0 \implies \mathbb{E}\angle(\mathbf{x}_1, \mathbf{x}_2) \to 90°$, corresponding to the *maximal* task dissimilarity in Evron et al. (2022), where forgetting is *minimal*.

**Comparison to Lin et al. (2023).** Lin et al. (2023) prove generalization bounds on forgetting which suggest that forgetting may not change monotonically with task similarity. However, their data model is not suitable for high overparameterization. For example, in the limit of high overparameterization, their model is performing as well as a null predictor. In contrast, we focus on the training error, and do not assume a specific data model for the first task, which allows us to generalize even in the highly overparameterized regime.

**A starting point for analysis.** Our work focuses on linear models and data permutation tasks. Exploring linear models using (stochastic) gradient descent is the most natural starting point for theoretical analysis, as an initial step towards understanding more complex systems. Moreover, recent work shows connections between extremely overparameterized neural networks and linear models via the neural tangent kernel (NTK) (Jacot et al. (2018); but also see Wenger et al. (2023)). We choose to study data permutation tasks for their well-defined mathematical relationship and generation of equally difficult tasks (for a fully connected model). However there is criticism that permutation tasks are relatively easy to solve in practice and only provide a best-case for real-world problems (Farquhar & Gal, 2018; Pfülb & Gepperth, 2019). Despite these critiques, we believe that the data permutation setting is the most amenable for initial theoretical results.

## 4.1 LIMITATIONS AND FUTURE WORK

Our analysis in Section 2 has centered around a continual *linear* regression model. An immediate next step is to explore the extension of our analysis and empirical findings to more intricate non-linear models (*e.g.,* MLPs, CNNs, and transformers) or to other notions of task similarity. Another avenue of investigation involves extending the analysis to continual *classification* models, possibly using the weak regularization approach suggested by Evron et al. (2023).

Our analysis has also primarily examined settings with $T = 2$ tasks. Extending these analytical results to $T \geq 3$ tasks poses an immediate challenge. The complexity of our analysis, which already required intricate techniques and proofs, suggests that tackling the extension may be considerably difficult. Moreover, the convergence analysis presented in a previous paper (Evron et al., 2022) for learning $T \geq 3$ tasks cyclically has proven to be notably more challenging than that for $T = 2$ tasks (and was further improved in a follow-up paper (Kong et al., 2023)).

Finally, since our models are linear, our proxy for overparameterization, *i.e.,* $\beta = 1 - \frac{d}{p}$, directly controls the overlap between the task subspaces (see also the geometric interpretation above). Clearly, this proxy is different than the width of deep networks. On the other hand, there are still relations between these two proxies through the theory of the NTK regime (Jacot et al., 2018). A further examination of these relations, both theoretically and empirically (perhaps in the spirit of our Appendix B and Wenger et al. (2023)), could benefit the continual learning literature.

---

[2]For simplicity, we discuss the principal angles between $\mathbf{x}_1, \mathbf{x}_2$ instead of between their nullspaces (*i.e.,* the solution spaces). In two-task scenarios, these are essentially equivalent (see Claim 19 in Evron et al. (2022)).

ACKNOWLEDGMENTS

We would like to thank the anonymous reviewers for their insightful feedback. DG is partially supported by NSF DMS-2053448. PH acknowledges support from NSF DMS-2053448, DMS-2022205, and DMS-1848087. The research of NW was partially supported by the Israel Science Foundation (ISF), grant no. 1782/22. The research of DS was Funded by the European Union (ERC, A-B-C-Deep, 101039436). Views and opinions expressed are however those of the author only and do not necessarily reflect those of the European Union or the European Research Council Executive Agency (ERCEA). Neither the European Union nor the granting authority can be held responsible for them. DS also acknowledges the support of the Schmidt Career Advancement Chair in AI.

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

Table 1: Hyperparameters for the neural network experiments

| Hyperparameter | Value |
|---|---|
| learning rate | 0.01 |
| batch size | 64 |
| epochs / task | 100 |
| momentum | 0 |
| dropout | 0 |

## A  NEURAL NETWORK IMPLEMENTATION DETAILS FOR SECTION 3

Table 1 reports the hyperparameters used in the neural network experiments. All architectures used ReLU activation functions for the hidden layers and softmax for the output layers. Weights were initialized as $Unif(\frac{-1}{\sqrt{i}}, \frac{1}{\sqrt{i}})$ where $i$ is the input dimension of the given layer. The experiment in Figure 6 used intermediate width 400 and the experiment in Figure 7 used intermediate width 20 for MNIST and 40 for EMNIST.

## B  NTK FEATURE SIMILARITY EXPERIMENTS

Let $\mathbf{a}, \mathbf{b} \in \mathbb{R}^p$ be two sets of NTK features and define correlation as $\frac{|\langle \mathbf{a}, \mathbf{b} \rangle|}{\|\mathbf{a}\|\|\mathbf{b}\|}$. Then the average correlation between datasets $\mathbf{A}, \mathbf{B} \in \mathbb{R}^{n \times p}$ is $\sum_{i=1}^{n} \frac{|\langle \mathbf{a}_i, \mathbf{b}_i \rangle|}{\|\mathbf{a}_i\|\|\mathbf{b}_i\|}/n$. Figure 8 plots the average correlation between original MNIST and permuted MNIST as a function of permutation size. We see that the average correlation is monotonic in permutation size — extremely high for low permutation size and extremely low for high permutation size. This provides evidence of the connection between the centered permuted MNIST problem and task similarity in the NTK regime.

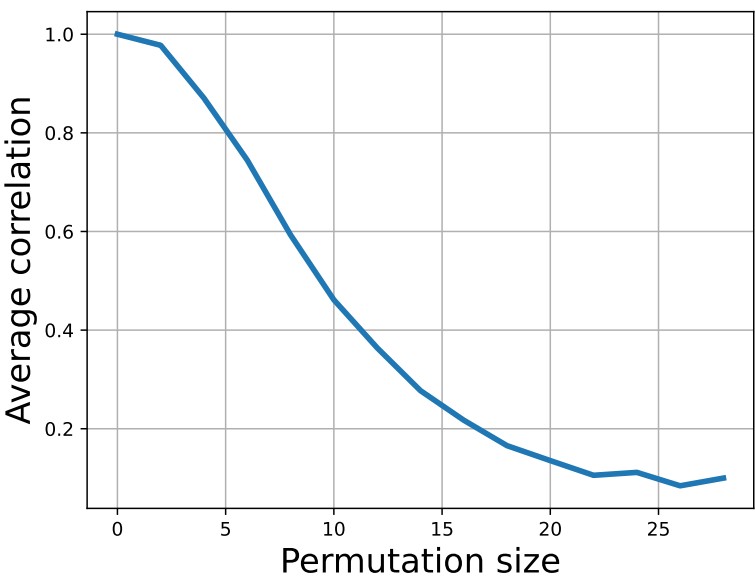

Figure 8: Results of the permuted MNIST NTK feature similarity experiment.

## C    SUPPLEMENTARY MATERIALS FOR OUR ANALYTIC RESULTS (SECTION 2)

Throughout the appendix, we denote the singular-value decomposition of a given $\mathbf{X} \in \mathbb{R}^{n \times p}$ by $\mathbf{X} = \mathbf{U}\boldsymbol{\Sigma}\mathbf{V}^\top$ for $\boldsymbol{\Sigma} \in \mathbb{R}^{n \times p}$. The number of nonzero entries on the diagonal of $\boldsymbol{\Sigma}$ is $d = \mathrm{rank}\left(\mathbf{X}\right)$.

**Recall Theorem 3.**
Let $p \geq 4, d \in \{1, \ldots, p\}, m \geq 2$. Define $\mathcal{X}_{p,d} \triangleq \{\mathbf{X} \in \mathbb{R}^{n \times p} \mid n \geq \mathrm{rank}(\mathbf{X}) = d\}$. Define the **D**imensionality **o**f **T**ransformed **S**ubspace $\alpha \triangleq \frac{m}{p}$ as our proxy for task dissimilarity and $\beta \triangleq 1 - \frac{d}{p}$ as our proxy for overparameterization. Then, for any solution $\mathbf{w}^\star \in \mathbb{R}^p$ (Assumption 1), the (normalized) worst-case expected forgetting per Def. 2 (obtained by Scheme 1) is

$$
\max_{\mathbf{X} \in \mathcal{X}_{p,d}} \frac{\mathbb{E}_{\mathbf{O}} F(\mathbf{O}; \mathbf{X}, \mathbf{w}^\star)}{\|\mathbf{X}\|^2 \|\mathbf{X}^+ \mathbf{X} \mathbf{w}^\star\|^2} = \alpha \Big( 2 + \beta \left( \alpha^3 + 11\alpha - 6\alpha^2 - 8 \right) + \beta^2 \left( -5\alpha^3 + 22\alpha^2 - 30\alpha + 12 \right) +
$$
$$
\beta^3 \left( 5\alpha^3 - 18\alpha^2 + 20\alpha - 6 \right) \Big) + \mathcal{O}\left( \frac{1}{p} \right),
$$

where $\mathbf{X}^+ \mathbf{X} \mathbf{w}^\star$ projects $\mathbf{w}^\star$ onto the column space of $\mathbf{X}$. Notice that $\|\mathbf{X}\|^2 \|\mathbf{X}^+ \mathbf{X} \mathbf{w}^\star\|^2$ is a necessary scaling factor, since the forgetting $\|\mathbf{X}\mathbf{w}_2 - \mathbf{y}\|^2 = \|\mathbf{X}\mathbf{w}_2 - \mathbf{X}\mathbf{w}^\star\|^2$ naturally scales with $\|\mathbf{X}\|^2$ and $\|\mathbf{X}^+ \mathbf{X} \mathbf{w}^\star\|^2$. The exact expression (without the $\mathcal{O}$ notation) appears in Eq. (8) in Appendix C.

**Proof outline.**    We start our proof (below) by showing that the expected forgetting is sharply upper bounded as,

$$
\frac{1}{\|\mathbf{X}\|^2} \mathbb{E}_{\mathbf{O}} F(\mathbf{O}; \mathbf{X}, \mathbf{w}^\star) \leq \sum_{i=1}^d \mathbb{E}_{\mathbf{O}} \left( \mathbf{e}_i^\top \mathbf{O}^\top \boldsymbol{\Sigma}^+ \boldsymbol{\Sigma} \left( \mathbf{I} - \mathbf{O} \right) \boldsymbol{\Sigma}^+ \boldsymbol{\Sigma} \mathbf{V}^\top \mathbf{w}^\star \right)^2
$$
$$
= \left( \mathbb{E}_{\mathbf{O}} \left( \mathbf{e}_1^\top \mathbf{O}^\top \boldsymbol{\Sigma}^+ \boldsymbol{\Sigma} \left( \mathbf{I} - \mathbf{O} \right) \mathbf{e}_1 \right)^2 + (d-1) \mathbb{E}_{\mathbf{O}} \left( \mathbf{e}_1^\top \mathbf{O}^\top \boldsymbol{\Sigma}^+ \boldsymbol{\Sigma} \left( \mathbf{I} - \mathbf{O} \right) \mathbf{e}_2 \right)^2 \right) \underbrace{\sum_{i=1}^d \left( \mathbf{V}^\top \mathbf{w}^\star \right)_i^2}_{=\|\mathbf{X}^+ \mathbf{X} \mathbf{w}^\star\|^2},
$$

where $\mathbf{e}_1, \mathbf{e}_2$ are the two first standard unit vectors in $\mathbb{R}^p$. This directly implies that,

$$
\frac{\mathbb{E}_{\mathbf{O}} F(\mathbf{O}; \mathbf{X}, \mathbf{w}^\star)}{\|\mathbf{X}\|^2 \|\mathbf{X}^+ \mathbf{X} \mathbf{w}^\star\|^2} \leq \underbrace{\mathbb{E}_{\mathbf{O}} \left( \mathbf{e}_1^\top \mathbf{O}^\top \boldsymbol{\Sigma}^+ \boldsymbol{\Sigma} \left( \mathbf{I} - \mathbf{O} \right) \mathbf{e}_1 \right)^2}_{\text{solved in Lemma 6}} + (d-1) \underbrace{\mathbb{E}_{\mathbf{O}} \left( \mathbf{e}_1^\top \mathbf{O}^\top \boldsymbol{\Sigma}^+ \boldsymbol{\Sigma} \left( \mathbf{I} - \mathbf{O} \right) \mathbf{e}_2 \right)^2}_{\text{solved in Lemma 8}}.
$$

Each of these two expectations is essentially a polynomial of the entries of our random orthogonal $\mathbf{Q}_p \in \mathbb{R}^{p \times p}, \mathbf{Q}_m \in \mathbb{R}^{m \times m}$ which form the random operator $\mathbf{O} = \mathbf{Q}_p \begin{bmatrix} \mathbf{Q}_m & \\ & \mathbf{I}_{p-m} \end{bmatrix} \mathbf{Q}_p^\top$ as explained in Eq. (1).

We compute these two expectations (in Lemmas 6 and 8) by employing *exact* formulas for the integrals of monomials over the orthogonal groups in $p$ and $m$ dimensions (Gorin, 2002). Our derivations often get complicated, and so we split them into three appendices:

1. Appendix C (below): Derivations of more complicated expressions involving the operator $\mathbf{O}$.

2. Appendix D: Derivations and properties of monomials of general random orthogonal matrices, mostly using results from Gorin (2002).

3. Appendix E: Derivations of auxiliary expressions (mostly involving $\mathbf{Q}_m, \mathbf{Q}_p$) that are used as building blocks in multiple derivations.

*Proof for Theorem 3.* Starting from Eq. (4), we show that for any given $\mathbf{X}, \mathbf{w}^\star$ it holds that

$$
\begin{aligned}
\tfrac{1}{\|\mathbf{X}\|^2}\mathbb{E}_{\mathbf{O}}F(\mathbf{O};\mathbf{X},\mathbf{w}^\star) &= \tfrac{1}{\|\mathbf{X}\|^2}\mathbb{E}_{\mathbf{O}}\big\|\mathbf{X}\mathbf{X}^+\mathbf{X}\mathbf{O}^\top\mathbf{X}^+\mathbf{X}\left(\mathbf{I}-\mathbf{O}\right)\mathbf{X}^+\mathbf{X}\mathbf{w}^\star\big\|^2 \\
&\leq \mathbb{E}_{\mathbf{O}}\big\|\mathbf{X}^+\mathbf{X}\mathbf{O}^\top\mathbf{X}^+\mathbf{X}\left(\mathbf{I}-\mathbf{O}\right)\mathbf{X}^+\mathbf{X}\mathbf{w}^\star\big\|^2,
\end{aligned}
\tag{6}
$$

where, importantly, the inequality saturates when all the nonzero singular values of $\mathbf{X}$ are identical. Plugging in the SVD, *i.e.,* of $\mathbf{X}=\mathbf{U}\boldsymbol{\Sigma}\mathbf{V}^\top$, we get

$$
\begin{aligned}
\tfrac{1}{\|\mathbf{X}\|^2}\mathbb{E}_{\mathbf{O}}F(\mathbf{O};\mathbf{X},\mathbf{w}^\star) &\leq \mathbb{E}_{\mathbf{O}}\big\|\mathbf{V}\boldsymbol{\Sigma}^+\boldsymbol{\Sigma}\mathbf{V}^\top\mathbf{O}^\top\mathbf{V}\boldsymbol{\Sigma}^+\boldsymbol{\Sigma}\mathbf{V}^\top\left(\mathbf{I}-\mathbf{O}\right)\mathbf{V}\boldsymbol{\Sigma}^+\boldsymbol{\Sigma}\mathbf{V}^\top\mathbf{w}^\star\big\|^2 \\
&= \mathbb{E}_{\mathbf{O}}\big\|\boldsymbol{\Sigma}^+\boldsymbol{\Sigma}\mathbf{V}^\top\mathbf{O}^\top\mathbf{V}\boldsymbol{\Sigma}^+\boldsymbol{\Sigma}\mathbf{V}^\top\left(\mathbf{I}-\mathbf{O}\right)\mathbf{V}\boldsymbol{\Sigma}^+\boldsymbol{\Sigma}\mathbf{V}^\top\mathbf{w}^\star\big\|^2,
\end{aligned}
$$

where the last equality stems from spectral norm properties (recall that $\mathbf{V}$ is an orthogonal matrix).

Following our definition of $\mathbf{O}=\mathbf{Q}_p\begin{bmatrix}\mathbf{Q}_m & \\ & \mathbf{I}_{p-m}\end{bmatrix}\mathbf{Q}_p^\top$ (Eq. (1)) and since $\mathbf{Q}_p$ is sampled *uniformly* from the orthogonal group $O(p)$, we notice that $\mathbf{O}$ and $\mathbf{V}^\top\mathbf{O}\mathbf{V}$ are identically distributed. Hence, we can rewrite the above expectation as

$$
\begin{aligned}
\tfrac{1}{\|\mathbf{X}\|^2}\mathbb{E}_{\mathbf{O}}F(\mathbf{O};\mathbf{X},\mathbf{w}^\star) &\leq \mathbb{E}_{\mathbf{O}}\big\|\boldsymbol{\Sigma}^+\boldsymbol{\Sigma}\mathbf{O}^\top\boldsymbol{\Sigma}^+\boldsymbol{\Sigma}\left(\mathbf{I}-\mathbf{O}\right)\underbrace{\boldsymbol{\Sigma}^+\boldsymbol{\Sigma}\mathbf{V}^\top\mathbf{w}^\star}_{\triangleq\mathbf{v}}\big\|^2 \\
[\boldsymbol{\Sigma}^+\boldsymbol{\Sigma}=(\boldsymbol{\Sigma}^+\boldsymbol{\Sigma})^2] &= \mathbb{E}\Big\|\sum_{i=1}^d \mathbf{e}_i\mathbf{e}_i^\top\mathbf{O}^\top\boldsymbol{\Sigma}^+\boldsymbol{\Sigma}\left(\mathbf{I}-\mathbf{O}\right)\boldsymbol{\Sigma}^+\boldsymbol{\Sigma}\mathbf{v}\Big\|^2 \\
[\text{Pythagorean theorem}] &= \sum_{i=1}^d \mathbb{E}\left(\mathbf{e}_i^\top\mathbf{O}^\top\boldsymbol{\Sigma}^+\boldsymbol{\Sigma}\left(\mathbf{I}-\mathbf{O}\right)\boldsymbol{\Sigma}^+\boldsymbol{\Sigma}\mathbf{v}\right)^2\underbrace{\|\mathbf{e}_i\|^2}_{=1}.
\end{aligned}
\tag{7}
$$

**Remark 4** (Ease of notation). *For simplicity, in the equation above and from now on, we omit the explicit subscript notation whenever it is clear from the context of our derivations what the expectation pertains to. For instance, instead of writing $\mathbb{E}_{\mathbf{O}}\left[F(\mathbf{O};\mathbf{X},\mathbf{w}^\star)\right]$ we can simply write $\mathbb{E}\left[F(\mathbf{O};\mathbf{X},\mathbf{w}^\star)\right]$.*

*As another example, we often analyze (for $i\neq j$) expressions of the following spirit,*

$$
\begin{aligned}
\mathbb{E}_{\mathbf{O}}\left(\mathbf{e}_i^\top\mathbf{O}\mathbf{e}_j\right)^2 &= \mathbb{E}_{\mathbf{Q}_p\sim O(p),\mathbf{Q}_m\sim O(m)}\left(\mathbf{e}_i^\top\mathbf{Q}_p\begin{bmatrix}\mathbf{Q}_m & \\ & \mathbf{I}_{p-m}\end{bmatrix}\mathbf{Q}_p^\top\mathbf{e}_j\right)^2 \\
&= \mathbb{E}_{\substack{\mathbf{u},\mathbf{v}\sim\mathcal{S}^{p-1}(p):\mathbf{u}\perp\mathbf{v}\\\mathbf{Q}_m\sim O(m)}}\left(\mathbf{u}^\top\begin{bmatrix}\mathbf{Q}_m & \\ & \mathbf{I}_{p-m}\end{bmatrix}\mathbf{v}\right)^2,
\end{aligned}
$$

*where the last step follows from the angle-preserving property of orthogonal operators ($\mathbf{Q}_p$). For the sake of simplicity, we take the liberty to also write the expectations above as,*

$$
\mathbb{E}\left(\mathbf{e}_i^\top\mathbf{O}\mathbf{e}_j\right)^2 = \mathbb{E}\left(\mathbf{e}_i^\top\mathbf{Q}_p\begin{bmatrix}\mathbf{Q}_m & \\ & \mathbf{I}_{p-m}\end{bmatrix}\mathbf{Q}_p^\top\mathbf{e}_j\right)^2 = \mathbb{E}_{\mathbf{u}\perp\mathbf{v}}\left(\mathbf{u}^\top\begin{bmatrix}\mathbf{Q}_m & \\ & \mathbf{I}_{p-m}\end{bmatrix}\mathbf{v}\right)^2.
$$

*Finally, when the dimensions are clear from the context, we interchangeably write matrices in the two following forms:* $\begin{bmatrix}\mathbf{0}_m & \\ & \mathbf{I}_{p-m}\end{bmatrix}=\begin{bmatrix}\mathbf{0} & \\ & \mathbf{I}_{p-m}\end{bmatrix}$, *and* $\begin{bmatrix}\mathbf{Q}_m & \\ & \mathbf{0}_{p-m}\end{bmatrix}=\begin{bmatrix}\mathbf{Q}_m & \\ & \mathbf{0}\end{bmatrix}$.

**Back to the proof.** Focusing on just one term from the above Eq. (7), we have,

$$\mathbb{E}\left(\mathbf{e}_i^\top \mathbf{O}^\top \boldsymbol{\Sigma}^+ \boldsymbol{\Sigma} \left(\mathbf{I} - \mathbf{O}\right) \boldsymbol{\Sigma}^+ \boldsymbol{\Sigma} \mathbf{v}\right)^2 = \mathbb{E}\left(\mathbf{e}_i^\top \mathbf{O}^\top \boldsymbol{\Sigma}^+ \boldsymbol{\Sigma} \left(\mathbf{I} - \mathbf{O}\right) \sum_{i=1}^d \mathbf{e}_j \mathbf{e}_j^\top \mathbf{v}\right)^2$$

$$= \mathbb{E}\left(\sum_{j=1}^d \left(\mathbf{e}_i^\top \mathbf{O}^\top \boldsymbol{\Sigma}^+ \boldsymbol{\Sigma} \left(\mathbf{I} - \mathbf{O}\right) \mathbf{e}_j\right)\left(\mathbf{e}_j^\top \mathbf{v}\right)\right)^2$$

$$= \sum_{j=1}^d \sum_{k=1}^d v_j v_k \mathbb{E}\left(\mathbf{e}_i^\top \mathbf{O}^\top \boldsymbol{\Sigma}^+ \boldsymbol{\Sigma} \left(\mathbf{I} - \mathbf{O}\right) \mathbf{e}_j\right)\left(\mathbf{e}_i^\top \mathbf{O}^\top \boldsymbol{\Sigma}^+ \boldsymbol{\Sigma} \left(\mathbf{I} - \mathbf{O}\right) \mathbf{e}_k\right)$$

$$= \sum_{j=1}^d v_j^2 \mathbb{E}\left(\mathbf{e}_i^\top \mathbf{O}^\top \boldsymbol{\Sigma}^+ \boldsymbol{\Sigma} \left(\mathbf{I} - \mathbf{O}\right) \mathbf{e}_j\right)^2 +$$

$$+ \sum_{j \neq k=1}^d v_j v_k \underbrace{\mathbb{E}\left(\mathbf{e}_i^\top \mathbf{O}^\top \boldsymbol{\Sigma}^+ \boldsymbol{\Sigma} \left(\mathbf{I} - \mathbf{O}\right) \mathbf{e}_j\right)\left(\mathbf{e}_i^\top \mathbf{O}^\top \boldsymbol{\Sigma}^+ \boldsymbol{\Sigma} \left(\mathbf{I} - \mathbf{O}\right) \mathbf{e}_k\right)}_{=0, \text{ by Lemma 5}}$$

$$= \sum_{j=1}^d v_j^2 \mathbb{E}\left(\mathbf{e}_i^\top \mathbf{O}^\top \boldsymbol{\Sigma}^+ \boldsymbol{\Sigma} \left(\mathbf{I} - \mathbf{O}\right) \mathbf{e}_j\right)^2$$

$$= \underbrace{v_i^2 \mathbb{E}\left(\mathbf{e}_i^\top \mathbf{O}^\top \boldsymbol{\Sigma}^+ \boldsymbol{\Sigma} \left(\mathbf{I} - \mathbf{O}\right) \mathbf{e}_i\right)^2}_{j=i} + \underbrace{\sum_{j \in [d] \setminus \{i\}} v_j^2 \mathbb{E}\left(\mathbf{e}_i^\top \mathbf{O}^\top \boldsymbol{\Sigma}^+ \boldsymbol{\Sigma} \left(\mathbf{I} - \mathbf{O}\right) \mathbf{e}_j\right)^2}_{j \neq i}$$

$$= v_i^2 \mathbb{E}\left(\mathbf{e}_1^\top \mathbf{O}^\top \boldsymbol{\Sigma}^+ \boldsymbol{\Sigma} \left(\mathbf{I} - \mathbf{O}\right) \mathbf{e}_1\right)^2 + \mathbb{E}\left(\mathbf{e}_1^\top \mathbf{O}^\top \boldsymbol{\Sigma}^+ \boldsymbol{\Sigma} \left(\mathbf{I} - \mathbf{O}\right) \mathbf{e}_2\right)^2 \sum_{j \in [d] \setminus \{i\}} v_j^2 .$$

We are free to use $\mathbf{e}_1, \mathbf{e}_2$ instead of $\mathbf{e}_i, \mathbf{e}_j$ (for $i, j \in [d]$) due to the exchangeability of different rows of $\mathbf{Q}_p$. For instance, notice that

$$\mathbb{E}\left(\mathbf{e}_i^\top \mathbf{O}^\top \boldsymbol{\Sigma}^+ \boldsymbol{\Sigma} \left(\mathbf{I} - \mathbf{O}\right) \mathbf{e}_j\right)^2 = \mathbb{E}\left(\sum_{k=1}^d \mathbf{e}_i^\top \mathbf{Q}_p \begin{bmatrix} \mathbf{Q}_m^\top & \\ & \mathbf{I}_{p-m} \end{bmatrix} \mathbf{Q}_p^\top \mathbf{e}_k \mathbf{e}_k^\top \mathbf{Q}_p \left(\mathbf{I} - \begin{bmatrix} \mathbf{Q}_m & \\ & \mathbf{I}_{p-m} \end{bmatrix}\right) \mathbf{Q}_p^\top \mathbf{e}_j\right)^2.$$

That is, the expression is a function of $\mathbf{Q}_p^\top \mathbf{e}_1, \ldots, \mathbf{Q}_p^\top \mathbf{e}_d$ which are the first $d$ rows of the random $\mathbf{Q}_p$ and are entirely exchangeable (see Prop. 9).

Going back to Eq. (7),

$$\frac{1}{\|\mathbf{X}\|^2} \mathbb{E}_\mathbf{O} F(\mathbf{O}; \mathbf{X}, \mathbf{w}^\star) \le \sum_{i=1}^{d} \mathbb{E} \left( \mathbf{e}_i^\top \mathbf{O}^\top \mathbf{\Sigma}^+ \mathbf{\Sigma} (\mathbf{I} - \mathbf{O}) \mathbf{\Sigma}^+ \mathbf{\Sigma} \mathbf{v} \right)^2$$

$$= \sum_{i=1}^{d} \left( v_i^2 \mathbb{E} \left( \mathbf{e}_1^\top \mathbf{O}^\top \mathbf{\Sigma}^+ \mathbf{\Sigma} (\mathbf{I} - \mathbf{O}) \mathbf{e}_1 \right)^2 + \mathbb{E} \left( \mathbf{e}_1^\top \mathbf{O}^\top \mathbf{\Sigma}^+ \mathbf{\Sigma} (\mathbf{I} - \mathbf{O}) \mathbf{e}_2 \right)^2 \sum_{j \in [d] \setminus \{i\}} v_j^2 \right)$$

$$= \mathbb{E} \left( \mathbf{e}_1^\top \mathbf{O}^\top \mathbf{\Sigma}^+ \mathbf{\Sigma} (\mathbf{I} - \mathbf{O}) \mathbf{e}_1 \right)^2 \sum_{i=1}^{d} v_i^2 + \mathbb{E} \left( \mathbf{e}_1^\top \mathbf{O}^\top \mathbf{\Sigma}^+ \mathbf{\Sigma} (\mathbf{I} - \mathbf{O}) \mathbf{e}_2 \right)^2 \sum_{i=1}^{d} \sum_{j \in [d] \setminus \{i\}} v_j^2$$

$$= \mathbb{E} \left( \mathbf{e}_1^\top \mathbf{O}^\top \mathbf{\Sigma}^+ \mathbf{\Sigma} (\mathbf{I} - \mathbf{O}) \mathbf{e}_1 \right)^2 \sum_{i=1}^{d} v_i^2 + \mathbb{E} \left( \mathbf{e}_1^\top \mathbf{O}^\top \mathbf{\Sigma}^+ \mathbf{\Sigma} (\mathbf{I} - \mathbf{O}) \mathbf{e}_2 \right)^2 \sum_{i=1}^{d} \left( \left( \sum_{j \in [d]} v_j^2 \right) - v_i^2 \right)$$

$$= \mathbb{E} \left( \mathbf{e}_1^\top \mathbf{O}^\top \mathbf{\Sigma}^+ \mathbf{\Sigma} (\mathbf{I} - \mathbf{O}) \mathbf{e}_1 \right)^2 \sum_{i=1}^{d} v_i^2 + \mathbb{E} \left( \mathbf{e}_1^\top \mathbf{O}^\top \mathbf{\Sigma}^+ \mathbf{\Sigma} (\mathbf{I} - \mathbf{O}) \mathbf{e}_2 \right)^2 \left( d \sum_{j=1}^{d} v_j^2 - \sum_{i=1}^{d} v_i^2 \right)$$

$$= \mathbb{E} \left( \mathbf{e}_1^\top \mathbf{O}^\top \mathbf{\Sigma}^+ \mathbf{\Sigma} (\mathbf{I} - \mathbf{O}) \mathbf{e}_1 \right)^2 \sum_{i=1}^{d} v_i^2 + \mathbb{E} \left( \mathbf{e}_1^\top \mathbf{O}^\top \mathbf{\Sigma}^+ \mathbf{\Sigma} (\mathbf{I} - \mathbf{O}) \mathbf{e}_2 \right)^2 (d-1) \sum_{i=1}^{d} v_i^2$$

$$= \left( \mathbb{E} \left( \mathbf{e}_1^\top \mathbf{O}^\top \mathbf{\Sigma}^+ \mathbf{\Sigma} (\mathbf{I} - \mathbf{O}) \mathbf{e}_1 \right)^2 + (d-1) \mathbb{E} \left( \mathbf{e}_1^\top \mathbf{O}^\top \mathbf{\Sigma}^+ \mathbf{\Sigma} (\mathbf{I} - \mathbf{O}) \mathbf{e}_2 \right)^2 \right) \sum_{i=1}^{d} v_i^2 .$$

Interestingly, in this worst-case scenario, the direction of $\mathbf{w}^\star$ that lies in the column span of $\mathbf{X}$, *i.e.*, $\mathbf{X}^+ \mathbf{X} \mathbf{w}^\star = \mathbf{V} \underbrace{\mathbf{\Sigma}^+ \mathbf{\Sigma} \mathbf{V}^\top \mathbf{w}^\star}_{=\mathbf{v}}$ does not play a role, but only its scale $\left\| \mathbf{V} \mathbf{\Sigma}^+ \mathbf{\Sigma} \mathbf{V}^\top \mathbf{w}^\star \right\| = \|\mathbf{V}\mathbf{v}\| = \|\mathbf{v}\| = \sum_{i=1}^{d} v_i^2$ (since $\mathbf{v}$ is only nonzero in its first $d$ entries). Normalizing by this scale, we get,

$$\frac{\mathbb{E}_\mathbf{O} F(\mathbf{O}; \mathbf{X}, \mathbf{w}^\star)}{\|\mathbf{X}\|^2 \|\mathbf{X}^+ \mathbf{X} \mathbf{w}^\star\|^2} \le \underbrace{\mathbb{E} \left( \mathbf{e}_1^\top \mathbf{O}^\top \mathbf{\Sigma}^+ \mathbf{\Sigma} (\mathbf{I} - \mathbf{O}) \mathbf{e}_1 \right)^2}_{\text{solved in Lemma 6}} + (d-1) \underbrace{\mathbb{E} \left( \mathbf{e}_1^\top \mathbf{O}^\top \mathbf{\Sigma}^+ \mathbf{\Sigma} (\mathbf{I} - \mathbf{O}) \mathbf{e}_2 \right)^2}_{\text{solved in Lemma 8}},$$

where we remind the reader that the inequality saturates when all the nonzero singular values of $\mathbf{X}$ are identical.

By plugging in the two lemmata and after some tedious algebraic steps, we get the final bound of,

$$
\max_{\mathbf{X}\in\mathcal{X}_{p,d}} \frac{\mathbb{E}_{\mathbf{O}}F(\mathbf{O};\mathbf{X},\mathbf{w}^\star)}{\|\mathbf{X}\|^2\|\mathbf{X}^+\mathbf{X}\mathbf{w}^\star\|^2}
$$

$$
= \frac{m^4p^2\big(2+p+p^2\big)-2m^3p\big(24+10p+13p^2+p^4\big)+m^2p\big(240+230p-15p^2+50p^3-2p^4+p^5\big)}{(p-3)(p-2)(p-1)p(p+1)(p+2)(p+4)(p+6)} +
$$

$$
= \frac{m\big(p^6-28p^5+47p^4-324p^3-60p^2-240p-576\big)+2p\big(288+120p-90p^2+71p^3-6p^4+p^5\big)}{(p-3)(p-2)(p-1)p(p+1)(p+2)(p+4)(p+6)} +
$$

$$
d\cdot\frac{m^4p\big(-7p-2-6p^2\big)+m^3\big(16p^4+18p^3+74p^2+92p+48\big)+m^2\big(-11p^5-223p^3-267p^2-302p-240\big)}{(p-3)(p-2)(p-1)p(p+1)(p+2)(p+4)(p+6)} +
$$

$$
d\cdot\frac{m\big(2p^6-3p^5+168p^4-33p^3+728p^2+1124p+192\big)+\big(74p^4-240p-576-490p^3-252p^2-24p^5\big)}{(p-3)(p-2)(p-1)p(p+1)(p+2)(p+4)(p+6)} +
$$
(8)
$$
2d^2\cdot\frac{m^4p(6+5p)-2m^3\big(8p^3+13p^2+9p+18\big)+m^2\big(15p^4+24p^3+106p^2+162p+36\big)}{(p-3)(p-2)(p-1)p(p+1)(p+2)(p+4)(p+6)} +
$$

$$
2d^2\cdot\frac{m\big(-3p^5+3p^4-129p^3-191p^2-198p-144\big)+p\big(288+246p-15p^2+26p^3-p^4\big)}{(p-3)(p-2)(p-1)p(p+1)(p+2)(p+4)(p+6)} +
$$

$$
d^3\cdot\frac{m^4(-5p-6)+m^3\big(18p^2+34p-12\big)+m^2\big(-20p^3-46p^2-39p-42\big)}{(p-3)(p-2)(p-1)p(p+1)(p+2)(p+4)(p+6)} +
$$

$$
d^3\cdot\frac{m\big(6p^4+10p^3+96p^2+154p+60\big)+\big(2p^4-30p^3-32p^2-144p-144\big)}{(p-3)(p-2)(p-1)p(p+1)(p+2)(p+4)(p+6)} .
$$

The above is the exact expression for the worst-case forgetting. To reach the $\mathcal{O}$ notation, we assume that $p\gg 1$, and so we are left with the most significant elements of each product. That is, we show that,

$$
\approx \frac{m^4p^2\big(p^2\big)-2m^3p\big(p^4\big)+m^2p\big(p^5\big)+m\big(p^6\big)+2p\big(p^5\big)}{p^8}+d\frac{m^4p\big(-6p^2\big)+m^3\big(16p^4\big)+m^2\big(-11p^5\big)+m\big(2p^6\big)-24p^5}{p^8}+
$$

$$
2d^2\frac{m^4p(5p)-2m^3\big(8p^3\big)+m^2\big(15p^4\big)+m\big(-3p^5\big)+p\big(-p^4\big)}{p^8}+
$$

$$
d^3\frac{m^4(-5p)+m^3\big(18p^2\big)+m^2\big(-20p^3\big)+m\big(6p^4\big)+\big(2p^4\big)}{p^8}
$$

$$
= \frac{m^4-2m^3p+m^2p^2+mp^2+2p^2}{p^4} + d\frac{-6m^4+16m^3p-11m^2p^2+2mp^3-24p^2}{p^5} +
$$

$$
2d^2\frac{5m^4-16m^3p+15m^2p^2-3mp^3-p^3}{p^6} + d^3\frac{-5m^4+18m^3-20m^2p^2+6mp^3+2p^3}{p^7}
$$

$$
= \frac{m^4-2m^3p+m^2p^2+mp^2+2p^2}{p^4} + \frac{d}{p}\frac{-6m^4+16m^3p-11m^2p^2+2mp^3-24p^2}{p^4} +
$$

$$
2\left(\frac{d}{p}\right)^2\frac{5m^4-16m^3p+15m^2p^2-3mp^3-p^3}{p^4} + \left(\frac{d}{p}\right)^3\frac{-5m^4+18m^3-20m^2p^2+6mp^3+2p^3}{p^4}
$$

$$
= \frac{m^4-2m^3p+\big(m^2+m+2\big)p^2}{p^4} + \left(\frac{d}{p}\right)\frac{-6m^4+16m^3p-11m^2p^2+2mp^3}{p^4} +
$$

$$
2\left(\frac{d}{p}\right)^2\frac{5m^4-16m^3p+15m^2p^2-(3m+1)p^3}{p^4} + \left(\frac{d}{p}\right)^3\frac{-5m^4+18m^3p-20m^2p^2+(6m+2)p^3}{p^4}
$$

$$
= \left(\frac{m}{p}\right)^4-2\left(\frac{m}{p}\right)^3+\left(\frac{m}{p}\right)^2+\left(\frac{m}{p}\right)\frac{1}{p}+\frac{2}{p^2}+\left(\frac{d}{p}\right)\left(-6\left(\frac{m}{p}\right)^4+16\left(\frac{m}{p}\right)^3-11\left(\frac{m}{p}\right)^2+2\left(\frac{m}{p}\right)\right)+
$$

$$
2\left(\frac{d}{p}\right)^2\left(5\left(\frac{m}{p}\right)^4-16\left(\frac{m}{p}\right)^3+15\left(\frac{m}{p}\right)^2-3\left(\frac{m}{p}\right)-\frac{1}{p}\right)+
$$

$$
\left(\frac{d}{p}\right)^3\left(-5\left(\frac{m}{p}\right)^4+18\left(\frac{m}{p}\right)^3-20\left(\frac{m}{p}\right)^2+6\left(\frac{m}{p}\right)+\frac{2}{p}\right)
$$

$$
\triangleq \alpha^4-2\alpha^3+\alpha^2+(1-\beta)\left(-6\alpha^4+16\alpha^3-11\alpha^2+2\alpha\right) +
$$

$$
2(1-\beta)^2\left(5\alpha^4-16\alpha^3+15\alpha^2-3\alpha\right)+(1-\beta)^3\left(-5\alpha^4+18\alpha^3-20\alpha^2+6\alpha\right)+\mathcal{O}\left(\frac{1}{p}\right)
$$

$$
= \alpha\Big(2+\beta^3\left(5\alpha^3-18\alpha^2+20\alpha-6\right)+\beta^2\left(-5\alpha^3+22\alpha^2-30\alpha+12\right)+
$$

$$
\beta\left(\alpha^3-6\alpha^2+11\alpha-8\right)\Big)+\mathcal{O}\left(\frac{1}{p}\right) .
$$

$\square$

**Lemma 5.** *Let $i \in [d]$ and let $j, k \in [d]$ such that $j \neq k$. It holds that*

$$\mathbb{E} \left( \mathbf{e}_i^\top \mathbf{O}^\top \mathbf{\Sigma}^+ \mathbf{\Sigma} \left( \mathbf{I} - \mathbf{O} \right) \mathbf{e}_j \right) \left( \mathbf{e}_i^\top \mathbf{O}^\top \mathbf{\Sigma}^+ \mathbf{\Sigma} \left( \mathbf{I} - \mathbf{O} \right) \mathbf{e}_k \right) = 0 \,.$$

*Proof.* The expectation can be decomposed as,

$$\begin{aligned}
&\mathbb{E} \left( \mathbf{e}_i^\top \mathbf{O}^\top \mathbf{\Sigma}^+ \mathbf{\Sigma} \left( \mathbf{I} - \mathbf{O} \right) \mathbf{e}_j \right) \left( \mathbf{e}_i^\top \mathbf{O}^\top \mathbf{\Sigma}^+ \mathbf{\Sigma} \left( \mathbf{I} - \mathbf{O} \right) \mathbf{e}_k \right) \\
&= \mathbb{E} \left[ \left( \mathbf{e}_i^\top \mathbf{O}^\top \mathbf{\Sigma}^+ \mathbf{\Sigma} \mathbf{e}_j \right) \left( \mathbf{e}_i^\top \mathbf{O}^\top \mathbf{\Sigma}^+ \mathbf{\Sigma} \mathbf{e}_k \right) \right] - \mathbb{E} \left[ \left( \mathbf{e}_i^\top \mathbf{O}^\top \mathbf{\Sigma}^+ \mathbf{\Sigma} \mathbf{e}_j \right) \left( \mathbf{e}_i^\top \mathbf{O}^\top \mathbf{\Sigma}^+ \mathbf{\Sigma} \mathbf{O} \mathbf{e}_k \right) \right] \\
&\quad - \mathbb{E} \left[ \left( \mathbf{e}_i^\top \mathbf{O}^\top \mathbf{\Sigma}^+ \mathbf{\Sigma} \mathbf{O} \mathbf{e}_j \right) \left( \mathbf{e}_i^\top \mathbf{O}^\top \mathbf{\Sigma}^+ \mathbf{\Sigma} \mathbf{e}_k \right) \right] + \mathbb{E} \left[ \left( \mathbf{e}_i^\top \mathbf{O}^\top \mathbf{\Sigma}^+ \mathbf{\Sigma} \mathbf{O} \mathbf{e}_j \right) \left( \mathbf{e}_i^\top \mathbf{O}^\top \mathbf{\Sigma}^+ \mathbf{\Sigma} \mathbf{O} \mathbf{e}_k \right) \right] \\
&= \mathbb{E} \left[ \left( \mathbf{e}_i^\top \mathbf{O}^\top \mathbf{e}_j \right) \left( \mathbf{e}_i^\top \mathbf{O}^\top \mathbf{e}_k \right) \right] - \mathbb{E} \left[ \left( \mathbf{e}_i^\top \mathbf{O}^\top \mathbf{e}_j \right) \left( \mathbf{e}_i^\top \mathbf{O}^\top \mathbf{\Sigma}^+ \mathbf{\Sigma} \mathbf{O} \mathbf{e}_k \right) \right] \\
&\quad - \mathbb{E} \left[ \left( \mathbf{e}_i^\top \mathbf{O}^\top \mathbf{\Sigma}^+ \mathbf{\Sigma} \mathbf{O} \mathbf{e}_j \right) \left( \mathbf{e}_i^\top \mathbf{O}^\top \mathbf{e}_k \right) \right] + \mathbb{E} \left[ \left( \mathbf{e}_i^\top \mathbf{O}^\top \mathbf{\Sigma}^+ \mathbf{\Sigma} \mathbf{O} \mathbf{e}_j \right) \left( \mathbf{e}_i^\top \mathbf{O}^\top \mathbf{\Sigma}^+ \mathbf{\Sigma} \mathbf{O} \mathbf{e}_k \right) \right] \,,
\end{aligned}$$

where the last step holds because $j, k \in [d]$ and therefore $\mathbf{\Sigma}^+ \mathbf{\Sigma} \mathbf{e}_j = \mathbf{e}_j$, $\mathbf{\Sigma}^+ \mathbf{\Sigma} \mathbf{e}_k = \mathbf{e}_k$.

Following the definition of $\mathbf{O}$ (Eq. (1)), the first expectation becomes

$$\mathbb{E} \left[ \left( \mathbf{e}_i^\top \mathbf{O}^\top \mathbf{e}_j \right) \left( \mathbf{e}_i^\top \mathbf{O}^\top \mathbf{e}_k \right) \right] = \mathbb{E}_{\mathbf{Q}_p, \mathbf{Q}_m} \left[ \mathbf{e}_j^\top \mathbf{Q}_p \begin{bmatrix} \mathbf{Q}_m & \\ & \mathbf{I}_{p-m} \end{bmatrix} \mathbf{Q}_p^\top \mathbf{e}_i \cdot \mathbf{e}_k^\top \mathbf{Q}_p \underbrace{\begin{bmatrix} \mathbf{Q}_m & \\ & \mathbf{I}_{p-m} \end{bmatrix}}_{\triangleq \mathbf{A}} \mathbf{Q}_p^\top \mathbf{e}_i \right] .$$

Since $j \neq k$, we must have either $j \notin \{i, k\}$ or $k \notin \{i, j\}$ (or both). Denote the relevant rows of $\mathbf{Q}_p$ by $\mathbf{q}_i \triangleq \mathbf{Q}_p^\top \mathbf{e}_i$, $\mathbf{q}_j \triangleq \mathbf{Q}_p^\top \mathbf{e}_j$, $\mathbf{q}_k \triangleq \mathbf{Q}_p^\top \mathbf{e}_k$ and notice that they are independent of $\mathbf{Q}_m$ (or $\mathbf{A}$). Without loss of generality, $j \notin \{i, k\}$. The above expectation becomes $\mathbb{E}_{\mathbf{Q}_p, \mathbf{Q}_m} \left[ \mathbf{q}_j^\top \mathbf{A} \mathbf{q}_i \cdot \mathbf{q}_k^\top \mathbf{A} \mathbf{q}_i \right]$, where $\mathbf{q}_j$ appears only once (an odd number). By Cor. 11, the expectation vanishes.

Quite similarly, the second expectation becomes

$$\begin{aligned}
\mathbb{E} \left[ \left( \mathbf{e}_i^\top \mathbf{O}^\top \mathbf{e}_j \right) \left( \mathbf{e}_i^\top \mathbf{O}^\top \mathbf{\Sigma}^+ \mathbf{\Sigma} \mathbf{O} \mathbf{e}_k \right) \right] &= \sum_{t=1}^d \mathbb{E} \left[ \mathbf{e}_i^\top \mathbf{O}^\top \mathbf{e}_j \cdot \mathbf{e}_i^\top \mathbf{O}^\top \mathbf{e}_t \cdot \mathbf{e}_t^\top \mathbf{O} \mathbf{e}_k \right] \\
&= \sum_{t=1}^d \mathbb{E} \left[ \mathbf{q}_j^\top \mathbf{A} \mathbf{q}_i \cdot \mathbf{q}_t^\top \mathbf{A} \mathbf{q}_i \cdot \mathbf{q}_t^\top \mathbf{A} \mathbf{q}_k \right] \,.
\end{aligned}$$

Notice that both $i, t$ appear an *even* number of times in (each of) the above expectation(s). Since $j \neq k$, at least one out of $j, k$ appears an odd number of times (either one, three, or five) in each of the above expectations. Again, by Cor. 11, this expectation vanishes. Clearly, the same holds for the third expectation.

The fourth expectation is,

$$\begin{aligned}
\mathbb{E} \left[ \left( \mathbf{e}_i^\top \mathbf{O}^\top \mathbf{\Sigma}^+ \mathbf{\Sigma} \mathbf{O} \mathbf{e}_j \right) \left( \mathbf{e}_i^\top \mathbf{O}^\top \mathbf{\Sigma}^+ \mathbf{\Sigma} \mathbf{O} \mathbf{e}_k \right) \right] &= \sum_{\ell, t=1}^d \mathbb{E} \left[ \left( \mathbf{e}_i^\top \mathbf{O}^\top \mathbf{e}_\ell \mathbf{e}_\ell^\top \mathbf{O} \mathbf{e}_j \right) \left( \mathbf{e}_i^\top \mathbf{O}^\top \mathbf{e}_t \mathbf{e}_t^\top \mathbf{O} \mathbf{e}_k \right) \right] \\
&= \sum_{\ell, t=1}^d \mathbb{E} \left[ \mathbf{q}_\ell^\top \mathbf{A} \mathbf{q}_i \cdot \mathbf{q}_\ell^\top \mathbf{A} \mathbf{q}_j \cdot \mathbf{q}_t^\top \mathbf{A} \mathbf{q}_i \cdot \mathbf{q}_t^\top \mathbf{A} \mathbf{q}_k \right] \,.
\end{aligned}$$

Notice that $i, \ell, t$ appear an *even* number of times in (each of) the above expectation(s). Since $j \neq k$, at least one out of $j, k$ appears an odd number of times (either one, three, five, or seven) in each of the above expectations. Again, by Cor. 11, this expectation vanishes.

$\square$

## C.1 Deriving $\mathbb{E}\left(\mathbf{e}_i^\top \mathbf{O}^\top \mathbf{\Sigma}^+ \mathbf{\Sigma} \left(\mathbf{I} - \mathbf{O}\right) \mathbf{e}_i\right)^2$

**Lemma 6.** *Let $p \geq 4, m \in \{2, \ldots, p\}, d \in [p]$, and let $\mathbf{O}$ be a random transformation sampled as described in Eq. (1). Then, $\forall i \in [d]$, it holds that*

$$
\mathbb{E}\left(\mathbf{e}_i^\top \mathbf{O}^\top \mathbf{\Sigma}^+ \mathbf{\Sigma} \left(\mathbf{I} - \mathbf{O}\right) \mathbf{e}_i\right)^2
$$
$$
= \frac{m^4\left(p^2+2p\right)-2m^3p^3+m^2\left(p^4-4p^3+20p^2-24\right)+m\left(3p^4-10p^3+63p^2+6p-72\right)+2(p+1)\left(p^3-11p^2+38p-24\right)}{(p-1)p(p+1)(p+2)(p+4)(p+6)} +
$$
$$
\frac{d(p+1)\left(-2m^4+6m^3p-2m^2(p-1)(2p-5)-12m\left(p^2-3p+3\right)-8\left(p^2-8p+6\right)\right)}{(p-1)p(p+1)(p+2)(p+4)(p+6)} +
$$
$$
\frac{d^2\left(4mp\left(-m^2+m+4\right)+4(m+1)(m+2)p^2+(m-6)(m-1)m(m+1)-8(2p+3)\right)}{(p-1)p(p+1)(p+2)(p+4)(p+6)}
$$

*Proof.* We decompose the expectation as,

$$
\mathbb{E}\left(\mathbf{e}_i^\top \mathbf{O}^\top \mathbf{\Sigma}^+ \mathbf{\Sigma} \left(\mathbf{I} - \mathbf{O}\right) \mathbf{e}_i\right)^2
$$
$$
= \mathbb{E}\left(\mathbf{e}_i^\top \mathbf{O}\mathbf{e}_i\right)^2 - 2\mathbb{E}\left[\left(\mathbf{e}_i^\top \mathbf{O}\mathbf{e}_i\right)\left(\mathbf{e}_i^\top \mathbf{O}^\top \mathbf{\Sigma}^+ \mathbf{\Sigma}\mathbf{O}\mathbf{e}_i\right)\right] + \mathbb{E}\left(\mathbf{e}_i^\top \mathbf{O}^\top \mathbf{\Sigma}^+ \mathbf{\Sigma}\mathbf{O}\mathbf{e}_i\right)^2,
$$

and derive each of its three terms separately in the following subsections.

By adding these three terms and by simple algebra, we get the required result,

$$
= \frac{m^2-2mp-m+p^2+2p+2}{p(p+2)} - 2\frac{(p-m)\left(d\left(-m^2+2mp+3m-6\right)+m^2p-2mp^2-3mp+p^3+5p^2+8p-8\right)}{(p-1)p(p+2)(p+4)} +
$$
$$
\frac{3(m+4)(m+6)+(p-m)(p-m+2)\left(m^2-2mp-4m+p^2+10p+36\right)}{p(p+2)(p+4)(p+6)} +
$$
$$
\frac{(d-1)\left(-72+m^4(-1-2p)-72p-20p^2-20p^3+m^3\left(6+16p+8p^2\right)+m^2\left(-47-70p-34p^2-10p^3\right)\right)}{(p-1)p(p+1)(p+2)(p+4)(p+6)} +
$$
$$
\frac{(d-1)\left(m\left(42+136p+114p^2+22p^3+4p^4\right)\right)}{(p-1)p(p+1)(p+2)(p+4)(p+6)} +
$$
$$
\frac{(d-1)d\left(4mp\left(-m^2+m+4\right)+4(m+1)(m+2)p^2+(m-6)(m-1)m(m+1)-8(2p+3)\right)}{(p-1)p(p+1)(p+2)(p+4)(p+6)}
$$
$$
= \frac{m^4\left(p^2+2p\right)-2m^3p^3+m^2\left(p^4-4p^3+20p^2-24\right)+m\left(3p^4-10p^3+63p^2+6p-72\right)+2(p+1)\left(p^3-11p^2+38p-24\right)}{(p-1)p(p+1)(p+2)(p+4)(p+6)} +
$$
$$
\frac{d(p+1)\left(-2m^4+6m^3p-2m^2(p-1)(2p-5)-12m\left(p^2-3p+3\right)-8\left(p^2-8p+6\right)\right)}{(p-1)p(p+1)(p+2)(p+4)(p+6)} +
$$
$$
\frac{d^2\left(4mp\left(-m^2+m+4\right)+4(m+1)(m+2)p^2+(m-6)(m-1)m(m+1)-8(2p+3)\right)}{(p-1)p(p+1)(p+2)(p+4)(p+6)}
$$

$\square$

**Remark 7** (Explaining proof techniques). *In this subsection, we explain the proof steps more thoroughly than in other places, since most of the techniques repeat themselves throughout the appendices.*

### C.1.1 TERM 1: $\mathbb{E}\left(\mathbf{e}_i^\top \mathbf{O}\mathbf{e}_i\right)^2$

Recalling Remark 4 on our simplified notations, we show that,

$$
\mathbb{E}\left(\mathbf{e}_i^\top \mathbf{O}\mathbf{e}_i\right)^2 = \mathbb{E}\left(\mathbf{e}_i^\top \mathbf{Q}_p \begin{bmatrix} \mathbf{Q}_m & \\ & \mathbf{I}_{p-m} \end{bmatrix} \mathbf{Q}_p^\top \mathbf{e}_i\right)^2 = \mathbb{E}_{\mathbf{u}\sim\mathcal{S}^{p-1}}\left(\mathbf{u}^\top \begin{bmatrix} \mathbf{Q}_m & \\ & \mathbf{I}_{p-m} \end{bmatrix} \mathbf{u}\right)^2
$$

$$
= \mathbb{E}\left[\mathbf{u}^\top \left(\begin{bmatrix} \mathbf{Q}_m & \\ & \mathbf{0} \end{bmatrix} + \begin{bmatrix} \mathbf{0} & \\ & \mathbf{I}_{p-m} \end{bmatrix}\right)\mathbf{u}\cdot\mathbf{u}^\top\left(\begin{bmatrix} \mathbf{Q}_m & \\ & \mathbf{0} \end{bmatrix} + \begin{bmatrix} \mathbf{0} & \\ & \mathbf{I}_{p-m} \end{bmatrix}\right)\mathbf{u}\right].
$$

Opening the product above, by Cor. 12 we are only left with the following terms:

$$
= \mathbb{E}\left[\mathbf{u}^\top \begin{bmatrix} \mathbf{0} & \\ & \mathbf{I}_{p-m} \end{bmatrix} \mathbf{u}\mathbf{u}^\top \begin{bmatrix} \mathbf{0} & \\ & \mathbf{I}_{p-m} \end{bmatrix} \mathbf{u}\right] + \mathbb{E}\left[\mathbf{u}^\top \begin{bmatrix} \mathbf{Q}_m & \\ & \mathbf{0} \end{bmatrix} \mathbf{u}\mathbf{u}^\top \begin{bmatrix} \mathbf{Q}_m & \\ & \mathbf{0} \end{bmatrix} \mathbf{u}\right]
$$

$$
= \mathbb{E}\left[\left(\mathbf{u}_b^\top \mathbf{u}_b\right)^2\right] + \mathbb{E}\left[\mathbf{u}_a^\top \mathbf{Q}_m \mathbf{u}_a \mathbf{u}_a^\top \mathbf{Q}_m \mathbf{u}_a\right],
$$

where, like we frequently do throughout the appendix, we decomposed $\mathbf{u}$ into $\mathbf{u} = \begin{bmatrix} \mathbf{u}_a \\ \mathbf{u}_b \end{bmatrix} \in \mathbb{R}^p$ with $\mathbf{u}_a \in \mathbb{R}^m$ and $\mathbf{u}_b \in \mathbb{R}^{p-m}$. This decomposition is often useful, since for two orthogonal unit vectors $\mathbf{u}, \mathbf{v}$, it holds that $0 = \mathbf{u}^\top \mathbf{v} = \mathbf{u}_a^\top \mathbf{v}_a + \mathbf{u}_b^\top \mathbf{v}_b \implies \mathbf{u}_a^\top \mathbf{v}_a = -\mathbf{u}_b^\top \mathbf{v}_b$ and $1 = \|\mathbf{u}\|^2 = \|\mathbf{u}_a\|^2 + \|\mathbf{u}_b\|^2 \implies \|\mathbf{u}_a\|^2 = 1 - \|\mathbf{u}_b\|^2$.

Another "trick" that we use often, is to reparameterize $\mathbf{Q}_m \mathbf{u}_a$ (for $\mathbf{Q}_m \sim O(m)$) as $\|\mathbf{u}_a\|\,\mathbf{r}$ for $\mathbf{r} \sim \mathcal{S}^{m-1}$. Then, the expectation above becomes,

$$
= \mathbb{E}\left[\|\mathbf{u}_b\|^4\right] + \mathbb{E}\left[\|\mathbf{u}_a\|^2 \mathbf{u}_a^\top \left(\frac{1}{m}\mathbf{I}_m\right)\mathbf{u}_a\right] = \mathbb{E}\left[\|\mathbf{u}_b\|^4\right] + \frac{1}{m}\mathbb{E}\left[\|\mathbf{u}_a\|^4\right]
$$

$$
= \sum_{i=m+1}^{p}\sum_{j=m+1}^{p}\mathbb{E}\left[u_i^2 u_j^2\right] + \frac{1}{m}\sum_{i=1}^{m}\sum_{j=1}^{m}\mathbb{E}\left[u_i^2 u_j^2\right]
$$

$$
= (p-m)\,\mathbb{E}\left[u_p^4\right] + (p-m)(p-m-1)\,\mathbb{E}\left[u_{p-1}^2 u_p^2\right] + \frac{1}{m}\left(m\mathbb{E}\left[u_1^4\right] + m(m-1)\,\mathbb{E}\left[u_1^2 u_2^2\right]\right)
$$

$$
= (p-m)\,\mathbb{E}\left[u_1^4\right] + (p-m)(p-m-1)\,\mathbb{E}\left[u_1^2 u_2^2\right] + \mathbb{E}\left[u_1^4\right] + (m-1)\,\mathbb{E}\left[u_1^2 u_2^2\right],
$$

where in the last step we used the fact that different entries of $\mathbf{u}$ are identically distributed (see also Prop. 9). Using simple algebraic steps and employing the notations presented in Appendix D for monomials of entries of random orthogonal matrices, we have

$$
= (p-m+1)\,\mathbb{E}\left[u_1^4\right] + \left(m^2 - 2mp + 2m + p^2 - p - 1\right)\mathbb{E}\left[u_1^2 u_2^2\right]
$$

$$
= (p-m+1)\left\langle \begin{array}{c} 4 \\ \overrightarrow{0} \end{array} \right\rangle + \left(m^2 - 2mp + 2m + p^2 - p - 1\right)\left\langle \begin{array}{c} 2 \\ 2 \\ \overrightarrow{0} \end{array} \right\rangle.
$$

Finally, we plug in the expectations (computed in Appendix D), and get

$$
= \frac{3(p-m+1)}{p(p+2)} + \frac{m^2 - 2mp + 2m + p^2 - p - 1}{p(p+2)} = \frac{m^2 - 2mp - m + p^2 + 2p + 2}{p(p+2)}.
$$

### C.1.2 TERM 2: $\mathbb{E}\left[\left(\mathbf{e}_i^\top \mathbf{O}\mathbf{e}_i\right)\left(\mathbf{e}_i^\top \mathbf{O}^\top \mathbf{\Sigma}^+ \mathbf{\Sigma}\mathbf{O}\mathbf{e}_i\right)\right]$

It holds that,

$$
\mathbb{E}\left[\left(\mathbf{e}_i^\top \mathbf{O}\mathbf{e}_i\right)\left(\mathbf{e}_i^\top \mathbf{O}^\top \mathbf{\Sigma}^+ \mathbf{\Sigma}\mathbf{O}\mathbf{e}_i\right)\right] = \mathbb{E}\left[\left(\mathbf{e}_1^\top \mathbf{O}\mathbf{e}_1\right)\left(\mathbf{e}_1^\top \mathbf{O}^\top \mathbf{\Sigma}^+ \mathbf{\Sigma}\mathbf{O}\mathbf{e}_1\right)\right]
$$

$$
= \mathbb{E}\left[\mathbf{e}_1^\top \mathbf{O}\mathbf{e}_1\mathbf{e}_1^\top \mathbf{O}^\top \sum_{k=1}^{d}\mathbf{e}_k\mathbf{e}_k^\top \mathbf{O}\mathbf{e}_1\right] = \sum_{k=1}^{d}\mathbb{E}\left[\mathbf{e}_1^\top \mathbf{O}\mathbf{e}_1\mathbf{e}_1^\top \mathbf{O}^\top \mathbf{e}_k\mathbf{e}_k^\top \mathbf{O}\mathbf{e}_1\right]
$$

$$
= \sum_{k=1}^{d}\mathbb{E}\left[\left(\mathbf{e}_k^\top \mathbf{O}\mathbf{e}_1\right)^2\mathbf{e}_1^\top \mathbf{O}\mathbf{e}_1\right] = \underbrace{\mathbb{E}\left[\left(\mathbf{e}_1^\top \mathbf{O}\mathbf{e}_1\right)^3\right]}_{\text{solved in Prop. 14}} + (d-1)\underbrace{\mathbb{E}\left[\left(\mathbf{e}_2^\top \mathbf{O}\mathbf{e}_1\right)^2\mathbf{e}_1^\top \mathbf{O}\mathbf{e}_1\right]}_{\text{solved in Prop. 15}}
$$

$$
= \frac{(p-m)\left(m^2-2mp-3m+p^2+6p+14\right)}{p\,(p+2)\,(p+4)} + (d-1)\frac{(p-m)\left(-m^2+2mp+3m-6\right)}{(p-1)\,p\,(p+2)\,(p+4)}
$$

$$
= \frac{(p-m)\left((p-1)\left(m^2-2mp-3m+p^2+6p+14\right)+(d-1)\left(-m^2+2mp+3m-6\right)\right)}{(p-1)\,p\,(p+2)\,(p+4)}
$$

$$
= \frac{(p-m)\left(d\left(-m^2+2mp+3m-6\right)+m^2p-2mp^2-3mp+p^3+5p^2+8p-8\right)}{(p-1)\,p\,(p+2)\,(p+4)}
$$

### C.1.3 TERM 3: $\mathbb{E}\left(\mathbf{e}_i^\top \mathbf{O}^\top \mathbf{\Sigma}^+ \mathbf{\Sigma}\mathbf{O}\mathbf{e}_i\right)^2$

It holds that,

$$
\mathbb{E}\left(\mathbf{e}_i^\top \mathbf{O}^\top \mathbf{\Sigma}^+ \mathbf{\Sigma}\mathbf{O}\mathbf{e}_i\right)^2 = \mathbb{E}\left(\mathbf{e}_1^\top \mathbf{O}^\top \mathbf{\Sigma}^+ \mathbf{\Sigma}\mathbf{O}\mathbf{e}_1\right)^2 = \mathbb{E}\left(\mathbf{e}_1^\top \mathbf{O}^\top \sum_{k=1}^{d}\mathbf{e}_k\mathbf{e}_k^\top \mathbf{O}\mathbf{e}_1\right)^2 = \mathbb{E}\left(\sum_{k=1}^{d}\left(\mathbf{e}_k^\top \mathbf{O}\mathbf{e}_1\right)^2\right)^2
$$

$$
= \sum_{k=1}^{d}\sum_{\ell=1}^{d}\mathbb{E}\left(\mathbf{e}_k^\top \mathbf{O}\mathbf{e}_1\right)^2\left(\mathbf{e}_\ell^\top \mathbf{O}\mathbf{e}_1\right)^2 = \underbrace{\sum_{k=1}^{d}\mathbb{E}\left(\mathbf{e}_k^\top \mathbf{O}\mathbf{e}_1\right)^4}_{k=\ell} + \underbrace{\sum_{k\neq\ell=1}^{d}\mathbb{E}\left(\mathbf{e}_k^\top \mathbf{O}\mathbf{e}_1\right)^2\left(\mathbf{e}_\ell^\top \mathbf{O}\mathbf{e}_1\right)^2}_{k\neq\ell}
$$

$$
= \underbrace{\underbrace{\mathbb{E}\left(\mathbf{e}_1^\top \mathbf{O}\mathbf{e}_1\right)^4}_{k=1,\text{ solved in Prop. 16}} + \underbrace{(d-1)\mathbb{E}\left(\mathbf{e}_2^\top \mathbf{O}\mathbf{e}_1\right)^4}_{k\geq 2,\text{ solved in Prop. 17}}}_{k=\ell} +
$$

$$
\underbrace{\underbrace{2\,(d-1)\mathbb{E}\left(\mathbf{e}_1^\top \mathbf{O}\mathbf{e}_1\right)^2\left(\mathbf{e}_2^\top \mathbf{O}\mathbf{e}_1\right)^2}_{\ell\neq k=1\,\vee\,k\neq\ell=1,\text{ solved in Prop. 18}} + \underbrace{(d-1)\,(d-2)\mathbb{E}\left(\mathbf{e}_2^\top \mathbf{O}\mathbf{e}_1\right)^2\left(\mathbf{e}_3^\top \mathbf{O}\mathbf{e}_1\right)^2}_{k,\ell\geq 2,\,k\neq\ell\text{ solved in Prop. 19}}}_{k\neq\ell}
$$

We note in passing that since $d\leq p$, then when $p=2 \Rightarrow d\leq 2$, the rightmost term is necessarily zero. Therefore, we can use Prop. 19 freely $\forall p\geq 2$, even though it requires that $p\geq 3$.

$$
\mathbb{E}\left(\mathbf{e}_i^\top \mathbf{O}^\top \mathbf{\Sigma}^+ \mathbf{\Sigma}\mathbf{O}\mathbf{e}_i\right)^2 = \frac{3(m+4)(m+6)+(p-m)(p-m+2)\left(m^2-2mp-4m+p^2+10p+36\right)}{p(p+2)(p+4)(p+6)} +
$$

$$
\frac{3(d-1)\left(m^4-2m^3(2p+3)+m^2\left(4p^2+4p-1\right)+2m\left(6p^2+8p+3\right)+8\left(p^2-2p-3\right)\right)}{(p-1)p(p+1)(p+2)(p+4)(p+6)} +
$$

$$
2\,(d-1)\frac{(m+4)\left(2mp+4p+m-m^2-6\right)-(p-m)(p-m+2)(m(m-2p-5)+10)}{(p-1)p(p+2)(p+4)(p+6)} +
$$

$$
\frac{(d-1)(d-2)\left(4mp\left(-m^2+m+4\right)+4(m+1)(m+2)p^2+(m-6)(m-1)m(m+1)-8(2p+3)\right)}{(p-1)p(p+1)(p+2)(p+4)(p+6)}
$$

$$
= \frac{3(m+4)(m+6)+(p-m)(p-m+2)\left(m^2-2mp-4m+p^2+10p+36\right)}{p(p+2)(p+4)(p+6)} +
$$

$$
\frac{(d-1)\left(-72+m^4(-1-2p)-72p-20p^2-20p^3+m^3\left(6+16p+8p^2\right)\right)}{(p-1)p(p+1)(p+2)(p+4)(p+6)} +
$$

$$
\frac{(d-1)\left(m^2\left(-47-70p-34p^2-10p^3\right)+m\left(42+136p+114p^2+22p^3+4p^4\right)\right)}{(p-1)p(p+1)(p+2)(p+4)(p+6)} +
$$

$$
\frac{(d-1)d\left(4mp\left(-m^2+m+4\right)+4(m+1)(m+2)p^2+(m-6)(m-1)m(m+1)-8(2p+3)\right)}{(p-1)p(p+1)(p+2)(p+4)(p+6)}
$$

## C.2  Deriving $\mathbb{E}\left(\mathbf{e}_i^\top \mathbf{O}^\top \mathbf{\Sigma}^+ \mathbf{\Sigma}\left(\mathbf{I}-\mathbf{O}\right)\mathbf{e}_j\right)^2$

**Lemma 8.** *Let $p \geq 4, m \in \{2,\ldots,p\}, d \in [p]$, and let $\mathbf{O}$ be a random transformation sampled as described in Eq. (1). Then, $\forall i, j \in [d]$ such that $i \neq j$, it holds that*

$$
\mathbb{E}\left(\mathbf{e}_i^\top \mathbf{O}^\top \mathbf{\Sigma}^+ \mathbf{\Sigma}\left(\mathbf{I}-\mathbf{O}\right)\mathbf{e}_j\right)^2
$$

$$
= \frac{\left(-4p^3-6p^2+12p\right)m^4+m^3\left(10p^4+14p^3+20p^2+48p\right)+m^2\left(-7p^5-4p^4-109p^3-134p^2-120p-144\right)}{(p-3)(p-2)(p-1)p(p+1)(p+2)(p+4)(p+6)}+
$$

$$
\frac{m\left(2p^6+3p^5+84p^4-45p^3+336p^2+636p+144\right)+\left(-18p^5+24p^4-182p^3-104p^2-168p-288\right)}{(p-3)(p-2)(p-1)p(p+1)(p+2)(p+4)(p+6)}+
$$

$$
d\cdot\frac{3m^4\left(-4+4p+3p^2\right)-4m^3(p+2)\left(7p^2-2p+6\right)+m^2\left(26p^4+44p^3+163p^2+256p+36\right)}{(p-3)(p-2)(p-1)p(p+1)(p+2)(p+4)(p+6)}+
$$

$$
d\cdot\frac{-2m\left(3p^5+102p^3+142p^2+154p+132\right)-2p(p+1)\left(p^3-24p^2+26p-204\right)}{(p-3)(p-2)(p-1)p(p+1)(p+2)(p+4)(p+6)}+
$$

$$
d^2\cdot\frac{m^4(-5p-6)+m^3\left(18p^2+34p-12\right)+m^2\left(-20p^3-46p^2-39p-42\right)}{(p-3)(p-2)(p-1)p(p+1)(p+2)(p+4)(p+6)}+
$$

$$
d^2\cdot\frac{m\left(6p^4+10p^3+96p^2+154p+60\right)+\left(2p^4-30p^3-32p^2-144p-144\right)}{(p-3)(p-2)(p-1)p(p+1)(p+2)(p+4)(p+6)}\,.
$$

*Proof.* We decompose the expectation as,

$$
\mathbb{E}\left(\mathbf{e}_i^\top \mathbf{O}^\top \mathbf{\Sigma}^+ \mathbf{\Sigma}\left(\mathbf{I}-\mathbf{O}\right)\mathbf{e}_j\right)^2
$$
$$
= \mathbb{E}\left(\mathbf{e}_i^\top \mathbf{O}\mathbf{e}_j\right)^2 - 2\mathbb{E}\left[\left(\mathbf{e}_j^\top \mathbf{O}\mathbf{e}_i\right)\left(\mathbf{e}_i^\top \mathbf{O}^\top \mathbf{\Sigma}^+ \mathbf{\Sigma}\mathbf{O}\mathbf{e}_j\right)\right] + \mathbb{E}\left(\mathbf{e}_i^\top \mathbf{O}^\top \mathbf{\Sigma}^+ \mathbf{\Sigma}\mathbf{O}\mathbf{e}_j\right)^2\,,
$$

and derive each of its three terms separately in the following subsections. The final result in the lemma is obtained by summing these three terms. $\square$

### C.2.1  Term 1: $\mathbb{E}\left(\mathbf{e}_i^\top \mathbf{O}\mathbf{e}_j\right)^2$

It holds that,

$$
\mathbb{E}\left(\mathbf{e}_i^\top \mathbf{O}\mathbf{e}_j\right)^2 = \mathbb{E}\left(\mathbf{u}^\top\left(\begin{bmatrix}\mathbf{Q}_m & \\ & \mathbf{0}_{p-m}\end{bmatrix}+\begin{bmatrix}\mathbf{0}_m & \\ & \mathbf{I}_{p-m}\end{bmatrix}\right)\mathbf{v}\right)^2 = \mathbb{E}\left(\mathbf{u}_a^\top \mathbf{Q}_m\mathbf{v}_a + \mathbf{u}_b^\top \mathbf{v}_b\right)^2
$$
$$
= \underbrace{\mathbb{E}\left(\mathbf{u}_a^\top \mathbf{Q}_m\mathbf{v}_a\right)^2}_{\text{solved in Eq. (22)}} + \underbrace{2\mathbb{E}\left(\mathbf{u}_a^\top \mathbf{Q}_m\mathbf{v}_a\mathbf{u}_b^\top \mathbf{v}_b\right)}_{=0,\text{ by Cor. 12}} + \underbrace{\mathbb{E}\left(\mathbf{u}_b^\top \mathbf{v}_b\right)^2}_{\text{solved in Eq. (23)}}
$$
$$
= \frac{mp+m-2}{(p-1)p(p+2)} + \frac{m(p-m)}{(p-1)p(p+2)} = \frac{mp+m-2+m(p-m)}{(p-1)p(p+2)}
$$
$$
= \frac{2mp+m-m^2-2}{(p-1)p(p+2)}
$$

### C.2.2 TERM 2: $2\left(\mathbf{e}_j^\top \mathbf{O}\mathbf{e}_i\right)\left(\mathbf{e}_i^\top \mathbf{O}^\top \mathbf{\Sigma}^+ \mathbf{\Sigma}\mathbf{O}\mathbf{e}_j\right)$

It holds that,

$$
2\mathbb{E}\left[\left(\mathbf{e}_j^\top \mathbf{O}\mathbf{e}_i\right)\left(\mathbf{e}_i^\top \mathbf{O}^\top \mathbf{\Sigma}^+ \mathbf{\Sigma}\mathbf{O}\mathbf{e}_j\right)\right] = 2\mathbb{E}\left[\mathbf{e}_2^\top \mathbf{O}\mathbf{e}_1 \mathbf{e}_1^\top \mathbf{O}^\top \sum_{k=1}^d \mathbf{e}_k \mathbf{e}_k^\top \mathbf{O}\mathbf{e}_2\right]
$$

$$
= 2\sum_{k=1}^d \mathbb{E}\left[\mathbf{e}_2^\top \mathbf{O}\mathbf{e}_1 \mathbf{e}_1^\top \mathbf{O}^\top \mathbf{e}_k \mathbf{e}_k^\top \mathbf{O}\mathbf{e}_2\right] = 2\sum_{k=1}^d \mathbb{E}\left[\mathbf{e}_2^\top \mathbf{O}\mathbf{e}_1 \mathbf{e}_k^\top \mathbf{O}\mathbf{e}_1 \mathbf{e}_k^\top \mathbf{O}\mathbf{e}_2\right]
$$

$$
= 2\left(\underbrace{\mathbb{E}\left[\mathbf{e}_2^\top \mathbf{O}\mathbf{e}_1 \mathbf{e}_1^\top \mathbf{O}\mathbf{e}_1 \mathbf{e}_1^\top \mathbf{O}\mathbf{e}_2\right]}_{\text{solved in Prop. 20}} + \underbrace{\mathbb{E}\left[\left(\mathbf{e}_2^\top \mathbf{O}\mathbf{e}_1\right)^2 \mathbf{e}_2^\top \mathbf{O}\mathbf{e}_2\right]}_{\text{solved in Prop. 15}} + (d-2)\underbrace{\mathbb{E}\left[\mathbf{e}_2^\top \mathbf{O}\mathbf{e}_1 \mathbf{e}_3^\top \mathbf{O}\mathbf{e}_1 \mathbf{e}_3^\top \mathbf{O}\mathbf{e}_2\right]}_{\text{solved in Prop. 21}}\right)
$$

$$
= 2\left(p-m\right)\left(\frac{-m^2-m+(m+1)p-2}{(p-1)p(p+2)(p+4)} + \frac{-m^2+2mp+3m-6}{(p-1)p(p+2)(p+4)} + \frac{(d-2)\left(2m^2-3mp-2m-p+8\right)}{(p-2)(p-1)p(p+2)(p+4)}\right)
$$

$$
= 2\left(p-m\right)\left(\frac{-2m^2+2m+(3m+1)p-8}{(p-1)p(p+2)(p+4)} + \frac{(d-2)\left(2m^2-3mp-2m-p+8\right)}{(p-2)(p-1)p(p+2)(p+4)}\right)
$$

$$
= \frac{2\left(p-m\right)\left(d\left(2m^2-3mp-2m-p+8\right)+p\left(-2m^2+3mp+2m+p-8\right)\right)}{(p-2)(p-1)p(p+2)(p+4)}
$$

### C.2.3 TERM 3: $\mathbb{E}\left(\mathbf{e}_i^\top \mathbf{O}^\top \mathbf{\Sigma}^+ \mathbf{\Sigma} \mathbf{O} \mathbf{e}_j\right)^2$

It holds that,

$$\mathbb{E}\left(\mathbf{e}_i^\top \mathbf{O}^\top \mathbf{\Sigma}^+ \mathbf{\Sigma} \mathbf{O} \mathbf{e}_j\right)^2 = \mathbb{E}\left(\mathbf{e}_1^\top \mathbf{O}^\top \mathbf{\Sigma}^+ \mathbf{\Sigma} \mathbf{O} \mathbf{e}_2\right)^2 = \mathbb{E}\left(\mathbf{e}_1^\top \mathbf{O}^\top \sum_{k=1}^{d} \mathbf{e}_k \mathbf{e}_k^\top \mathbf{O} \mathbf{e}_2\right)^2$$

$$= \sum_{k,\ell=1}^{d} \mathbb{E}\left(\mathbf{e}_1^\top \mathbf{O}^\top \mathbf{e}_k \mathbf{e}_k^\top \mathbf{O} \mathbf{e}_2\right)\left(\mathbf{e}_1^\top \mathbf{O}^\top \mathbf{e}_\ell \mathbf{e}_\ell^\top \mathbf{O} \mathbf{e}_2\right) = \sum_{k,\ell=1}^{d} \mathbb{E}\left(\mathbf{e}_k^\top \mathbf{O} \mathbf{e}_1 \cdot \mathbf{e}_k^\top \mathbf{O} \mathbf{e}_2\right)\left(\mathbf{e}_\ell^\top \mathbf{O} \mathbf{e}_1 \cdot \mathbf{e}_\ell^\top \mathbf{O} \mathbf{e}_2\right)$$

$$= \underbrace{\sum_{k=1}^{d} \mathbb{E}\left(\mathbf{e}_k^\top \mathbf{O} \mathbf{e}_1 \cdot \mathbf{e}_k^\top \mathbf{O} \mathbf{e}_2\right)^2}_{k=\ell} + \underbrace{\sum_{k\neq\ell=1}^{d} \mathbb{E}\left(\mathbf{e}_k^\top \mathbf{O} \mathbf{e}_1 \cdot \mathbf{e}_k^\top \mathbf{O} \mathbf{e}_2\right)\left(\mathbf{e}_\ell^\top \mathbf{O} \mathbf{e}_1 \cdot \mathbf{e}_\ell^\top \mathbf{O} \mathbf{e}_2\right)}_{k\neq\ell}$$

We now show that,

$$\sum_{k=1}^{d} \mathbb{E}\left(\mathbf{e}_k^\top \mathbf{O} \mathbf{e}_1 \cdot \mathbf{e}_k^\top \mathbf{O} \mathbf{e}_2\right)^2 = \underbrace{2\mathbb{E}\left(\mathbf{e}_1^\top \mathbf{O} \mathbf{e}_1 \cdot \mathbf{e}_1^\top \mathbf{O} \mathbf{e}_2\right)^2}_{k=1,2,\text{ solved in Prop. 18}} + \underbrace{(d-2)\mathbb{E}\left(\mathbf{e}_3^\top \mathbf{O} \mathbf{e}_1 \cdot \mathbf{e}_3^\top \mathbf{O} \mathbf{e}_2\right)^2}_{k\geq 3,\text{ solved in Prop. 19}}$$

$$= \frac{2\left((m+4)\left(2mp+4p+m-m^2-6\right)-(p-m)(p-m+2)(m(m-2p-5)+10)\right)}{(p-1)p(p+2)(p+4)(p+6)} +$$

$$\frac{(d-2)\left(4mp\left(-m^2+m+4\right)+4(m+1)(m+2)p^2+(m-6)(m-1)m(m+1)-8(2p+3)\right)}{(p-1)p(p+1)(p+2)(p+4)(p+6)}$$

$$= \frac{-2(m-p-1)(m-p)(m(p+2)(m-2p-5)+2(5p+6))}{(p-1)p(p+1)(p+2)(p+4)(p+6)} +$$

$$d \cdot \frac{4m\left(-m^2+m+4\right)p+4(m+1)(m+2)p^2+(m-6)(m-1)m(m+1)-8(2p+3)}{(p-1)p(p+1)(p+2)(p+4)(p+6)}$$

Moreover, we have that,

$$\sum_{k\neq\ell=1}^{d} \mathbb{E}\left(\mathbf{e}_k^\top \mathbf{O} \mathbf{e}_1 \cdot \mathbf{e}_k^\top \mathbf{O} \mathbf{e}_2\right)\left(\mathbf{e}_\ell^\top \mathbf{O} \mathbf{e}_1 \cdot \mathbf{e}_\ell^\top \mathbf{O} \mathbf{e}_2\right)$$

$$= \underbrace{2\mathbb{E}\left(\mathbf{e}_1^\top \mathbf{O} \mathbf{e}_1 \mathbf{e}_1^\top \mathbf{O} \mathbf{e}_2\right)\left(\mathbf{e}_2^\top \mathbf{O} \mathbf{e}_1 \mathbf{e}_2^\top \mathbf{O} \mathbf{e}_2\right)}_{k=1,\ell=2 \vee k=2,\ell=1,\text{ solved in Prop. 22}} + \underbrace{4(d-2)\mathbb{E}\left(\mathbf{e}_1^\top \mathbf{O} \mathbf{e}_1 \mathbf{e}_1^\top \mathbf{O} \mathbf{e}_2\right)\left(\mathbf{e}_3^\top \mathbf{O} \mathbf{e}_1 \mathbf{e}_3^\top \mathbf{O} \mathbf{e}_2\right)}_{k\leq 2,\ell\geq 3 \vee k\geq 3,\ell\leq 2,\text{ solved in Prop. 24.}} +$$

$$\underbrace{(d-2)(d-3)\mathbb{E}\left(\mathbf{e}_3^\top \mathbf{O} \mathbf{e}_1 \mathbf{e}_3^\top \mathbf{O} \mathbf{e}_2\right)\left(\mathbf{e}_4^\top \mathbf{O} \mathbf{e}_1 \mathbf{e}_4^\top \mathbf{O} \mathbf{e}_2\right)}_{k\neq\ell\geq 3,\text{ solved in Prop. 25}}$$

By summing all of the above and by some tedious algebraic steps, we finally get that,

$$\mathbb{E}\left(\mathbf{e}_i^\top \mathbf{O}^\top \mathbf{\Sigma}^+ \mathbf{\Sigma} \mathbf{O} \mathbf{e}_j\right)^2$$

$$= \frac{2p\left(m^4\left(6-3p-2p^2\right)+m^3\left(-12-20p+15p^2+7p^3\right)+m^2\left(18+13p+5p^2-21p^3-8p^4\right)\right)}{(p-3)(p-2)(p-1)p(p+1)(p+2)(p+4)(p+6)} +$$

$$\frac{2p\left(m\left(3p^5+8p^4+21p^3+28p^2-14p-12\right)+p\left(12-4p-29p^2-12p^3+p^4\right)\right)}{(p-3)(p-2)(p-1)p(p+1)(p+2)(p+4)(p+6)} +$$

$$d \cdot \frac{3m^4\left(-4+4p+3p^2\right)-4m^3\left(-6-13p+16p^2+8p^3\right)+m^2\left(-36+16p+29p^2+88p^3+36p^4\right)}{(p-3)(p-2)(p-1)p(p+1)(p+2)(p+4)(p+6)} +$$

$$d \cdot \frac{-2m\left(6p^5+13p^4+69p^3+105p^2+16p-12\right)+\left(-4p^5+54p^4+90p^3+152p^2+120p\right)}{(p-3)(p-2)(p-1)p(p+1)(p+2)(p+4)(p+6)} +$$

$$d^2 \cdot \frac{m^4(-5p-6)+m^3\left(18p^2+34p-12\right)+m^2\left(-20p^3-46p^2-39p-42\right)}{(p-3)(p-2)(p-1)p(p+1)(p+2)(p+4)(p+6)} +$$

$$d^2 \cdot \frac{m\left(6p^4+10p^3+96p^2+154p+60\right)+\left(2p^4-30p^3-32p^2-144p-144\right)}{(p-3)(p-2)(p-1)p(p+1)(p+2)(p+4)(p+6)} .$$

## C.3 EXTENDING FIGURE 2: MORE RANDOM SYNTHETIC DATA EXPERIMENTS

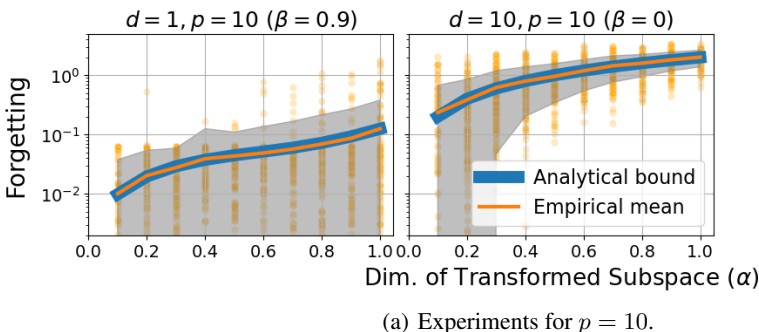

(a) Experiments for $p = 10$.

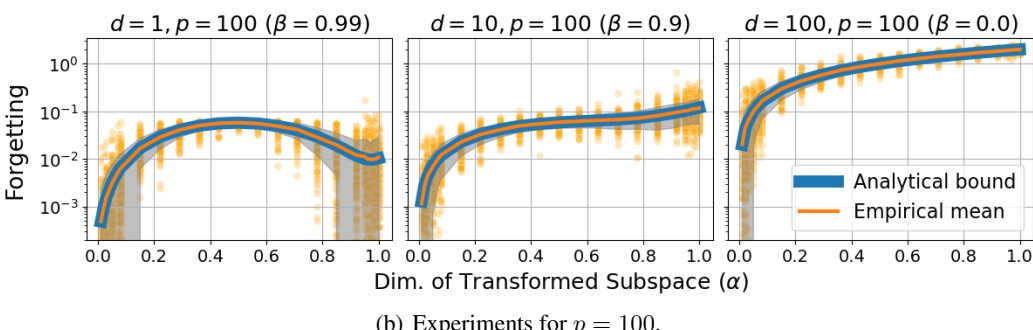

(b) Experiments for $p = 100$.

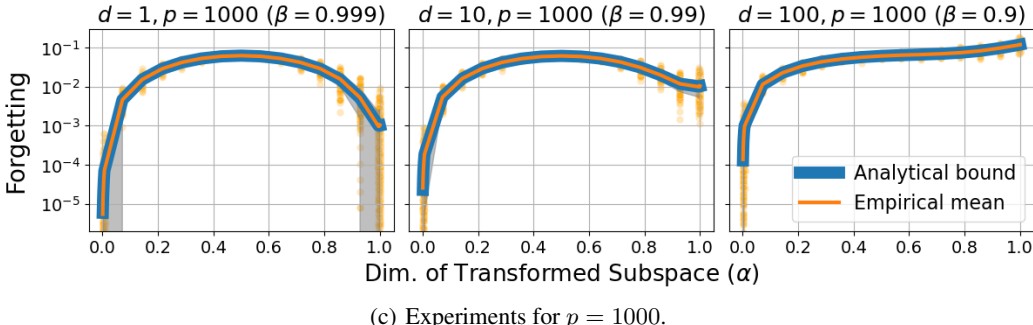

(c) Experiments for $p = 1000$.

Figure 9: Empirically illustrating the worst-case forgetting under different overparameterization levels. Points indicate the forgetting under many sampled random transformations applied on a (single) random data matrix $\mathbf{X}$. Their mean is shown in the thin orange line, with the standard deviation represented by a gray band. The thick blue line depicts the analytical expression of Theorem 3. The analytical bound matches the empirical mean, thus exemplifying the tightness of our analysis.

## D  Monomials of Entries of Random Orthogonal Matrices

Throughout the supplementary materials, we often wish to compute the expectation of an arbitrary monomial of the entries of a random orthogonal matrix $\mathbf{Q} \sim O(p)$ sampled uniformly from the orthogonal group, *e.g.*, $\mathbb{E}_{\mathbf{Q}}[q_{1,1}^2 q_{1,2}^4 q_{2,2}^6]$. Following Gorin (2002), we define a "power matrix" $\mathbf{M} \in \mathbb{Z}_{\geq 0}^{p \times R}$ (for some $R \leq p$) that maps into a monomial $\prod_{i,j=1}^{p,R} q_{i,j}^{m_{i,j}}$ constructed from the entries of the first $R$ columns of the random $\mathbf{Q} \in \mathbb{R}^{p \times p}$. We denote the expected value of this monomial by

$$\mathbb{E}_{\mathbf{Q}}\left[\prod_{i,j=1}^{p,R} q_{i,j}^{m_{i,j}}\right] \triangleq \langle \mathbf{M} \rangle, \quad \text{for example,} \quad \mathbb{E}_{\mathbf{Q}}[q_{1,1}^2 q_{2,1}^4 q_{2,2}^6] \triangleq \left\langle \begin{smallmatrix} 2 & 0 \\ 4 & 6 \\ \vec{0} & \vec{0} \end{smallmatrix} \right\rangle.$$

We employ the notation $\vec{0}$ to complement $\mathbf{M}$ to have $p$ rows. For instance, in the example above, $\vec{0}$ is a vector with $p-2$ zero entries.

The following are helpful properties of the integral (expectation) over the orthogonal group, as mentioned in Gorin (2002).

**Property 9** (Invariance of the integral over the orthogonal group). *Let $\mathbf{Q} \sim O(p)$ and $\mathbf{M} \in \mathbb{Z}_{\geq 0}^{p \times R}$.*

1.  ***Invariance w.r.t. transpose.*** *Since $\mathbf{Q}$ and $\mathbf{Q}^\top$ are identically distributed, it holds that $\langle \mathbf{M} \rangle = \langle \mathbf{M}^\top \rangle$. For example,*

$$\left\langle \begin{smallmatrix} 2 & 0 \\ 4 & 2 \\ 4 & 6 \\ \vec{0} & \vec{0} \end{smallmatrix} \right\rangle = \mathbb{E}_{\mathbf{Q}}[q_{1,1}^2 q_{2,1}^4 q_{2,2}^2 q_{3,1}^4 q_{3,2}^6] = \mathbb{E}_{\mathbf{Q}}[q_{1,1}^2 q_{1,2}^4 q_{1,3}^4 q_{2,2}^2 q_{2,3}^6] = \left\langle \begin{smallmatrix} 2 & 4 & 4 \\ 0 & 2 & 6 \\ \vec{0} & \vec{0} & \vec{0} \end{smallmatrix} \right\rangle.$$

2.  ***Invariance w.r.t. row and column permutations.*** *Since different rows/columns of $\mathbf{Q}$ are identically distributed, the integral over the orthogonal group $O(p)$ is invariant under permutations of columns or rows of the power matrix $\mathbf{M}$ (see also Ullah (1964)). For example,*

$$\left\langle \begin{smallmatrix} 2 & 0 \\ 4 & 6 \\ \vec{0} & \vec{0} \end{smallmatrix} \right\rangle = \mathbb{E}_{\mathbf{Q}}[q_{1,1}^2 q_{2,1}^4 q_{2,2}^6] = \mathbb{E}_{\mathbf{Q}}[q_{1,1}^6 q_{1,2}^4 q_{2,2}^2] = \left\langle \begin{smallmatrix} 6 & 4 \\ 0 & 2 \\ \vec{0} & \vec{0} \end{smallmatrix} \right\rangle.$$

The following is a known property of odd moments in integrals (expectations) over the orthogonal group (see Brody et al. (1981); Gorin (2002)).

**Property 10.** *If the sum over* any *row or column of the power matrix $\mathbf{M}$ is odd, the integral vanishes, i.e., $\langle \mathbf{M} \rangle = 0$. For example,* $\mathbb{E}_{\mathbf{Q}}[q_{1,1} q_{2,1}^4 q_{2,2}^6] = \left\langle \begin{smallmatrix} 1 & 0 \\ 4 & 6 \\ \vec{0} & \vec{0} \end{smallmatrix} \right\rangle = 0.$

*(In contrast, generally, it holds that $\left\langle \begin{smallmatrix} 1 & 1 \\ \vec{0} & \vec{0} \end{smallmatrix} \right\rangle \neq 0$).*

**Corollary 11.** *Let $\mathbf{q}_i$ be the $i^{th}$ column (or row) of $\mathbf{Q}$. Let $\mathbf{A}^{(1)}, \dots, \mathbf{A}^{(N)} \in \mathbb{R}^{p \times p}$ be $N$ matrices independent on $\mathbf{Q}$ and let $i_1, \dots, i_{2N} \in [p]$ be $2N$ indices. Then, if there exists an index $i \in [p]$ that appears an* odd *number of times in $i_1, \dots, i_{2N}$, the following expectation vanishes:*

$$\mathbb{E}[\mathbf{q}_{i_1} \mathbf{A}^{(i_1)} \mathbf{q}_{i_{N+1}} \cdot \mathbf{q}_{i_2} \mathbf{A}^{(i_2)} \mathbf{q}_{i_{N+2}} \cdots \mathbf{q}_{i_N} \mathbf{A}^{(i_N)} \mathbf{q}_{i_{2N}}] = 0.$$

*For example, the above dictates that,*

$$\mathbb{E}_{\mathbf{Q}_p, \mathbf{Q}_m}\left[ \mathbf{e}_1^\top \mathbf{Q}_p \begin{bmatrix} \mathbf{Q}_m \\ & \mathbf{I}_{p-m} \end{bmatrix} \mathbf{Q}_p^\top \mathbf{e}_1 \cdot \mathbf{e}_1^\top \mathbf{Q}_p \begin{bmatrix} \mathbf{0}_m \\ & \mathbf{I}_{p-m} \end{bmatrix} \mathbf{Q}_p^\top \mathbf{e}_2 \cdot \mathbf{e}_3^\top \mathbf{Q}_p \begin{bmatrix} \mathbf{Q}_m \\ & \mathbf{0}_{p-m} \end{bmatrix} \mathbf{Q}_p^\top \mathbf{e}_2 \right] = 0,$$

*since $\mathbf{q}_1$ appears 3 times (also because $\mathbf{q}_3$ appears once).*

*Proof.* The expectation can be rewritten as,

$$\mathbb{E}[\mathbf{q}_{i_1} \mathbf{A}^{(i_1)} \mathbf{q}_{i_{N+1}} \cdots \mathbf{q}_{i_N} \mathbf{A}^{(i_N)} \mathbf{q}_{i_{2N}}] = \sum_{j_1,\dots,j_N=1}^{p} \sum_{k_1,\dots,k_N=1}^{p} \mathbb{E}\left[\prod_{\ell=1}^{N} q_{j_\ell, i_\ell} a_{j_\ell, k_\ell}^{(i_\ell)} q_{k_\ell, i_{N+\ell}}\right].$$

Let $t \in [p]$ be an index that appears an odd number of times in $i_1, \dots, i_{2N}$. The $t^{\text{th}}$ column (or row) appears the same number of times in each of the (summand) expectations. Thus, the sum of the $t^{\text{th}}$ column (or row) corresponding to $\mathbf{q}_t$ in the power matrix $\mathbf{M}$ corresponding to that expectation will be an odd number. Thus, by Prop. 10, the expectation vanishes. $\square$

**Corollary 12.** *Let* $\mathbf{v}_1, \ldots, \mathbf{v}_{2N} \in \mathbb{R}^p$ *and* $c \in \mathbb{R}$ *be random variables independent of* $\mathbf{Q}_m \sim O(m)$. *Then, if* $N$ *is an odd number, the following expectation vanishes:*

$$\mathbb{E}[c \cdot \mathbf{v}_1^\top \begin{bmatrix} \mathbf{Q}_m & \mathbf{0}_{p-m} \end{bmatrix} \mathbf{v}_{N+1} \cdot \mathbf{v}_2^\top \begin{bmatrix} \mathbf{Q}_m & \mathbf{0}_{p-m} \end{bmatrix} \mathbf{v}_{N+2} \cdots \mathbf{v}_N^\top \begin{bmatrix} \mathbf{Q}_m & \mathbf{0}_{p-m} \end{bmatrix} \mathbf{v}_{2N}] = 0 \,.$$

*For example, the above dictates that the following expectation vanishes,*

$$\mathbb{E}_{\mathbf{Q}_p, \mathbf{Q}_m} \left[ \mathbf{e}_1^\top \mathbf{Q}_p \begin{bmatrix} \mathbf{Q}_m & \mathbf{0}_{p-m} \end{bmatrix} \mathbf{Q}_p^\top \mathbf{e}_1 \cdot \mathbf{e}_1^\top \mathbf{Q}_p \begin{bmatrix} \mathbf{Q}_m & \mathbf{0}_{p-m} \end{bmatrix} \mathbf{Q}_p^\top \mathbf{e}_1 \cdot \mathbf{e}_2^\top \mathbf{Q}_p \begin{bmatrix} \mathbf{Q}_m & \mathbf{0}_{p-m} \end{bmatrix} \mathbf{Q}_p^\top \mathbf{e}_2 \right] = 0 \,,$$

*since here* $N = 3$.
*In contrast, the above does* not *imply that the following expectation vanishes,*

$$\mathbb{E}_{\mathbf{Q}_p, \mathbf{Q}_m} \left[ \mathbf{e}_1^\top \mathbf{Q}_p \begin{bmatrix} \mathbf{Q}_m & \mathbf{0}_{p-m} \end{bmatrix} \mathbf{Q}_p^\top \mathbf{e}_1 \cdot \underbrace{\mathbf{e}_1^\top \mathbf{Q}_p \begin{bmatrix} \mathbf{0}_m & \mathbf{I}_{p-m} \end{bmatrix} \mathbf{Q}_p^\top \mathbf{e}_1}_{\text{here, this is considered as } c} \cdot \mathbf{e}_2^\top \mathbf{Q}_p \begin{bmatrix} \mathbf{Q}_m & \mathbf{0}_{p-m} \end{bmatrix} \mathbf{Q}_p^\top \mathbf{e}_2 \right] \,,$$

*since here* $N = 2$.

*Proof.* The expectations become,

$$\mathbb{E}\left[ c \cdot \prod_{\ell=1}^N \mathbf{v}_\ell^\top \begin{bmatrix} \mathbf{Q}_m & \mathbf{0}_\ell \end{bmatrix} \mathbf{v}_{N+\ell} \right] = \mathbb{E}\left[ c \cdot \prod_{\ell=1}^N \left( \sum_{i,j=1}^m (\mathbf{v}_\ell)_i \underbrace{(\mathbf{Q}_m)_{i,j}}_{\triangleq q_{i,j}} (\mathbf{v}_{N+\ell})_j \right) \right]$$

$$= \mathbb{E}\left[ c \cdot \prod_{\ell=1}^N \left( \sum_{i,j=1}^m q_{i,j} (\mathbf{v}_\ell)_i (\mathbf{v}_{N+\ell})_j \right) \right]$$

$$= \sum_{i_1, \ldots, i_N = 1}^m \sum_{j_1, \ldots, j_N = 1}^m \mathbb{E}\left[ c \cdot \prod_{\ell=1}^N q_{i_\ell, j_\ell} (\mathbf{v}_\ell)_{i_\ell} (\mathbf{v}_{N+\ell})_{j_\ell} \right] \,.$$

We notice that in each of the (summand) expectations the entries of $\mathbf{Q}_m$ appear exactly $N$ times. However, since $N$ is odd, at least one row or column of the power matrix corresponding to the monomial in the expectation must have an odd sum. Then, by Prop. 10, all these expectations vanish. □

The main result we need from Gorin (2002) is their Eq. (23), providing a recursive formula to compute $\langle \mathbf{M} \rangle$ for any power matrix $\mathbf{M}$. We bring this formula here for the sake of completeness.

**Lemma 13** (Recursive formula for computing expectations of monomials over the orthogonal group). *Define the Pochhammer symbol* $(z)_n = \frac{\Gamma(z+n)}{\Gamma(z)}$. *Denote* $\binom{\overrightarrow{n}}{k} = \prod_{i=1}^{p} \binom{n_i}{k_i}$. *Denote* $(\overrightarrow{n} | K) = \prod_{i=1}^{p} (n_i \mid K_{i,1}, \dots, K_{i,R-1})$.

*Then, the one-vector integral is given by* $\langle \overrightarrow{m} \rangle = \left( \frac{p}{2} \right)_{\overline{m}/2}^{-1} \prod_{i=1}^{p} \left( \frac{1}{2} \right)_{m_i/2}$.

*Moreover, the $R$-vector integral (corresponding to a matrix $\mathbf{M} \in \mathbb{Z}_{\geq 0}^{p \times R}$, is given by*

$$\langle \mathbf{M} \rangle = \left( \frac{p-R+1}{2} \right)_{\overline{m}_R/2}^{-1} \cdot$$

$$\cdot \sum_{\overrightarrow{\kappa}} \left( \binom{\overline{m}_R}{\overrightarrow{\kappa}} (-1)^{(\overline{m}_R - \overline{\kappa})/2} \cdot \prod_{i=1}^{p} \left( \tfrac{1}{2} \right)_{\kappa_i/2} \cdot \sum_{K} (\overrightarrow{m}_R - \overrightarrow{\kappa} | K) \cdot \prod_{j=1}^{R-1} \left( \tfrac{1}{2} \right)_{\overline{k}_j/2} \cdot \langle \mathbf{M}^{(R-1)} + K \rangle \right)$$

*where the first sum runs over all $\overrightarrow{\kappa}$ with all even entries (less or equal to the corresponding entries of the last column $\overrightarrow{m}_R$). The second sum runs over all $K \in \mathbb{Z}_{\geq 0}^{p, R-1}$ for which all sums of columns are even, i.e., $\overline{k}_j \triangleq \sum_{i=1}^{p} k_{i,j}$ is even $\forall j \in [R-1]$. Finally, $\mathbf{M}^{(R-1)}$ stands for the first $R-1$ columns of $\mathbf{M}$, and $\overline{m}_R = \sum_{i=1}^{p} m_{i,R}$ and $\overline{\kappa} = \sum_{i=1}^{p} \kappa_i$.*

In the pages to come, we present many calculations of expectations of different monomials that we use throughout the supplementary materials. We include these calculations since the recursive formula of Lemma 13 is somewhat complicated to apply, and we wish our derivations to be reproducible and easily followed.

## D.1 MONOMIALS OF ONE ORTHOGONAL VECTOR

$$\mathbb{E}[u_1^4] = \left\langle \tfrac{4}{0} \right\rangle = \left( \tfrac{p}{2} \right)_2^{-1} \left( \tfrac{1}{2} \right)_2 = \left( \tfrac{p}{2} \left( \tfrac{p+2}{2} \right) \right)^{-1} \tfrac{3}{4} = \tfrac{3}{p(p+2)}$$

$$\mathbb{E}[u_1^6] = \left\langle \tfrac{6}{0} \right\rangle = \left( \tfrac{p}{2} \right)_3^{-1} \left( \tfrac{1}{2} \right)_3 = \tfrac{15}{p(p+2)(p+4)}$$

$$\mathbb{E}[u_1^2 u_2^2] = \left\langle \begin{smallmatrix} 2 \\ 2 \\ 0 \end{smallmatrix} \right\rangle = \left( \tfrac{p}{2} \right)_2^{-1} \left( \tfrac{1}{2} \right)_1 \left( \tfrac{1}{2} \right)_1 = \left( \tfrac{p}{2} \left( \tfrac{p+2}{2} \right) \right)^{-1} \tfrac{1}{4} = \tfrac{1}{p(p+2)}$$

$$\mathbb{E}[u_1^4 u_2^2] = \left\langle \begin{smallmatrix} 4 \\ 2 \\ 0 \end{smallmatrix} \right\rangle = \left( \tfrac{p}{2} \right)_3^{-1} \left( \tfrac{1}{2} \right)_2 \left( \tfrac{1}{2} \right)_1 = \tfrac{3}{p(p+2)(p+4)}$$

$$\mathbb{E}[u_1^2 u_2^2 u_3^2] = \left\langle \begin{smallmatrix} 2 \\ 2 \\ 2 \\ 0 \end{smallmatrix} \right\rangle = \left( \tfrac{p}{2} \right)_3^{-1} \left( \tfrac{1}{2} \right)_1^3 = \tfrac{1}{p(p+2)(p+4)}$$

### D.2 MONOMIALS OF TWO ORTHOGONAL VECTOR

### D.2.1 ONE INDEX (ROW)

$$\left\langle {\textstyle\begin{smallmatrix}4\\0\end{smallmatrix}}\ {\textstyle\begin{smallmatrix}4\\0\end{smallmatrix}}\right\rangle = \left(\frac{p-1}{2}\left(\frac{p+1}{2}\right)\right)^{-1}\left((-1)^2\left(\tfrac{1}{2}\right)_2\left\langle{\textstyle\begin{smallmatrix}8\\0\end{smallmatrix}}\right\rangle + \binom{4}{2}(-1)^1\left(\tfrac{1}{2}\right)_1^2\left\langle{\textstyle\begin{smallmatrix}6\\0\end{smallmatrix}}\right\rangle + (-1)^0\left(\tfrac{1}{2}\right)_2\left\langle{\textstyle\begin{smallmatrix}4\\0\end{smallmatrix}}\right\rangle\right)$$

$$= \frac{1}{(p-1)(p+1)}\left(3\left\langle{\textstyle\begin{smallmatrix}8\\0\end{smallmatrix}}\right\rangle - 6\left\langle{\textstyle\begin{smallmatrix}6\\0\end{smallmatrix}}\right\rangle + 3\left\langle{\textstyle\begin{smallmatrix}4\\0\end{smallmatrix}}\right\rangle\right)$$

$$= \frac{3}{(p-1)(p+1)}\left(\left(\tfrac{p}{2}\right)_4^{-1}\left(\tfrac{1}{2}\right)_4 - 2\left(\tfrac{p}{2}\right)_3^{-1}\left(\tfrac{1}{2}\right)_3 + \left(\tfrac{p}{2}\right)_2^{-1}\left(\tfrac{1}{2}\right)_2\right)$$

$$= \frac{3}{(p-1)(p+1)}\left(\frac{105}{p(p+2)(p+4)(p+6)} - \frac{30}{p(p+2)(p+4)} + \frac{3}{p(p+2)}\right)$$

$$= \frac{3}{(p-1)(p+1)}\left(\frac{3(p-1)(p+1)}{p(p+2)(p+4)(p+6)}\right) = \frac{9}{p(p+2)(p+4)(p+6)}$$

$$\left\langle{\textstyle\begin{smallmatrix}2\\0\end{smallmatrix}}\ {\textstyle\begin{smallmatrix}2\\0\end{smallmatrix}}\right\rangle = \frac{1}{p(p+2)}$$

$$\left\langle{\textstyle\begin{smallmatrix}4\\0\end{smallmatrix}}\ {\textstyle\begin{smallmatrix}2\\0\end{smallmatrix}}\right\rangle = \left(\frac{p-1}{2}\right)^{-1}\left((-1)^1\left(\tfrac{1}{2}\right)_1\left\langle{\textstyle\begin{smallmatrix}6\\0\end{smallmatrix}}\right\rangle + (-1)^0\left(\tfrac{1}{2}\right)_1\left\langle{\textstyle\begin{smallmatrix}4\\0\end{smallmatrix}}\right\rangle\right)$$

$$= \frac{1}{(p-1)}\left(\left\langle{\textstyle\begin{smallmatrix}4\\0\end{smallmatrix}}\right\rangle - \left\langle{\textstyle\begin{smallmatrix}6\\0\end{smallmatrix}}\right\rangle\right) = \frac{1}{(p-1)}\left(\left(\tfrac{p}{2}\right)_2^{-1}\left(\tfrac{1}{2}\right)_2 - \left(\tfrac{p}{2}\right)_3^{-1}\left(\tfrac{1}{2}\right)_3\right)$$

$$= \frac{1}{(p-1)}\left(\frac{3}{p(p+2)} - \frac{15}{p(p+2)(p+4)}\right) = \frac{3(p-1)}{(p-1)p(p+2)(p+4)} = \frac{3}{p(p+2)(p+4)}$$

$$\left\langle{\textstyle\begin{smallmatrix}6\\0\end{smallmatrix}}\ {\textstyle\begin{smallmatrix}2\\0\end{smallmatrix}}\right\rangle = \left\langle{\textstyle\begin{smallmatrix}6\\2\\0\end{smallmatrix}}\right\rangle = \left(\tfrac{p}{2}\right)_4^{-1}\left(\tfrac{1}{2}\right)_3\left(\tfrac{1}{2}\right)_1 = \frac{15}{p(p+2)(p+4)(p+6)}$$

### D.2.2 TWO INDICES (ROWS)

$$\left\langle \begin{smallmatrix} 2 & 2 \\ 2 & 2 \\ 0 & 0 \end{smallmatrix} \right\rangle = \tfrac{1}{4}\left(\tfrac{p-1}{2}\left(\tfrac{p+1}{2}\right)\right)^{-1}\left(3\left\langle \begin{smallmatrix} 4 \\ 4 \\ 0 \end{smallmatrix} \right\rangle + \left\langle \begin{smallmatrix} 2 \\ 2 \\ 0 \end{smallmatrix} \right\rangle - \left\langle \begin{smallmatrix} 2 \\ 4 \\ 0 \end{smallmatrix} \right\rangle - \left\langle \begin{smallmatrix} 4 \\ 2 \\ 0 \end{smallmatrix} \right\rangle\right)$$

$$= \tfrac{1}{(p-1)(p+1)}\left(3\left(\tfrac{p}{2}\right)_4^{-1}\left(\tfrac{1}{2}\right)_2\left(\tfrac{1}{2}\right)_2 + \left(\tfrac{p}{2}\right)_2^{-1}\left(\tfrac{1}{2}\right)_1\left(\tfrac{1}{2}\right)_1 - 2\left(\tfrac{p}{2}\right)_3^{-1}\left(\tfrac{1}{2}\right)_1\left(\tfrac{1}{2}\right)_2\right)$$

$$= \tfrac{1}{(p-1)(p+1)}\left(3\tfrac{9}{16}\left(\tfrac{p}{2}\right)_4^{-1} + \tfrac{1}{4}\left(\tfrac{p}{2}\right)_2^{-1} - \tfrac{2}{4}\cdot\tfrac{3}{4}\left(\tfrac{p}{2}\right)_3^{-1}\right)$$

$$= \tfrac{1}{(p-1)(p+1)}\left(\tfrac{27}{16}\left(\tfrac{p}{2}\cdot\tfrac{p+2}{2}\cdot\tfrac{p+4}{2}\cdot\tfrac{p+6}{2}\right)^{-1} + \tfrac{1}{4}\left(\tfrac{p}{2}\cdot\tfrac{p+2}{2}\right)^{-1} - \tfrac{3}{4}\left(\tfrac{p}{2}\cdot\tfrac{p+2}{2}\cdot\tfrac{p+4}{2}\right)^{-1}\right)$$

$$= \tfrac{1}{(p-1)(p+1)}\left(\tfrac{27}{p(p+2)(p+4)(p+6)} + \tfrac{1}{p(p+2)} - \tfrac{3}{4}\tfrac{8}{p(p+2)(p+4)}\right)$$

$$= \tfrac{1}{(p-1)(p+1)}\left(\tfrac{27+(p+4)(p+6)-6(p+6)}{p(p+2)(p+4)(p+6)}\right) = \tfrac{p^2+4p+15}{(p-1)p(p+1)(p+2)(p+4)(p+6)}$$

$$\left\langle \begin{smallmatrix} 4 & 0 \\ 0 & 4 \\ 0 & 0 \end{smallmatrix} \right\rangle = \left(\tfrac{p-1}{2}\left(\tfrac{p+1}{2}\right)\right)^{-1}\left((-1)^2\left(\tfrac{1}{2}\right)_2\left\langle \begin{smallmatrix} 4 \\ 4 \\ 0 \end{smallmatrix} \right\rangle + \binom{4}{2}(-1)^1\left(\tfrac{1}{2}\right)_1^2\left\langle \begin{smallmatrix} 2 \\ 2 \\ 0 \end{smallmatrix} \right\rangle + (-1)^0\left(\tfrac{1}{2}\right)_2\left\langle \begin{smallmatrix} 0 \\ 0 \\ 0 \end{smallmatrix} \right\rangle\right)$$

$$= \tfrac{1}{(p-1)(p+1)}\left(3\left\langle \begin{smallmatrix} 4 \\ 4 \\ 0 \end{smallmatrix} \right\rangle - 6\left\langle \begin{smallmatrix} 2 \\ 2 \\ 0 \end{smallmatrix} \right\rangle + 3\left\langle \begin{smallmatrix} 0 \\ 0 \\ 0 \end{smallmatrix} \right\rangle\right)$$

$$= \tfrac{3}{(p-1)(p+1)}\left(\left(\tfrac{p}{2}\right)_4^{-1}\left(\tfrac{1}{2}\right)_2^2 - 2\left(\tfrac{p}{2}\right)_3^{-1}\left(\tfrac{1}{2}\right)_2\left(\tfrac{1}{2}\right)_1 + \left(\tfrac{p}{2}\right)_2^{-1}\left(\tfrac{1}{2}\right)_2\right)$$

$$= \tfrac{3}{(p-1)(p+1)}\left(\tfrac{9}{p(p+2)(p+4)(p+6)} - \tfrac{6}{p(p+2)(p+4)} + \tfrac{3}{p(p+2)}\right)$$

$$= \tfrac{3}{(p-1)(p+1)}\left(\tfrac{3(p+3)(p+5)}{p(p+2)(p+4)(p+6)}\right)$$

$$= \tfrac{9(p+3)(p+5)}{(p-1)p(p+1)(p+2)(p+4)(p+6)}$$

$$\left\langle \begin{smallmatrix} 4 & 0 \\ 0 & 2 \\ 0 & 0 \end{smallmatrix} \right\rangle = \left(\tfrac{p-1}{2}\right)^{-1}\left((-1)^1\left(\tfrac{1}{2}\right)_1\left\langle \begin{smallmatrix} 2 \\ 2 \\ 0 \end{smallmatrix} \right\rangle + (-1)^0\left(\tfrac{1}{2}\right)_1\left\langle \begin{smallmatrix} 0 \\ 0 \\ 0 \end{smallmatrix} \right\rangle\right)$$

$$= \tfrac{1}{(p-1)}\left(\left\langle \begin{smallmatrix} 0 \\ 0 \\ 0 \end{smallmatrix} \right\rangle - \left\langle \begin{smallmatrix} 2 \\ 2 \\ 0 \end{smallmatrix} \right\rangle\right) = \tfrac{1}{(p-1)}\left(\left(\tfrac{p}{2}\right)_2^{-1}\left(\tfrac{1}{2}\right)_2 - \left(\tfrac{p}{2}\right)_3^{-1}\left(\tfrac{1}{2}\right)_2\left(\tfrac{1}{2}\right)_1\right)$$

$$= \tfrac{1}{(p-1)}\left(\tfrac{3}{p(p+2)} - \tfrac{3}{p(p+2)(p+4)}\right) = \tfrac{3(p+3)}{(p-1)p(p+2)(p+4)}$$

$$\left\langle \begin{smallmatrix} 6 & 0 \\ 0 & 2 \\ 0 & 0 \end{smallmatrix} \right\rangle = \left(\tfrac{p-1}{2}\right)^{-1}\left((-1)^1\left(\tfrac{1}{2}\right)_1\left\langle \begin{smallmatrix} 2 \\ 2 \\ 0 \end{smallmatrix} \right\rangle + (-1)^0\left(\tfrac{1}{2}\right)_1\left\langle \begin{smallmatrix} 6 \\ 0 \\ 0 \end{smallmatrix} \right\rangle\right)$$

$$= \tfrac{1}{p-1}\left(\left\langle \begin{smallmatrix} 6 \\ 0 \\ 0 \end{smallmatrix} \right\rangle - \left\langle \begin{smallmatrix} 6 \\ 2 \\ 0 \end{smallmatrix} \right\rangle\right) = \tfrac{1}{p-1}\left(\left(\tfrac{p}{2}\right)_3^{-1}\left(\tfrac{1}{2}\right)_3 - \left(\tfrac{p}{2}\right)_4^{-1}\left(\tfrac{1}{2}\right)_3\left(\tfrac{1}{2}\right)_1\right)$$

$$= \tfrac{1}{p-1}\left(\tfrac{15}{p(p+2)(p+4)} - \tfrac{15}{p(p+2)(p+4)(p+6)}\right) = \tfrac{15(p+5)}{(p-1)p(p+2)(p+4)(p+6)}$$

$$\left\langle \begin{smallmatrix} 3 & 3 \\ 1 & 1 \\ 0 & 0 \end{smallmatrix} \right\rangle = \left\langle \begin{smallmatrix} 3 & 1 \\ 3 & 1 \\ 0 & 0 \end{smallmatrix} \right\rangle = \left(\tfrac{p-1}{2}\left(\tfrac{p+1}{2}\right)\right)^{-1}\left((-1)^2\left(\tfrac{1}{2}\right)_2\left\langle \begin{smallmatrix} 6 \\ 2 \\ 0 \end{smallmatrix} \right\rangle + \binom{3}{2}(-1)^1\left(\tfrac{1}{2}\right)_1\left(\tfrac{1}{2}\right)_1\left\langle \begin{smallmatrix} 4 \\ 2 \\ 0 \end{smallmatrix} \right\rangle\right)$$

$$= \tfrac{4}{(p-1)(p+1)}\left(\tfrac{3}{4}\left(\tfrac{p}{2}\right)_4^{-1}\left(\tfrac{1}{2}\right)_3\left(\tfrac{1}{2}\right)_1 - \tfrac{3!}{2!1!}\cdot\tfrac{3}{8}\left(\tfrac{p}{2}\right)_3^{-1}\left(\tfrac{1}{2}\right)_1\left(\tfrac{1}{2}\right)_1\right)$$

$$= \tfrac{4}{(p-1)(p+1)}\left(\tfrac{3}{4}\cdot\tfrac{15}{16}\left(\tfrac{p}{2}\right)_4^{-1} - \tfrac{9}{8}\cdot\tfrac{1}{4}\left(\tfrac{p}{2}\right)_3^{-1}\right)$$

$$= \tfrac{4}{(p-1)(p+1)}\left(\tfrac{45}{64}\cdot\left(\tfrac{p}{2}\cdot\tfrac{p+2}{2}\cdot\tfrac{p+4}{2}\cdot\tfrac{p+6}{2}\right)^{-1} - \tfrac{9}{32}\cdot\left(\tfrac{p}{2}\cdot\tfrac{p+2}{2}\cdot\tfrac{p+4}{2}\right)^{-1}\right)$$

$$= \tfrac{1}{(p-1)(p+1)}\left(\tfrac{45}{p(p+2)(p+4)(p+6)} - \tfrac{9}{p(p+2)(p+4)}\right) = \tfrac{-9(p+1)}{(p-1)p(p+1)(p+2)(p+4)(p+6)}$$

$$\left\langle \begin{smallmatrix}3&1\\1&3\\\vec0&\vec0\end{smallmatrix}\right\rangle = \left(\tfrac{p-1}{2}\left(\tfrac{p+1}{2}\right)\right)^{-1}\left((-1)^2\left(\tfrac12\right)_2\left\langle\begin{smallmatrix}4\\4\\\vec0\end{smallmatrix}\right\rangle + \binom{3}{2}(-1)^1\left(\tfrac12\right)_1^2\left\langle\begin{smallmatrix}4\\2\\\vec0\end{smallmatrix}\right\rangle\right)$$

$$= \frac{3}{(p-1)(p+1)}\left(\left\langle\begin{smallmatrix}4\\4\\\vec0\end{smallmatrix}\right\rangle - \left\langle\begin{smallmatrix}4\\2\\\vec0\end{smallmatrix}\right\rangle\right) = \frac{3}{(p-1)(p+1)}\left(\left(\tfrac{p}{2}\right)_4^{-1}\left(\tfrac12\right)_2^2 - \left(\tfrac{p}{2}\right)_3^{-1}\left(\tfrac12\right)_1\left(\tfrac12\right)_2\right)$$

$$= \frac{3}{(p-1)(p+1)}\left(\frac{9}{p(p+2)(p+4)(p+6)} - \frac{3}{p(p+2)(p+4)}\right) = \frac{-9(p+3)}{(p-1)p(p+1)(p+2)(p+4)(p+6)}$$

$$\left\langle\begin{smallmatrix}3&1\\1&1\\\vec0&\vec0\end{smallmatrix}\right\rangle = \frac{1}{p-1}\left((-1)^1\left(\tfrac12\right)_1\left\langle\begin{smallmatrix}4\\2\\\vec0\end{smallmatrix}\right\rangle\right) = \frac{-1}{p-1}\left(\left(\tfrac{p}{2}\right)_3^{-1}\left(\tfrac12\right)_2\left(\tfrac12\right)_1\right) = \frac{-3}{(p-1)p(p+2)(p+4)}$$

$$\left\langle\begin{smallmatrix}5&1\\1&1\\\vec0&\vec0\end{smallmatrix}\right\rangle = \left(\tfrac{p-1}{2}\right)^{-1}\left((-1)^1\left(\tfrac12\right)_1\left\langle\begin{smallmatrix}6\\2\\\vec0\end{smallmatrix}\right\rangle\right) = \frac{1}{p-1}\left(-\left(\tfrac{p}{2}\right)_4^{-1}\left(\tfrac12\right)_1\left(\tfrac12\right)_3\right) = \frac{-15}{(p-1)p(p+2)(p+4)(p+6)}$$

$$\left\langle\begin{smallmatrix}4&2\\0&2\\\vec0&\vec0\end{smallmatrix}\right\rangle = \frac{4}{(p-1)(p+1)}\Big((-1)^2\left(\tfrac12\right)_2\left\langle\begin{smallmatrix}6\\2\\\vec0\end{smallmatrix}\right\rangle + (-1)^1\left(\tfrac12\right)_1^2\left\langle\begin{smallmatrix}4\\2\\\vec0\end{smallmatrix}\right\rangle +$$

$$(-1)^1\left(\tfrac12\right)_1^2\left\langle\begin{smallmatrix}6\\0\\\vec0\end{smallmatrix}\right\rangle + (-1)^0\left(\tfrac12\right)_1^2\left\langle\begin{smallmatrix}4\\0\\\vec0\end{smallmatrix}\right\rangle\Big)$$

$$= \frac{1}{(p-1)(p+1)}\left(3\left\langle\begin{smallmatrix}6\\2\\\vec0\end{smallmatrix}\right\rangle - \left\langle\begin{smallmatrix}4\\2\\\vec0\end{smallmatrix}\right\rangle - \left\langle\begin{smallmatrix}6\\0\\\vec0\end{smallmatrix}\right\rangle + \left\langle\begin{smallmatrix}4\\0\\\vec0\end{smallmatrix}\right\rangle\right)$$

$$= \frac{1}{(p-1)(p+1)}\left(3\left(\tfrac{p}{2}\right)_4^{-1}\left(\tfrac12\right)_3\left(\tfrac12\right)_1 - \left(\tfrac{p}{2}\right)_3^{-1}\left(\tfrac12\right)_2\left(\tfrac12\right)_1 - \left(\tfrac{p}{2}\right)_3^{-1}\left(\tfrac12\right)_3 + \left(\tfrac{p}{2}\right)_2^{-1}\left(\tfrac12\right)_2\right)$$

$$= \frac{1}{(p-1)(p+1)}\left(\frac{45}{p(p+2)(p+4)(p+6)} + \frac{-3-15}{p(p+2)(p+4)} + \frac{3}{p(p+2)}\right)$$

$$= \frac{3(p+1)(p+3)}{(p-1)p(p+1)(p+2)(p+4)(p+6)} = \frac{3(p+3)}{(p-1)p(p+2)(p+4)(p+6)}$$

$$\left\langle\begin{smallmatrix}4&2\\2&0\\\vec0&\vec0\end{smallmatrix}\right\rangle = \left(\tfrac{p-1}{2}\right)^{-1}\left((-1)^1\left(\tfrac12\right)_1\left\langle\begin{smallmatrix}6\\2\\\vec0\end{smallmatrix}\right\rangle + (-1)^0\left(\tfrac12\right)_1\left\langle\begin{smallmatrix}4\\2\\\vec0\end{smallmatrix}\right\rangle\right) = \frac{1}{p-1}\left(\left\langle\begin{smallmatrix}4\\2\\\vec0\end{smallmatrix}\right\rangle - \left\langle\begin{smallmatrix}6\\2\\\vec0\end{smallmatrix}\right\rangle\right)$$

$$= \frac{1}{p-1}\left(\left(\tfrac{p}{2}\right)_3^{-1}\left(\tfrac12\right)_2\left(\tfrac12\right)_1 - \left(\tfrac{p}{2}\right)_4^{-1}\left(\tfrac12\right)_3\left(\tfrac12\right)_1\right)$$

$$= \frac{1}{p-1}\left(\frac{3}{p(p+2)(p+4)} - \frac{15}{p(p+2)(p+4)(p+6)}\right) = \frac{3(p+1)}{(p-1)p(p+2)(p+4)(p+6)}$$

$$\left\langle\begin{smallmatrix}2&1\\0&1\\\vec0&\vec0\end{smallmatrix}\right\rangle = \frac{1}{p-1}\left(-\left\langle\begin{smallmatrix}3\\1\\\vec0\end{smallmatrix}\right\rangle\right) = 0$$

$$\left\langle\begin{smallmatrix}2&0\\0&2\\\vec0&\vec0\end{smallmatrix}\right\rangle = \frac{2}{p-1}\left((-1)^1\left(\tfrac12\right)_1\left\langle\begin{smallmatrix}2\\2\\\vec0\end{smallmatrix}\right\rangle + (-1)^0\left(\tfrac12\right)_1\left\langle\begin{smallmatrix}2\\0\\\vec0\end{smallmatrix}\right\rangle\right) = \frac{1}{p-1}\left(\left\langle\begin{smallmatrix}2\\0\\\vec0\end{smallmatrix}\right\rangle - \left\langle\begin{smallmatrix}2\\2\\\vec0\end{smallmatrix}\right\rangle\right)$$

$$= \frac{1}{p-1}\left(\frac{1}{p} - \frac{1}{p(p+2)}\right) = \frac{p+1}{(p-1)p(p+2)}$$

$$\left\langle\begin{smallmatrix}1&1\\1&1\\\vec0&\vec0\end{smallmatrix}\right\rangle = \frac{1}{p-1}\left(-\left\langle\begin{smallmatrix}2\\2\\\vec0\end{smallmatrix}\right\rangle\right) = \frac{1}{p-1}\left(-\left(\tfrac{p}{2}\right)_2^{-1}\left(\tfrac12\right)_1^2\right) = \frac{-1}{(p-1)p(p+2)}$$

$$\left\langle\begin{smallmatrix}2&2\\2&0\\\vec0&\vec0\end{smallmatrix}\right\rangle = \left(\tfrac{p-1}{2}\right)_1^{-1}\left((-1)^1\left(\tfrac12\right)_1\left\langle\begin{smallmatrix}4\\2\\\vec0\end{smallmatrix}\right\rangle + (-1)^0\left(\tfrac12\right)_1\left(\tfrac12\right)_0\left\langle\begin{smallmatrix}2\\2\\\vec0\end{smallmatrix}\right\rangle\right)$$

$$= \frac{1}{p-1}\left(\left\langle\begin{smallmatrix}2\\2\\\vec0\end{smallmatrix}\right\rangle - \left\langle\begin{smallmatrix}4\\2\\\vec0\end{smallmatrix}\right\rangle\right) = \frac{1}{p-1}\left(\left(\tfrac{p}{2}\right)_2^{-1}\left(\tfrac12\right)_1^2 - \left(\tfrac{p}{2}\right)_3^{-1}\left(\tfrac12\right)_1\left(\tfrac12\right)_2\right)$$

$$= \frac{(p+4)-3}{(p-1)p(p+2)(p+4)} = \frac{p+1}{(p-1)p(p+2)(p+4)}$$

$$\left\langle\begin{smallmatrix}2&2\\4&0\\\vec0&\vec0\end{smallmatrix}\right\rangle = \left\langle\begin{smallmatrix}4&2\\0&2\\\vec0&\vec0\end{smallmatrix}\right\rangle = \frac{3p+9}{(p-1)p(p+2)(p+4)(p+6)}$$

### D.2.3 THREE INDICES (ROWS)

$$\left\langle \begin{smallmatrix} 4 & 0 \\ 0 & 2 \\ 0 & 2 \\ \rightarrow & \rightarrow \\ 0 & 0 \end{smallmatrix} \right\rangle = \left( \tfrac{p-1}{2} \left( \tfrac{p+1}{2} \right) \right)^{-1} \left( (-1)^2 \left( \tfrac{1}{2} \right)_2 \left\langle \begin{smallmatrix} 4 \\ 2 \\ 2 \\ \rightarrow \\ 0 \end{smallmatrix} \right\rangle + \right.$$

$$\left. (-1)^1 \left( \tfrac{1}{2} \right)_1^2 \left\langle \begin{smallmatrix} 4 \\ 0 \\ 2 \\ \rightarrow \\ 0 \end{smallmatrix} \right\rangle + (-1)^1 \left( \tfrac{1}{2} \right)_1^2 \left\langle \begin{smallmatrix} 4 \\ 2 \\ 0 \\ \rightarrow \\ 0 \end{smallmatrix} \right\rangle + (-1)^0 \left( \tfrac{1}{2} \right)_1^2 \left\langle \begin{smallmatrix} 4 \\ 0 \\ 0 \\ \rightarrow \\ 0 \end{smallmatrix} \right\rangle \right)$$

$$= \tfrac{1}{(p-1)(p+1)} \left( 3 \left\langle \begin{smallmatrix} 4 \\ 2 \\ 2 \\ \rightarrow \\ 0 \end{smallmatrix} \right\rangle - 2 \left\langle \begin{smallmatrix} 4 \\ 2 \\ 0 \\ \rightarrow \\ 0 \end{smallmatrix} \right\rangle + \left\langle \begin{smallmatrix} 4 \\ 0 \\ 0 \\ \rightarrow \\ 0 \end{smallmatrix} \right\rangle \right)$$

$$= \tfrac{1}{(p-1)(p+1)} \left( 3 \left( \tfrac{p}{2} \right)_4^{-1} \left( \tfrac{1}{2} \right)_1^2 \left( \tfrac{1}{2} \right)_2 - 2 \left( \tfrac{p}{2} \right)_3^{-1} \left( \tfrac{1}{2} \right)_1 \left( \tfrac{1}{2} \right)_2 + \left( \tfrac{p}{2} \right)_2^{-1} \left( \tfrac{1}{2} \right)_2 \right)$$

$$= \tfrac{1}{(p-1)(p+1)} \left( \tfrac{9}{p(p+2)(p+4)(p+6)} - \tfrac{6}{p(p+2)(p+4)} + \tfrac{3}{p(p+2)} \right)$$

$$= \tfrac{3(p+3)(p+5)}{(p-1)p(p+1)(p+2)(p+4)(p+6)}$$

$$\left\langle \begin{smallmatrix} 2 & 0 \\ 4 & 0 \\ 0 & 2 \\ \rightarrow & \rightarrow \\ 0 & 0 \end{smallmatrix} \right\rangle = \left( \tfrac{p-1}{2} \right)^{-1} \left( (-1)^1 \left( \tfrac{1}{2} \right)_1 \left\langle \begin{smallmatrix} 2 \\ 4 \\ 2 \\ \rightarrow \\ 0 \end{smallmatrix} \right\rangle + (-1)^0 \left( \tfrac{1}{2} \right)_1 \left\langle \begin{smallmatrix} 2 \\ 4 \\ 0 \\ \rightarrow \\ 0 \end{smallmatrix} \right\rangle \right) = \tfrac{1}{p-1} \left( \left\langle \begin{smallmatrix} 2 \\ 4 \\ 0 \\ \rightarrow \\ 0 \end{smallmatrix} \right\rangle - \left\langle \begin{smallmatrix} 2 \\ 4 \\ 2 \\ \rightarrow \\ 0 \end{smallmatrix} \right\rangle \right)$$

$$= \tfrac{1}{p-1} \left( \left( \tfrac{p}{2} \right)_3^{-1} \left( \tfrac{1}{2} \right)_1 \left( \tfrac{1}{2} \right)_2 - \left( \tfrac{p}{2} \right)_4^{-1} \left( \tfrac{1}{2} \right)_1^2 \left( \tfrac{1}{2} \right)_2 \right) = \tfrac{3(p+5)}{(p-1)p(p+2)(p+4)(p+6)}$$

$$\left\langle \begin{smallmatrix} 2 & 2 \\ 1 & 1 \\ 1 & 1 \\ \rightarrow & \rightarrow \\ 0 & 0 \end{smallmatrix} \right\rangle = \left( \tfrac{p-1}{2} \right)_2^{-1} \left( (-1)^2 \left( \tfrac{1}{2} \right)_2 \left\langle \begin{smallmatrix} 4 \\ 2 \\ 2 \\ \rightarrow \\ 0 \end{smallmatrix} \right\rangle + (-1)^1 \left( \tfrac{1}{2} \right)_1^2 \left\langle \begin{smallmatrix} 2 \\ 2 \\ 2 \\ \rightarrow \\ 0 \end{smallmatrix} \right\rangle \right)$$

$$= \tfrac{4}{(p-1)(p+1)} \left( \tfrac{3}{4} \left\langle \begin{smallmatrix} 4 \\ 2 \\ 2 \\ \rightarrow \\ 0 \end{smallmatrix} \right\rangle - \tfrac{1}{4} \left\langle \begin{smallmatrix} 2 \\ 2 \\ 2 \\ \rightarrow \\ 0 \end{smallmatrix} \right\rangle \right) = \tfrac{1}{(p-1)(p+1)} \left( 3 \left\langle \begin{smallmatrix} 4 \\ 2 \\ 2 \\ \rightarrow \\ 0 \end{smallmatrix} \right\rangle - \left\langle \begin{smallmatrix} 2 \\ 2 \\ 2 \\ \rightarrow \\ 0 \end{smallmatrix} \right\rangle \right)$$

$$= \tfrac{1}{(p-1)(p+1)} \left( 3 \left( \tfrac{p}{2} \right)_4^{-1} \left( \tfrac{1}{2} \right)_2 \left( \tfrac{1}{2} \right)_1^2 - \left( \tfrac{p}{2} \right)_3^{-1} \left( \tfrac{1}{2} \right)_1^3 \right) = \tfrac{-p+3}{(p-1)p(p+1)(p+2)(p+4)(p+6)}$$

$$\left\langle \begin{smallmatrix} 0 & 2 \\ 2 & 0 \\ 2 & 0 \\ \rightarrow & \rightarrow \\ 0 & 0 \end{smallmatrix} \right\rangle = \left( \tfrac{p-1}{2} \right)^{-1} \left( (-1)^1 \left( \tfrac{1}{2} \right)_1 \left\langle \begin{smallmatrix} 2 \\ 2 \\ 2 \\ \rightarrow \\ 0 \end{smallmatrix} \right\rangle + (-1)^0 \left( \tfrac{1}{2} \right)_1 \left\langle \begin{smallmatrix} 0 \\ 2 \\ 2 \\ \rightarrow \\ 0 \end{smallmatrix} \right\rangle \right) = \tfrac{1}{(p-1)} \left( \left\langle \begin{smallmatrix} 0 \\ 2 \\ 2 \\ \rightarrow \\ 0 \end{smallmatrix} \right\rangle - \left\langle \begin{smallmatrix} 2 \\ 2 \\ 2 \\ \rightarrow \\ 0 \end{smallmatrix} \right\rangle \right)$$

$$= \tfrac{1}{(p-1)} \left( \left( \tfrac{p}{2} \right)_2^{-1} \left( \tfrac{1}{2} \right)_1^2 - \left( \tfrac{p}{2} \right)_3^{-1} \left( \tfrac{1}{2} \right)_1^3 \right) = \tfrac{1}{(p-1)} \left( \tfrac{(p+4)-1}{p(p+2)(p+4)} \right) = \tfrac{p+3}{(p-1)p(p+2)(p+4)}$$

$$\left\langle \begin{smallmatrix} 2 & 2 \\ 2 & 0 \\ 2 & 0 \\ \rightarrow & \rightarrow \\ 0 & 0 \end{smallmatrix} \right\rangle = \left( \tfrac{p-1}{2} \right)_1^{-1} \left( (-1)^1 \left( \tfrac{1}{2} \right)_1 \left\langle \begin{smallmatrix} 4 \\ 2 \\ 2 \\ \rightarrow \\ 0 \end{smallmatrix} \right\rangle + (-1)^0 \left( \tfrac{1}{2} \right)_1 \left( \tfrac{1}{2} \right)_0 \left\langle \begin{smallmatrix} 2 \\ 2 \\ 2 \\ \rightarrow \\ 0 \end{smallmatrix} \right\rangle \right) = \tfrac{1}{p-1} \left( \left\langle \begin{smallmatrix} 2 \\ 2 \\ 2 \\ \rightarrow \\ 0 \end{smallmatrix} \right\rangle - \left\langle \begin{smallmatrix} 4 \\ 2 \\ 2 \\ \rightarrow \\ 0 \end{smallmatrix} \right\rangle \right)$$

$$= \tfrac{1}{p-1} \left( \left( \tfrac{p}{2} \right)_3^{-1} \left( \tfrac{1}{2} \right)_1^3 - \left( \tfrac{p}{2} \right)_4^{-1} \left( \tfrac{1}{2} \right)_1^2 \left( \tfrac{1}{2} \right)_2 \right) = \tfrac{p+3}{(p-1)p(p+2)(p+4)(p+6)}$$

$$\left\langle \begin{smallmatrix} 3 & 1 \\ 1 & 1 \\ 2 & 0 \\ \rightarrow & \rightarrow \\ 0 & 0 \end{smallmatrix} \right\rangle = \left( \tfrac{p-1}{2} \right)^{-1} \left( (-1)^1 \left( \tfrac{1}{2} \right)_1 \left\langle \begin{smallmatrix} 4 \\ 2 \\ 2 \\ \rightarrow \\ 0 \end{smallmatrix} \right\rangle \right) = -\tfrac{3}{p-1} \left( \left( \tfrac{p}{2} \right)_4^{-1} \left( \tfrac{1}{2} \right)_2 \left( \tfrac{1}{2} \right)_1^2 \right)$$

$$= -\tfrac{3}{(p-1)p(p+2)(p+4)(p+6)}$$

$$\left\langle \begin{smallmatrix} 3 & 1 \\ 1 & 1 \\ 0 & 2 \\ \rightarrow & \rightarrow \\ 0 & 0 \end{smallmatrix} \right\rangle = \left( \tfrac{p-1}{2} \left( \tfrac{p+1}{2} \right) \right)^{-1} \left( (-1)^2 \left( \tfrac{1}{2} \right)_2 \left\langle \begin{smallmatrix} 4 \\ 2 \\ 2 \\ \rightarrow \\ 0 \end{smallmatrix} \right\rangle + (-1)^1 \left( \tfrac{1}{2} \right)_1^2 \left\langle \begin{smallmatrix} 4 \\ 2 \\ 0 \\ \rightarrow \\ 0 \end{smallmatrix} \right\rangle \right)$$

$$= \tfrac{1}{(p-1)(p+1)} \left( 3 \left( \tfrac{p}{2} \right)_4^{-1} \left( \tfrac{1}{2} \right)_2 \left( \tfrac{1}{2} \right)_1^2 - \left( \tfrac{p}{2} \right)_3^{-1} \left( \tfrac{1}{2} \right)_2 \left( \tfrac{1}{2} \right)_1 \right)$$

$$= \tfrac{1}{(p-1)(p+1)} \left( \tfrac{9}{p(p+2)(p+4)(p+6)} - \tfrac{3}{p(p+2)(p+4)} \right)$$

$$= \tfrac{-3(p+3)}{(p-1)p(p+1)(p+2)(p+4)(p+6)}$$

$$\left\langle \begin{smallmatrix} 4 & 0 \\ 1 & 1 \\ 1 & 1 \\ 0 & 0 \end{smallmatrix} \right\rangle = \left( \tfrac{p-1}{2} \right)_1^{-1} (-1)^1 \left( \tfrac{1}{2} \right)_1 \left\langle \begin{smallmatrix} 4 \\ 2 \\ 2 \\ 0 \end{smallmatrix} \right\rangle = -\tfrac{1}{p-1} \left\langle \begin{smallmatrix} 4 \\ 2 \\ 2 \\ 0 \end{smallmatrix} \right\rangle = -\tfrac{1}{p-1} \left( \tfrac{p}{2} \right)_4^{-1} \left( \tfrac{1}{2} \right)_2 \left( \tfrac{1}{2} \right)_1^2$$

$$= -\tfrac{3}{(p-1)p(p+2)(p+4)(p+6)}$$

$$\left\langle \begin{smallmatrix} 2 & 2 \\ 2 & 0 \\ 0 & 2 \\ 0 & 0 \end{smallmatrix} \right\rangle = \left( \tfrac{p-1}{2} \right)_2^{-1} \left( (-1)^2 \left( \tfrac{1}{2} \right)_2 \left\langle \begin{smallmatrix} 4 \\ 2 \\ 2 \\ 0 \end{smallmatrix} \right\rangle + \right.$$

$$\left. (-1)^1 \left( \tfrac{1}{2} \right)_1^2 \left\langle \begin{smallmatrix} 2 \\ 2 \\ 2 \\ 0 \end{smallmatrix} \right\rangle + (-1)^1 \left( \tfrac{1}{2} \right)_1^2 \left\langle \begin{smallmatrix} 4 \\ 2 \\ 0 \\ 0 \end{smallmatrix} \right\rangle + (-1)^0 \left( \tfrac{1}{2} \right)_1^2 \left\langle \begin{smallmatrix} 2 \\ 2 \\ 0 \\ 0 \end{smallmatrix} \right\rangle \right)$$

$$= \tfrac{1}{(p-1)(p+1)} \left( 3 \left\langle \begin{smallmatrix} 4 \\ 2 \\ 2 \\ 0 \end{smallmatrix} \right\rangle - \left\langle \begin{smallmatrix} 2 \\ 2 \\ 2 \\ 0 \end{smallmatrix} \right\rangle - \left\langle \begin{smallmatrix} 4 \\ 2 \\ 0 \\ 0 \end{smallmatrix} \right\rangle + \left\langle \begin{smallmatrix} 2 \\ 2 \\ 0 \\ 0 \end{smallmatrix} \right\rangle \right)$$

$$= \tfrac{1}{(p-1)(p+1)} \left( 3 \left( \tfrac{p}{2} \right)_4^{-1} \left( \tfrac{1}{2} \right)_2 \left( \tfrac{1}{2} \right)_1^2 - \left( \tfrac{p}{2} \right)_3^{-1} \left( \tfrac{1}{2} \right)_1^3 - \left( \tfrac{p}{2} \right)_3^{-1} \left( \tfrac{1}{2} \right)_1 \left( \tfrac{1}{2} \right)_2 + \left( \tfrac{p}{2} \right)_2^{-1} \left( \tfrac{1}{2} \right)_1^2 \right)$$

$$= \tfrac{9-(p+6)-3(p+6)+(p+4)(p+6)}{(p-1)p(p+1)(p+2)(p+4)(p+6)} = \tfrac{(p+3)^2}{(p-1)p(p+1)(p+2)(p+4)(p+6)}$$

$$\left\langle \begin{smallmatrix} 2 & 0 \\ 1 & 1 \\ 1 & 1 \\ 0 & 0 \end{smallmatrix} \right\rangle = \left( \tfrac{p-1}{2} \right)_1^{-1} (-1)^1 \left( \tfrac{1}{2} \right)_1 \left\langle \begin{smallmatrix} 2 \\ 2 \\ 2 \\ 0 \end{smallmatrix} \right\rangle = -\tfrac{1}{p-1} \left\langle \begin{smallmatrix} 2 \\ 2 \\ 2 \\ 0 \end{smallmatrix} \right\rangle = -\tfrac{1}{p-1} \left( \tfrac{p}{2} \right)_3^{-1} \left( \tfrac{1}{2} \right)_1^3$$

$$= -\tfrac{1}{(p-1)p(p+2)(p+4)}$$

### D.2.4 Four indices (rows)

$$\left\langle \begin{smallmatrix} 2 & 0 \\ 0 & 2 \\ 1 & 1 \\ \overrightarrow{0} & \overrightarrow{0} \end{smallmatrix} \right\rangle = \left(\tfrac{p-1}{2}\right)_2^{-1}\left((-1)^2 \left(\tfrac{1}{2}\right)_2 \left\langle \begin{smallmatrix} 2 \\ 2 \\ 2 \\ \overrightarrow{0} \end{smallmatrix}\right\rangle + (-1)^1 \left(\tfrac{1}{2}\right)_1^2 \left\langle \begin{smallmatrix} 2 \\ 0 \\ 2 \\ \overrightarrow{0} \end{smallmatrix}\right\rangle\right)$$

$$= \tfrac{1}{(p-1)(p+1)}\left(3\left\langle \begin{smallmatrix} 2 \\ 2 \\ 2 \\ \overrightarrow{0} \end{smallmatrix}\right\rangle - \left\langle \begin{smallmatrix} 2 \\ 0 \\ 2 \\ \overrightarrow{0} \end{smallmatrix}\right\rangle\right) = \tfrac{1}{(p-1)(p+1)}\left(3\left(\tfrac{p}{2}\right)_4^{-1}\left(\tfrac{1}{2}\right)_1^4 - \left(\tfrac{p}{2}\right)_3^{-1}\left(\tfrac{1}{2}\right)_1^3\right)$$

$$= \tfrac{-p-3}{(p-1)p(p+1)(p+2)(p+4)(p+6)}$$

$$\left\langle \begin{smallmatrix} 2 & 0 \\ 2 & 0 \\ 1 & 1 \\ \overrightarrow{0} & \overrightarrow{0} \end{smallmatrix} \right\rangle = \left(\tfrac{p-1}{2}\right)_1^{-1}(-1)^1\left(\tfrac{1}{2}\right)_1\left\langle \begin{smallmatrix} 2 \\ 2 \\ 2 \\ \overrightarrow{0} \end{smallmatrix}\right\rangle = -\tfrac{1}{p-1}\left\langle \begin{smallmatrix} 2 \\ 2 \\ 2 \\ \overrightarrow{0} \end{smallmatrix}\right\rangle = -\tfrac{1}{p-1}\left(\tfrac{p}{2}\right)_4^{-1}\left(\tfrac{1}{2}\right)_1^4$$

$$= -\tfrac{1}{(p-1)p(p+2)(p+4)(p+6)}$$

$$\left\langle \begin{smallmatrix} 2 & 0 \\ 2 & 0 \\ 0 & 2 \\ \overrightarrow{0} & \overrightarrow{0} \end{smallmatrix} \right\rangle = \left(\tfrac{p-1}{2}\left(\tfrac{p+1}{2}\right)\right)^{-1}\left((-1)^2\left(\tfrac{1}{2}\right)_2\left\langle \begin{smallmatrix} 2 \\ 2 \\ 2 \\ \overrightarrow{0} \end{smallmatrix}\right\rangle + 2(-1)^1\left(\tfrac{1}{2}\right)_1^2\left\langle \begin{smallmatrix} 2 \\ 2 \\ 0 \\ \overrightarrow{0} \end{smallmatrix}\right\rangle + (-1)^0\left(\tfrac{1}{2}\right)_1^2\left\langle \begin{smallmatrix} 2 \\ 0 \\ 0 \\ \overrightarrow{0} \end{smallmatrix}\right\rangle\right)$$

$$= \tfrac{1}{(p-1)(p+1)}\left(3\left\langle \begin{smallmatrix} 2 \\ 2 \\ 2 \\ \overrightarrow{0} \end{smallmatrix}\right\rangle - 2\left\langle \begin{smallmatrix} 2 \\ 2 \\ 0 \\ \overrightarrow{0} \end{smallmatrix}\right\rangle + \left\langle \begin{smallmatrix} 2 \\ 0 \\ 0 \\ \overrightarrow{0} \end{smallmatrix}\right\rangle\right)$$

$$= \tfrac{1}{(p-1)(p+1)}\left(3\left(\tfrac{p}{2}\right)_4^{-1}\left(\tfrac{1}{2}\right)_1^4 - 2\left(\tfrac{p}{2}\right)_3^{-1}\left(\tfrac{1}{2}\right)_1^3 + \left(\tfrac{p}{2}\right)_2^{-1}\left(\tfrac{1}{2}\right)_1^2\right)$$

$$= \tfrac{1}{(p-1)(p+1)}\left(\tfrac{3}{p(p+2)(p+4)(p+6)} - \tfrac{2}{p(p+2)(p+4)} + \tfrac{1}{p(p+2)}\right)$$

$$= \tfrac{(p+3)(p+5)}{(p-1)p(p+1)(p+2)(p+4)(p+6)}$$

$$\left\langle \begin{smallmatrix} 1 & 1 \\ 1 & 1 \\ 1 & 1 \\ \overrightarrow{0} & \overrightarrow{0} \end{smallmatrix} \right\rangle = \left(\tfrac{p-1}{2}\right)_2^{-1}(-1)^2\left(\tfrac{1}{2}\right)_2\left\langle \begin{smallmatrix} 2 \\ 2 \\ 2 \\ \overrightarrow{0} \end{smallmatrix}\right\rangle = \tfrac{3}{(p-1)(p+1)}\left(\tfrac{p}{2}\right)_4^{-1}\left(\tfrac{1}{2}\right)_1^4 = \tfrac{3}{(p-1)p(p+1)(p+2)(p+4)(p+6)}$$

$$\left\langle \begin{smallmatrix} 2 & 0 \\ 2 & 0 \\ 0 & 2 \\ \overrightarrow{0} & \overrightarrow{0} \end{smallmatrix} \right\rangle = \left\langle \begin{smallmatrix} 2 & 0 \\ 2 & 0 \\ 2 & 0 \\ \overrightarrow{0} & \overrightarrow{0} \end{smallmatrix} \right\rangle = \left(\tfrac{p-1}{2}\right)^{-1}\left((-1)^1\left(\tfrac{1}{2}\right)_1\left\langle \begin{smallmatrix} 2 \\ 2 \\ 2 \\ \overrightarrow{0} \end{smallmatrix}\right\rangle + (-1)^0\left(\tfrac{1}{2}\right)_1\left\langle \begin{smallmatrix} 2 \\ 0 \\ 2 \\ \overrightarrow{0} \end{smallmatrix}\right\rangle\right)$$

$$= \tfrac{1}{p-1}\left(\left\langle \begin{smallmatrix} 2 \\ 0 \\ 2 \\ \overrightarrow{0} \end{smallmatrix}\right\rangle - \left\langle \begin{smallmatrix} 2 \\ 2 \\ 2 \\ \overrightarrow{0} \end{smallmatrix}\right\rangle\right) = \tfrac{1}{p-1}\left(\left(\tfrac{p}{2}\right)_3^{-1}\left(\tfrac{1}{2}\right)_1^3 - \left(\tfrac{p}{2}\right)_4^{-1}\left(\tfrac{1}{2}\right)_1^4\right)$$

$$= \tfrac{1}{p-1}\left(\tfrac{1}{p(p+2)(p+4)} - \tfrac{1}{p(p+2)(p+4)(p+6)}\right) = \tfrac{p+5}{(p-1)p(p+2)(p+4)(p+6)}$$

### D.3 Monomials of Three Orthogonal Vector

#### D.3.1 One index (row)

$$\left\langle \begin{smallmatrix} 2 \\ 0 \end{smallmatrix} \begin{smallmatrix} 2 \\ 0 \end{smallmatrix} \begin{smallmatrix} 2 \\ 0 \end{smallmatrix} \right\rangle = \frac{1}{p(p+2)(p+4)}$$

$$\left\langle \begin{smallmatrix} 4 \\ 0 \end{smallmatrix} \begin{smallmatrix} 2 \\ 0 \end{smallmatrix} \begin{smallmatrix} 2 \\ 0 \end{smallmatrix} \right\rangle = \frac{1}{p-2}\left( (-1)^1 \left( \left\langle \begin{smallmatrix} 6 \\ 0 \end{smallmatrix} \begin{smallmatrix} 2 \\ 0 \end{smallmatrix} \right\rangle + \left\langle \begin{smallmatrix} 4 \\ 0 \end{smallmatrix} \begin{smallmatrix} 4 \\ 0 \end{smallmatrix} \right\rangle \right) + (-1)^0 \left\langle \begin{smallmatrix} 4 \\ 0 \end{smallmatrix} \begin{smallmatrix} 2 \\ 0 \end{smallmatrix} \right\rangle \right)$$

$$= \frac{1}{p-2}\left( \left\langle \begin{smallmatrix} 4 \\ 0 \end{smallmatrix} \begin{smallmatrix} 2 \\ 0 \end{smallmatrix} \right\rangle - \left\langle \begin{smallmatrix} 6 \\ 0 \end{smallmatrix} \begin{smallmatrix} 2 \\ 0 \end{smallmatrix} \right\rangle - \left\langle \begin{smallmatrix} 4 \\ 0 \end{smallmatrix} \begin{smallmatrix} 4 \\ 0 \end{smallmatrix} \right\rangle \right) = \frac{1}{p-2}\left( \frac{3}{p(p+2)(p+4)} + \frac{-15-9}{p(p+2)(p+4)(p+6)} \right)$$

$$= \frac{3(p-2)}{(p-2)p(p+2)(p+4)(p+6)} = \frac{3}{p(p+2)(p+4)(p+6)}$$

#### D.3.2 Two indices (rows)

$$\left\langle \begin{smallmatrix} 2 \\ 2 \\ 0 \end{smallmatrix} \begin{smallmatrix} 2 \\ 0 \\ 0 \end{smallmatrix} \begin{smallmatrix} 2 \\ 0 \\ 0 \end{smallmatrix} \right\rangle = \left\langle \begin{smallmatrix} 2 \\ 2 \\ 0 \end{smallmatrix} \begin{smallmatrix} 2 \\ 0 \\ 0 \end{smallmatrix} \right\rangle = \frac{p+3}{(p-1)p(p+2)(p+4)(p+6)}$$

$$\left\langle \begin{smallmatrix} 2 \\ 0 \\ 0 \end{smallmatrix} \begin{smallmatrix} 2 \\ 0 \\ 0 \end{smallmatrix} \begin{smallmatrix} 0 \\ 2 \\ 0 \end{smallmatrix} \right\rangle = \left\langle \begin{smallmatrix} 0 \\ 2 \\ 0 \end{smallmatrix} \begin{smallmatrix} 2 \\ 0 \\ 0 \end{smallmatrix} \right\rangle = \frac{(p+3)}{(p-1)p(p+2)(p+4)}$$

$$\left\langle \begin{smallmatrix} 0 \\ 4 \\ 0 \end{smallmatrix} \begin{smallmatrix} 2 \\ 0 \\ 0 \end{smallmatrix} \begin{smallmatrix} 2 \\ 0 \\ 0 \end{smallmatrix} \right\rangle = \left\langle \begin{smallmatrix} 4 \\ 0 \\ 0 \end{smallmatrix} \begin{smallmatrix} 0 \\ 2 \\ 0 \end{smallmatrix} \right\rangle = \frac{3(p+3)(p+5)}{(p-1)p(p+1)(p+2)(p+4)(p+6)}$$

$$\left\langle \begin{smallmatrix} 2 \\ 0 \\ 0 \end{smallmatrix} \begin{smallmatrix} 1 \\ 0 \\ 0 \end{smallmatrix} \begin{smallmatrix} 1 \\ 0 \\ 0 \end{smallmatrix} \right\rangle = \frac{1}{p-2}\left( -\left\langle \begin{smallmatrix} 2 \\ 0 \\ 0 \end{smallmatrix} \begin{smallmatrix} 2 \\ 0 \\ 0 \end{smallmatrix} \right\rangle - \left\langle \begin{smallmatrix} 3 \\ 1 \\ 0 \end{smallmatrix} \begin{smallmatrix} 1 \\ 0 \\ 0 \end{smallmatrix} \right\rangle \right)$$

$$= \frac{1}{p-2}\left( -\frac{p+1}{(p-1)p(p+2)(p+4)} + \frac{3}{(p-1)p(p+2)(p+4)} \right) = \frac{-1}{(p-1)p(p+2)(p+4)}$$

$$\left\langle \begin{smallmatrix} 4 \\ 0 \\ 0 \end{smallmatrix} \begin{smallmatrix} 1 \\ 0 \\ 0 \end{smallmatrix} \begin{smallmatrix} 1 \\ 0 \\ 0 \end{smallmatrix} \right\rangle = \left\langle \begin{smallmatrix} 4 \\ 1 \\ 0 \end{smallmatrix} \begin{smallmatrix} 0 \\ 1 \\ 0 \end{smallmatrix} \right\rangle = \frac{-3}{(p-1)p(p+2)(p+4)(p+6)}$$

$$\left\langle \begin{smallmatrix} 2 \\ 2 \\ 0 \end{smallmatrix} \begin{smallmatrix} 1 \\ 1 \\ 0 \end{smallmatrix} \begin{smallmatrix} 1 \\ 1 \\ 0 \end{smallmatrix} \right\rangle = \left\langle \begin{smallmatrix} 2 \\ 1 \\ 0 \end{smallmatrix} \begin{smallmatrix} 2 \\ 1 \\ 0 \end{smallmatrix} \right\rangle = \frac{-(p-3)}{(p-1)p(p+1)(p+2)(p+4)(p+6)}$$

$$\left\langle \begin{smallmatrix} 3 \\ 1 \\ 0 \end{smallmatrix} \begin{smallmatrix} 1 \\ 1 \\ 0 \end{smallmatrix} \begin{smallmatrix} 2 \\ 0 \\ 0 \end{smallmatrix} \right\rangle = \left\langle \begin{smallmatrix} 3 \\ 1 \\ 2 \\ 0 \end{smallmatrix} \begin{smallmatrix} 1 \\ 1 \\ 0 \end{smallmatrix} \right\rangle = \frac{-3}{(p-1)p(p+2)(p+4)(p+6)}$$

$$\left\langle \begin{smallmatrix} 1 \\ 3 \\ 0 \end{smallmatrix} \begin{smallmatrix} 1 \\ 1 \\ 0 \end{smallmatrix} \begin{smallmatrix} 2 \\ 0 \\ 0 \end{smallmatrix} \right\rangle = \left\langle \begin{smallmatrix} 3 \\ 1 \\ 0 \end{smallmatrix} \begin{smallmatrix} 1 \\ 1 \\ 2 \\ 0 \end{smallmatrix} \right\rangle = \frac{-3(p+3)}{(p-1)p(p+1)(p+2)(p+4)(p+6)}$$

$$\left\langle \begin{smallmatrix} 2 \\ 2 \\ 0 \end{smallmatrix} \begin{smallmatrix} 2 \\ 0 \\ 0 \end{smallmatrix} \begin{smallmatrix} 0 \\ 2 \\ 0 \end{smallmatrix} \right\rangle = \left\langle \begin{smallmatrix} 2 \\ 2 \\ 0 \end{smallmatrix} \begin{smallmatrix} 2 \\ 0 \\ 2 \\ 0 \end{smallmatrix} \right\rangle = \frac{(p+3)^2}{(p-1)p(p+1)(p+2)(p+4)(p+6)}$$

$$\left\langle \begin{smallmatrix} 4 \\ 0 \\ 0 \end{smallmatrix} \begin{smallmatrix} 2 \\ 0 \\ 0 \end{smallmatrix} \begin{smallmatrix} 0 \\ 2 \\ 0 \end{smallmatrix} \right\rangle = \frac{1}{p-2}\left( (-1)^1 \left( \left\langle \begin{smallmatrix} 4 \\ 0 \\ 0 \end{smallmatrix} \begin{smallmatrix} 2 \\ 2 \\ 0 \end{smallmatrix} \right\rangle + \left\langle \begin{smallmatrix} 4 \\ 2 \\ 0 \end{smallmatrix} \begin{smallmatrix} 2 \\ 0 \\ 0 \end{smallmatrix} \right\rangle \right) + (-1)^0 \left\langle \begin{smallmatrix} 4 \\ 0 \\ 0 \end{smallmatrix} \begin{smallmatrix} 2 \\ 0 \\ 0 \end{smallmatrix} \right\rangle \right)$$

$$= \frac{1}{p-2}\left( \left\langle \begin{smallmatrix} 4 \\ 0 \\ 0 \end{smallmatrix} \begin{smallmatrix} 2 \\ 0 \\ 0 \end{smallmatrix} \right\rangle - \left\langle \begin{smallmatrix} 4 \\ 0 \\ 0 \end{smallmatrix} \begin{smallmatrix} 2 \\ 2 \\ 0 \end{smallmatrix} \right\rangle - \left\langle \begin{smallmatrix} 4 \\ 2 \\ 0 \end{smallmatrix} \begin{smallmatrix} 2 \\ 0 \\ 0 \end{smallmatrix} \right\rangle \right)$$

$$= \frac{1}{p-2}\left( \frac{3(p-1)(p+6)-3(p+3)-3(p+1)}{(p-1)p(p+2)(p+4)(p+6)} \right) = \frac{3(p+5)}{(p-1)p(p+2)(p+4)(p+6)}$$

### D.3.3 THREE INDICES (ROWS)

$$\left\langle \begin{smallmatrix} 2 & 0 & 0 \\ 0 & 2 & 0 \\ 0 & 0 & 2 \\ \to & \to & \to \\ 0 & 0 & 0 \end{smallmatrix} \right\rangle = \tfrac{1}{p-2}\left( \left\langle \begin{smallmatrix} 2 & 0 \\ 0 & 2 \\ 0 & 0 \\ \to & \to \\ 0 & 0 \end{smallmatrix} \right\rangle - \left\langle \begin{smallmatrix} 2 & 0 \\ 0 & 2 \\ 2 & 0 \\ \to & \to \\ 0 & 0 \end{smallmatrix} \right\rangle - \left\langle \begin{smallmatrix} 2 & 0 \\ 0 & 2 \\ 0 & 2 \\ \to & \to \\ 0 & 0 \end{smallmatrix} \right\rangle \right) = \tfrac{1}{p-2}\left( \left\langle \begin{smallmatrix} 2 & 0 \\ 0 & 2 \\ 0 & 0 \\ \to & \to \\ 0 & 0 \end{smallmatrix} \right\rangle - 2\left\langle \begin{smallmatrix} 2 & 0 \\ 0 & 2 \\ 0 & 2 \\ \to & \to \\ 0 & 0 \end{smallmatrix} \right\rangle \right)$$

$$= \tfrac{1}{p-2}\left( \tfrac{p+1}{(p-1)p(p+2)} - \tfrac{2(p+3)}{(p-1)p(p+2)(p+4)} \right) = \tfrac{p^2+3p-2}{(p-2)(p-1)p(p+2)(p+4)}$$

$$\left\langle \begin{smallmatrix} 4 & 0 & 0 \\ 0 & 2 & 0 \\ 0 & 0 & 2 \\ \to & \to & \to \\ 0 & 0 & 0 \end{smallmatrix} \right\rangle = \tfrac{1}{p-2}\left( \left\langle \begin{smallmatrix} 4 & 0 \\ 0 & 2 \\ 0 & 0 \\ \to & \to \\ 0 & 0 \end{smallmatrix} \right\rangle - \left\langle \begin{smallmatrix} 4 & 0 \\ 0 & 2 \\ 2 & 0 \\ \to & \to \\ 0 & 0 \end{smallmatrix} \right\rangle - \left\langle \begin{smallmatrix} 4 & 0 \\ 0 & 2 \\ 0 & 2 \\ \to & \to \\ 0 & 0 \end{smallmatrix} \right\rangle \right)$$

$$= \tfrac{1}{p-2}\left( \tfrac{3(p+3)}{(p-1)p(p+2)(p+4)} - \tfrac{3(p+5)}{(p-1)p(p+2)(p+4)(p+6)} - \tfrac{3(p+3)(p+5)}{(p-1)p(p+1)(p+2)(p+4)(p+6)} \right)$$

$$= \tfrac{1}{p-2}\left( \tfrac{3(p+1)(p+3)(p+6)-3(p+1)(p+5)-3(p+3)(p+5)}{(p-1)p(p+1)(p+2)(p+4)(p+6)} \right)$$

$$= \tfrac{3\left(p^3+8p^2+13p-2\right)}{(p-2)(p-1)p(p+1)(p+2)(p+4)(p+6)}$$

$$\left\langle \begin{smallmatrix} 0 & 2 & 2 \\ 2 & 0 & 0 \\ 2 & 0 & 0 \\ \to & \to & \to \\ 0 & 0 & 0 \end{smallmatrix} \right\rangle = \tfrac{1}{p-2}\left( (-1)^1\left( \left\langle \begin{smallmatrix} 2 & 2 \\ 2 & 0 \\ 2 & 0 \\ \to & \to \\ 0 & 0 \end{smallmatrix} \right\rangle + \left\langle \begin{smallmatrix} 0 & 4 \\ 2 & 0 \\ 2 & 0 \\ \to & \to \\ 0 & 0 \end{smallmatrix} \right\rangle \right) + (-1)^0\left\langle \begin{smallmatrix} 0 & 2 \\ 2 & 0 \\ 2 & 0 \\ \to & \to \\ 0 & 0 \end{smallmatrix} \right\rangle \right)$$

$$= \tfrac{1}{p-2}\left( \left\langle \begin{smallmatrix} 0 & 2 \\ 2 & 0 \\ 2 & 0 \\ \to & \to \\ 0 & 0 \end{smallmatrix} \right\rangle - \left\langle \begin{smallmatrix} 2 & 2 \\ 2 & 0 \\ 2 & 0 \\ \to & \to \\ 0 & 0 \end{smallmatrix} \right\rangle - \left\langle \begin{smallmatrix} 0 & 4 \\ 2 & 0 \\ 2 & 0 \\ \to & \to \\ 0 & 0 \end{smallmatrix} \right\rangle \right)$$

$$= \tfrac{1}{p-2}\left( \tfrac{p+3}{(p-1)p(p+2)(p+4)} - \tfrac{p+3}{(p-1)p(p+2)(p+4)(p+6)} - \tfrac{3(p+3)(p+5)}{(p-1)p(p+1)(p+2)(p+4)(p+6)} \right)$$

$$= \tfrac{(p+3)((p+1)(p+6)-(p+1)-3(p+5))}{(p-2)(p-1)p(p+1)(p+2)(p+4)(p+6)} = \tfrac{(p-2)(p+3)(p+5)}{(p-2)(p-1)p(p+1)(p+2)(p+4)(p+6)}$$

$$= \tfrac{(p+3)(p+5)}{(p-1)p(p+1)(p+2)(p+4)(p+6)}$$

$$\left\langle \begin{smallmatrix} 2 & 0 & 0 \\ 0 & 2 & 0 \\ 2 & 0 & 2 \\ \to & \to & \to \\ 0 & 0 & 0 \end{smallmatrix} \right\rangle = \tfrac{1}{p-2}\left( \left\langle \begin{smallmatrix} 2 & 0 \\ 0 & 2 \\ 2 & 0 \\ \to & \to \\ 0 & 0 \end{smallmatrix} \right\rangle - \left\langle \begin{smallmatrix} 2 & 0 \\ 0 & 2 \\ 4 & 0 \\ \to & \to \\ 0 & 0 \end{smallmatrix} \right\rangle - \left\langle \begin{smallmatrix} 2 & 0 \\ 0 & 2 \\ 2 & 2 \\ \to & \to \\ 0 & 0 \end{smallmatrix} \right\rangle \right)$$

$$= \tfrac{1}{p-2}\left( \tfrac{p+3}{(p-1)p(p+2)(p+4)} - \tfrac{3(p+5)}{(p-1)p(p+2)(p+4)(p+6)} - \tfrac{(p+3)^2}{(p-1)p(p+1)(p+2)(p+4)(p+6)} \right)$$

$$= \tfrac{(p+1)((p+3)(p+6)-3(p+5))-(p+3)^2}{(p-2)(p-1)p(p+1)(p+2)(p+4)(p+6)} = \tfrac{p^3+6p^2+3p-6}{(p-2)(p-1)p(p+1)(p+2)(p+4)(p+6)}$$

$$\left\langle \begin{smallmatrix} 2 & 1 & 1 \\ 0 & 1 & 1 \\ 2 & 0 & 0 \\ \to & \to & \to \\ 0 & 0 & 0 \end{smallmatrix} \right\rangle = \tfrac{1}{p-2}\left( -\left\langle \begin{smallmatrix} 3 & 1 \\ 1 & 1 \\ 2 & 0 \\ \to & \to \\ 0 & 0 \end{smallmatrix} \right\rangle - \left\langle \begin{smallmatrix} 2 & 2 \\ 0 & 2 \\ 2 & 0 \\ \to & \to \\ 0 & 0 \end{smallmatrix} \right\rangle \right)$$

$$= \tfrac{1}{p-2}\left( \tfrac{3}{(p-1)p(p+2)(p+4)(p+6)} - \tfrac{(p+3)^2}{(p-1)p(p+1)(p+2)(p+4)(p+6)} \right)$$

$$= \tfrac{3(p+1)-(p+3)^2}{(p-2)(p-1)p(p+1)(p+2)(p+4)(p+6)} = \tfrac{-\left(p^2+3p+6\right)}{(p-2)(p-1)p(p+1)(p+2)(p+4)(p+6)}$$

$$\left\langle \begin{smallmatrix} 0 & 1 & 1 \\ 0 & 1 & 1 \\ 2 & 0 & 0 \\ \to & \to & \to \\ 0 & 0 & 0 \end{smallmatrix} \right\rangle = \tfrac{1}{p-2}\left( -\left\langle \begin{smallmatrix} 1 & 1 \\ 1 & 1 \\ 2 & 0 \\ \to & \to \\ 0 & 0 \end{smallmatrix} \right\rangle - \left\langle \begin{smallmatrix} 0 & 2 \\ 0 & 2 \\ 2 & 0 \\ \to & \to \\ 0 & 0 \end{smallmatrix} \right\rangle \right)$$

$$= \tfrac{1}{p-2}\left( \tfrac{1}{(p-1)p(p+2)(p+4)} - \tfrac{p+3}{(p-1)p(p+2)(p+4)} \right) = \tfrac{-(p+2)}{(p-2)(p-1)p(p+2)(p+4)}$$

$$\left\langle \begin{smallmatrix} 0 & 1 & 1 \\ 0 & 1 & 1 \\ 4 & 0 & 0 \\ \to & \to & \to \\ 0 & 0 & 0 \end{smallmatrix} \right\rangle = \tfrac{1}{p-2}\left( -\left\langle \begin{smallmatrix} 1 & 1 \\ 1 & 1 \\ 4 & 0 \\ \to & \to \\ 0 & 0 \end{smallmatrix} \right\rangle - \left\langle \begin{smallmatrix} 0 & 2 \\ 0 & 2 \\ 4 & 0 \\ \to & \to \\ 0 & 0 \end{smallmatrix} \right\rangle \right)$$

$$= \tfrac{1}{p-2}\left( \tfrac{3}{(p-1)p(p+2)(p+4)(p+6)} - \tfrac{3(p+3)(p+5)}{(p-1)p(p+1)(p+2)(p+4)(p+6)} \right)$$

$$= \tfrac{1}{p-2}\left( \tfrac{3(p+1)-3(p+3)(p+5)}{(p-1)p(p+1)(p+2)(p+4)(p+6)} \right) = \tfrac{-3\left(p^2+7p+14\right)}{(p-2)(p-1)p(p+1)(p+2)(p+4)(p+6)}$$

$$\left\langle \begin{smallmatrix} 1 & 2 & 1 \\ 1 & 0 & 1 \\ 0 & 0 & 2 \\ \vec{0} & \vec{0} & \vec{0} \end{smallmatrix} \right\rangle = \frac{1}{p-2}\left( -\left\langle \begin{smallmatrix} 3 & 1 \\ 1 & 1 \\ 0 & 2 \\ \vec{0} & \vec{0} \end{smallmatrix} \right\rangle - \left\langle \begin{smallmatrix} 2 & 2 \\ 0 & 2 \\ 0 & 2 \\ \vec{0} & \vec{0} \end{smallmatrix} \right\rangle \right)$$

$$= \frac{1}{p-2}\left( \frac{3(p+3)}{(p-1)p(p+1)(p+2)(p+4)(p+6)} - \frac{p+3}{(p-1)p(p+2)(p+4)(p+6)} \right)$$

$$= \frac{-(p+3)}{(p-1)p(p+1)(p+2)(p+4)(p+6)}$$

$$\left\langle \begin{smallmatrix} 1 & 3 & 0 \\ 1 & 1 & 0 \\ 0 & 0 & 2 \\ \vec{0} & \vec{0} & \vec{0} \end{smallmatrix} \right\rangle = \frac{1}{p-2}\left( -\left\langle \begin{smallmatrix} 4 & 0 \\ 2 & 0 \\ 0 & 2 \\ \vec{0} & \vec{0} \end{smallmatrix} \right\rangle - \left\langle \begin{smallmatrix} 3 & 1 \\ 1 & 1 \\ 0 & 2 \\ \vec{0} & \vec{0} \end{smallmatrix} \right\rangle \right)$$

$$= \frac{1}{p-2}\left( -\frac{3(p+5)}{(p-1)p(p+2)(p+4)(p+6)} + \frac{3(p+3)}{(p-1)p(p+1)(p+2)(p+4)(p+6)} \right)$$

$$= \frac{1}{p-2}\left( \frac{3(p+3)-3(p+1)(p+5)}{(p-1)p(p+1)(p+2)(p+4)(p+6)} \right)$$

$$= \frac{-3\left(p^2+5p+2\right)}{(p-2)(p-1)p(p+1)(p+2)(p+4)(p+6)}$$

$$\left\langle \begin{smallmatrix} 2 & 2 & 0 \\ 0 & 1 & 1 \\ 0 & 1 & 1 \\ \vec{0} & \vec{0} & \vec{0} \end{smallmatrix} \right\rangle = \left\langle \begin{smallmatrix} 2 & 1 & 1 \\ 0 & 1 & 1 \\ 2 & 0 & 0 \\ \vec{0} & \vec{0} & \vec{0} \end{smallmatrix} \right\rangle = \frac{-\left(p^2+3p+6\right)}{(p-2)(p-1)p(p+1)(p+2)(p+4)(p+6)}$$

$$\left\langle \begin{smallmatrix} 1 & 1 & 0 \\ 1 & 0 & 1 \\ 2 & 1 & 1 \\ \vec{0} & \vec{0} & \vec{0} \end{smallmatrix} \right\rangle = \left\langle \begin{smallmatrix} 2 & 1 & 1 \\ 1 & 1 & 0 \\ 1 & 0 & 1 \\ \vec{0} & \vec{0} & \vec{0} \end{smallmatrix} \right\rangle = \frac{1}{p-2}\left( -\left\langle \begin{smallmatrix} 3 & 1 \\ 2 & 0 \\ 0 & 0 \\ \vec{0} & \vec{0} \end{smallmatrix} \right\rangle - \left\langle \begin{smallmatrix} 2 & 2 \\ 1 & 1 \\ 0 & 0 \\ \vec{0} & \vec{0} \end{smallmatrix} \right\rangle \right)$$

$$= \frac{1}{p-2}\left( \frac{3}{(p-1)p(p+2)(p+4)(p+6)} - \frac{-p+3}{(p-1)p(p+1)(p+2)(p+4)(p+6)} \right)$$

$$= \frac{p-3+3(p+1)}{(p-2)(p-1)p(p+1)(p+2)(p+4)(p+6)}$$

$$= \frac{4p}{(p-2)(p-1)p(p+1)(p+2)(p+4)(p+6)}$$

$$\left\langle \begin{smallmatrix} 3 & 1 & 0 \\ 0 & 1 & 1 \\ 1 & 0 & 1 \\ \vec{0} & \vec{0} & \vec{0} \end{smallmatrix} \right\rangle = \frac{1}{p-2}\left( -\left\langle \begin{smallmatrix} 3 & 1 \\ 1 & 1 \\ 2 & 0 \\ \vec{0} & \vec{0} \end{smallmatrix} \right\rangle - \left\langle \begin{smallmatrix} 3 & 1 \\ 0 & 2 \\ 1 & 1 \\ \vec{0} & \vec{0} \end{smallmatrix} \right\rangle \right)$$

$$= \frac{1}{p-2}\left( \frac{3}{(p-1)p(p+2)(p+4)(p+6)} + \frac{3(p+3)}{(p-1)p(p+1)(p+2)(p+4)(p+6)} \right)$$

$$= \frac{1}{p-2}\left( \frac{3(p+1)+3(p+3)}{(p-1)p(p+1)(p+2)(p+4)(p+6)} \right)$$

$$= \frac{6(p+2)}{(p-2)(p-1)p(p+1)(p+2)(p+4)(p+6)}$$

$$\left\langle \begin{smallmatrix} 0 & 1 & 1 \\ 1 & 1 & 0 \\ 1 & 0 & 1 \\ \vec{0} & \vec{0} & \vec{0} \end{smallmatrix} \right\rangle = \frac{1}{p-2}\left( -\left\langle \begin{smallmatrix} 1 & 1 \\ 2 & 0 \\ 0 & 0 \\ \vec{0} & \vec{0} \end{smallmatrix} \right\rangle - \left\langle \begin{smallmatrix} 0 & 2 \\ 1 & 1 \\ 1 & 1 \\ \vec{0} & \vec{0} \end{smallmatrix} \right\rangle \right) = \frac{1}{p-2}\left( -2\left\langle \begin{smallmatrix} 1 & 1 \\ 2 & 0 \\ 0 & 0 \\ \vec{0} & \vec{0} \end{smallmatrix} \right\rangle \right) = \frac{2}{(p-2)(p-1)p(p+2)(p+4)}$$

### D.3.4 Four indices (rows)

$$\left\langle \begin{smallmatrix} 0 & 2 & 0 \\ 0 & 0 & 2 \\ 2 & 0 & 0 \\ 2 & 0 & 0 \\ \vec{0} & \vec{0} & \vec{0} \end{smallmatrix} \right\rangle = \frac{1}{p-2}\left( \left\langle \begin{smallmatrix} 0 & 2 \\ 0 & 0 \\ 2 & 0 \\ 2 & 0 \\ \vec{0} & \vec{0} \end{smallmatrix} \right\rangle - \left\langle \begin{smallmatrix} 0 & 2 \\ 2 & 0 \\ 2 & 0 \\ 2 & 0 \\ \vec{0} & \vec{0} \end{smallmatrix} \right\rangle - \left\langle \begin{smallmatrix} 0 & 2 \\ 0 & 2 \\ 2 & 0 \\ 2 & 0 \\ \vec{0} & \vec{0} \end{smallmatrix} \right\rangle \right)$$

$$= \frac{1}{p-2}\left( \frac{p+3}{(p-1)p(p+2)(p+4)} - \frac{p+5}{(p-1)p(p+2)(p+4)(p+6)} - \frac{(p+3)(p+5)}{(p-1)p(p+1)(p+2)(p+4)(p+6)} \right)$$

$$= \frac{(p+1)(p+3)(p+6)-(p+1)(p+5)-(p+3)(p+5)}{(p-2)(p-1)p(p+1)(p+2)(p+4)(p+6)}$$

$$= \frac{p^3+8p^2+13p-2}{(p-2)(p-1)p(p+1)(p+2)(p+4)(p+6)}$$

$$\left\langle \begin{smallmatrix} 0 & 1 & 1 \\ 0 & 1 & 1 \\ 2 & 0 & 0 \\ 2 & 0 & 0 \\ \vec{0} & \vec{0} & \vec{0} \end{smallmatrix} \right\rangle = \frac{1}{p-2}\left( -\left\langle \begin{smallmatrix} 1 & 1 \\ 1 & 1 \\ 2 & 0 \\ 2 & 0 \\ \vec{0} & \vec{0} \end{smallmatrix} \right\rangle - \left\langle \begin{smallmatrix} 0 & 2 \\ 0 & 2 \\ 2 & 0 \\ 2 & 0 \\ \vec{0} & \vec{0} \end{smallmatrix} \right\rangle \right) = \frac{1}{p-2}\left( \frac{(p+1)-(p+3)(p+5)}{(p-1)p(p+1)(p+2)(p+4)(p+6)} \right)$$

$$= \frac{-\left(p^2+7p+14\right)}{(p-2)(p-1)p(p+1)(p+2)(p+4)(p+6)}$$

$$\left\langle \begin{smallmatrix} 2 & 0 & 0 \\ 0 & 2 & 0 \\ 1 & 0 & 1 \\ 1 & 0 & 1 \\ \vec{0} & \vec{0} & \vec{0} \end{smallmatrix} \right\rangle = \frac{1}{p-2}\left( -\left\langle \begin{smallmatrix} 2 & 0 \\ 0 & 2 \\ 2 & 0 \\ 2 & 0 \\ \vec{0} & \vec{0} \end{smallmatrix} \right\rangle - \left\langle \begin{smallmatrix} 2 & 0 \\ 0 & 2 \\ 1 & 1 \\ 1 & 1 \\ \vec{0} & \vec{0} \end{smallmatrix} \right\rangle \right)$$

$$= \frac{1}{p-2}\left( \frac{-(p+5)}{(p-1)p(p+2)(p+4)(p+6)} + \frac{p+3}{(p-1)p(p+1)(p+2)(p+4)(p+6)} \right)$$

$$= \frac{1}{p-2}\left( \frac{p+3-(p+5)(p+1)}{(p-1)p(p+1)(p+2)(p+4)(p+6)} \right)$$

$$= \frac{-\left(p^2+5p+2\right)}{(p-2)(p-1)p(p+1)(p+2)(p+4)(p+6)}$$

$$\left\langle \begin{smallmatrix} 1 & 0 & 1 \\ 1 & 0 & 1 \\ 1 & 1 & 0 \\ 1 & 1 & 0 \\ \vec{0} & \vec{0} & \vec{0} \end{smallmatrix} \right\rangle = \frac{1}{p-2}\left( -\left\langle \begin{smallmatrix} 2 & 0 \\ 2 & 0 \\ 1 & 1 \\ 1 & 1 \\ \vec{0} & \vec{0} \end{smallmatrix} \right\rangle - \left\langle \begin{smallmatrix} 1 & 1 \\ 1 & 1 \\ 1 & 1 \\ 1 & 1 \\ \vec{0} & \vec{0} \end{smallmatrix} \right\rangle \right)$$

$$= \frac{1}{p-2}\left( \frac{1}{(p-1)p(p+2)(p+4)(p+6)} - \frac{3}{(p-1)p(p+1)(p+2)(p+4)(p+6)} \right)$$

$$= \frac{p-2}{(p-2)(p-1)p(p+1)(p+2)(p+4)(p+6)} = \frac{1}{(p-1)p(p+1)(p+2)(p+4)(p+6)}$$

$$\left\langle \begin{smallmatrix} 2 & 0 & 0 \\ 0 & 1 & 1 \\ 1 & 1 & 0 \\ 1 & 0 & 1 \\ \vec{0} & \vec{0} & \vec{0} \end{smallmatrix} \right\rangle = \frac{1}{p-2}\left( \frac{1}{(p-1)p(p+2)(p+4)(p+6)} + \frac{p+3}{(p-1)p(p+1)(p+2)(p+4)(p+6)} \right)$$

$$= \frac{2(p+2)}{(p-2)(p-1)p(p+1)(p+2)(p+4)(p+6)}$$

### D.4 MONOMIALS OF FOUR ORTHOGONAL VECTOR

$$\left\langle \tfrac{2}{0}\ \tfrac{2}{0}\ \tfrac{2}{0}\ \tfrac{2}{0} \right\rangle = \tfrac{1}{p-3}\left( (-1)^1 \left( \left\langle \tfrac{4}{0}\ \tfrac{2}{0}\ \tfrac{2}{0} \right\rangle + \left\langle \tfrac{2}{0}\ \tfrac{4}{0}\ \tfrac{2}{0} \right\rangle + \left\langle \tfrac{2}{0}\ \tfrac{2}{0}\ \tfrac{4}{0} \right\rangle \right) + (-1)^0 \left\langle \tfrac{2}{0}\ \tfrac{2}{0}\ \tfrac{2}{0} \right\rangle \right)$$

$$= \tfrac{1}{p-3}\left( \left\langle \tfrac{2}{0}\ \tfrac{2}{0}\ \tfrac{2}{0} \right\rangle - 3\left\langle \tfrac{4}{0}\ \tfrac{2}{0}\ \tfrac{2}{0} \right\rangle \right)$$

$$= \tfrac{1}{p-3}\left( \tfrac{1}{p(p+2)(p+4)} - \tfrac{9}{p(p+2)(p+4)(p+6)} \right)$$

$$= \tfrac{p-3}{(p-3)p(p+2)(p+4)(p+6)} = \tfrac{1}{p(p+2)(p+4)(p+6)}$$

$$\left\langle \begin{smallmatrix} 2 & 2 & 2 & 0 \\ 0 & 0 & 0 & 2 \\ 0 & 0 & 0 & 0 \end{smallmatrix} \right\rangle = \left\langle \begin{smallmatrix} 2 & 0 \\ 0 & 2 \\ 2 & 0 \\ 0 & 0 \end{smallmatrix} \right\rangle = \tfrac{p+5}{(p-1)p(p+2)(p+4)(p+6)}$$

$$\left\langle \begin{smallmatrix} 2 & 2 & 0 & 0 \\ 0 & 0 & 2 & 2 \\ 0 & 0 & 0 & 0 \end{smallmatrix} \right\rangle = \left\langle \begin{smallmatrix} 2 & 0 \\ 2 & 0 \\ 0 & 2 \\ 0 & 0 \end{smallmatrix} \right\rangle = \tfrac{(p+3)(p+5)}{(p-1)p(p+1)(p+2)(p+4)(p+6)}$$

$$\left\langle \begin{smallmatrix} 2 & 2 & 0 & 0 \\ 0 & 0 & 2 & 0 \\ 0 & 0 & 0 & 2 \\ 0 & 0 & 0 & 0 \end{smallmatrix} \right\rangle = \left\langle \begin{smallmatrix} 0 & 2 & 0 \\ 0 & 0 & 2 \\ 2 & 0 & 0 \\ 0 & 0 & 0 \end{smallmatrix} \right\rangle = \tfrac{\left(p^3+8p^2+13p-2\right)}{(p-2)(p-1)p(p+1)(p+2)(p+4)(p+6)}$$

$$\left\langle \begin{smallmatrix} 2 & 0 & 0 & 0 \\ 0 & 2 & 0 & 0 \\ 0 & 0 & 2 & 0 \\ 0 & 0 & 0 & 2 \\ 0 & 0 & 0 & 0 \end{smallmatrix} \right\rangle = \tfrac{1}{p-3}\left( (-1)^1 \left( \left\langle \begin{smallmatrix} 2 & 0 & 0 \\ 0 & 2 & 0 \\ 0 & 0 & 2 \\ 2 & 0 & 0 \\ 0 & 0 & 0 \end{smallmatrix} \right\rangle + \left\langle \begin{smallmatrix} 2 & 0 & 0 \\ 0 & 2 & 0 \\ 0 & 0 & 2 \\ 0 & 2 & 0 \\ 0 & 0 & 0 \end{smallmatrix} \right\rangle + \left\langle \begin{smallmatrix} 2 & 0 & 0 \\ 0 & 2 & 0 \\ 0 & 0 & 2 \\ 0 & 0 & 2 \\ 0 & 0 & 0 \end{smallmatrix} \right\rangle \right) + (-1)^0 \left\langle \begin{smallmatrix} 2 & 0 & 0 \\ 0 & 2 & 0 \\ 0 & 0 & 2 \\ 0 & 0 & 0 \\ 0 & 0 & 0 \end{smallmatrix} \right\rangle \right)$$

$$= \tfrac{1}{p-3}\left( \left\langle \begin{smallmatrix} 2 & 0 & 0 \\ 0 & 2 & 0 \\ 0 & 0 & 2 \\ 0 & 0 & 0 \\ 0 & 0 & 0 \end{smallmatrix} \right\rangle - 3\left\langle \begin{smallmatrix} 2 & 0 & 0 \\ 0 & 2 & 0 \\ 0 & 0 & 2 \\ 0 & 0 & 2 \\ 0 & 0 & 0 \end{smallmatrix} \right\rangle \right)$$

$$= \tfrac{1}{p-3}\left( \tfrac{p^2+3p-2}{(p-2)(p-1)p(p+2)(p+4)} - \tfrac{3\left(p^3+8p^2+13p-2\right)}{(p-2)(p-1)p(p+1)(p+2)(p+4)(p+6)} \right)$$

$$= \tfrac{(p-2)(p+3)\left(p^2+6p+1\right)}{(p-3)(p-2)(p-1)p(p+1)(p+2)(p+4)(p+6)}$$

$$\left\langle \begin{smallmatrix} 1 & 1 & 2 & 2 \\ 1 & 1 & 0 & 0 \\ 0 & 0 & 0 & 0 \end{smallmatrix} \right\rangle = \left\langle \begin{smallmatrix} 2 & 0 \\ 2 & 0 \\ 1 & 1 \\ 1 & 1 \\ 0 & 0 \end{smallmatrix} \right\rangle = \tfrac{-1}{(p-1)p(p+2)(p+4)(p+6)}$$

$$\left\langle \begin{smallmatrix} 1 & 1 & 2 & 0 \\ 1 & 1 & 0 & 2 \\ 0 & 0 & 0 & 0 \end{smallmatrix} \right\rangle = \left\langle \begin{smallmatrix} 2 & 0 \\ 0 & 2 \\ 1 & 1 \\ 1 & 1 \\ 0 & 0 \end{smallmatrix} \right\rangle = \tfrac{-(p+3)}{(p-1)p(p+1)(p+2)(p+4)(p+6)}$$

$$\left\langle \begin{smallmatrix} 1 & 1 & 2 & 0 \\ 1 & 1 & 0 & 0 \\ 0 & 0 & 0 & 2 \\ 0 & 0 & 0 & 0 \end{smallmatrix} \right\rangle = \left\langle \begin{smallmatrix} 2 & 0 & 0 \\ 0 & 2 & 0 \\ 1 & 0 & 1 \\ 1 & 0 & 1 \\ 0 & 0 & 0 \end{smallmatrix} \right\rangle = \tfrac{-\left(p^2+5p+2\right)}{(p-2)(p-1)p(p+1)(p+2)(p+4)(p+6)}$$

$$\left\langle \begin{smallmatrix} 1 & 1 & 0 & 0 \\ 1 & 1 & 0 & 0 \\ 0 & 0 & 2 & 2 \\ 0 & 0 & 0 & 0 \end{smallmatrix} \right\rangle = \left\langle \begin{smallmatrix} 0 & 1 & 1 \\ 0 & 1 & 1 \\ 2 & 0 & 0 \\ 2 & 0 & 0 \\ 0 & 0 & 0 \end{smallmatrix} \right\rangle = \tfrac{-\left(p^2+7p+14\right)}{(p-2)(p-1)p(p+1)(p+2)(p+4)(p+6)}$$

$$\left\langle \begin{smallmatrix} 2 & 1 & 1 & 0 \\ 0 & 1 & 0 & 1 \\ 0 & 0 & 1 & 1 \\ 0 & 0 & 0 & 0 \end{smallmatrix} \right\rangle = \left\langle \begin{smallmatrix} 2 & 0 & 0 \\ 0 & 1 & 1 \\ 1 & 1 & 0 \\ 1 & 0 & 1 \\ 0 & 0 & 0 \end{smallmatrix} \right\rangle = \tfrac{2(p+2)}{(p-2)(p-1)p(p+1)(p+2)(p+4)(p+6)}$$

$$\left\langle \begin{smallmatrix} 1 & 1 & 0 & 0 \\ 1 & 1 & 0 & 0 \\ 0 & 0 & 2 & 0 \\ 0 & 0 & 0 & 2 \\ \vec{0} & \vec{0} & \vec{0} & \vec{0} \end{smallmatrix} \right\rangle = \frac{1}{p-3} \left( - \left\langle \begin{smallmatrix} 2 & 0 & 0 \\ 2 & 0 & 0 \\ 0 & 2 & 0 \\ 0 & 0 & 2 \\ \vec{0} & \vec{0} & \vec{0} \end{smallmatrix} \right\rangle - \left\langle \begin{smallmatrix} 1 & 1 & 0 \\ 1 & 1 & 0 \\ 0 & 2 & 0 \\ 0 & 0 & 2 \\ \vec{0} & \vec{0} & \vec{0} \end{smallmatrix} \right\rangle - \left\langle \begin{smallmatrix} 1 & 0 & 1 \\ 1 & 0 & 1 \\ 0 & 2 & 0 \\ 0 & 0 & 2 \\ \vec{0} & \vec{0} & \vec{0} \end{smallmatrix} \right\rangle \right)$$

$$= \frac{1}{p-3} \left( - \left\langle \begin{smallmatrix} 2 & 0 & 0 \\ 2 & 0 & 0 \\ 0 & 2 & 0 \\ 0 & 0 & 2 \\ \vec{0} & \vec{0} & \vec{0} \end{smallmatrix} \right\rangle - 2 \left\langle \begin{smallmatrix} 1 & 1 & 0 \\ 1 & 1 & 0 \\ 0 & 2 & 0 \\ 0 & 0 & 2 \\ \vec{0} & \vec{0} & \vec{0} \end{smallmatrix} \right\rangle \right)$$

$$= \frac{1}{p-3} \left( - \frac{p^3 + 8p^2 + 13p - 2}{(p-2)(p-1)p(p+1)(p+2)(p+4)(p+6)} + \frac{2\left(p^2 + 5p + 2\right)}{(p-2)(p-1)p(p+1)(p+2)(p+4)(p+6)} \right)$$

$$= \frac{-\left(p^3 + 6p^2 + 3p - 6\right)}{(p-3)(p-2)(p-1)p(p+1)(p+2)(p+4)(p+6)}$$

$$\left\langle \begin{smallmatrix} 2 & 0 & 0 & 0 \\ 0 & 1 & 1 & 0 \\ 0 & 0 & 1 & 1 \\ 0 & 0 & 1 & 1 \\ \vec{0} & \vec{0} & \vec{0} & \vec{0} \end{smallmatrix} \right\rangle = \frac{1}{p-3} \left( - \left\langle \begin{smallmatrix} 2 & 0 & 0 \\ 0 & 1 & 1 \\ 1 & 1 & 0 \\ 1 & 0 & 1 \\ \vec{0} & \vec{0} & \vec{0} \end{smallmatrix} \right\rangle - \underbrace{ \left\langle \begin{smallmatrix} 2 & 0 & 0 \\ 0 & 1 & 1 \\ 0 & 2 & 0 \\ 0 & 1 & 1 \\ \vec{0} & \vec{0} & \vec{0} \end{smallmatrix} \right\rangle - \left\langle \begin{smallmatrix} 2 & 0 & 0 \\ 0 & 1 & 1 \\ 0 & 1 & 1 \\ 0 & 0 & 2 \\ \vec{0} & \vec{0} & \vec{0} \end{smallmatrix} \right\rangle }_{\text{equal}} \right)$$

$$= \frac{1}{p-3} \left( \frac{2\left(p^2 + 5p + 2\right) - 2(p+2)}{(p-2)(p-1)p(p+1)(p+2)(p+4)(p+6)} \right)$$

$$= \frac{2p(p+4)}{(p-3)(p-2)(p-1)p(p+1)(p+2)(p+4)(p+6)}$$

$$\left\langle \begin{smallmatrix} 1 & 1 & 1 & 1 \\ 1 & 1 & 1 & 1 \\ \vec{0} & \vec{0} & \vec{0} & \vec{0} \end{smallmatrix} \right\rangle = \left\langle \begin{smallmatrix} 1 & 1 \\ 1 & 1 \\ 1 & 1 \\ \vec{0} & \vec{0} \end{smallmatrix} \right\rangle = \frac{3}{(p-1)p(p+1)(p+2)(p+4)(p+6)}$$

$$\left\langle \begin{smallmatrix} 1 & 1 & 1 & 1 \\ 1 & 1 & 0 & 0 \\ 0 & 0 & 1 & 1 \\ \vec{0} & \vec{0} & \vec{0} & \vec{0} \end{smallmatrix} \right\rangle = \frac{1}{p-3} \left( - \left\langle \begin{smallmatrix} 2 & 1 & 1 \\ 1 & 1 & 0 \\ 1 & 0 & 1 \\ \vec{0} & \vec{0} & \vec{0} \end{smallmatrix} \right\rangle - \left\langle \begin{smallmatrix} 1 & 2 & 1 \\ 1 & 1 & 0 \\ 0 & 1 & 1 \\ \vec{0} & \vec{0} & \vec{0} \end{smallmatrix} \right\rangle - \left\langle \begin{smallmatrix} 1 & 1 & 2 \\ 1 & 1 & 0 \\ 0 & 0 & 2 \\ \vec{0} & \vec{0} & \vec{0} \end{smallmatrix} \right\rangle \right)$$

$$= \frac{1}{p-3} \left( -2 \left\langle \begin{smallmatrix} 2 & 1 & 1 \\ 1 & 1 & 0 \\ 1 & 0 & 1 \\ \vec{0} & \vec{0} & \vec{0} \end{smallmatrix} \right\rangle - \left\langle \begin{smallmatrix} 1 & 1 & 2 \\ 1 & 1 & 0 \\ 0 & 0 & 2 \\ \vec{0} & \vec{0} & \vec{0} \end{smallmatrix} \right\rangle \right)$$

$$= \frac{1}{p-3} \left( \frac{\left(p^2 + 3p + 6\right) - 8p}{(p-2)(p-1)p(p+1)(p+2)(p+4)(p+6)} \right) = \frac{1}{(p-1)p(p+1)(p+2)(p+4)(p+6)}$$

$$\left\langle \begin{smallmatrix} 1 & 1 & 0 & 0 \\ 1 & 1 & 0 & 0 \\ 0 & 0 & 1 & 1 \\ 0 & 0 & 1 & 1 \\ \vec{0} & \vec{0} & \vec{0} & \vec{0} \end{smallmatrix} \right\rangle = \frac{1}{p-3} \left( - \left\langle \begin{smallmatrix} 1 & 1 & 0 \\ 1 & 1 & 0 \\ 1 & 0 & 1 \\ \vec{0} & \vec{0} & \vec{0} \end{smallmatrix} \right\rangle - \left\langle \begin{smallmatrix} 1 & 1 & 0 \\ 1 & 1 & 0 \\ 0 & 1 & 1 \\ \vec{0} & \vec{0} & \vec{0} \end{smallmatrix} \right\rangle - \left\langle \begin{smallmatrix} 1 & 1 & 0 \\ 1 & 1 & 0 \\ 0 & 0 & 2 \\ \vec{0} & \vec{0} & \vec{0} \end{smallmatrix} \right\rangle \right)$$

$$= \frac{1}{p-3} \left( -2 \left\langle \begin{smallmatrix} 1 & 1 & 0 \\ 1 & 1 & 0 \\ 0 & 1 & 1 \\ 0 & 1 & 1 \\ \vec{0} & \vec{0} & \vec{0} \end{smallmatrix} \right\rangle - \left\langle \begin{smallmatrix} 1 & 1 & 0 \\ 1 & 1 & 0 \\ 0 & 0 & 2 \\ \vec{0} & \vec{0} & \vec{0} \end{smallmatrix} \right\rangle \right)$$

$$= \frac{1}{p-3} \left( \frac{-2}{(p-1)p(p+1)(p+2)(p+4)(p+6)} + \frac{p^2 + 7p + 14}{(p-2)(p-1)p(p+1)(p+2)(p+4)(p+6)} \right)$$

$$= \frac{p^2 + 5p + 18}{(p-3)(p-2)(p-1)p(p+1)(p+2)(p+4)(p+6)}$$

$$\left\langle \begin{smallmatrix} 1 & 0 & 1 & 0 \\ 0 & 1 & 0 & 1 \\ 1 & 1 & 0 & 0 \\ 0 & 0 & 1 & 1 \\ \vec{0} & \vec{0} & \vec{0} & \vec{0} \end{smallmatrix} \right\rangle = \frac{1}{p-3} \left( - \left\langle \begin{smallmatrix} 1 & 0 & 1 \\ 1 & 1 & 0 \\ 1 & 1 & 0 \\ 1 & 0 & 1 \\ \vec{0} & \vec{0} & \vec{0} \end{smallmatrix} \right\rangle - \left\langle \begin{smallmatrix} 1 & 0 & 1 \\ 0 & 2 & 0 \\ 1 & 1 & 0 \\ 0 & 1 & 1 \\ \vec{0} & \vec{0} & \vec{0} \end{smallmatrix} \right\rangle - \left\langle \begin{smallmatrix} 1 & 0 & 1 \\ 0 & 1 & 1 \\ 1 & 1 & 0 \\ 0 & 0 & 2 \\ \vec{0} & \vec{0} & \vec{0} \end{smallmatrix} \right\rangle \right)$$

$$= \frac{1}{p-3} \left( - \frac{1}{(p-1)p(p+1)(p+2)(p+4)(p+6)} + \frac{-4(p+2)}{(p-2)(p-1)p(p+1)(p+2)(p+4)(p+6)} \right)$$

$$= \frac{-5p - 6}{(p-3)(p-2)(p-1)p(p+1)(p+2)(p+4)(p+6)}$$

## E  Auxiliary Expectation Derivations

In this section, we attach many auxiliary derivations of simple and complicated polynomials that we need in our main propositions and lemmas.

**Proposition 14.** *For $p \geq 2, m \in \{2, \ldots, p\}$ and a random transformation $\mathbf{O}$ sampled as described in Eq. (1), it holds that,*

$$\mathbb{E}\left[\left(\mathbf{e}_1^\top \mathbf{O} \mathbf{e}_1\right)^3\right] = \frac{(p-m)\left(m^2 - 2mp - 3m + p^2 + 6p + 14\right)}{p(p+2)(p+4)}.$$

*Proof.* We notice that the expectation can be written as

$$\mathbb{E}\left[\left(\mathbf{e}_1^\top \mathbf{O} \mathbf{e}_1\right)^3\right] = \mathbb{E}\left[\left(\mathbf{u}^\top \left[\begin{smallmatrix}\mathbf{Q}_m & \\ & \mathbf{0}\end{smallmatrix}\right]\mathbf{u} + \mathbf{u}^\top \left[\begin{smallmatrix}\mathbf{0} & \\ & \mathbf{I}_{p-m}\end{smallmatrix}\right]\mathbf{u}\right)^3\right].$$

Employing the algebraic identity that $(a+b)^3 = a^3 + 3a^2 b + 3ab^2 + b^3$ and Cor. 12 (canceling terms with an odd number of $\mathbf{Q}_m$ appearances), it can be readily seen that in our case we are left with $(a+b)^3 = \cancel{a^3} + 3a^2 b + \cancel{3ab^2} + b^3$. We thus get,

$$\mathbb{E}\left[\left(\mathbf{e}_1^\top \mathbf{O} \mathbf{e}_1\right)^3\right] = 3\mathbb{E}\left[\left(\mathbf{u}^\top \left[\begin{smallmatrix}\mathbf{Q}_m & \\ & \mathbf{0}\end{smallmatrix}\right]\mathbf{u}\right)^2 \mathbf{u}^\top \left[\begin{smallmatrix}\mathbf{0} & \\ & \mathbf{I}_{p-m}\end{smallmatrix}\right]\mathbf{u}\right] + \mathbb{E}\left[\left(\mathbf{u}^\top \left[\begin{smallmatrix}\mathbf{0} & \\ & \mathbf{I}_{p-m}\end{smallmatrix}\right]\mathbf{u}\right)^3\right]$$

$$= 3\mathbb{E}\left[\left(\mathbf{u}^\top \left[\begin{smallmatrix}\mathbf{Q}_m & \\ & \mathbf{0}\end{smallmatrix}\right]\mathbf{u}\right)^2 \sum_{i=m+1}^{p} u_i^2\right] + \mathbb{E}\left[\left(\sum_{i=m+1}^{p} u_i^2\right)^3\right]$$

$$= 3\sum_{i=m+1}^{p}\mathbb{E}\left[\left(\mathbf{u}^\top \left[\begin{smallmatrix}\mathbf{Q}_m & \\ & \mathbf{0}\end{smallmatrix}\right]\mathbf{u}\right)^2 u_i^2\right] + \sum_{i=m+1}^{p}\sum_{j=m+1}^{p}\sum_{k=m+1}^{p}\mathbb{E}\left[u_i^2 u_j^2 u_k^2\right]$$

$$= (p-m)\left(3\mathbb{E}\left[u_p^2 \cdot \mathbf{u}_a^\top \mathbf{Q}_m \mathbf{u}_a \cdot \mathbf{u}_a^\top \mathbf{Q}_m \mathbf{u}_a\right] + \sum_{j=m+1}^{p}\sum_{k=m+1}^{p}\mathbb{E}\left[u_p^2 u_j^2 u_k^2\right]\right)$$

The inner left term becomes

$$\mathbb{E}\left[u_p^2 \left\|\mathbf{u}_a\right\|^2 \cdot \mathbf{u}_a^\top \mathbb{E}_{\mathbf{r}\sim\mathcal{S}^{m-1}}\left[\mathbf{r}\mathbf{r}^\top\right]\mathbf{u}_a\right] = \frac{3}{m}\mathbb{E}\left[u_p^2 \left\|\mathbf{u}_a\right\|^4\right] = \frac{3}{m}\sum_{i=1}^{m}\sum_{j=1}^{m}\mathbb{E}\left[u_p^2 u_i^2 u_j^2\right]$$

$$= \frac{3}{m}\left(\underbrace{m\mathbb{E}\left[u_p^2 u_1^4\right]}_{i=j} + \underbrace{m(m-1)\mathbb{E}\left[u_p^2 u_1^2 u_2^2\right]}_{i\neq j}\right) = 3\left(\left\langle \begin{smallmatrix}4\\2\\0\end{smallmatrix}\right\rangle + (m-1)\left\langle \begin{smallmatrix}2\\2\\2\\0\end{smallmatrix}\right\rangle\right),$$

while the inner right term becomes

$$\underbrace{\sum_{k=m+1}^{p}\mathbb{E}\left[u_p^4 u_k^2\right]}_{j=p} + (p-m-1)\underbrace{\sum_{k=m+1}^{p}\mathbb{E}\left[u_{p-1}^2 u_p^2 u_k^2\right]}_{m+1\leq j\leq p-1}$$

$$= \underbrace{\mathbb{E}\left[u_p^6\right]}_{k=p} + (p-m-1)\underbrace{\mathbb{E}\left[u_{p-1}^2 u_p^4\right]}_{m+1\leq k\leq p-1} + (p-m-1)\left(\underbrace{2\mathbb{E}\left[u_{p-1}^2 u_p^4\right]}_{k=p \vee k=p-1} + (p-m-2)\underbrace{\mathbb{E}\left[u_{p-2}^2 u_{p-1}^2 u_p^2\right]}_{m+1\leq j\leq p-2}\right)$$

$$= \left\langle \begin{smallmatrix}6\\0\end{smallmatrix}\right\rangle + (p-m-1)\left(3\left\langle \begin{smallmatrix}4\\2\\0\end{smallmatrix}\right\rangle + (p-m-2)\left\langle \begin{smallmatrix}2\\2\\2\\0\end{smallmatrix}\right\rangle\right).$$

Combining these two inner terms, we get,

$$\mathbb{E}\left[\left(\mathbf{e}_1^\top \mathbf{O} \mathbf{e}_1\right)^3\right]$$

$$= (p-m)\left(\left\langle \begin{smallmatrix}6\\0\end{smallmatrix}\right\rangle + 3(p-m)\left\langle \begin{smallmatrix}4\\2\\0\end{smallmatrix}\right\rangle + \left(m^2 - 2mp + 6m + p^2 - 3p - 1\right)\left\langle \begin{smallmatrix}2\\2\\2\\0\end{smallmatrix}\right\rangle\right)$$

$$= (p - m) \left( \frac{15}{p\,(p+2)\,(p+4)} + \frac{9\,(p-m)}{p\,(p+2)\,(p+4)} + \frac{m^2 - 2mp + 6m + p^2 - 3p - 1}{p\,(p+2)\,(p+4)} \right)$$

$$= \frac{(p-m)\left(m^2 - 2mp - 3m + p^2 + 6p + 14\right)}{p\,(p+2)\,(p+4)}.$$

$\square$

**Proposition 15.** *For $p \geq 2, m \in \{2, \ldots, p\}$ and a random transformation $\mathbf{O}$ sampled as described in Eq. (1), it holds that,*

$$\mathbb{E}\left[\left(\mathbf{e}_2^\top \mathbf{O} \mathbf{e}_1\right)^2 \mathbf{e}_2^\top \mathbf{O} \mathbf{e}_2\right] = \frac{(p-m)\left(-m^2 + 2mp + 3m - 6\right)}{(p-1)\,p\,(p+2)\,(p+4)}.$$

*Proof.* By decomposing the expectation into the two additive terms below, we get that

$$\mathbb{E}\left[\left(\mathbf{e}_2^\top \mathbf{O} \mathbf{e}_1\right)^2 \mathbf{e}_2^\top \mathbf{O} \mathbf{e}_2\right] = \mathbb{E}\left[\left(\mathbf{u}^\top \left[\begin{smallmatrix} \mathbf{Q}_m & \\ & \mathbf{I}_{p-m} \end{smallmatrix}\right] \mathbf{v}\right)^2 \mathbf{u}^\top \left[\begin{smallmatrix} \mathbf{Q}_m & \\ & \mathbf{I}_{p-m} \end{smallmatrix}\right] \mathbf{u}\right]$$

$$= \mathbb{E}\left[\left(\mathbf{u}^\top \left[\begin{smallmatrix} \mathbf{Q}_m & \mathbf{0} \end{smallmatrix}\right] \mathbf{v} + \mathbf{u}^\top \left[\begin{smallmatrix} \mathbf{0} & \mathbf{I}_{p-m} \end{smallmatrix}\right] \mathbf{v}\right)^2 \left(\mathbf{u}^\top \left[\begin{smallmatrix} \mathbf{Q}_m & \mathbf{0} \end{smallmatrix}\right] \mathbf{u} + \mathbf{u}^\top \left[\begin{smallmatrix} \mathbf{0} & \mathbf{I}_{p-m} \end{smallmatrix}\right] \mathbf{u}\right)\right]$$

$$= \mathbb{E}\left[\left(\mathbf{u}_a \mathbf{Q}_m \mathbf{v}_a + \mathbf{u}_b^\top \mathbf{v}_b\right)^2 \left(\mathbf{u}_a \mathbf{Q}_m \mathbf{u}_a + \mathbf{u}_b^\top \mathbf{u}_b\right)\right]$$

$$= \underbrace{\mathbb{E}\left[\mathbf{u}_a \mathbf{Q}_m \mathbf{v}_a \left(\mathbf{u}_a \mathbf{Q}_m \mathbf{v}_a + \mathbf{u}_b^\top \mathbf{v}_b\right)\left(\mathbf{u}_a \mathbf{Q}_m \mathbf{u}_a + \mathbf{u}_b^\top \mathbf{u}_b\right)\right]}_{\text{below}} +$$

$$\underbrace{\mathbb{E}\left[\mathbf{u}_b^\top \mathbf{v}_b \left(\mathbf{u}_a \mathbf{Q}_m \mathbf{v}_a + \mathbf{u}_b^\top \mathbf{v}_b\right)\left(\mathbf{u}_a \mathbf{Q}_m \mathbf{u}_a + \mathbf{u}_b^\top \mathbf{u}_b\right)\right]}_{\text{below}}$$

$$= \frac{(p-m)\left(m\,(p+2) - 4 - m^2 + mp + m - 2\right)}{(p-1)\,p\,(p+2)\,(p+4)} = \frac{(p-m)\left(-m^2 + 2mp + 3m - 6\right)}{(p-1)\,p\,(p+2)\,(p+4)}.$$

Using Cor. 12 in the first step below, we show that the first term is,

$$\mathbb{E}\left[\mathbf{u}_a \mathbf{Q}_m \mathbf{v}_a \left(\mathbf{u}_a \mathbf{Q}_m \mathbf{v}_a + \mathbf{u}_b^\top \mathbf{v}_b\right)\left(\mathbf{u}_a \mathbf{Q}_m \mathbf{u}_a + \mathbf{u}_b^\top \mathbf{u}_b\right)\right]$$

$$= \mathbb{E}\left[\mathbf{u}_a \mathbf{Q}_m \mathbf{v}_a \mathbf{u}_b^\top \mathbf{v}_b \mathbf{u}_a \mathbf{Q}_m \mathbf{u}_a\right] + \mathbb{E}\left[\mathbf{u}_a \mathbf{Q}_m \mathbf{v}_a \mathbf{u}_a \mathbf{Q}_m \mathbf{v}_a \mathbf{u}_b^\top \mathbf{u}_b\right]$$

$$= \frac{1}{m}\mathbb{E}\left[\|\mathbf{u}_a\|^2 \mathbf{u}_a^\top \mathbf{v}_a \cdot \mathbf{u}_b^\top \mathbf{v}_b\right] + \frac{1}{m}\mathbb{E}\left[\|\mathbf{u}_a\|^2 \|\mathbf{v}_a\|^2 \|\mathbf{u}_b\|^2\right]$$

$$= -\frac{1}{m}\underbrace{\mathbb{E}\left[\|\mathbf{u}_a\|^2 \left(\mathbf{u}_b^\top \mathbf{v}_b\right)^2\right]}_{\text{solved in Eq. (17)}} + \frac{1}{m}\underbrace{\mathbb{E}\left[\|\mathbf{u}_a\|^2 \|\mathbf{v}_a\|^2 \|\mathbf{u}_b\|^2\right]}_{\text{solved in Eq. (19)}}$$

$$= -\frac{1}{m}\frac{(p-m)m(m+2)}{(p-1)p(p+2)(p+4)} + \frac{1}{m}\frac{m(p-m)(m(p+3)-2)}{(p-1)p(p+2)(p+4)} = \frac{(p-m)\left(m\,(p+2)-4\right)}{(p-1)\,p\,(p+2)\,(p+4)}.$$

From Eq. (9), we know that the second term is

$$\mathbb{E}\left[\mathbf{u}_b^\top \mathbf{v}_b \left(\mathbf{u}_a \mathbf{Q}_m \mathbf{v}_a + \mathbf{u}_b^\top \mathbf{v}_b\right)\left(\mathbf{u}_a \mathbf{Q}_m \mathbf{u}_a + \mathbf{u}_b^\top \mathbf{u}_b\right)\right]$$

$$= \underbrace{\mathbb{E}\left[\left(\mathbf{u}_b^\top \mathbf{v}_b\right)^2\right]}_{\text{solved in Eq. (23)}} - \left(1 + \frac{1}{m}\right)\underbrace{\mathbb{E}\left[\|\mathbf{u}_a\|^2 \left(\mathbf{u}_b^\top \mathbf{v}_b\right)^2\right]}_{\text{solved in Eq. (17)}} = \frac{(p-m)m}{(p-1)p(p+2)} - \frac{m+1}{m}\frac{(p-m)m(m+2)}{(p-1)p(p+2)(p+4)}$$

$$= \frac{(p-m)\left(-m^2 + mp + m - 2\right)}{(p-1)\,p\,(p+2)\,(p+4)}.$$

$\square$

**Proposition 16.** *For $p \geq 2, m \in \{2, \ldots, p\}$ and a random transformation $\mathbf{O}$ sampled as described in Eq. (1), it holds that,*

$$\mathbb{E}\left(\mathbf{e}_1^\top \mathbf{O} \mathbf{e}_1\right)^4 = \frac{3\,(m+4)\,(m+6) + (p-m)\,(p-m+2)\,\left(m^2 - 2mp - 4m + p^2 + 10p + 36\right)}{p\,(p+2)\,(p+4)\,(p+6)}$$

*Proof.* We start by showing,

$$\mathbb{E}\left(\mathbf{e}_1^\top \mathbf{O} \mathbf{e}_1\right)^4 = \mathbb{E}\left(\mathbf{u}^\top \begin{bmatrix} \mathbf{Q}_m & \\ & \mathbf{I}_{p-m} \end{bmatrix} \mathbf{u}\right)^4 = \mathbb{E}\left(\mathbf{u}_a^\top \mathbf{Q}_m \mathbf{u}_a + \mathbf{u}_b^\top \mathbf{u}_b\right)^4 .$$

Employing an algebraic identity and Cor. 12 to cancel terms with an odd number of $\mathbf{Q}_m$ appearances, it can be readily seen that in our case we are left with $(a+b)^4 = a^4 + \cancel{4a^3b} + 6a^2b^2 + \cancel{4ab^3} + b^4$. We thus get,

$$= \mathbb{E}\left(\mathbf{u}_a^\top \mathbf{Q}_m \mathbf{u}_a\right)^4 + 6\mathbb{E}\left(\mathbf{u}_a^\top \mathbf{Q}_m \mathbf{u}_a\right)^2 \left(\mathbf{u}_a^\top \mathbf{u}_a\right)^2 + \mathbb{E}\left(\mathbf{u}_b^\top \mathbf{u}_b\right)^4$$

$$= \mathbb{E}_{\mathbf{r}\sim\mathcal{S}^{m-1},\mathbf{u}\sim\mathcal{S}^{p-1}}\left(\|\mathbf{u}_a\|\,\mathbf{r}^\top\mathbf{u}_a\right)^4 + 6\mathbb{E}\left[\left(\mathbf{u}_b^\top \mathbf{u}_b\right)^2 \mathbb{E}_{\mathbf{r}\sim\mathcal{S}^{m-1}}\left(\|\mathbf{u}_a\|\,\mathbf{r}^\top\mathbf{u}_a\right)^2\right] + \mathbb{E}\left(\mathbf{u}_b^\top \mathbf{u}_b\right)^4$$

$$= \mathbb{E}\left[\|\mathbf{u}_a\|^4\,\mathbb{E}_{\mathbf{r}\sim\mathcal{S}^{m-1}}\left(\mathbf{r}^\top\mathbf{u}_a\right)^4\right] + 6\mathbb{E}\left[\|\mathbf{u}_b\|^4\,\|\mathbf{u}_a\|^2\,\mathbf{u}_a^\top\,\mathbb{E}_{\mathbf{r}\sim\mathcal{S}^{m-1}}\left[\mathbf{r}\mathbf{r}^\top\right]\mathbf{u}_a\right] + \mathbb{E}\|\mathbf{u}_b\|^8$$

We employ the isotropy of $\mathbf{r}$ and show that,

$$= \mathbb{E}\left[\|\mathbf{u}_a\|^4\,\mathbb{E}_{\mathbf{r}\sim\mathcal{S}^{m-1}}\left(\|\mathbf{u}_a\|\,\mathbf{r}^\top\mathbf{e}_1\right)^4\right] + \frac{6}{m}\mathbb{E}\left[\|\mathbf{u}_b\|^4\,\|\mathbf{u}_a\|^2\,\mathbf{u}_a^\top\mathbf{u}_a\right] + \mathbb{E}\|\mathbf{u}_b\|^8$$

$$= \mathbb{E}_{\mathbf{r}\sim\mathcal{S}^{m-1}}\left[r_1^4\right]\mathbb{E}\|\mathbf{u}_a\|^8 + \frac{6}{m}\mathbb{E}\left[\|\mathbf{u}_b\|^4\,\|\mathbf{u}_a\|^4\right] + \mathbb{E}\|\mathbf{u}_b\|^8$$

$$= \left\langle \tfrac{4}{0} \right\rangle_m \mathbb{E}\|\mathbf{u}_a\|^8 + \frac{6}{m}\mathbb{E}\left[\left(1 - \|\mathbf{u}_a\|^2\right)^2 \|\mathbf{u}_a\|^4\right] + \mathbb{E}\|\mathbf{u}_b\|^8$$

$$= \frac{3}{m\,(m+2)}\mathbb{E}\|\mathbf{u}_a\|^8 + \frac{6}{m}\left(\mathbb{E}\|\mathbf{u}_a\|^4 - 2\mathbb{E}\left[\|\mathbf{u}_a\|^2\,\|\mathbf{u}_a\|^4\right] + \mathbb{E}\|\mathbf{u}_a\|^8\right) + \mathbb{E}\|\mathbf{u}_b\|^8$$

$$= \left(\frac{3}{m\,(m+2)} + \frac{6}{m}\right)\mathbb{E}\|\mathbf{u}_a\|^8 + \frac{6}{m}\mathbb{E}\|\mathbf{u}_a\|^4 - \frac{12}{m}\mathbb{E}\|\mathbf{u}_a\|^6 + \mathbb{E}\|\mathbf{u}_b\|^8$$

$$= \frac{6m + 15}{m\,(m+2)}\mathbb{E}\|\mathbf{u}_a\|^8 + \frac{6}{m}\mathbb{E}\|\mathbf{u}_a\|^4 - \frac{12}{m}\mathbb{E}\|\mathbf{u}_a\|^6 + \mathbb{E}\|\mathbf{u}_b\|^8 ,$$

where we use the subscript in $\left\langle \tfrac{4}{0} \right\rangle_m$ to indicate that, unlike in most places, the corresponding random vector is in $\mathcal{S}^{m-1}$ rather than $\mathcal{S}^{p-1}$.

Next, we derive these expected norms in Prop. 26, and obtain

$$= \frac{6m + 15}{m\,(m+2)}\frac{m\,(m+2)\,(m+4)\,(m+6)}{p\,(p+2)\,(p+4)\,(p+6)} + \frac{6}{m}\frac{m\,(m+2)}{p\,(p+2)} - \frac{12}{m}\frac{m\,(m+2)\,(m+4)}{p\,(p+2)\,(p+4)} +$$

$$\frac{(p-m)\,(p-m+2)\,(p-m+4)\,(p-m+6)}{p\,(p+2)\,(p+4)\,(p+6)}$$

$$= \frac{(6m + 15)\,(m+4)\,(m+6)}{p\,(p+2)\,(p+4)\,(p+6)} + \frac{6\,(m+2)}{p\,(p+2)} - \frac{12\,(m+2)\,(m+4)}{p\,(p+2)\,(p+4)} +$$

$$\frac{(p-m)\,(p-m+2)\,(p-m+4)\,(p-m+6)}{p\,(p+2)\,(p+4)\,(p+6)}$$

$$= \frac{(6m + 15)\,(m+4)\,(m+6) + 6\,(m+2)\,(p+4)\,(p+6) - 12\,(m+2)\,(m+4)\,(p+6)}{p\,(p+2)\,(p+4)\,(p+6)}$$

$$\frac{(p-m)\,(p-m+2)\,(p-m+4)\,(p-m+6)}{p\,(p+2)\,(p+4)\,(p+6)}$$

$$= \frac{3\,(m+4)\,(m+6) + (p-m)\,(p-m+2)\,\left(m^2 - 2mp - 4m + p^2 + 10p + 36\right)}{p\,(p+2)\,(p+4)\,(p+6)} .$$

$\square$

**Proposition 17.** *For $p \geq 2, m \in \{2, \ldots, p\}$ and a random transformation $\mathbf{O}$ sampled as described in Eq. (1), it holds that,*

$$\mathbb{E}\left(\mathbf{e}_1^\top \mathbf{O} \mathbf{e}_2\right)^4 = \frac{3\left(m^4 - 2m^3(2p+3) + m^2\left(4p^2 + 4p - 1\right) + 2m\left(6p^2 + 8p + 3\right) + 8\left(p^2 - 2p - 3\right)\right)}{(p-1)p(p+1)(p+2)(p+4)(p+6)}.$$

*Proof.* We begin by showing that,

$$\mathbb{E}\left(\mathbf{e}_1^\top \mathbf{O} \mathbf{e}_2\right)^4 = \mathbb{E}\left(\mathbf{e}_1^\top \mathbf{Q}_p \begin{bmatrix} \mathbf{Q}_m & \\ & \mathbf{I}_{p-m} \end{bmatrix} \mathbf{Q}_p^\top \mathbf{e}_2\right)^4 = \mathbb{E}_{\mathbf{u}\perp\mathbf{v}}\left(\mathbf{u}^\top \begin{bmatrix} \mathbf{Q}_m & \\ & \mathbf{I}_{p-m} \end{bmatrix} \mathbf{v}\right)^4$$

$$= \mathbb{E}_{\mathbf{u}\perp\mathbf{v}}\left(\mathbf{u}_a^\top \mathbf{Q}_m \mathbf{v}_a + \mathbf{u}_b^\top \mathbf{v}_b\right)^4 = \mathbb{E}\left(\mathbf{u}_a^\top \mathbf{Q}_m \mathbf{v}_a + \mathbf{u}_b^\top \mathbf{v}_b\right)^4,$$

where the last step simply employs our simplifying notations (Remark 4).

Employing an algebraic identity and Cor. 12 to cancel terms with an odd number of $\mathbf{Q}_m$ appearances, it can be readily seen that in our case we are left with $(a+b)^4 = a^4 + \cancel{4a^3b} + 6a^2b^2 + \cancel{4ab^3} + b^4$. We thus get,

$$= \underbrace{\mathbb{E}\left[\left(\mathbf{u}_a^\top \mathbf{Q}_m \mathbf{v}_a\right)^4\right]}_{\text{solved in Eq. (25)}} + 6\underbrace{\mathbb{E}\left[\left(\mathbf{u}_a^\top \mathbf{Q}_m \mathbf{v}_a\right)^2 \left(\mathbf{u}_b^\top \mathbf{v}_b\right)^2\right]}_{\text{solved in Eq. (26)}} + \underbrace{\mathbb{E}\left[\left(\mathbf{u}_b^\top \mathbf{v}_b\right)^4\right]}_{\text{solved in Eq. (24)}}$$

$$= \frac{3\left(m^2(p+3)(p+5) + 2m(p+1)(p+3) - 8(2p+3)\right)}{(p-1)p(p+1)(p+2)(p+4)(p+6)} + 6\frac{(m+2)(mp+2p+3m)(p-m)}{(p-1)p(p+1)(p+2)(p+4)(p+6)} +$$

$$\frac{3m^4 - 6m^3p + 3m^2p^2 - 6m^2p - 12m^2 + 6mp^2 + 12mp}{(p-1)p(p+1)(p+2)(p+4)(p+6)}.$$

After some tedious algebra, we get, as required,

$$\mathbb{E}\left(\mathbf{e}_1^\top \mathbf{O} \mathbf{e}_2\right)^4 = \frac{3\left(m^4 - 2m^3(2p+3) + m^2\left(4p^2 + 4p - 1\right) + 2m\left(6p^2 + 8p + 3\right) + 8\left(p^2 - 2p - 3\right)\right)}{(p-1)p(p+1)(p+2)(p+4)(p+6)}.$$

$\square$

**Proposition 18.** *For $p \geq 2, m \in \{2, \ldots, p\}$ and a random transformation $\mathbf{O}$ sampled as described in Eq. (1), it holds that,*

$$\mathbb{E}\left(\mathbf{e}_1^\top \mathbf{O}\mathbf{e}_1 \cdot \mathbf{e}_2^\top \mathbf{O}\mathbf{e}_1\right)^2 = \mathbb{E}\left(\mathbf{e}_2^\top \mathbf{O}\mathbf{e}_2 \cdot \mathbf{e}_2^\top \mathbf{O}\mathbf{e}_1\right)^2$$

$$= \frac{(m+4)\left(2mp+4p+m-m^2-6\right)-(p-m)(p-m+2)(m(m-2p-5)+10)}{(p-1)\,p\,(p+2)\,(p+4)\,(p+6)}.$$

*Proof.* First, due to the exchangeability (Prop. 9), we have,

$$\mathbb{E}\left(\mathbf{e}_1^\top \mathbf{O}\mathbf{e}_1 \cdot \mathbf{e}_2^\top \mathbf{O}\mathbf{e}_1\right)^2 = \mathbb{E}\left(\mathbf{e}_2^\top \mathbf{O}\mathbf{e}_2 \mathbf{e}_1^\top \mathbf{O}\mathbf{e}_2\right)^2 = \mathbb{E}\left(\mathbf{e}_2^\top \mathbf{O}^\top \mathbf{e}_2 \mathbf{e}_2^\top \mathbf{O}^\top \mathbf{e}_1\right)^2 = \mathbb{E}\left(\mathbf{e}_2^\top \mathbf{O}\mathbf{e}_2 \cdot \mathbf{e}_2^\top \mathbf{O}\mathbf{e}_1\right)^2$$

Then, we show that,

$$\mathbb{E}\left(\mathbf{e}_2^\top \mathbf{O}\mathbf{e}_2 \cdot \mathbf{e}_2^\top \mathbf{O}\mathbf{e}_1\right)^2 = \mathbb{E}\left(\mathbf{u}^\top \left(\begin{bmatrix}\mathbf{Q}_m & \mathbf{0}\end{bmatrix} + \begin{bmatrix}\mathbf{0} & \mathbf{I}_{p-m}\end{bmatrix}\right)\mathbf{u} \cdot \mathbf{u}^\top \left(\begin{bmatrix}\mathbf{Q}_m & \mathbf{0}\end{bmatrix} + \begin{bmatrix}\mathbf{0} & \mathbf{I}_{p-m}\end{bmatrix}\right)\mathbf{v}\right)^2$$

$$= \mathbb{E}\left(\mathbf{u}_a^\top \mathbf{Q}_m \mathbf{u}_a + \mathbf{u}_b^\top \mathbf{u}_b\right)^2 \left(\mathbf{u}_a^\top \mathbf{Q}_m \mathbf{v}_a + \mathbf{u}_b^\top \mathbf{v}_b\right)^2$$

$$= \mathbb{E}\left(\left(\mathbf{u}_a^\top \mathbf{Q}_m \mathbf{u}_a\right)^2 + 2\mathbf{u}_a^\top \mathbf{Q}_m \mathbf{u}_a \mathbf{u}_b^\top \mathbf{u}_b + \|\mathbf{u}_b\|^4\right)\left(\mathbf{u}_a^\top \mathbf{Q}_m \mathbf{v}_a + \mathbf{u}_b^\top \mathbf{v}_b\right)^2$$

We now partition the above into three terms that we solve separately.

The first term is,

$$\mathbb{E}\left(\mathbf{u}_a^\top \mathbf{Q}_m \mathbf{u}_a\right)^2 \left(\left(\mathbf{u}_a^\top \mathbf{Q}_m \mathbf{v}_a\right)^2 + 2\mathbf{u}_a^\top \mathbf{Q}_m \mathbf{v}_a \mathbf{u}_b^\top \mathbf{v}_b + \left(\mathbf{u}_b^\top \mathbf{v}_b\right)^2\right)$$

$$= \underbrace{\mathbb{E}\left(\mathbf{u}_a^\top \mathbf{Q}_m \mathbf{u}_a\right)^2 \left(\mathbf{u}_a^\top \mathbf{Q}_m \mathbf{v}_a\right)^2}_{\text{solved in Eq. (14)}} + 2\underbrace{\mathbb{E}\left(\mathbf{u}_a^\top \mathbf{Q}_m \mathbf{u}_a\right)^2 \mathbf{u}_a^\top \mathbf{Q}_m \mathbf{v}_a \mathbf{u}_b^\top \mathbf{v}_b}_{=0,\text{ by Prop. 12}} + \underbrace{\mathbb{E}\left(\mathbf{u}_a^\top \mathbf{Q}_m \mathbf{u}_a\right)^2 \left(\mathbf{u}_b^\top \mathbf{v}_b\right)^2}_{\text{solved in Eq. (16)}}$$

$$= \frac{(m+4)(m(p+3)+2(p-3))}{(p-1)\,p\,(p+2)\,(p+4)\,(p+6)} + \frac{(m+2)(m+4)(p-m)}{(p-1)\,p\,(p+2)\,(p+4)\,(p+6)}$$

$$= \frac{(m+4)\left(2mp+4p+m-m^2-6\right)}{(p-1)\,p\,(p+2)\,(p+4)\,(p+6)}.$$

The second term is,

$$\mathbb{E}\left[\left(2\mathbf{u}_a^\top \mathbf{Q}_m \mathbf{u}_a \mathbf{u}_b^\top \mathbf{u}_b\right)\left(\left(\mathbf{u}_a^\top \mathbf{Q}_m \mathbf{v}_a\right)^2 + 2\mathbf{u}_a^\top \mathbf{Q}_m \mathbf{v}_a \mathbf{u}_b^\top \mathbf{v}_b + \left(\mathbf{u}_b^\top \mathbf{v}_b\right)^2\right)\right]$$

$$\overset{[\text{Prop. 12}]}{=} 4\mathbb{E}\left[\mathbf{u}_a^\top \mathbf{Q}_m \mathbf{u}_a \mathbf{u}_b^\top \mathbf{u}_b \cdot \mathbf{u}_a^\top \mathbf{Q}_m \mathbf{v}_a \mathbf{u}_b^\top \mathbf{v}_b\right]$$

$$= 4\mathbb{E}_{\mathbf{u}\perp\mathbf{v}}\left[\mathbb{E}_{\mathbf{Q}_m}\left[\mathbf{u}_a^\top \mathbf{Q}_m \mathbf{u}_a \mathbf{u}_a^\top \mathbf{Q}_m \mathbf{v}_a\right] \cdot \|\mathbf{u}_b\|^2 \cdot \mathbf{u}_b^\top \mathbf{v}_b\right]$$

$$= 4\mathbb{E}_{\mathbf{u}\perp\mathbf{v}}\left[\mathbb{E}_{\mathbf{r}\sim\mathcal{S}^{m-1}}\left[\left(\|\mathbf{u}_a\|\,\mathbf{u}_a^\top \mathbf{r}\right)\left(\|\mathbf{u}_a\|\,\mathbf{r}^\top \mathbf{v}_a\right)\right] \cdot \|\mathbf{u}_b\|^2 \cdot \mathbf{u}_b^\top \mathbf{v}_b\right]$$

$$= 4\mathbb{E}_{\mathbf{u}\perp\mathbf{v}}\left[\|\mathbf{u}_a\|^2 \|\mathbf{u}_b\|^2 \cdot \mathbf{u}_a^\top \mathbb{E}_{\mathbf{r}\sim\mathcal{S}^{m-1}}\left[\mathbf{r}\mathbf{r}^\top\right]\mathbf{v}_a \cdot \mathbf{u}_b^\top \mathbf{v}_b\right]$$

$$= \frac{4}{m}\mathbb{E}_{\mathbf{u}\perp\mathbf{v}}\left[\|\mathbf{u}_a\|^2 \|\mathbf{u}_b\|^2 \cdot \mathbf{u}_a^\top \mathbf{v}_a \cdot \mathbf{u}_b^\top \mathbf{v}_b\right] = \frac{4}{m}\mathbb{E}\left[\|\mathbf{u}_a\|^2 \left(1 - \|\mathbf{u}_a\|^2\right)\left(-\mathbf{u}_b^\top \mathbf{v}_b\right)\mathbf{u}_b^\top \mathbf{v}_b\right]$$

$$= \frac{4}{m}\underbrace{\mathbb{E}\left[\|\mathbf{u}_a\|^4 \left(\mathbf{u}_b^\top \mathbf{v}_b\right)^2\right]}_{\text{solved in Eq. (15)}} - \frac{4}{m}\underbrace{\mathbb{E}\left[\|\mathbf{u}_a\|^2 \left(\mathbf{u}_b^\top \mathbf{v}_b\right)^2\right]}_{\text{solved in Eq. (17)}}$$

$$= \frac{4}{m}\frac{m(m+2)(m+4)(p-m)}{(p-1)\,p\,(p+2)\,(p+4)\,(p+6)} - \frac{4}{m}\frac{(p-m)\,m\,(m+2)}{(p-1)\,p\,(p+2)\,(p+4)}$$

$$= \frac{4(p-m)(m+2)((m+4)-(p+6))}{(p-1)\,p\,(p+2)\,(p+4)\,(p+6)} = \frac{-4(p-m)\left(mp+2p+4-m^2\right)}{(p-1)\,p\,(p+2)\,(p+4)\,(p+6)}$$

The third term is,

$$\mathbb{E}\left[\|\mathbf{u}_b\|^4 \left(\left(\mathbf{u}_a^\top \mathbf{Q}_m \mathbf{v}_a\right)^2 + 2\mathbf{u}_a^\top \mathbf{Q}_m \mathbf{v}_a \mathbf{u}_b^\top \mathbf{v}_b + \left(\mathbf{u}_b^\top \mathbf{v}_b\right)^2\right)\right]$$

$$= \mathbb{E}\left[\|\mathbf{u}_b\|^4 \left(\mathbf{u}_a^\top \mathbf{Q}_m \mathbf{v}_a\right)^2\right] + 2\underbrace{\mathbb{E}\left[\|\mathbf{u}_b\|^4 \mathbf{u}_a^\top \mathbf{Q}_m \mathbf{v}_a \mathbf{u}_b^\top \mathbf{v}_b\right]}_{=0,\text{ by Prop. 12}} + \mathbb{E}\left[\|\mathbf{u}_b\|^4 \left(\mathbf{u}_b^\top \mathbf{v}_b\right)^2\right]$$

$$= \underbrace{\mathbb{E}\left[\|\mathbf{u}_b\|^4 \left(\mathbf{u}_a^\top \mathbf{Q}_m \mathbf{v}_a\right)^2\right]}_{\text{solved in Eq. (28)}} + \underbrace{\mathbb{E}\left[\|\mathbf{u}_b\|^4 \left(\mathbf{u}_a^\top \mathbf{v}_a\right)^2\right]}_{\substack{\text{solved in Eq. (15)} \\ \text{(plugging in } m \leftarrow p - m\text{)}}}$$

$$= \frac{(p-m)\left(mp^2 - m^2 p - 5m^2 + 7mp + 12m - 2p - 4\right)}{(p-1)p(p+2)(p+4)(p+6)} + \frac{(p-m)m\left(m^2 - 2mp - 6m + p^2 + 6p + 8\right)}{(p-1)p(p+2)(p+4)(p+6)}$$

$$= \frac{(p-m)\left(m^3 - 3m^2 p - 11m^2 + 2mp^2 + 13mp + 20m - 2p - 4\right)}{(p-1)p(p+2)(p+4)(p+6)}$$

Finally, summing the three terms (and after some tedious algebra), we get the required

$$\mathbb{E}\left(\mathbf{e}_2^\top \mathbf{O}\mathbf{e}_2 \cdot \mathbf{e}_2^\top \mathbf{O}\mathbf{e}_1\right)^2 = \frac{(m+4)\left(2mp + 4p + m - m^2 - 6\right) - (p-m)(p-m+2)(m(m-2p-5)+10)}{(p-1)p(p+2)(p+4)(p+6)} .$$

$\square$

**Proposition 19.** *For $p \geq 3, m \in \{2, 3, \ldots, p\}$ and a random transformation $\mathbf{O}$ sampled as described in Eq. (1), it holds that,*

$$\mathbb{E}\left(\mathbf{e}_2^\top \mathbf{O} \mathbf{e}_3 \cdot \mathbf{e}_2^\top \mathbf{O} \mathbf{e}_1\right)^2 = \frac{4mp(-m^2+m+4)+4(m+1)(m+2)p^2+(m-6)(m-1)m(m+1)-8(2p+3)}{(p-1)p(p+1)(p+2)(p+4)(p+6)}.$$

*Proof.* We show that,

$$\mathbb{E}\left(\mathbf{e}_2^\top \mathbf{O} \mathbf{e}_3 \cdot \mathbf{e}_2^\top \mathbf{O} \mathbf{e}_1\right)^2 = \mathbb{E}\left(\mathbf{u}^\top \left(\begin{bmatrix} \mathbf{Q}_m & \\ & \mathbf{0} \end{bmatrix} + \begin{bmatrix} \mathbf{0} & \\ & \mathbf{I}_{p-m} \end{bmatrix}\right) \mathbf{z} \cdot \mathbf{u}^\top \left(\begin{bmatrix} \mathbf{Q}_m & \\ & \mathbf{0} \end{bmatrix} + \begin{bmatrix} \mathbf{0} & \\ & \mathbf{I}_{p-m} \end{bmatrix}\right) \mathbf{v}\right)^2$$

$$= \mathbb{E}\left(\mathbf{u}_a^\top \mathbf{Q}_m \mathbf{z}_a + \mathbf{u}_b^\top \mathbf{z}_b\right)^2 \left(\mathbf{u}_a^\top \mathbf{Q}_m \mathbf{v}_a + \mathbf{u}_b^\top \mathbf{v}_b\right)^2$$

$$= \mathbb{E}\left(\left(\mathbf{u}_a^\top \mathbf{Q}_m \mathbf{z}_a\right)^2 + 2\mathbf{u}_a^\top \mathbf{Q}_m \mathbf{z}_a \mathbf{u}_b^\top \mathbf{z}_b + \left(\mathbf{u}_b^\top \mathbf{z}_b\right)^2\right)\left(\mathbf{u}_a^\top \mathbf{Q}_m \mathbf{v}_a + \mathbf{u}_b^\top \mathbf{v}_b\right)^2,$$

where the expectation is computed over the orthogonal matrix $\mathbf{Q}_m \sim O(m)$ and three random (isotropic) orthogonal unit vectors $\mathbf{u} \perp \mathbf{v} \perp \mathbf{z} \in \mathcal{S}^{p-1}$.

Again, starting from the first additive term,

$$\mathbb{E}\left[\left(\mathbf{u}_a^\top \mathbf{Q}_m \mathbf{z}_a\right)^2 \left(\left(\mathbf{u}_a^\top \mathbf{Q}_m \mathbf{v}_a\right)^2 + 2\mathbf{u}_a^\top \mathbf{Q}_m \mathbf{v}_a \mathbf{u}_b^\top \mathbf{v}_b + \left(\mathbf{u}_b^\top \mathbf{v}_b\right)^2\right)\right]$$

$$= \mathbb{E}\left[\underbrace{\left(\mathbf{u}_a^\top \mathbf{Q}_m \mathbf{z}_a\right)^2 \left(\mathbf{u}_a^\top \mathbf{Q}_m \mathbf{v}_a\right)^2}_{\text{solved in Eq. (34)}} + \underbrace{2\left(\mathbf{u}_a^\top \mathbf{Q}_m \mathbf{z}_a\right)^2 \mathbf{u}_a^\top \mathbf{Q}_m \mathbf{v}_a \mathbf{u}_b^\top \mathbf{v}_b}_{=0, \text{ by Cor. 12}} + \underbrace{\left(\mathbf{u}_a^\top \mathbf{Q}_m \mathbf{z}_a\right)^2 \left(\mathbf{u}_b^\top \mathbf{v}_b\right)^2}_{\text{solved in Eq. (35)}}\right]$$

$$= \frac{m(p+3)((m+2)p+5m+2)-16p-24}{(p-1)p(p+1)(p+2)(p+4)(p+6)} + \frac{(p-m)(m+2)\left(m\left(p^2+5p+2\right)-6p-4\right)}{(p-2)(p-1)p(p+1)(p+2)(p+4)(p+6)}$$

$$= \frac{(p-2)(m(p+3)((m+2)p+5m+2)-16p-24)+(p-m)(m+2)\left(m\left(p^2+5p+2\right)-6p-4\right)}{(p-2)(p-1)p(p+1)(p+2)(p+4)(p+6)}.$$

The second term is,

$$\mathbb{E}\left[2\mathbf{u}_a^\top \mathbf{Q}_m \mathbf{z}_a \mathbf{u}_b^\top \mathbf{z}_b \left(\left(\mathbf{u}_a^\top \mathbf{Q}_m \mathbf{v}_a\right)^2 + 2\mathbf{u}_a^\top \mathbf{Q}_m \mathbf{v}_a \mathbf{u}_b^\top \mathbf{v}_b + \left(\mathbf{u}_b^\top \mathbf{v}_b\right)^2\right)\right]$$

$$= \underbrace{\mathbb{E}\left[2\mathbf{u}_a^\top \mathbf{Q}_m \mathbf{z}_a \mathbf{u}_b^\top \mathbf{z}_b \left(\mathbf{u}_a^\top \mathbf{Q}_m \mathbf{v}_a\right)^2\right]}_{=0, \text{ by Cor. 12}} + 4\mathbb{E}\left[\mathbf{u}_a^\top \mathbf{Q}_m \mathbf{z}_a \mathbf{u}_b^\top \mathbf{z}_b \mathbf{u}_a^\top \mathbf{Q}_m \mathbf{v}_a \mathbf{u}_b^\top \mathbf{v}_b\right] +$$

$$\underbrace{\mathbb{E}\left[2\mathbf{u}_a^\top \mathbf{Q}_m \mathbf{z}_a \mathbf{u}_b^\top \mathbf{z}_b \left(\mathbf{u}_b^\top \mathbf{v}_b\right)^2\right]}_{=0, \text{ by Cor. 12}}$$

$$= 4\mathbb{E}_{\mathbf{u}\perp\mathbf{v}\perp\mathbf{z},\mathbf{Q}_m}\left[\mathbf{u}_a^\top \mathbf{Q}_m \mathbf{z}_a \mathbf{u}_b^\top \mathbf{z}_b \mathbf{u}_a^\top \mathbf{Q}_m \mathbf{v}_a \mathbf{u}_b^\top \mathbf{v}_b\right]$$

$$= 4\mathbb{E}_{\mathbf{u}\perp\mathbf{v}\perp\mathbf{z}}\left[\|\mathbf{u}_a\|^2 \mathbb{E}_{\mathbf{r}\sim\mathcal{S}^{m-1}}\left(\mathbf{v}_a^\top \mathbf{r}\mathbf{r}^\top \mathbf{z}_a\right) \mathbf{u}_b^\top \mathbf{z}_b \mathbf{u}_b^\top \mathbf{v}_b\right] = \frac{4}{m}\underbrace{\mathbb{E}_{\mathbf{u}\perp\mathbf{v}\perp\mathbf{z}}\left[\|\mathbf{u}_a\|^2 \mathbf{v}_a^\top \mathbf{z}_a \mathbf{u}_b^\top \mathbf{z}_b \mathbf{u}_b^\top \mathbf{v}_b\right]}_{\text{solved in Eq. (39)}}$$

$$= \frac{4(p-m)\left((m+2)p^2+(2-3m)p-2m^2(p+2)-6m+4\right)}{(p-2)(p-1)p(p+1)(p+2)(p+4)(p+6)}.$$

And the third additive term is,

$$\mathbb{E}\left[\left(\mathbf{u}_b^\top \mathbf{z}_b\right)^2 \left(\left(\mathbf{u}_a^\top \mathbf{Q}_m \mathbf{v}_a\right)^2 + 2\mathbf{u}_a^\top \mathbf{Q}_m \mathbf{v}_a \mathbf{u}_b^\top \mathbf{v}_b + \left(\mathbf{u}_b^\top \mathbf{v}_b\right)^2\right)\right]$$

$$= \underbrace{\mathbb{E}\left[\left(\mathbf{u}_b^\top \mathbf{z}_b\right)^2 \left(\mathbf{u}_a^\top \mathbf{Q}_m \mathbf{v}_a\right)^2\right]}_{\substack{=\mathbb{E}\left[\left(\mathbf{u}_a^\top \mathbf{Q}_m \mathbf{z}_a\right)^2\left(\mathbf{u}_b^\top \mathbf{v}_b\right)^2\right] \\ \text{due to the invariance (Prop. 9),} \\ \text{already solved in Eq. (35)}}} + 2\underbrace{\mathbb{E}\left[\left(\mathbf{u}_b^\top \mathbf{z}_b\right)^2 \mathbf{u}_a^\top \mathbf{Q}_m \mathbf{v}_a \mathbf{u}_b^\top \mathbf{v}_b\right]}_{=0, \text{ by Cor. 12}} + \underbrace{\mathbb{E}\left[\left(\mathbf{u}_b^\top \mathbf{z}_b\right)^2 \left(\mathbf{u}_b^\top \mathbf{v}_b\right)^2\right]}_{\text{solved in Eq. (40)}}$$

$$= \frac{(p-m)(m+2)\left(m\left(p^2+5p+2\right)-6p-4\right)}{(p-2)(p-1)p(p+1)(p+2)(p+4)(p+6)} + \frac{m(p-m)\left(-m^2+mp+2p+4\right)}{(p-1)p(p+1)(p+2)(p+4)(p+6)}$$

$$= \frac{(p-m)\left((m+2)\left(m\left(p^2+5p+2\right)-6p-4\right)+(p-2)m\left(-m^2+mp+2p+4\right)\right)}{(p-2)(p-1)p(p+1)(p+2)(p+4)(p+6)}.$$

By adding these three terms and after some tedious algebra, we get the required proposition. $\quad\square$

**Proposition 20.** *For $p \geq 2, m \in \{2, \ldots, p\}$ and a random transformation $\mathbf{O}$ sampled as described in Eq. (1), it holds that,*

$$\mathbb{E}\left[\mathbf{e}_2^\top \mathbf{O}\mathbf{e}_1 \mathbf{e}_1^\top \mathbf{O}\mathbf{e}_1 \mathbf{e}_1^\top \mathbf{O}\mathbf{e}_2\right] = \frac{(p-m)\left(-m^2 + mp - m + p - 2\right)}{(p-1)\,p\,(p+2)\,(p+4)}$$

*Proof.* First, we notice that

$$\mathbb{E}_{\mathbf{u}\perp\mathbf{v},\mathbf{Q}_m}\left[\mathbf{v}_a^\top \mathbf{Q}_m \mathbf{u}_a \mathbf{u}_a^\top \mathbf{Q}_m \mathbf{u}_a \mathbf{u}_b^\top \mathbf{v}_b\right] = \mathbb{E}_{\mathbf{u}\perp\mathbf{v}}\left[\mathbf{v}_a^\top \mathbb{E}_{\mathbf{r}\sim\mathcal{S}^{m-1}}\left[\left(\|\mathbf{u}_a\|\,\mathbf{r}\right)\left(\|\mathbf{u}_a\|\,\mathbf{r}\right)^\top\right]\mathbf{u}_a \mathbf{u}_b^\top \mathbf{v}_b\right]$$

$$= \mathbb{E}_{\mathbf{u}\perp\mathbf{v}}\left[\|\mathbf{u}_a\|^2\,\mathbf{v}_a^\top \mathbb{E}_{\mathbf{r}\sim\mathcal{S}^{m-1}}\left[\mathbf{r}\mathbf{r}^\top\right]\mathbf{u}_a \mathbf{u}_b^\top \mathbf{v}_b\right]$$

$$= \frac{1}{m}\mathbb{E}_{\mathbf{u}\perp\mathbf{v}}\left[\|\mathbf{u}_a\|^2\,\mathbf{v}_a^\top \mathbf{u}_a \mathbf{u}_b^\top \mathbf{v}_b\right] = -\frac{1}{m}\mathbb{E}\left[\|\mathbf{u}_a\|^2\left(\mathbf{u}_b^\top \mathbf{v}_b\right)^2\right].$$

Then, we focus on our quantity of interest here,

$$\mathbb{E}\left(\mathbf{e}_2^\top \mathbf{O}\mathbf{e}_1 \mathbf{e}_1^\top \mathbf{O}\mathbf{e}_1 \mathbf{e}_1^\top \mathbf{O}\mathbf{e}_2\right)$$

$$= \mathbb{E}_{\mathbf{u}\perp\mathbf{v},\mathbf{Q}_m}\left[\mathbf{v}^\top \begin{bmatrix} \mathbf{Q}_m & \\ & \mathbf{I}_{p-m} \end{bmatrix}\mathbf{u}\mathbf{u}^\top \begin{bmatrix} \mathbf{Q}_m & \\ & \mathbf{I}_{p-m} \end{bmatrix}\mathbf{u}\mathbf{u}^\top \begin{bmatrix} \mathbf{Q}_m & \\ & \mathbf{I}_{p-m} \end{bmatrix}\mathbf{v}\right]$$

$$= \mathbb{E}\left[\left(\mathbf{v}_a^\top \mathbf{Q}_m \mathbf{u}_a + \mathbf{v}_b^\top \mathbf{u}_b\right)\left(\mathbf{u}_a^\top \mathbf{Q}_m \mathbf{u}_a + \mathbf{u}_b^\top \mathbf{u}_b\right)\left(\mathbf{u}_a^\top \mathbf{Q}_m \mathbf{v}_a + \mathbf{u}_b^\top \mathbf{v}_b\right)\right].$$

Splitting the first multiplicative term, we get,

$$\mathbb{E}\left[\mathbf{v}_a^\top \mathbf{Q}_m \mathbf{u}_a\left(\mathbf{u}_a^\top \mathbf{Q}_m \mathbf{u}_a + \mathbf{u}_b^\top \mathbf{u}_b\right)\left(\mathbf{u}_a^\top \mathbf{Q}_m \mathbf{v}_a + \mathbf{u}_b^\top \mathbf{v}_b\right)\right]$$

$$\text{Cor. 12} = \mathbb{E}\left[\mathbf{v}_a^\top \mathbf{Q}_m \mathbf{u}_a \mathbf{u}_a^\top \mathbf{Q}_m \mathbf{u}_a \mathbf{u}_b^\top \mathbf{v}_b\right] + \mathbb{E}\left[\mathbf{v}_a^\top \mathbf{Q}_m \mathbf{u}_a \mathbf{u}_b^\top \mathbf{u}_b \mathbf{u}_a^\top \mathbf{Q}_m \mathbf{v}_a\right]$$

$$= \mathbb{E}\left[\mathbf{v}_a^\top \mathbf{Q}_m \mathbf{u}_a \mathbf{u}_a^\top \mathbf{Q}_m \mathbf{u}_a \mathbf{u}_b^\top \mathbf{v}_b\right] + \underbrace{\mathbb{E}\left[\|\mathbf{u}_b\|^2\,\mathbf{v}_a^\top \mathbf{Q}_m \mathbf{u}_a \mathbf{u}_a^\top \mathbf{Q}_m \mathbf{v}_a\right]}_{\text{solved in Eq. (27)}},$$

$$[\text{above}] = -\frac{1}{m}\mathbb{E}_{\mathbf{u}\perp\mathbf{v}}\left[\|\mathbf{u}_a\|^2\left(\mathbf{u}_b^\top \mathbf{v}_b\right)^2\right] + \frac{(p-m)(p-m+2)}{(p-1)\,p\,(p+2)\,(p+4)},$$

and

$$\mathbb{E}\left[\mathbf{v}_b^\top \mathbf{u}_b\left(\mathbf{u}_a^\top \mathbf{Q}_m \mathbf{u}_a + \mathbf{u}_b^\top \mathbf{u}_b\right)\left(\mathbf{u}_a^\top \mathbf{Q}_m \mathbf{v}_a + \mathbf{u}_b^\top \mathbf{v}_b\right)\right]$$

$$[\text{Cor. 12}] = \mathbb{E}\left[\mathbf{v}_b^\top \mathbf{u}_b \mathbf{u}_a^\top \mathbf{Q}_m \mathbf{u}_a \mathbf{u}_a^\top \mathbf{Q}_m \mathbf{v}_a\right] + \mathbb{E}\left[\mathbf{v}_b^\top \mathbf{u}_b \mathbf{u}_b^\top \mathbf{u}_b \mathbf{u}_b^\top \mathbf{v}_b\right]$$

$$= \mathbb{E}\left[\mathbf{v}_b^\top \mathbf{u}_b \mathbf{u}_a^\top \mathbf{Q}_m \mathbf{u}_a \mathbf{u}_a^\top \mathbf{Q}_m \mathbf{v}_a\right] + \mathbb{E}\left[\|\mathbf{u}_b\|^2\left(\mathbf{u}_b^\top \mathbf{v}_b\right)^2\right]$$

$$[\text{invariance of } \mathbf{Q}_m \text{ w.r.t. transpose (Prop. 9)}] \tag{9}$$

$$= \mathbb{E}\left[\mathbf{v}_a^\top \mathbf{Q}_m \mathbf{u}_a \mathbf{u}_a^\top \mathbf{Q}_m \mathbf{u}_a \mathbf{u}_b^\top \mathbf{v}_b\right] + \mathbb{E}\left[\left(1 - \|\mathbf{u}_a\|^2\right)\left(\mathbf{u}_b^\top \mathbf{v}_b\right)^2\right]$$

$$[\text{above}] = -\frac{1}{m}\mathbb{E}_{\mathbf{u}\perp\mathbf{v}}\left[\|\mathbf{u}_a\|^2\left(\mathbf{u}_b^\top \mathbf{v}_b\right)^2\right] + \mathbb{E}\left[\left(\mathbf{u}_b^\top \mathbf{v}_b\right)^2\right] - \mathbb{E}\left[\|\mathbf{u}_a\|^2\left(\mathbf{u}_b^\top \mathbf{v}_b\right)^2\right].$$

Combining the above, we get

$$\mathbb{E}\left(\mathbf{e}_2^\top \mathbf{O}\mathbf{e}_1 \mathbf{e}_1^\top \mathbf{O}\mathbf{e}_1 \mathbf{e}_1^\top \mathbf{O}\mathbf{e}_2\right) = \frac{(p-m)(p-m+2)}{(p-1)p(p+2)(p+4)} + \underbrace{\mathbb{E}\left[\left(\mathbf{u}_b^\top \mathbf{v}_b\right)^2\right]}_{\text{solved in Eq. (23)}} - \left(1 + \frac{2}{m}\right)\underbrace{\mathbb{E}\left[\|\mathbf{u}_a\|^2\left(\mathbf{u}_b^\top \mathbf{v}_b\right)^2\right]}_{\text{solved in Eq. (17)}}$$

$$= \frac{(p-m)(p-m+2)}{(p-1)p(p+2)(p+4)} + \frac{(p-m)m}{(p-1)p(p+2)} - \frac{m+2}{m}\frac{(p-m)m(m+2)}{(p-1)p(p+2)(p+4)}$$

$$= \frac{(p-m)\left(-m^2 + mp - m + p - 2\right)}{(p-1)\,p\,(p+2)\,(p+4)}.$$

$\square$

**Proposition 21.** *For $p \geq 3, m \in \{2, 3, \ldots, p\}$ and a random transformation $\mathbf{O}$ sampled as described in Eq. (1), it holds that,*

$$\mathbb{E}\left[\mathbf{e}_2^\top \mathbf{O} \mathbf{e}_1 \mathbf{e}_3^\top \mathbf{O} \mathbf{e}_1 \mathbf{e}_3^\top \mathbf{O} \mathbf{e}_2\right] = \frac{(p-m)\left(2m^2 - 3mp - 2m - p + 8\right)}{(p-2)(p-1)p(p+2)(p+4)}$$

*Proof.* We start by decomposing the expectation into the following,

$$\mathbb{E}\left[\mathbf{e}_2^\top \mathbf{O} \mathbf{e}_1 \mathbf{e}_3^\top \mathbf{O} \mathbf{e}_1 \mathbf{e}_3^\top \mathbf{O} \mathbf{e}_2\right]$$

$$= \mathbb{E}\left[\mathbf{v}^\top \begin{bmatrix} \mathbf{Q}_m & \\ & \mathbf{I}_{p-m} \end{bmatrix} \mathbf{u} \cdot \mathbf{z}^\top \begin{bmatrix} \mathbf{Q}_m & \\ & \mathbf{I}_{p-m} \end{bmatrix} \mathbf{u} \cdot \mathbf{z}^\top \begin{bmatrix} \mathbf{Q}_m & \\ & \mathbf{I}_{p-m} \end{bmatrix} \mathbf{v}\right]$$

$$= \mathbb{E}\left[\mathbf{v}_a^\top \mathbf{Q}_m \mathbf{u}_a \cdot \left(\mathbf{z}_a^\top \mathbf{Q}_m \mathbf{u}_a + \mathbf{z}_b^\top \mathbf{u}_b\right) \cdot \left(\mathbf{z}_a^\top \mathbf{Q}_m \mathbf{v}_a + \mathbf{z}_b^\top \mathbf{v}_b\right)\right] +$$

$$\mathbb{E}\left[\mathbf{v}_b^\top \mathbf{u}_b \cdot \left(\mathbf{z}_a^\top \mathbf{Q}_m \mathbf{u}_a + \mathbf{z}_b^\top \mathbf{u}_b\right) \cdot \left(\mathbf{z}_a^\top \mathbf{Q}_m \mathbf{v}_a + \mathbf{z}_b^\top \mathbf{v}_b\right)\right]$$

Focusing on the first additive term and by employing Cor. 12 (in the first step below), we get,

$$\mathbb{E}\left[\mathbf{v}_a^\top \mathbf{Q}_m \mathbf{u}_a \left(\mathbf{z}_a^\top \mathbf{Q}_m \mathbf{u}_a + \mathbf{z}_b^\top \mathbf{u}_b\right) \left(\mathbf{z}_a^\top \mathbf{Q}_m \mathbf{v}_a + \mathbf{z}_b^\top \mathbf{v}_b\right)\right]$$

$$= \mathbb{E}\left[\mathbf{v}_a^\top \mathbf{Q}_m \mathbf{u}_a \cdot \mathbf{z}_a^\top \mathbf{Q}_m \mathbf{u}_a \cdot \mathbf{z}_b^\top \mathbf{v}_b\right] + \mathbb{E}\left[\mathbf{v}_a^\top \mathbf{Q}_m \mathbf{u}_a \cdot \mathbf{z}_b^\top \mathbf{u}_b \cdot \mathbf{z}_a^\top \mathbf{Q}_m \mathbf{v}_a\right]$$

$$= \mathbb{E}\left[\|\mathbf{u}_a\|^2 \mathbf{v}_a^\top \mathbb{E}_{\mathbf{r} \sim \mathcal{S}^{m-1}}\left[\mathbf{r}\mathbf{r}^\top\right] \mathbf{z}_a \cdot \mathbf{z}_b^\top \mathbf{v}_b\right] + \mathbb{E}\left[\mathbf{z}_b^\top \mathbf{u}_b \cdot \left(\sum_{i,j=1}^m v_i q_{ij} u_j\right)\left(\sum_{k,\ell=1}^m z_k q_{k\ell} v_\ell\right)\right]$$

$$= \frac{1}{m}\mathbb{E}\left[\|\mathbf{u}_a\|^2 \underbrace{\mathbf{v}_a^\top \mathbf{z}_a}_{=-\mathbf{z}_b^\top \mathbf{v}_b} \cdot \mathbf{z}_b^\top \mathbf{v}_b\right] + \mathbb{E}\left[\left(\sum_{s=m+1}^p u_s z_s\right) \cdot \left(\sum_{i,j=1}^m v_i q_{ij} u_j\right)\left(\sum_{k,\ell=1}^m z_k q_{k\ell} v_\ell\right)\right]$$

$$= -\frac{1}{m}\underbrace{\mathbb{E}\left[\|\mathbf{u}_a\|^2 \left(\mathbf{v}_b^\top \mathbf{z}_b\right)^2\right]}_{\text{solved in Eq. (41)}} + (p-m)\sum_{i,j=1}^m \sum_{k,\ell=1}^m \mathbb{E}\left[u_j u_p v_i v_\ell z_k z_p\right] \mathbb{E}\left[q_{ij} q_{k\ell}\right]$$

[By Prop. 10, most summands are zero]

$$= -\frac{1}{m}\frac{m(p-m)(mp+2m-4)}{(p-2)(p-1)p(p+2)(p+4)} + (p-m)\sum_{i,j=1}^m \mathbb{E}\left[u_j u_p v_i v_j z_i z_p\right] \mathbb{E}\left[q_{ij}^2\right]$$

$$= -\frac{(p-m)(mp+2m-4)}{(p-2)(p-1)p(p+2)(p+4)} + \frac{p-m}{m}\left(\underbrace{m\mathbb{E}\left[u_1 u_p v_1^2 z_1 z_p\right]}_{i=j} + \underbrace{m(m-1)\mathbb{E}\left[u_2 u_p v_1 v_2 z_1 z_p\right]}_{i\neq j}\right)$$

$$= -\frac{(p-m)(mp+2m-4)}{(p-2)(p-1)p(p+2)(p+4)} + (p-m)\left(\left\langle \frac{1}{\vec{0}}\frac{2}{\vec{0}}\frac{1}{\vec{0}}\right\rangle + (m-1)\left\langle \frac{1}{\vec{0}}\frac{1}{\vec{0}}\frac{0}{\vec{0}}\frac{1}{\vec{0}}\right\rangle\right)$$

$$= (p-m)\left(-\frac{mp+2m-4}{(p-2)(p-1)p(p+2)(p+4)} + \frac{-1}{(p-1)p(p+2)(p+4)} + \frac{2(m-1)}{(p-2)(p-1)p(p+2)(p+4)}\right)$$

$$= (p-m)\left(\frac{-mp+2}{(p-2)(p-1)p(p+2)(p+4)} + \frac{-1}{(p-1)p(p+2)(p+4)}\right)$$

$$= \frac{(p-m)(-mp-p+4)}{(p-2)(p-1)p(p+2)(p+4)}$$

In addition, the second term is,

$$\mathbb{E}\left[\mathbf{v}_b^\top \mathbf{u}_b \left(\mathbf{z}_a^\top \mathbf{Q}_m \mathbf{u}_a + \mathbf{z}_b^\top \mathbf{u}_b\right)\left(\mathbf{z}_a^\top \mathbf{Q}_m \mathbf{v}_a + \mathbf{z}_b^\top \mathbf{v}_b\right)\right]$$

$$= \mathbb{E}\left[\mathbf{v}_b^\top \mathbf{u}_b \cdot \mathbf{z}_a^\top \mathbf{Q}_m \mathbf{u}_a \cdot \mathbf{z}_a^\top \mathbf{Q}_m \mathbf{v}_a\right] + \mathbb{E}\left[\mathbf{v}_b^\top \mathbf{u}_b \cdot \mathbf{z}_b^\top \mathbf{u}_b \cdot \mathbf{z}_b^\top \mathbf{v}_b\right]$$

$$= \mathbb{E}\left[\|\mathbf{z}_a\|^2 \mathbf{v}_b^\top \mathbf{u}_b \mathbf{v}_a^\top \mathbb{E}_{\mathbf{r} \sim \mathcal{S}^{m-1}}\left[\mathbf{r}\mathbf{r}^\top\right] \mathbf{u}_a\right] + \mathbb{E}\left[\mathbf{v}_b^\top \mathbf{u}_b \mathbf{z}_b^\top \mathbf{u}_b \mathbf{z}_b^\top \mathbf{v}_b\right]$$

$$= \frac{1}{m}\mathbb{E}\left[\|\mathbf{z}_a\|^2 \mathbf{v}_b^\top \mathbf{u}_b \underbrace{\mathbf{v}_a^\top \mathbf{u}_a}_{=-\mathbf{v}_b^\top \mathbf{u}_b}\right] + \mathbb{E}\left[\mathbf{v}_b^\top \mathbf{u}_b \mathbf{z}_b^\top \mathbf{u}_b \underbrace{\mathbf{z}_b^\top \mathbf{v}_b}_{=-\mathbf{z}_a^\top \mathbf{v}_a}\right]$$

$$= -\frac{1}{m} \underbrace{\mathbb{E}\left[\|\mathbf{z}_a\|^2 \left(\mathbf{v}_b^\top \mathbf{u}_b\right)^2\right]}_{\text{solved in Eq. (41)}} - \mathbb{E}\left[\mathbf{v}_b^\top \mathbf{u}_b \mathbf{z}_b^\top \mathbf{u}_b \mathbf{z}_a^\top \mathbf{v}_a\right]$$

$$= -\frac{1}{m} \frac{m(p-m)(mp+2m-4)}{(p-2)(p-1)p(p+2)(p+4)} - \sum_{i=m+1}^{p} \sum_{k=m+1}^{p} \sum_{\ell=1}^{m} \mathbb{E}\left[u_i v_i u_k z_k v_\ell z_\ell\right]$$

$$= -\frac{(p-m)(mp+2m-4)}{(p-2)(p-1)p(p+2)(p+4)} - m \sum_{i=m+1}^{p} \sum_{k=m+1}^{p} \mathbb{E}\left[v_1 z_1 u_i v_i u_k z_k\right]$$

$$= -\frac{(p-m)(mp+2m-4)}{(p-2)(p-1)p(p+2)(p+4)} - (p-m)m \sum_{k=m+1}^{p} \mathbb{E}\left[v_1 z_1 u_p v_p u_k z_k\right]$$

$$= -(p-m)\left(\frac{mp+2m-4}{(p-2)(p-1)p(p+2)(p+4)}\right.$$

$$\left. + m\left(\underbrace{\mathbb{E}\left[u_p^2 v_1 v_p z_1 z_p\right]}_{k=p} + \underbrace{(p-m-1)\mathbb{E}\left[u_{p-1} u_p v_1 v_{p-1} z_1 z_p\right]}_{m+1 \le k \le p-1}\right)\right)$$

$$= -(p-m)\left(\frac{mp+2m-4}{(p-2)(p-1)p(p+2)(p+4)} + m\left(\left\langle \begin{smallmatrix} 0 & 1 & 1 \\ 2 & 1 & 1 \\ 0 & 0 & 0 \end{smallmatrix} \right\rangle + (p-m-1)\left\langle \begin{smallmatrix} 0 & 1 & 1 \\ 1 & 1 & 0 \\ 1 & 0 & 1 \end{smallmatrix} \right\rangle\right)\right)$$

$$= -(p-m)\left(\frac{mp+2m-4}{(p-2)(p-1)p(p+2)(p+4)} + m\left(\frac{-1}{(p-1)p(p+2)(p+4)} + \frac{2(p-m-1)}{(p-2)(p-1)p(p+2)(p+4)}\right)\right)$$

$$= \frac{-2(p-m)\left(-m^2 + mp + m - 2\right)}{(p-2)(p-1)p(p+2)(p+4)}.$$

Overall, we get that

$$\mathbb{E}\left[\mathbf{e}_2^\top \mathbf{O}\mathbf{e}_1 \mathbf{e}_3^\top \mathbf{O}\mathbf{e}_1 \mathbf{e}_3^\top \mathbf{O}\mathbf{e}_2\right] = \frac{(p-m)(-mp-p+4)}{(p-2)(p-1)p(p+2)(p+4)} + \frac{-2(p-m)\left(-m^2+mp+m-2\right)}{(p-2)(p-1)p(p+2)(p+4)}$$

$$= \frac{(p-m)\left(2m^2 - 3mp - 2m - p + 8\right)}{(p-2)(p-1)p(p+2)(p+4)}.$$

$\square$

**Proposition 22.** *For $p \geq 2, m \in \{2, \ldots, p\}$ and a random transformation $\mathbf{O}$ sampled as described in Eq. (1), it holds that,*

$$\mathbb{E}\left(\mathbf{e}_1^\top \mathbf{O} \mathbf{e}_1 \cdot \mathbf{e}_1^\top \mathbf{O} \mathbf{e}_2\right)\left(\mathbf{e}_2^\top \mathbf{O} \mathbf{e}_1 \cdot \mathbf{e}_2^\top \mathbf{O} \mathbf{e}_2\right)$$

$$= \tfrac{-m^4(p+3)+m^3\left(3p^2+8p-6\right)+m^2\left(-3p^3-6p^2+13p+3\right)+m\left(p^4-7p^2+4p+6\right)+p^4-p^3-18p^2-40p-24}{(p-1)p(p+1)(p+2)(p+4)(p+6)} \, .$$

*Proof.* We start by decomposing the quantity as follows,

$$\mathbb{E}\left(\mathbf{e}_1^\top \mathbf{O} \mathbf{e}_1 \cdot \mathbf{e}_1^\top \mathbf{O} \mathbf{e}_2\right)\left(\mathbf{e}_2^\top \mathbf{O} \mathbf{e}_1 \cdot \mathbf{e}_2^\top \mathbf{O} \mathbf{e}_2\right)$$

$$= \mathbb{E}\left[\left(\mathbf{u}_a^\top \mathbf{Q}_m \mathbf{u}_a + \mathbf{u}_b^\top \mathbf{u}_b\right)\left(\mathbf{u}_a^\top \mathbf{Q}_m \mathbf{v}_a + \mathbf{u}_b^\top \mathbf{v}_b\right)\left(\mathbf{v}_a^\top \mathbf{Q}_m \mathbf{u}_a + \mathbf{v}_b^\top \mathbf{u}_b\right)\left(\mathbf{v}_a^\top \mathbf{Q}_m \mathbf{v}_a + \mathbf{v}_b^\top \mathbf{v}_b\right)\right]$$

$$= \mathbb{E}\left[\mathbf{u}_a^\top \mathbf{Q}_m \mathbf{u}_a \left(\mathbf{u}_a^\top \mathbf{Q}_m \mathbf{v}_a + \mathbf{u}_b^\top \mathbf{v}_b\right)\left(\mathbf{v}_a^\top \mathbf{Q}_m \mathbf{u}_a + \mathbf{v}_b^\top \mathbf{u}_b\right)\left(\mathbf{v}_a^\top \mathbf{Q}_m \mathbf{v}_a + \mathbf{v}_b^\top \mathbf{v}_b\right)\right] +$$
$$\mathbb{E}\left[\mathbf{u}_b^\top \mathbf{u}_b \left(\mathbf{u}_a^\top \mathbf{Q}_m \mathbf{v}_a + \mathbf{u}_b^\top \mathbf{v}_b\right)\left(\mathbf{v}_a^\top \mathbf{Q}_m \mathbf{u}_a + \mathbf{v}_b^\top \mathbf{u}_b\right)\left(\mathbf{v}_a^\top \mathbf{Q}_m \mathbf{v}_a + \mathbf{v}_b^\top \mathbf{v}_b\right)\right]$$

We show that the first term is,

$$\mathbb{E}\left[\mathbf{u}_a^\top \mathbf{Q}_m \mathbf{u}_a \left(\mathbf{u}_a^\top \mathbf{Q}_m \mathbf{v}_a + \mathbf{u}_b^\top \mathbf{v}_b\right)\left(\mathbf{v}_a^\top \mathbf{Q}_m \mathbf{u}_a + \mathbf{v}_b^\top \mathbf{u}_b\right)\left(\mathbf{v}_a^\top \mathbf{Q}_m \mathbf{v}_a + \mathbf{v}_b^\top \mathbf{v}_b\right)\right]$$

$$= \mathbb{E}\left[\mathbf{u}_a^\top \mathbf{Q}_m \mathbf{u}_a \mathbf{u}_a^\top \mathbf{Q}_m \mathbf{v}_a \left(\mathbf{v}_a^\top \mathbf{Q}_m \mathbf{u}_a + \mathbf{v}_b^\top \mathbf{u}_b\right)\left(\mathbf{v}_a^\top \mathbf{Q}_m \mathbf{v}_a + \mathbf{v}_b^\top \mathbf{v}_b\right)\right] +$$
$$\mathbb{E}\left[\mathbf{u}_a^\top \mathbf{Q}_m \mathbf{u}_a \mathbf{u}_b^\top \mathbf{v}_b \left(\mathbf{v}_a^\top \mathbf{Q}_m \mathbf{u}_a + \mathbf{v}_b^\top \mathbf{u}_b\right)\left(\mathbf{v}_a^\top \mathbf{Q}_m \mathbf{v}_a + \mathbf{v}_b^\top \mathbf{v}_b\right)\right]$$

$$[\text{Cor. 12}] = \mathbb{E}\left[\mathbf{u}_a^\top \mathbf{Q}_m \mathbf{u}_a \mathbf{u}_a^\top \mathbf{Q}_m \mathbf{v}_a \mathbf{v}_a^\top \mathbf{Q}_m \mathbf{u}_a \mathbf{v}_a^\top \mathbf{Q}_m \mathbf{v}_a\right] + \mathbb{E}\left[\mathbf{u}_a^\top \mathbf{Q}_m \mathbf{u}_a \mathbf{u}_b^\top \mathbf{v}_b \mathbf{v}_b^\top \mathbf{u}_b \mathbf{v}_a^\top \mathbf{Q}_m \mathbf{v}_a\right] +$$
$$\underbrace{\mathbb{E}\left[\mathbf{u}_a^\top \mathbf{Q}_m \mathbf{u}_a \mathbf{u}_a^\top \mathbf{Q}_m \mathbf{v}_a \mathbf{v}_b^\top \mathbf{u}_b \mathbf{v}_b^\top \mathbf{v}_b\right] + \mathbb{E}\left[\mathbf{u}_a^\top \mathbf{Q}_m \mathbf{u}_a \mathbf{u}_b^\top \mathbf{v}_b \mathbf{v}_a^\top \mathbf{Q}_m \mathbf{u}_a \mathbf{v}_b^\top \mathbf{v}_b\right]}_{\text{equivalent}}$$

$$= \mathbb{E}\left[\mathbf{u}_a^\top \mathbf{Q}_m \mathbf{u}_a \mathbf{u}_a^\top \mathbf{Q}_m \mathbf{v}_a \mathbf{v}_a^\top \mathbf{Q}_m \mathbf{u}_a \mathbf{v}_a^\top \mathbf{Q}_m \mathbf{v}_a\right] + 2\mathbb{E}\left[\mathbf{u}_a^\top \mathbf{Q}_m \mathbf{u}_a \mathbf{u}_a^\top \mathbf{Q}_m \mathbf{v}_a \mathbf{v}_b^\top \mathbf{u}_b \mathbf{v}_b^\top \mathbf{v}_b\right] +$$
$$\mathbb{E}\left[\mathbf{u}_a^\top \mathbf{Q}_m \mathbf{u}_a \mathbf{v}_a^\top \mathbf{Q}_m \mathbf{v}_a \left(\mathbf{u}_b^\top \mathbf{v}_b\right)^2\right]$$

$$= \mathbb{E}\left[\mathbf{u}_a^\top \mathbf{Q}_m \mathbf{u}_a \mathbf{u}_a^\top \mathbf{Q}_m \mathbf{v}_a \mathbf{v}_a^\top \mathbf{Q}_m \mathbf{u}_a \mathbf{v}_a^\top \mathbf{Q}_m \mathbf{v}_a\right] +$$
$$2\mathbb{E}\left[\|\mathbf{u}_a\|^2 \, \mathbb{E}_{\mathbf{r} \sim \mathcal{S}^{m-1}}\left(\mathbf{r}^\top \mathbf{u}_a \mathbf{r}^\top \mathbf{v}_a\right) \mathbf{v}_b^\top \mathbf{u}_b \mathbf{v}_b^\top \mathbf{v}_b\right] + \mathbb{E}\left[\mathbf{u}_a^\top \mathbf{Q}_m \mathbf{u}_a \mathbf{v}_a^\top \mathbf{Q}_m \mathbf{v}_a \left(\mathbf{u}_b^\top \mathbf{v}_b\right)^2\right]$$

$$= \underbrace{\mathbb{E}\left[\mathbf{u}_a^\top \mathbf{Q}_m \mathbf{u}_a \mathbf{u}_a^\top \mathbf{Q}_m \mathbf{v}_a \mathbf{v}_a^\top \mathbf{Q}_m \mathbf{u}_a \mathbf{v}_a^\top \mathbf{Q}_m \mathbf{v}_a\right]}_{\text{solved in Prop. 23}} + \underbrace{\tfrac{2}{m}\, \mathbb{E}\left[\|\mathbf{u}_a\|^2 \, \mathbf{u}_a^\top \mathbf{v}_a \cdot \mathbf{v}_b^\top \mathbf{u}_b \, \|\mathbf{v}_b\|^2\right]}_{\text{solved in Eq. (29)}} +$$

$$\underbrace{\mathbb{E}\left[\mathbf{u}_a^\top \mathbf{Q}_m \mathbf{u}_a \mathbf{v}_a^\top \mathbf{Q}_m \mathbf{v}_a \left(\mathbf{v}_b^\top \mathbf{u}_b\right)^2\right]}_{\text{solved in Eq. (30)}}$$

$$= \tfrac{-\left(m^2(2p+3)+m\left(-p^2+14p+30\right)-6p^2+4p+24\right)}{(p-1)p(p+1)(p+2)(p+4)(p+6)} + \tfrac{2m(p-m)(p+3)\left(m^2-mp-2(p+2)\right)}{m(p-1)p(p+1)(p+2)(p+4)(p+6)} +$$
$$\tfrac{3(p-m)\left(-m^2+mp+2p+4\right)}{(p-1)p(p+1)(p+2)(p+4)(p+6)}$$

$$= \tfrac{-m^3(2p+3)+m^2\left(4p^2+4p-3\right)-2m\left(p^3-p^2+9\right)-4\left(p^3+2p^2+4p+6\right)}{(p-1)p(p+1)(p+2)(p+4)(p+6)} \, .$$

Moreover, we show that the second term is,

$$\mathbb{E}\left[\mathbf{u}_b^\top \mathbf{u}_b \left(\mathbf{u}_a^\top \mathbf{Q}_m \mathbf{v}_a + \mathbf{u}_b^\top \mathbf{v}_b\right)\left(\mathbf{v}_a^\top \mathbf{Q}_m \mathbf{u}_a + \mathbf{v}_b^\top \mathbf{u}_b\right)\left(\mathbf{v}_a^\top \mathbf{Q}_m \mathbf{v}_a + \mathbf{v}_b^\top \mathbf{v}_b\right)\right]$$

$$[\text{Cor. 12}] = \mathbb{E}\left[\mathbf{u}_b^\top \mathbf{u}_b \mathbf{u}_b^\top \mathbf{v}_b \mathbf{v}_b^\top \mathbf{u}_b \mathbf{v}_b^\top \mathbf{v}_b\right] + \mathbb{E}\left[\mathbf{u}_b^\top \mathbf{u}_b \mathbf{u}_a^\top \mathbf{Q}_m \mathbf{v}_a \mathbf{v}_a^\top \mathbf{Q}_m \mathbf{u}_a \mathbf{v}_b^\top \mathbf{v}_b\right] +$$
$$\mathbb{E}\left[\mathbf{u}_b^\top \mathbf{u}_b \mathbf{u}_a^\top \mathbf{Q}_m \mathbf{v}_a \mathbf{v}_b^\top \mathbf{u}_b \mathbf{v}_a^\top \mathbf{Q}_m \mathbf{v}_a\right] + \mathbb{E}\left[\mathbf{u}_b^\top \mathbf{u}_b \mathbf{u}_b^\top \mathbf{v}_b \mathbf{v}_a^\top \mathbf{Q}_m \mathbf{u}_a \mathbf{v}_a^\top \mathbf{Q}_m \mathbf{v}_a\right]$$

$$= \mathbb{E}\left[\mathbf{u}_b^\top \mathbf{u}_b \left(\mathbf{v}_a^\top \mathbf{u}_a\right)^2 \mathbf{v}_b^\top \mathbf{v}_b\right] + \mathbb{E}\left[\mathbf{u}_b^\top \mathbf{u}_b \mathbf{u}_a^\top \mathbf{Q}_m \mathbf{v}_a \mathbf{v}_a^\top \mathbf{Q}_m \mathbf{u}_a \mathbf{v}_b^\top \mathbf{v}_b\right] +$$
$$\underbrace{\mathbb{E}\left[\mathbf{u}_b^\top \mathbf{u}_b \mathbf{u}_b^\top \mathbf{v}_b \cdot \mathbf{u}_a^\top \mathbf{Q}_m \mathbf{v}_a \mathbf{v}_a^\top \mathbf{Q}_m \mathbf{v}_a\right] + \mathbb{E}\left[\mathbf{u}_b^\top \mathbf{u}_b \mathbf{u}_b^\top \mathbf{v}_b \cdot \mathbf{v}_a^\top \mathbf{Q}_m \mathbf{u}_a \mathbf{v}_a^\top \mathbf{Q}_m \mathbf{v}_a\right]}_{\text{equal due to the invariance of } \mathbf{Q}_m \text{ w.r.t. transpose (Prop. 9)}}$$

$$= \underbrace{\mathbb{E}\left[\mathbf{u}_b^\top \mathbf{u}_b \left(\mathbf{v}_a^\top \mathbf{u}_a\right)^2 \mathbf{v}_b^\top \mathbf{v}_b\right]}_{\text{solved in Eq. (31)}} + \underbrace{\mathbb{E}\left[\mathbf{u}_b^\top \mathbf{u}_b \mathbf{u}_a^\top \mathbf{Q}_m \mathbf{v}_a \mathbf{v}_a^\top \mathbf{Q}_m \mathbf{u}_a \mathbf{v}_b^\top \mathbf{v}_b\right]}_{\text{solved in Eq. (32)}} +$$

$$\underbrace{2\mathbb{E}\left[\mathbf{u}_b^\top \mathbf{u}_b \mathbf{u}_b^\top \mathbf{v}_b \cdot \mathbf{u}_a^\top \mathbf{Q}_m \mathbf{v}_a \mathbf{v}_a^\top \mathbf{Q}_m \mathbf{v}_a\right]}_{\text{solved in Eq. (33)}}$$

$$= \frac{(p-m)m\left(m^2(p+3)-2m\left(p^2+5p+3\right)+p\left(p^2+7p+10\right)\right)}{(p-1)p(p+1)(p+2)(p+4)(p+6)}+$$
$$\frac{(p-m)\left(m^2p+3m^2-2mp^2-10mp-6m+p^3+7p^2+10p\right)}{(p-1)p(p+1)(p+2)(p+4)(p+6)} + \frac{-2(p-m)(p+3)(3(p+1)+(m-1)(p-m-1))}{(p-1)p(p+1)(p+2)(p+4)(p+6)}$$

$$= \frac{-(p-m)(p-m+2)\left(m^2p+3m^2-mp^2-2mp+9m-p^2-p+12\right)}{(p-1)p(p+1)(p+2)(p+4)(p+6)} .$$

Overall, we conclude that

$$\mathbb{E}\left(\mathbf{e}_1^\top \mathbf{O}\mathbf{e}_1 \cdot \mathbf{e}_1^\top \mathbf{O}\mathbf{e}_2\right)\left(\mathbf{e}_2^\top \mathbf{O}\mathbf{e}_1 \cdot \mathbf{e}_2^\top \mathbf{O}\mathbf{e}_2\right)$$
$$= \frac{-m^3(2p+3)+m^2\left(4p^2+4p-3\right)-2m\left(p^3-p^2+9\right)-4\left(p^3+2p^2+4p+6\right)}{(p-1)p(p+1)(p+2)(p+4)(p+6)}+$$
$$\frac{-(p-m)(p-m+2)\left(m^2p+3m^2-mp^2-2mp+9m-p^2-p+12\right)}{(p-1)p(p+1)(p+2)(p+4)(p+6)}$$
$$= \frac{-m^4(p+3)+m^3\left(3p^2+8p-6\right)+m^2\left(-3p^3-6p^2+13p+3\right)+m\left(p^4-7p^2+4p+6\right)+p^4-p^3-18p^2-40p-24}{(p-1)p(p+1)(p+2)(p+4)(p+6)} .$$

$\square$

**Proposition 23.** *For $p \geq 2, m \in \{2, \ldots, p\}$ and a random transformation $\mathbf{O}$ sampled as described in Eq. (1), it holds that,*

$$\mathbb{E}\left[\mathbf{u}_a^\top \mathbf{Q}_m \mathbf{u}_a \mathbf{u}_a^\top \mathbf{Q}_m \mathbf{v}_a \mathbf{v}_a^\top \mathbf{Q}_m \mathbf{u}_a \mathbf{v}_a^\top \mathbf{Q}_m \mathbf{v}_a\right] = \frac{-\left(m^2(2p+3) + m\left(-p^2 + 14p + 30\right) - 6p^2 + 4p + 24\right)}{(p-1)p(p+1)(p+2)(p+4)(p+6)}.$$

*Proof.*

$$\mathbb{E}\left[\mathbf{u}_a^\top \mathbf{Q}_m \mathbf{u}_a \mathbf{u}_a^\top \mathbf{Q}_m \mathbf{v}_a \mathbf{v}_a^\top \mathbf{Q}_m \mathbf{u}_a \mathbf{v}_a^\top \mathbf{Q}_m \mathbf{v}_a\right]$$

$$= \sum_{i,j=1}^{m} \sum_{k,\ell=1}^{m} \sum_{n,r=1}^{m} \sum_{s,t=1}^{m} \mathbb{E}\left[u_i u_j u_k v_\ell v_n u_r v_s v_t\right] \mathbb{E}\left[q_{ij} q_{k\ell} q_{nr} q_{st}\right]$$

$$= \underbrace{\sum_{i,n=1}^{m} \sum_{j,\ell,r,t=1}^{m} \mathbb{E}\left[u_i u_j u_i v_\ell v_n u_r v_n v_t\right] \mathbb{E}\left[q_{ij} q_{i\ell} q_{nr} q_{nt}\right] +}_{i=k \Longrightarrow n=s}$$

$$\underbrace{\sum_{i \neq k=1}^{m} \sum_{j,\ell,r,t=1}^{m} \mathbb{E}\left[u_i u_j u_k v_\ell v_i u_r v_k v_t\right] \left(\mathbb{E}\left[q_{ij} q_{k\ell} q_{ir} q_{kt}\right] + \mathbb{E}\left[q_{ij} q_{k\ell} q_{kr} q_{it}\right]\right)}_{i \neq k \Longrightarrow (n=i \neq s=k) \vee (n=k \neq s=i) \text{ due to symmetry w.r.t. } n, s}$$

$$= \sum_{i,n=1}^{m} \sum_{j,\ell,r,t=1}^{m} \mathbb{E}\left[u_i u_j u_i v_\ell u_r v_n^2 v_t\right] \mathbb{E}\left[q_{ij} q_{i\ell} q_{nr} q_{nt}\right] +$$

$$\sum_{i \neq k=1}^{m} \sum_{j,\ell,r,t=1}^{m} \mathbb{E}\left[u_i u_j u_k v_\ell v_i u_r v_k v_t\right] \left(\mathbb{E}\left[q_{ij} q_{k\ell} q_{ir} q_{kt}\right] + \mathbb{E}\left[q_{ij} q_{k\ell} q_{kr} q_{it}\right]\right)$$

$$= m \underbrace{\sum_{n=1}^{m} \sum_{j,\ell,r,t=1}^{m} \mathbb{E}\left[u_1^2 u_j u_r v_\ell v_t v_n^2\right] \mathbb{E}\left[q_{1,j} q_{1,\ell} q_{nr} q_{nt}\right] +}_{\triangleq (A) \text{ below}}$$

$$m \underbrace{\sum_{k=2}^{m} \sum_{j,\ell,r,t=1}^{m} \mathbb{E}\left[u_1 v_1 u_r u_j u_k v_k v_\ell v_t\right] \left(\mathbb{E}\left[q_{1,j} q_{k\ell} q_{1,r} q_{kt}\right] + \mathbb{E}\left[q_{1,j} q_{k\ell} q_{kr} q_{1,t}\right]\right)}_{\triangleq (B) \text{ below}}$$

$$= \frac{-m^3(p+3) + m^2\left(p^2 - 5p - 12\right) + m\left(6p^2 - 26p - 60\right) + 16p^2 - 8p - 48}{(m+2)(p-1)p(p+1)(p+2)(p+4)(p+6)} +$$

$$\frac{-m^3 p - 13m^2 p - 24m^2 + 2mp^2 - 6mp - 24m - 4p^2}{(m+2)(p-1)p(p+1)(p+2)(p+4)(p+6)}$$

$$= \frac{-\left(m^2\left(2p+3\right) + m\left(-p^2 + 14p + 30\right) - 6p^2 + 4p + 24\right)}{(p-1)p(p+1)(p+2)(p+4)(p+6)}$$

**Deriving (A).**

$$(A) = \sum_{n=1}^{m} \sum_{j,\ell,r,t=1}^{m} \mathbb{E}\left[u_1^2 u_j u_r v_\ell v_t v_n^2\right] \mathbb{E}\left[q_{1,j} q_{1,\ell} q_{nr} q_{nt}\right]$$

$$= \underbrace{\sum_{j,\ell,r,t=1}^{m} \mathbb{E}\left[u_1^2 u_j u_r v_\ell v_t v_1^2\right] \mathbb{E}\left[q_{1,j} q_{1,\ell} q_{1,r} q_{1,t}\right]}_{n=1,\text{ solved below}} +$$

$$(m-1) \underbrace{\sum_{j,\ell,r,t=1}^{m} \mathbb{E}\left[u_1^2 u_j u_r v_\ell v_t v_2^2\right] \mathbb{E}\left[q_{1,j} q_{1,\ell} q_{2,r} q_{2,t}\right]}_{n \geq 2,\text{ solved below}}$$

$$= \frac{m^2\left(p^2+4p+15\right)+6m(p-3)(p+1)+4\left(5p^2+2p-6\right)}{m(m+2)(p-1)p(p+1)(p+2)(p+4)(p+6)} +$$

$$\frac{-m^3 p-3m^3-9m^2 p-27m^2-14mp-42m-4p^2-16p-24}{m(m+2)(p-1)p(p+1)(p+2)(p+4)(p+6)}$$

$$= \frac{-m^3(p+3)+m^2\left(p^2-5p-12\right)+m\left(6p^2-26p-60\right)+16p^2-8p-48}{m(m+2)(p-1)p(p+1)(p+2)(p+4)(p+6)}$$

When $n=1$: The only nonzero options are $\left\langle \begin{smallmatrix} 4 \\ \vec{0} \end{smallmatrix} \begin{smallmatrix} 0 \\ \vec{0} \end{smallmatrix} \right\rangle$ and $\left\langle \begin{smallmatrix} 2 \\ \vec{0} \end{smallmatrix} \begin{smallmatrix} 0 \\ \vec{0} \end{smallmatrix} \right\rangle$. We have,

$$\sum_{j,\ell,r,t=1}^{m} \mathbb{E}\left[u_1^2 u_j u_r v_\ell v_t v_1^2\right] \mathbb{E}\left[q_{1,j} q_{1,\ell} q_{1,r} q_{1,t}\right]$$

$$= \left\langle \begin{smallmatrix} 4 \\ \vec{0} \end{smallmatrix} \begin{smallmatrix} 0 \\ \vec{0} \end{smallmatrix} \right\rangle_m \sum_{j=1}^{m} \mathbb{E}\left[u_1^2 u_j^2 v_j^2 v_1^2\right] +$$

$$\left\langle \begin{smallmatrix} 2 \\ \vec{0} \end{smallmatrix} \begin{smallmatrix} 0 \\ \vec{0} \end{smallmatrix} \right\rangle_m \left( \sum_{j\neq\ell=1}^{m} \underbrace{\mathbb{E}\left[u_1^2 u_j^2 v_\ell^2 v_1^2\right]}_{j=r\neq\ell=t} + \sum_{j\neq r=1}^{m} \underbrace{\mathbb{E}\left[u_1^2 u_j u_r v_j v_r v_1^2\right]}_{j=\ell\neq r=t} + \sum_{j\neq\ell=1}^{m} \underbrace{\mathbb{E}\left[u_1^2 u_j u_\ell v_j v_\ell v_1^2\right]}_{j=t\neq\ell=r} \right)$$

$$= \left\langle \begin{smallmatrix} 4 \\ \vec{0} \end{smallmatrix} \begin{smallmatrix} 0 \\ \vec{0} \end{smallmatrix} \right\rangle_m \sum_{j=1}^{m} \mathbb{E}\left[u_1^2 u_j^2 v_1^2 v_j^2\right] + \left\langle \begin{smallmatrix} 2 \\ \vec{0} \end{smallmatrix} \begin{smallmatrix} 0 \\ \vec{0} \end{smallmatrix} \right\rangle_m \sum_{j\neq\ell=1}^{m} \left( \mathbb{E}\left[u_1^2 u_j^2 v_1^2 v_\ell^2\right] + 2\mathbb{E}\left[u_1^2 u_j u_\ell v_1^2 v_j v_\ell\right] \right)$$

$$= \left\langle \begin{smallmatrix} 4 \\ \vec{0} \end{smallmatrix} \begin{smallmatrix} 0 \\ \vec{0} \end{smallmatrix} \right\rangle_m \left( \mathbb{E}\left[u_1^4 v_1^4\right] + (m-1)\mathbb{E}\left[u_1^2 u_2^2 v_1^2 v_2^2\right] \right) +$$

$$\left\langle \begin{smallmatrix} 2 \\ \vec{0} \end{smallmatrix} \begin{smallmatrix} 0 \\ \vec{0} \end{smallmatrix} \right\rangle_m \left( \sum_{j\neq\ell=1}^{m} \mathbb{E}\left[u_1^2 u_j^2 v_1^2 v_\ell^2\right] + 2\sum_{j\neq\ell=1}^{m} \mathbb{E}\left[u_1^2 u_j u_\ell v_1^2 v_j v_\ell\right] \right)$$

$$= \left\langle \begin{smallmatrix} 4 \\ \vec{0} \end{smallmatrix} \begin{smallmatrix} 0 \\ \vec{0} \end{smallmatrix} \right\rangle_m \left( \left\langle \begin{smallmatrix} 4 \\ \vec{0} \end{smallmatrix} \begin{smallmatrix} 4 \\ \vec{0} \end{smallmatrix} \right\rangle_p + (m-1)\left\langle \begin{smallmatrix} 2 \\ \vec{0} \end{smallmatrix} \begin{smallmatrix} 2 \\ \vec{0} \end{smallmatrix} \right\rangle_p \right) + \left\langle \begin{smallmatrix} 2 \\ \vec{0} \end{smallmatrix} \begin{smallmatrix} 0 \\ \vec{0} \end{smallmatrix} \right\rangle_m \left( 2\sum_{j\neq\ell=1}^{m} \mathbb{E}\left[u_1^2 u_j u_\ell v_j v_\ell v_1^2\right] \right)$$

$$\left\langle \begin{smallmatrix} 2 \\ \vec{0} \end{smallmatrix} \begin{smallmatrix} 0 \\ \vec{0} \end{smallmatrix} \right\rangle_m \left( \underbrace{(m-1)\mathbb{E}\left[u_1^4 v_2^2 v_1^2\right]}_{j=1,\ell\geq 2} + \underbrace{(m-1)\mathbb{E}\left[u_1^2 u_2^2 v_1^4\right]}_{j\geq 2,\ell=1} + \underbrace{(m-1)(m-2)\mathbb{E}\left[u_1^2 u_2^2 v_1^2 v_3^2\right]}_{j\neq\ell\geq 2} \right)$$

$$= \left\langle \begin{smallmatrix} 4 \\ \vec{0} \end{smallmatrix} \begin{smallmatrix} 0 \\ \vec{0} \end{smallmatrix} \right\rangle_m \left( \left\langle \begin{smallmatrix} 4 \\ \vec{0} \end{smallmatrix} \begin{smallmatrix} 4 \\ \vec{0} \end{smallmatrix} \right\rangle_p + (m-1)\left\langle \begin{smallmatrix} 2 \\ \vec{0} \end{smallmatrix} \begin{smallmatrix} 2 \\ \vec{0} \end{smallmatrix} \right\rangle_p \right) +$$

$$\left\langle \begin{smallmatrix} 2 \\ \vec{0} \end{smallmatrix} \begin{smallmatrix} 0 \\ \vec{0} \end{smallmatrix} \right\rangle_m \left( 2(m-1)\left\langle \begin{smallmatrix} 4 \\ \vec{0} \end{smallmatrix} \begin{smallmatrix} 2 \\ \vec{0} \end{smallmatrix} \right\rangle_p + (m-1)(m-2)\left\langle \begin{smallmatrix} 2 & 2 \\ 0 & \\ \vec{0} & \vec{0} \end{smallmatrix} \right\rangle_p \right) +$$

$$\left\langle \begin{smallmatrix} 2 \\ \vec{0} \end{smallmatrix} \begin{smallmatrix} 0 \\ \vec{0} \end{smallmatrix} \right\rangle_m \left( 2(m-1)\left( \underbrace{2\mathbb{E}\left[u_1^3 u_2 v_2 v_1^3\right]}_{j=1,\ell\geq 2 \vee j\geq 2,\ell=1} + \underbrace{(m-2)\mathbb{E}\left[u_1^2 u_2 u_3 v_2 v_3 v_1^2\right]}_{j\neq\ell\geq 2} \right) \right)$$

$$= \left\langle \begin{smallmatrix} 4 & 0 \\ 0 & 0 \end{smallmatrix} \right\rangle_m \left( \left\langle \begin{smallmatrix} 4 & 4 \\ 0 & 0 \end{smallmatrix} \right\rangle_p + (m-1) \left\langle \begin{smallmatrix} 2 & 2 \\ 2 & 2 \\ 0 & 0 \end{smallmatrix} \right\rangle_p \right) +$$

$$(m-1) \left\langle \begin{smallmatrix} 2 & 0 \\ 2 & 0 \\ 0 & 0 \end{smallmatrix} \right\rangle_m \left( 2 \left\langle \begin{smallmatrix} 4 & 2 \\ 0 & 2 \\ 0 & 0 \end{smallmatrix} \right\rangle_p + (m-2) \left\langle \begin{smallmatrix} 2 & 2 \\ 2 & 0 \\ 0 & 2 \\ 0 & 0 \end{smallmatrix} \right\rangle_p + 4 \left\langle \begin{smallmatrix} 3 & 3 \\ 1 & 1 \\ 0 & 0 \end{smallmatrix} \right\rangle_p + 2 (m-2) \left\langle \begin{smallmatrix} 2 & 2 \\ 1 & 1 \\ 1 & 1 \\ 0 & 0 \end{smallmatrix} \right\rangle_p \right)$$

$$= \frac{3}{m (m+2)} \left( \left\langle \begin{smallmatrix} 4 & 4 \\ 0 & 0 \end{smallmatrix} \right\rangle_p + (m-1) \left\langle \begin{smallmatrix} 2 & 2 \\ 2 & 2 \\ 0 & 0 \end{smallmatrix} \right\rangle_p \right) +$$

$$\frac{(m-1)}{m (m+2)} \left( 2 \left\langle \begin{smallmatrix} 4 & 2 \\ 0 & 2 \\ 0 & 0 \end{smallmatrix} \right\rangle_p + 4 \left\langle \begin{smallmatrix} 3 & 3 \\ 1 & 1 \\ 0 & 0 \end{smallmatrix} \right\rangle_p + (m-2) \left( 2 \left\langle \begin{smallmatrix} 2 & 2 \\ 1 & 1 \\ 1 & 1 \\ 0 & 0 \end{smallmatrix} \right\rangle_p + \left\langle \begin{smallmatrix} 2 & 2 \\ 2 & 0 \\ 0 & 2 \\ 0 & 0 \end{smallmatrix} \right\rangle_p \right) \right)$$

$$= \frac{3}{m(m+2)} \left( \frac{9(p-1)(p+1) + (m-1)\left(p^2 + 4p + 15\right)}{(p-1)p(p+1)(p+2)(p+4)(p+6)} \right) +$$

$$\frac{(m-1)}{m(m+2)} \left( \frac{2(p+1) \cdot 3(p+3) - 4 \cdot 9(p+1)}{(p-1)p(p+1)(p+2)(p+4)(p+6)} + \frac{(m-2)\left(2(-p+3) + (p+3)^2\right)}{(p-1)p(p+1)(p+2)(p+4)(p+6)} \right)$$

$$= \frac{m^2 \left(p^2 + 4p + 15\right) + 6m (p-3)(p+1) + 4 \left(5p^2 + 2p - 6\right)}{m (m+2)(p-1)p(p+1)(p+2)(p+4)(p+6)}.$$

When $n \geq 2$: The only nonzero options are $\left\langle \begin{smallmatrix} 2 & 2 \\ \overrightarrow{0} & \overrightarrow{0} \end{smallmatrix} \right\rangle, \left\langle \begin{smallmatrix} 1 & 1 \\ \overrightarrow{0} & \overrightarrow{0} \end{smallmatrix} \right\rangle, \left\langle \begin{smallmatrix} 2 & 0 \\ \overrightarrow{0} & \overrightarrow{0} \end{smallmatrix} \right\rangle$. We have,

$$\sum_{j,\ell,r,t=1}^{m} \mathbb{E}\left[u_1^2 u_j u_r v_\ell v_t v_2^2\right] \mathbb{E}\left[q_{1,j} q_{1,\ell} q_{2,r} q_{2,t}\right]$$

$$= \underbrace{\left\langle \begin{smallmatrix} 2 & 2 \\ \overrightarrow{0} & \overrightarrow{0} \end{smallmatrix} \right\rangle_m \sum_{j=1}^{m} \mathbb{E}\left[u_1^2 u_j^2 v_j^2 v_2^2\right]}_{j=\ell=r=t} + \underbrace{\left\langle \begin{smallmatrix} 1 & 1 \\ \overrightarrow{0} & \overrightarrow{0} \end{smallmatrix} \right\rangle_m \sum_{j\neq\ell=1}^{m} \left(\mathbb{E}\left[u_1^2 u_j u_\ell v_j v_\ell v_2^2\right] + \mathbb{E}\left[u_1^2 u_j^2 v_\ell^2 v_2^2\right]\right)}_{j=t\neq r=\ell \,\vee\, j=r\neq t=\ell} +$$

$$\underbrace{\left\langle \begin{smallmatrix} 2 & 0 \\ \overrightarrow{0} & \overrightarrow{0} \end{smallmatrix} \right\rangle_m \sum_{j\neq r=1}^{m} \mathbb{E}\left[u_1^2 u_j u_r v_j v_r v_2^2\right]}_{j=\ell\neq r=t}$$

$$= \left\langle \begin{smallmatrix} 2 & 2 \\ \overrightarrow{0} & \overrightarrow{0} \end{smallmatrix} \right\rangle_m \sum_{j=1}^{m} \mathbb{E}\left[u_1^2 u_j^2 v_j^2 v_2^2\right] + \left\langle \begin{smallmatrix} 1 & 1 \\ \overrightarrow{0} & \overrightarrow{0} \end{smallmatrix} \right\rangle_m \sum_{j\neq\ell=1}^{m} \mathbb{E}\left[u_1^2 u_j^2 v_\ell^2 v_2^2\right] +$$

$$\underbrace{\left(\left\langle \begin{smallmatrix} 1 & 1 \\ \overrightarrow{0} & \overrightarrow{0} \end{smallmatrix} \right\rangle_m + \left\langle \begin{smallmatrix} 2 & 0 \\ \overrightarrow{0} & \overrightarrow{0} \end{smallmatrix} \right\rangle_m\right)}_{=\frac{1}{(m-1)(m+2)}} \sum_{j\neq r=1}^{m} \mathbb{E}\left[u_1^2 u_j u_r v_j v_r v_2^2\right]$$

$$= \frac{1}{m(m+2)}\left(\underbrace{\mathbb{E}\left[u_1^4 v_1^2 v_2^2\right]}_{j=1} + \underbrace{\mathbb{E}\left[u_1^2 u_2^2 v_2^4\right]}_{j=2} + \underbrace{(m-2)\,\mathbb{E}\left[u_1^2 u_3^2 v_3^2 v_2^2\right]}_{j\geq 3}\right) +$$

$$\frac{-1}{(m-1)m(m+2)}\left(\underbrace{\mathbb{E}\left[u_1^4 v_2^4\right]}_{j=1,\ell=2} + \underbrace{\mathbb{E}\left[u_1^2 u_2^2 v_1^2 v_2^2\right]}_{j=2,\ell=1}\right) +$$

$$\frac{-1}{(m-1)m(m+2)}(m-2)\left(\underbrace{\mathbb{E}\left[u_1^4 v_2^2 v_3^2\right]}_{j=1,\ell\geq 3} + \underbrace{\mathbb{E}\left[u_1^2 u_3^2 v_1^2 v_2^2\right]}_{j\geq 3,\ell=1} + \underbrace{\mathbb{E}\left[u_1^2 u_2^2 v_2^2 v_3^2\right]}_{j=2,\ell\geq 3} + \underbrace{\mathbb{E}\left[u_1^2 u_3^2 v_2^4\right]}_{j\geq 3,\ell=2} + \right.$$

$$\left. \underbrace{(m-3)\,\mathbb{E}\left[u_1^2 u_3^2 v_2^2 v_4^2\right]}_{j\neq\ell\geq 3}\right) +$$

$$\frac{1}{(m-1)(m+2)}\left(\underbrace{2\,\mathbb{E}\left[u_1^3 u_2 v_1 v_2^3\right]}_{j=1,r=2 \,\vee\, j=2,r=1} + \underbrace{2(m-2)\,\mathbb{E}\left[u_1^3 u_3 v_1 v_2^2 v_3\right]}_{j=1,r\geq 3 \,\vee\, j\geq 3,r=1} + \underbrace{2(m-2)\,\mathbb{E}\left[u_1^2 u_2 u_3 v_2^3 v_3\right]}_{j=2,r\geq 3 \,\vee\, j\geq 3,r=2} + \right.$$

$$\left. \underbrace{(m-2)(m-3)\,\mathbb{E}\left[u_1^2 u_3 u_4 v_2^2 v_3 v_4\right]}_{j\neq r\geq 3}\right)$$

$$= \frac{1}{m(m+2)}\left(2\left\langle \begin{smallmatrix} 4 & 2 \\ \overrightarrow{0} & \overrightarrow{0} \end{smallmatrix} \right\rangle + (m-2)\left\langle \begin{smallmatrix} 2 & 0 \\ 0 & 2 \\ \overrightarrow{2} & \overrightarrow{2} \\ 0 & 0 \end{smallmatrix} \right\rangle\right) + \frac{-1}{(m-1)m(m+2)}\left(\left\langle \begin{smallmatrix} 4 & 0 \\ \overrightarrow{0} & \overrightarrow{4} \end{smallmatrix} \right\rangle + \left\langle \begin{smallmatrix} 2 & 2 \\ \overrightarrow{0} & \overrightarrow{0} \end{smallmatrix} \right\rangle\right) +$$

$$\frac{-(m-2)}{(m-1)m(m+2)}\left(\left\langle \begin{smallmatrix} 4 & 0 \\ 0 & 2 \\ \overrightarrow{0} & \overrightarrow{0} \end{smallmatrix} \right\rangle + \left\langle \begin{smallmatrix} 2 & 2 \\ 2 & 0 \\ \overrightarrow{0} & \overrightarrow{0} \end{smallmatrix} \right\rangle + \left\langle \begin{smallmatrix} 2 & 0 \\ 0 & 2 \\ \overrightarrow{0} & \overrightarrow{0} \end{smallmatrix} \right\rangle + \left\langle \begin{smallmatrix} 2 & 0 \\ 2 & 0 \\ \overrightarrow{0} & \overrightarrow{0} \end{smallmatrix} \right\rangle + (m-3)\left\langle \begin{smallmatrix} 2 & 0 \\ 0 & 2 \\ 2 & 0 \\ \overrightarrow{0} & \overrightarrow{2} \end{smallmatrix} \right\rangle\right) +$$

$$\frac{1}{(m-1)(m+2)}\left(2\left\langle \begin{smallmatrix} 3 & 1 \\ \overrightarrow{1} & \overrightarrow{3} \\ 0 & 0 \end{smallmatrix} \right\rangle + 2(m-2)\left\langle \begin{smallmatrix} 3 & 1 \\ 0 & 2 \\ \overrightarrow{1} & \overrightarrow{1} \\ 0 & 0 \end{smallmatrix} \right\rangle + 2(m-2)\left\langle \begin{smallmatrix} 2 & 0 \\ 1 & 3 \\ \overrightarrow{1} & \overrightarrow{1} \\ 0 & 0 \end{smallmatrix} \right\rangle + \right.$$

$$\left. (m-2)(m-3)\left\langle \begin{smallmatrix} 2 & 0 \\ 0 & 2 \\ 1 & 1 \\ \overrightarrow{1} & \overrightarrow{1} \\ 0 & 0 \end{smallmatrix} \right\rangle\right)$$

$$= \frac{1}{m(m+2)} \left( 2 \left\langle \begin{smallmatrix} 4 & 2 \\ 0 & 2 \\ \vec{0} & \vec{0} \end{smallmatrix} \right\rangle + (m-2) \left\langle \begin{smallmatrix} 2 & 0 \\ 0 & 2 \\ 2 & 2 \\ \vec{0} & \vec{0} \end{smallmatrix} \right\rangle \right) +$$

$$\frac{-1}{(m-1)m(m+2)} \left( \left\langle \begin{smallmatrix} 4 & 0 \\ 0 & 4 \\ \vec{0} & \vec{0} \end{smallmatrix} \right\rangle + \left\langle \begin{smallmatrix} 2 & 2 \\ 2 & 2 \\ \vec{0} & \vec{0} \end{smallmatrix} \right\rangle + \right.$$

$$\left. (m-2) \left( 2 \left\langle \begin{smallmatrix} 4 & 0 \\ 0 & 2 \\ \vec{0} & \vec{2} \\ \vec{0} & \vec{0} \end{smallmatrix} \right\rangle + 2 \left\langle \begin{smallmatrix} 2 & 2 \\ 0 & 2 \\ \vec{2} & \vec{0} \\ \vec{0} & \vec{0} \end{smallmatrix} \right\rangle + (m-3) \left\langle \begin{smallmatrix} 2 & 0 \\ 0 & 2 \\ 2 & 0 \\ 0 & 2 \\ \vec{0} & \vec{0} \end{smallmatrix} \right\rangle \right) \right) +$$

$$\frac{1}{(m-1)(m+2)} \left( 2 \left\langle \begin{smallmatrix} 3 & 1 \\ 1 & 3 \\ \vec{0} & \vec{0} \end{smallmatrix} \right\rangle + 4(m-2) \left\langle \begin{smallmatrix} 3 & 1 \\ 0 & 2 \\ 1 & 1 \\ \vec{0} & \vec{0} \end{smallmatrix} \right\rangle + (m-2)(m-3) \left\langle \begin{smallmatrix} 2 & 0 \\ 0 & 2 \\ 1 & 1 \\ 1 & 1 \\ \vec{0} & \vec{0} \end{smallmatrix} \right\rangle \right)$$

$$= \frac{1}{m(m+2)} \left( \frac{2(p+1)\cdot 3(p+3) + (m-2)(p+3)^2}{(p-1)p(p+1)(p+2)(p+4)(p+6)} \right) +$$

$$\frac{1}{(m-1)(m+2)} \left( \frac{-2\cdot 9(p+3) + 4(m-2)(-3(p+3)) + (m-2)(m-3)(-p-3)}{(p-1)p(p+1)(p+2)(p+4)(p+6)} \right) +$$

$$\frac{-1}{(m-1)m(m+2)} \left( \frac{9(p+3)(p+5) + \left( p^2 + 4p + 15 \right)}{(p-1)p(p+1)(p+2)(p+4)(p+6)} + (m-2) \frac{2\cdot 3(p+3)(p+5) + 2(p+3)^2 + (m-3)(p+3)(p+5)}{(p-1)p(p+1)(p+2)(p+4)(p+6)} \right)$$

$$= \frac{-m^3 p - 3m^3 - 9m^2 p - 27m^2 - 14mp - 42m - 4p^2 - 16p - 24}{(m-1)m(m+2)(p-1)p(p+1)(p+2)(p+4)(p+6)}$$

$\square$

**Deriving (B).**

$$
\text{(B)} = \sum_{k=2}^{m} \sum_{j,\ell,r,t=1}^{m} \mathbb{E}\left[u_1 v_1 u_r u_j u_k v_k v_\ell v_t\right] \left(\mathbb{E}\left[q_{1,j} q_{k\ell} q_{1,r} q_{kt}\right] + \mathbb{E}\left[q_{1,j} q_{k\ell} q_{kr} q_{1,t}\right]\right)
$$

$$
= (m-1) \sum_{j,\ell,r,t=1}^{m} \mathbb{E}\left[u_1 u_2 u_j u_r v_1 v_2 v_\ell v_t\right] \left(\mathbb{E}\left[q_{1,j} q_{1,r} q_{2,\ell} q_{2,t}\right] + \mathbb{E}\left[q_{1,j} q_{1,t} q_{2,\ell} q_{2,r}\right]\right)
$$

$$
= (m-1)\, 2 \left\langle \begin{smallmatrix} 2 \\ 0 \end{smallmatrix}\, \begin{smallmatrix} 2 \\ 0 \end{smallmatrix} \right\rangle_m \underbrace{\sum_{j=1}^{m} \mathbb{E}\left[u_1 u_2 u_j^2 v_1 v_2 v_j^2\right]}_{j=r=\ell=t} +
$$

$$
(m-1) \underbrace{\sum_{j\neq\ell=1}^{m} \mathbb{E}\left[u_1 u_2 u_j^2 v_1 v_2 v_\ell^2\right] \left(\left\langle \begin{smallmatrix} 2 \\ 0 \end{smallmatrix}\, \begin{smallmatrix} 0 \\ 0 \end{smallmatrix} \right\rangle_m + \left\langle \begin{smallmatrix} 1 \\ 0 \end{smallmatrix}\, \begin{smallmatrix} 1 \\ 0 \end{smallmatrix} \right\rangle_m\right)}_{j=r\neq\ell=t} +
$$

$$
(m-1)\, 2 \left\langle \begin{smallmatrix} 1 \\ 0 \end{smallmatrix}\, \begin{smallmatrix} 1 \\ 0 \end{smallmatrix} \right\rangle_m \underbrace{\sum_{j\neq r=1}^{m} \mathbb{E}\left[u_1 u_2 u_j u_r v_1 v_2 v_j v_r\right]}_{j=\ell\neq r=t} +
$$

$$
(m-1) \underbrace{\sum_{j\neq\ell=1}^{m} \mathbb{E}\left[u_1 u_2 u_j u_\ell v_1 v_2 v_j v_\ell\right] \left(\left\langle \begin{smallmatrix} 1 \\ 0 \end{smallmatrix}\, \begin{smallmatrix} 1 \\ 0 \end{smallmatrix} \right\rangle_m + \left\langle \begin{smallmatrix} 2 \\ 0 \end{smallmatrix}\, \begin{smallmatrix} 0 \\ 0 \end{smallmatrix} \right\rangle_m\right)}_{j=t\neq\ell=r}
$$

$$
= (m-1) \left( 2 \left\langle \begin{smallmatrix} 2 \\ 0 \end{smallmatrix}\, \begin{smallmatrix} 2 \\ 0 \end{smallmatrix} \right\rangle_m \sum_{j=1}^{m} \mathbb{E}\left[u_1 u_2 u_j^2 v_1 v_2 v_j^2\right] + \right.
$$

$$
\left. \left(\left\langle \begin{smallmatrix} 2 \\ 0 \end{smallmatrix}\, \begin{smallmatrix} 0 \\ 0 \end{smallmatrix} \right\rangle_m + \left\langle \begin{smallmatrix} 1 \\ 0 \end{smallmatrix}\, \begin{smallmatrix} 1 \\ 0 \end{smallmatrix} \right\rangle_m\right) \sum_{j\neq\ell=1}^{m} \mathbb{E}\left[u_1 u_2 u_j^2 v_1 v_2 v_\ell^2\right] \right) +
$$

$$
(m-1) \left( 3 \left\langle \begin{smallmatrix} 1 \\ 0 \end{smallmatrix}\, \begin{smallmatrix} 1 \\ 0 \end{smallmatrix} \right\rangle_m + \left\langle \begin{smallmatrix} 2 \\ 0 \end{smallmatrix}\, \begin{smallmatrix} 0 \\ 0 \end{smallmatrix} \right\rangle_m\right) \sum_{j\neq r=1}^{m} \mathbb{E}\left[u_1 u_2 u_j u_r v_1 v_2 v_j v_r\right]
$$

$$
= (m-1) \left( \frac{2}{m(m+2)} \sum_{j=1}^{m} \mathbb{E}\left[u_1 u_2 u_j^2 v_1 v_2 v_j^2\right] + \frac{(m+1)-1}{(m-1)m(m+2)} \sum_{j\neq\ell=1}^{m} \mathbb{E}\left[u_1 u_2 u_j^2 v_1 v_2 v_\ell^2\right] \right) +
$$

$$
(m-1) \frac{-3+(m+1)}{(m-1)m(m+2)} \sum_{j\neq r=1}^{m} \mathbb{E}\left[u_1 u_2 u_j u_r v_1 v_2 v_j v_r\right]
$$

$$
= \frac{(m-1)}{(m-1)m(m+2)} \left( 2(m-1) \sum_{j=1}^{m} \mathbb{E}\left[u_1 u_2 u_j^2 v_1 v_2 v_j^2\right] + m \sum_{j\neq\ell=1}^{m} \mathbb{E}\left[u_1 u_2 u_j^2 v_1 v_2 v_\ell^2\right] \right.
$$

$$
\left. + (m-2) \sum_{j\neq r=1}^{m} \mathbb{E}\left[u_1 u_2 u_j u_r v_1 v_2 v_j v_r\right] \right)
$$

$$
= \frac{1}{m(m+2)} \left( 2(m-1) \sum_{j=1}^{m} \mathbb{E}\left[u_1 u_2 u_j^2 v_1 v_2 v_j^2\right] + m \sum_{j\neq\ell=1}^{m} \mathbb{E}\left[u_1 u_2 u_j^2 v_1 v_2 v_\ell^2\right] + \right.
$$

$$
\left. (m-2) \sum_{j\neq r=1}^{m} \mathbb{E}\left[u_1 u_2 u_j u_r v_1 v_2 v_j v_r\right] \right)
$$

$$
= \frac{1}{m(m+2)} \left( 2(m-1) \left( 2 \underbrace{\mathbb{E}\left[u_1^3 u_2 v_1^3 v_2\right]}_{j=1,2} + \underbrace{(m-2)\,\mathbb{E}\left[u_1 u_2 u_3^2 v_1 v_2 v_3^2\right]}_{j\geq 3} \right) + \right.
$$

$$m \sum_{j \neq \ell = 1}^{m} \mathbb{E}\left[u_1 u_2 u_j^2 v_1 v_2 v_\ell^2\right] \Bigg) +$$

$$\frac{(m-2)}{m(m+2)} \Bigg( 2\underbrace{\mathbb{E}\left[u_1^2 u_2^2 v_1^2 v_2^2\right]}_{j=1,r=2 \vee r=1,j=2} + 4(m-2)\underbrace{\mathbb{E}\left[u_1^2 u_2 u_3 v_1^2 v_2 v_3\right]}_{j \leq 2, r \geq 3 \vee r \leq 2, j \geq 3} +$$

$$\underbrace{(m-2)(m-3)\mathbb{E}\left[u_1 u_2 u_3 u_4 v_1 v_2 v_3 v_4\right]}_{j \neq r \geq 3} \Bigg)$$

$$= \frac{2(m-1)}{m(m+2)}\left( 2\left\langle \begin{smallmatrix} 3 & 3 \\ 1 & 1 \\ 0 & 0 \end{smallmatrix} \right\rangle + (m-2)\left\langle \begin{smallmatrix} 1 & 1 \\ 2 & 2 \\ 0 & 0 \end{smallmatrix} \right\rangle \right) +$$

$$\frac{(m-2)}{m(m+2)}\left( 2\left\langle \begin{smallmatrix} 2 & 2 \\ 2 & 2 \\ 0 & 0 \end{smallmatrix} \right\rangle + 4(m-2)\left\langle \begin{smallmatrix} 2 & 2 \\ 1 & 1 \\ 0 & 0 \end{smallmatrix} \right\rangle + (m-2)(m-3)\left\langle \begin{smallmatrix} 1 & 1 \\ 1 & 1 \\ 1 & 1 \\ 0 & 0 \end{smallmatrix} \right\rangle \right) +$$

$$\frac{m}{m(m+2)} \Bigg( 2\underbrace{\mathbb{E}\left[u_1^3 u_2 v_1 v_2^3\right]}_{j=1,\ell=2 \vee j=2,\ell=1} + 2(m-2)\underbrace{\mathbb{E}\left[u_1^3 u_2 v_1 v_2 v_3^2\right]}_{j \in \{1,2\}, \ell \geq 3} + 2(m-2)\underbrace{\mathbb{E}\left[u_1 u_2 u_3^2 v_1^3 v_2\right]}_{\ell \in \{1,2\}, j \geq 3} +$$

$$\underbrace{(m-2)(m-3)\mathbb{E}\left[u_1 u_2 u_3^2 v_1 v_2 v_4^2\right]}_{\ell \neq j \geq 3} \Bigg)$$

$$= \frac{2(m-1)}{m(m+2)}\left( 2\left\langle \begin{smallmatrix} 3 & 3 \\ 1 & 1 \\ 0 & 0 \end{smallmatrix} \right\rangle + (m-2)\left\langle \begin{smallmatrix} 1 & 1 \\ 2 & 2 \\ 0 & 0 \end{smallmatrix} \right\rangle \right) +$$

$$\frac{(m-2)}{m(m+2)}\left( 2\left\langle \begin{smallmatrix} 2 & 2 \\ 2 & 2 \\ 0 & 0 \end{smallmatrix} \right\rangle + 4(m-2)\left\langle \begin{smallmatrix} 2 & 2 \\ 1 & 1 \\ 0 & 0 \end{smallmatrix} \right\rangle + (m-2)(m-3)\left\langle \begin{smallmatrix} 1 & 1 \\ 1 & 1 \\ 1 & 1 \\ 0 & 0 \end{smallmatrix} \right\rangle \right) +$$

$$\frac{m}{m(m+2)}\left( 2\left\langle \begin{smallmatrix} 3 & 1 \\ 1 & 3 \\ 0 & 0 \end{smallmatrix} \right\rangle + 2(m-2)\left\langle \begin{smallmatrix} 3 & 1 \\ 1 & 1 \\ 0 & 2 \\ 0 & 0 \end{smallmatrix} \right\rangle + \right.$$

$$\left. 2(m-2)\left\langle \begin{smallmatrix} 1 & 3 \\ 1 & 1 \\ 2 & 0 \\ 0 & 0 \end{smallmatrix} \right\rangle + (m-2)(m-3)\left\langle \begin{smallmatrix} 1 & 1 \\ 1 & 1 \\ 2 & 0 \\ 0 & 2 \\ 0 & 0 \end{smallmatrix} \right\rangle \right)$$

$$= \frac{4(m-1)}{m(m+2)}\left\langle \begin{smallmatrix} 3 & 3 \\ 1 & 1 \\ 0 & 0 \end{smallmatrix} \right\rangle + \left( \frac{2(m-1)(m-2)}{m(m+2)} + \frac{4(m-2)^2}{m(m+2)} \right)\left\langle \begin{smallmatrix} 2 & 2 \\ 1 & 1 \\ 1 & 1 \\ 0 & 0 \end{smallmatrix} \right\rangle +$$

$$\frac{(m-2)}{m(m+2)}\left( 2\left\langle \begin{smallmatrix} 2 & 2 \\ 2 & 2 \\ 0 & 0 \end{smallmatrix} \right\rangle + (m-2)(m-3)\left\langle \begin{smallmatrix} 1 & 1 \\ 1 & 1 \\ 1 & 1 \\ 0 & 0 \end{smallmatrix} \right\rangle \right) +$$

$$\frac{m}{m(m+2)}\left( 2\left\langle \begin{smallmatrix} 3 & 1 \\ 1 & 3 \\ 0 & 0 \end{smallmatrix} \right\rangle + 4(m-2)\left\langle \begin{smallmatrix} 3 & 1 \\ 1 & 1 \\ 0 & 2 \\ 0 & 0 \end{smallmatrix} \right\rangle + (m-2)(m-3)\left\langle \begin{smallmatrix} 1 & 1 \\ 1 & 1 \\ 2 & 0 \\ 0 & 2 \\ 0 & 0 \end{smallmatrix} \right\rangle \right)$$

$$= \frac{-4(m-1)\cdot 9(p+1) + 2(m-2)(3m-5)(-p+3)}{m(m+2)(p-1)p(p+1)(p+2)(p+4)(p+6)} + \frac{(m-2)}{m(m+2)}\left( \frac{2\left(p^2+4p+15\right)+3(m-2)(m-3)}{(p-1)p(p+1)(p+2)(p+4)(p+6)} \right) +$$

$$\frac{m}{m(m+2)}\left( \frac{-2\cdot 9(p+3) - 4(m-2)\cdot 3(p+3) - (m-2)(m-3)(p+3)}{(p-1)p(p+1)(p+2)(p+4)(p+6)} \right)$$

$$= \frac{-m^3 p - 13m^2 p - 24m^2 + 2mp^2 - 6mp - 24m - 4p^2}{m(m+2)(p-1)p(p+1)(p+2)(p+4)(p+6)}$$

**Proposition 24.** *For $p \geq 3, m \in \{2, 3, \ldots, p\}$ and a random transformation $\mathbf{O}$ sampled as described in Eq. (1), it holds that,*

$$\mathbb{E}\left(\mathbf{e}_1^\top \mathbf{O}\mathbf{e}_1 \cdot \mathbf{e}_1^\top \mathbf{O}\mathbf{e}_2\right)\left(\mathbf{e}_3^\top \mathbf{O}\mathbf{e}_1 \cdot \mathbf{e}_3^\top \mathbf{O}\mathbf{e}_2\right)$$

$$= \frac{2m^4p + 4m^4 - 7m^3p^2 - 18m^3p + 8m^2p^3 + 25m^2p^2 + 24m^2p}{(p-2)(p-1)p(p+1)(p+2)(p+4)(p+6)} +$$

$$\frac{20m^2 - 3mp^4 - 11mp^3 - 44mp^2 - 64mp - 24m - p^4 + 11p^3 + 32p^2 + 68p + 48}{(p-2)(p-1)p(p+1)(p+2)(p+4)(p+6)}$$

*Proof.* Like in previous proofs, we decompose the expression into,

$$\mathbb{E}\left(\mathbf{e}_1^\top \mathbf{O}\mathbf{e}_1 \cdot \mathbf{e}_1^\top \mathbf{O}\mathbf{e}_2\right)\left(\mathbf{e}_3^\top \mathbf{O}\mathbf{e}_1 \cdot \mathbf{e}_3^\top \mathbf{O}\mathbf{e}_2\right)$$

$$= \mathbb{E}\left[\left(\mathbf{u}_a^\top \mathbf{Q}_m \mathbf{u}_a + \mathbf{u}_b^\top \mathbf{u}_b\right)\left(\mathbf{u}_a^\top \mathbf{Q}_m \mathbf{v}_a + \mathbf{u}_b^\top \mathbf{v}_b\right)\left(\mathbf{z}_a^\top \mathbf{Q}_m \mathbf{u}_a + \mathbf{z}_b^\top \mathbf{u}_b\right)\left(\mathbf{z}_a^\top \mathbf{Q}_m \mathbf{v}_a + \mathbf{z}_b^\top \mathbf{v}_b\right)\right]$$

$$= \mathbb{E}\left[\mathbf{u}_a^\top \mathbf{Q}_m \mathbf{u}_a \left(\mathbf{u}_a^\top \mathbf{Q}_m \mathbf{v}_a + \mathbf{u}_b^\top \mathbf{v}_b\right)\left(\mathbf{z}_a^\top \mathbf{Q}_m \mathbf{u}_a + \mathbf{z}_b^\top \mathbf{u}_b\right)\left(\mathbf{z}_a^\top \mathbf{Q}_m \mathbf{v}_a + \mathbf{z}_b^\top \mathbf{v}_b\right)\right] +$$

$$\mathbb{E}\left[\mathbf{u}_b^\top \mathbf{u}_b \left(\mathbf{u}_a^\top \mathbf{Q}_m \mathbf{v}_a + \mathbf{u}_b^\top \mathbf{v}_b\right)\left(\mathbf{z}_a^\top \mathbf{Q}_m \mathbf{u}_a + \mathbf{z}_b^\top \mathbf{u}_b\right)\left(\mathbf{z}_a^\top \mathbf{Q}_m \mathbf{v}_a + \mathbf{z}_b^\top \mathbf{v}_b\right)\right] \tag{10}$$

Focusing on the first term, and employing Cor. 12, we see that,

$$\mathbb{E}\left[\mathbf{u}_a^\top \mathbf{Q}_m \mathbf{u}_a \left(\mathbf{u}_a^\top \mathbf{Q}_m \mathbf{v}_a + \mathbf{u}_b^\top \mathbf{v}_b\right)\left(\mathbf{z}_a^\top \mathbf{Q}_m \mathbf{u}_a + \mathbf{z}_b^\top \mathbf{u}_b\right)\left(\mathbf{z}_a^\top \mathbf{Q}_m \mathbf{v}_a + \mathbf{z}_b^\top \mathbf{v}_b\right)\right]$$

$$= \mathbb{E}\left[\mathbf{u}_a^\top \mathbf{Q}_m \mathbf{u}_a \mathbf{u}_a^\top \mathbf{Q}_m \mathbf{v}_a \mathbf{z}_a^\top \mathbf{Q}_m \mathbf{u}_a \mathbf{z}_a^\top \mathbf{Q}_m \mathbf{v}_a\right] + \mathbb{E}\left[\mathbf{u}_a^\top \mathbf{Q}_m \mathbf{u}_a \mathbf{u}_a^\top \mathbf{Q}_m \mathbf{v}_a \mathbf{z}_b^\top \mathbf{u}_b \mathbf{z}_b^\top \mathbf{v}_b\right] +$$

$$\mathbb{E}\left[\mathbf{u}_a^\top \mathbf{Q}_m \mathbf{u}_a \mathbf{u}_b^\top \mathbf{v}_b \mathbf{z}_a^\top \mathbf{Q}_m \mathbf{u}_a \mathbf{z}_b^\top \mathbf{v}_b\right] + \mathbb{E}\left[\mathbf{u}_a^\top \mathbf{Q}_m \mathbf{u}_a \mathbf{u}_b^\top \mathbf{v}_b \mathbf{z}_b^\top \mathbf{u}_b \mathbf{z}_a^\top \mathbf{Q}_m \mathbf{v}_a\right]$$

The polynomial in the first summand can be derived tediously, very much like in the proof of the former Prop. 23, and shown to hold

$$\mathbb{E}\left[\mathbf{u}_a^\top \mathbf{Q}_m \mathbf{u}_a \mathbf{u}_a^\top \mathbf{Q}_m \mathbf{v}_a \mathbf{z}_a^\top \mathbf{Q}_m \mathbf{u}_a \mathbf{z}_a^\top \mathbf{Q}_m \mathbf{v}_a\right] = \frac{-m^2\left(2p^2+9p+6\right) + m(p-2)\left(p^2-6\right) + 2p^3 + 32p + 48}{(p-2)(p-1)p(p+1)(p+2)(p+4)(p+6)} .$$

The sum of the three rightmost summands is,

$$\mathbb{E}\left[\mathbf{u}_a^\top \mathbf{Q}_m \mathbf{u}_a \mathbf{u}_a^\top \mathbf{Q}_m \mathbf{v}_a \mathbf{z}_b^\top \mathbf{u}_b \mathbf{z}_b^\top \mathbf{v}_b\right] + \mathbb{E}\left[\mathbf{u}_a^\top \mathbf{Q}_m \mathbf{u}_a \mathbf{u}_b^\top \mathbf{v}_b \mathbf{z}_a^\top \mathbf{Q}_m \mathbf{u}_a \mathbf{z}_b^\top \mathbf{v}_b\right] +$$

$$+ \mathbb{E}\left[\mathbf{u}_a^\top \mathbf{Q}_m \mathbf{u}_a \mathbf{u}_b^\top \mathbf{v}_b \mathbf{z}_b^\top \mathbf{u}_b \mathbf{z}_a^\top \mathbf{Q}_m \mathbf{v}_a\right]$$

$$= \sum_{i,j,k,\ell=1}^{m} \left(\mathbb{E}\left[u_i u_j q_{ij} u_k v_\ell q_{k\ell} \mathbf{z}_b^\top \mathbf{u}_b \mathbf{z}_b^\top \mathbf{v}_b\right] + \mathbb{E}\left[u_i u_j q_{ij} \mathbf{u}_b^\top \mathbf{v}_b z_k u_\ell q_{k\ell} \mathbf{z}_b^\top \mathbf{v}_b\right]\right) +$$

$$\sum_{i,j,k,\ell=1}^{m} \left(\mathbb{E}\left[u_i u_j q_{ij} z_k v_\ell q_{k\ell} \mathbf{u}_b^\top \mathbf{v}_b \mathbf{z}_b^\top \mathbf{u}_b\right]\right)$$

$$= \sum_{i,j,k,\ell=1}^{m} \mathbb{E}\left[q_{ij} q_{k\ell}\right]\left(\mathbb{E}\left[u_i u_j u_k v_\ell \mathbf{z}_b^\top \mathbf{u}_b \mathbf{z}_b^\top \mathbf{v}_b\right] + \mathbb{E}\left[u_i u_j \mathbf{u}_b^\top \mathbf{v}_b z_k u_\ell \mathbf{z}_b^\top \mathbf{v}_b\right]\right) +$$

$$\sum_{i,j,k,\ell=1}^{m} \mathbb{E}\left[q_{ij} q_{k\ell}\right]\left(\mathbb{E}\left[u_i u_j z_k v_\ell \mathbf{u}_b^\top \mathbf{v}_b \mathbf{z}_b^\top \mathbf{u}_b\right]\right)$$

$$= \sum_{i,j=1}^{m} \underbrace{\mathbb{E}\left[q_{ij}^2\right]}_{=1/m}\left(\mathbb{E}\left[u_i u_j u_i v_j \mathbf{z}_b^\top \mathbf{u}_b \mathbf{z}_b^\top \mathbf{v}_b\right] + \mathbb{E}\left[u_i u_j z_i u_j \mathbf{u}_b^\top \mathbf{v}_b \mathbf{z}_b^\top \mathbf{v}_b\right] + \mathbb{E}\left[u_i u_j z_i v_j \mathbf{u}_b^\top \mathbf{v}_b \mathbf{z}_b^\top \mathbf{u}_b\right]\right)$$

$$= \frac{1}{m}\left(\mathbb{E}\left[\|\mathbf{u}_a\|^2 \mathbf{u}_a^\top \mathbf{v}_a \mathbf{z}_b^\top \mathbf{u}_b \mathbf{z}_b^\top \mathbf{v}_b\right] + \mathbb{E}\left[\|\mathbf{u}_a\|^2 \mathbf{u}_a^\top \mathbf{z}_a \mathbf{u}_b^\top \mathbf{v}_b \mathbf{z}_b^\top \mathbf{v}_b\right] + \mathbb{E}\left[\mathbf{u}_a^\top \mathbf{v}_a \mathbf{u}_a^\top \mathbf{z}_a \mathbf{u}_b^\top \mathbf{v}_b \mathbf{z}_b^\top \mathbf{u}_b\right]\right)$$

$$= \frac{1}{m}\left(\underbrace{2\mathbb{E}\left[\|\mathbf{u}_a\|^2 \mathbf{v}_b^\top \mathbf{z}_b \mathbf{u}_b^\top \mathbf{z}_b \mathbf{u}_a^\top \mathbf{v}_a\right]}_{\substack{\text{obtained from Eq. (39) by plugging in } m \leftarrow p-m}} + \underbrace{\mathbb{E}\left[\mathbf{u}_a^\top \mathbf{v}_a \mathbf{u}_a^\top \mathbf{z}_a \mathbf{u}_b^\top \mathbf{v}_b \mathbf{z}_b^\top \mathbf{u}_b\right]}_{\substack{\text{solved in Eq. (40)} \\ \text{since } \mathbf{u}_a^\top \mathbf{v}_a = -\mathbf{u}_b^\top \mathbf{v}_b \text{ and } \mathbf{u}_a^\top \mathbf{z}_a = -\mathbf{z}_b^\top \mathbf{u}_b}}\right)$$

$$= 2\frac{(p-m)(m+2)\left(2(p-m)p + 4(p-m) - p^2 - 3p + 2\right)}{(p-2)(p-1)p(p+1)(p+2)(p+4)(p+6)} + \frac{(p-m)\left(-m^2 + mp + 2p + 4\right)}{(p-1)p(p+1)(p+2)(p+4)(p+6)}$$

$$= \frac{-(p-m)(m+2)\left(5mp + 6m - 3p^2 - 2p\right)}{(p-2)(p-1)p(p+1)(p+2)(p+4)(p+6)} .$$

Overall, the first term of Eq. (10) equals,

$$\mathbb{E}\left[\mathbf{u}_a^\top\mathbf{Q}_m\mathbf{u}_a\left(\mathbf{u}_a^\top\mathbf{Q}_m\mathbf{v}_a+\mathbf{u}_b^\top\mathbf{v}_b\right)\left(\mathbf{z}_a^\top\mathbf{Q}_m\mathbf{u}_a+\mathbf{z}_b^\top\mathbf{u}_b\right)\left(\mathbf{z}_a^\top\mathbf{Q}_m\mathbf{v}_a+\mathbf{z}_b^\top\mathbf{v}_b\right)\right]$$
$$=\frac{6\left(m^3+m^2+2m+8\right)+4(m+2)p^3-2\left(5m^2+8m-2\right)p^2+(m-2)\left(5m^2+3m-16\right)p}{(p-2)(p-1)p(p+1)(p+2)(p+4)(p+6)}\;.$$

The second term holds,

$$\mathbb{E}\left[\mathbf{u}_b^\top\mathbf{u}_b\left(\mathbf{u}_a^\top\mathbf{Q}_m\mathbf{v}_a+\mathbf{u}_b^\top\mathbf{v}_b\right)\left(\mathbf{z}_a^\top\mathbf{Q}_m\mathbf{u}_a+\mathbf{z}_b^\top\mathbf{u}_b\right)\left(\mathbf{z}_a^\top\mathbf{Q}_m\mathbf{v}_a+\mathbf{z}_b^\top\mathbf{v}_b\right)\right]$$
$$=\mathbb{E}\left[\mathbf{u}_b^\top\mathbf{u}_b\mathbf{u}_b^\top\mathbf{v}_b\mathbf{z}_b^\top\mathbf{u}_b\mathbf{z}_b^\top\mathbf{v}_b\right]+\mathbb{E}\left[\mathbf{u}_b^\top\mathbf{u}_b\mathbf{u}_a^\top\mathbf{Q}_m\mathbf{v}_a\mathbf{z}_a^\top\mathbf{Q}_m\mathbf{u}_a\mathbf{z}_b^\top\mathbf{v}_b\right]+$$
$$\underbrace{\mathbb{E}\left[\mathbf{u}_b^\top\mathbf{u}_b\mathbf{u}_a^\top\mathbf{Q}_m\mathbf{v}_a\mathbf{z}_b^\top\mathbf{u}_b\mathbf{z}_a^\top\mathbf{Q}_m\mathbf{v}_a\right]+\mathbb{E}\left[\mathbf{u}_b^\top\mathbf{u}_b\mathbf{u}_b^\top\mathbf{v}_b\mathbf{z}_a^\top\mathbf{Q}_m\mathbf{u}_a\mathbf{z}_a^\top\mathbf{Q}_m\mathbf{v}_a\right]}_{\text{equal (swap }\mathbf{v},\mathbf{z}\text{ and map }\mathbf{Q}\to\mathbf{Q}^\top)}$$
$$=\mathbb{E}\left[\mathbf{u}_b^\top\mathbf{u}_b\mathbf{u}_b^\top\mathbf{v}_b\mathbf{z}_b^\top\mathbf{u}_b\mathbf{z}_b^\top\mathbf{v}_b\right]+\mathbb{E}\left[\mathbf{u}_b^\top\mathbf{u}_b\mathbf{z}_b^\top\mathbf{v}_b\cdot\mathbf{u}_a^\top\mathbf{Q}_m\mathbf{v}_a\mathbf{z}_a^\top\mathbf{Q}_m\mathbf{u}_a\right]+$$
$$2\mathbb{E}\left[\mathbf{u}_b^\top\mathbf{u}_b\mathbf{u}_b^\top\mathbf{v}_b\cdot\mathbf{z}_a^\top\mathbf{Q}_m\mathbf{u}_a\mathbf{z}_a^\top\mathbf{Q}_m\mathbf{v}_a\right]$$
$$=\underbrace{\mathbb{E}\left[\mathbf{u}_b^\top\mathbf{u}_b\mathbf{u}_b^\top\mathbf{v}_b\mathbf{z}_a^\top\mathbf{u}_a\mathbf{z}_a^\top\mathbf{v}_a\right]}_{\text{solved in Eq. (42)}}+\underbrace{\mathbb{E}\left[\mathbf{u}_b^\top\mathbf{u}_b\mathbf{z}_b^\top\mathbf{v}_b\cdot\mathbf{u}_a^\top\mathbf{Q}_m\mathbf{v}_a\mathbf{z}_a^\top\mathbf{Q}_m\mathbf{u}_a\right]}_{\text{solved in Eq. (45)}}+$$
$$2\underbrace{\mathbb{E}\left[\mathbf{u}_b^\top\mathbf{u}_b\mathbf{u}_b^\top\mathbf{v}_b\cdot\mathbf{z}_a^\top\mathbf{Q}_m\mathbf{u}_a\mathbf{z}_a^\top\mathbf{Q}_m\mathbf{v}_a\right]}_{\text{solved in Eq. (46)}}$$
$$=\frac{(p-m)\left(\left(1+\frac{1}{m}\right)m(p-m+2)\left(2mp+4m-p^2-3p+2\right)\right)}{(p-2)(p-1)p(p+1)(p+2)(p+4)(p+6)}+$$
$$\frac{(p-m)\left(2\left(m^2p^2+5m^2p+2m^2-mp^3-7mp^2-16mp-12m+4p^2+16p+16\right)\right)}{(p-2)(p-1)p(p+1)(p+2)(p+4)(p+6)}$$
$$=\frac{(p-m)(p-m+2)\left(2m^2p+4m^2-3mp^2-11mp+2m-p^2+5p+18\right)}{(p-2)(p-1)p(p+1)(p+2)(p+4)(p+6)}\;.$$

Finally, the overall expression holds,

$$\mathbb{E}\left(\mathbf{e}_1^\top\mathbf{O}\mathbf{e}_1\cdot\mathbf{e}_1^\top\mathbf{O}\mathbf{e}_2\right)\left(\mathbf{e}_3^\top\mathbf{O}\mathbf{e}_1\cdot\mathbf{e}_3^\top\mathbf{O}\mathbf{e}_2\right)$$
$$=\frac{6\left(m^3+m^2+2m+8\right)+4(m+2)p^3-2\left(5m^2+8m-2\right)p^2+(m-2)\left(5m^2+3m-16\right)p}{(p-2)(p-1)p(p+1)(p+2)(p+4)(p+6)}+$$
$$\frac{(p-m)(p-m+2)\left(2m^2p+4m^2-3mp^2-11mp+2m-p^2+5p+18\right)}{(p-2)(p-1)p(p+1)(p+2)(p+4)(p+6)}$$
$$=\frac{2m^4p+4m^4-7m^3p^2-18m^3p+8m^2p^3+25m^2p^2+24m^2p}{(p-2)(p-1)p(p+1)(p+2)(p+4)(p+6)}+$$
$$\frac{20m^2-3mp^4-11mp^3-44mp^2-64mp-24m-p^4+11p^3+32p^2+68p+48}{(p-2)(p-1)p(p+1)(p+2)(p+4)(p+6)}\;.$$

$$\square$$

**Proposition 25.** *For $p \geq 4, m \in \{2, 3, \ldots, p\}$ and a random transformation $\mathbf{O}$ sampled as described in Eq. (1), it holds that,*

$$\mathbb{E}\left(\mathbf{e}_3^\top \mathbf{O} \mathbf{e}_4 \cdot \mathbf{e}_3^\top \mathbf{O} \mathbf{e}_1\right)\left(\mathbf{e}_2^\top \mathbf{O} \mathbf{e}_4 \cdot \mathbf{e}_2^\top \mathbf{O} \mathbf{e}_1\right)$$

$$= \frac{-5m^4p - 6m^4 + 18m^3p^2 + 34m^3p - 12m^3 - 20m^2p^3 - 46m^2p^2 - 39m^2p - 42m^2 + 6mp^4 + 10mp^3 + 96mp^2}{(p-3)(p-2)(p-1)p(p+1)(p+2)(p+4)(p+6)} +$$

$$\frac{154mp + 60m + 2p^4 - 30p^3 - 32p^2 - 144p - 144}{(p-3)(p-2)(p-1)p(p+1)(p+2)(p+4)(p+6)}.$$

*Proof.* We begin by decomposing the expression into four terms,

$$\mathbb{E}\left(\mathbf{e}_3^\top \mathbf{O} \mathbf{e}_p \cdot \mathbf{e}_3^\top \mathbf{O} \mathbf{e}_1\right)\left(\mathbf{e}_2^\top \mathbf{O} \mathbf{e}_p \cdot \mathbf{e}_2^\top \mathbf{O} \mathbf{e}_1\right)$$

$$= \mathbb{E}\left[\left(\mathbf{u}_a^\top \mathbf{Q}_m \mathbf{z}_a + \mathbf{u}_b^\top \mathbf{z}_b\right)\left(\mathbf{u}_a^\top \mathbf{Q}_m \mathbf{v}_a + \mathbf{u}_b^\top \mathbf{v}_b\right)\left(\mathbf{x}_a^\top \mathbf{Q}_m \mathbf{z}_a + \mathbf{x}_b^\top \mathbf{z}_b\right)\left(\mathbf{x}_a^\top \mathbf{Q}_m \mathbf{v}_a + \mathbf{x}_b^\top \mathbf{v}_b\right)\right]$$

$$= \mathbb{E}\left[\mathbf{u}_a^\top \mathbf{Q}_m \mathbf{z}_a \cdot \mathbf{u}_a^\top \mathbf{Q}_m \mathbf{v}_a \left(\mathbf{x}_a^\top \mathbf{Q}_m \mathbf{z}_a + \mathbf{x}_b^\top \mathbf{z}_b\right)\left(\mathbf{x}_a^\top \mathbf{Q}_m \mathbf{v}_a + \mathbf{x}_b^\top \mathbf{v}_b\right)\right] +$$

$$\mathbb{E}\left[\mathbf{u}_b^\top \mathbf{z}_b \cdot \mathbf{u}_b^\top \mathbf{v}_b \left(\mathbf{x}_a^\top \mathbf{Q}_m \mathbf{z}_a + \mathbf{x}_b^\top \mathbf{z}_b\right)\left(\mathbf{x}_a^\top \mathbf{Q}_m \mathbf{v}_a + \mathbf{x}_b^\top \mathbf{v}_b\right)\right] +$$

$$\mathbb{E}\left[\mathbf{u}_a^\top \mathbf{Q}_m \mathbf{z}_a \cdot \mathbf{u}_b^\top \mathbf{v}_b \left(\mathbf{x}_a^\top \mathbf{Q}_m \mathbf{z}_a + \mathbf{x}_b^\top \mathbf{z}_b\right)\left(\mathbf{x}_a^\top \mathbf{Q}_m \mathbf{v}_a + \mathbf{x}_b^\top \mathbf{v}_b\right)\right] +$$

$$\mathbb{E}\left[\mathbf{u}_b^\top \mathbf{z}_b \cdot \mathbf{u}_a^\top \mathbf{Q}_m \mathbf{v}_a \left(\mathbf{x}_a^\top \mathbf{Q}_m \mathbf{z}_a + \mathbf{x}_b^\top \mathbf{z}_b\right)\left(\mathbf{x}_a^\top \mathbf{Q}_m \mathbf{v}_a + \mathbf{x}_b^\top \mathbf{v}_b\right)\right]$$

Below we compute each of these terms separately. The result in the proposition is given by summing these 4 terms.

**Term 1.** Employing 12 once again, the term decomposes as

$$\mathbb{E}\left[\mathbf{u}_a^\top \mathbf{Q}_m \mathbf{z}_a \cdot \mathbf{u}_a^\top \mathbf{Q}_m \mathbf{v}_a \left(\mathbf{x}_a^\top \mathbf{Q}_m \mathbf{z}_a + \mathbf{x}_b^\top \mathbf{z}_b\right)\left(\mathbf{x}_a^\top \mathbf{Q}_m \mathbf{v}_a + \mathbf{x}_b^\top \mathbf{v}_b\right)\right]$$

$$= \mathbb{E}\left[\mathbf{u}_a^\top \mathbf{Q}_m \mathbf{z}_a \mathbf{u}_a^\top \mathbf{Q}_m \mathbf{v}_a \mathbf{x}_a^\top \mathbf{Q}_m \mathbf{z}_a \mathbf{x}_a^\top \mathbf{Q}_m \mathbf{v}_a\right] + \underbrace{\mathbb{E}\left[\mathbf{u}_a^\top \mathbf{Q}_m \mathbf{z}_a \mathbf{u}_a^\top \mathbf{Q}_m \mathbf{v}_a \cdot \mathbf{x}_b^\top \mathbf{z}_b \mathbf{x}_b^\top \mathbf{v}_b\right]}_{\text{solved in Eq. (44)}}.$$

The polynomial in the left inner term (the first summand) can be derived tediously, very much like in the proof of Prop. 23, and shown to hold

$$\mathbb{E}\left[\mathbf{u}_a^\top \mathbf{Q}_m \mathbf{z}_a \mathbf{u}_a^\top \mathbf{Q}_m \mathbf{v}_a \mathbf{x}_a^\top \mathbf{Q}_m \mathbf{z}_a \mathbf{x}_a^\top \mathbf{Q}_m \mathbf{v}_a\right]$$

$$= \frac{-m^3\left(2p^3 + 11p^2 + p - 30\right) + m^2\left(p^4 + p^3 + 2p^2 + 60p + 72\right) + 2m\left(p^4 + p^3 + 18p^2 + 14p - 60\right) - 8(2p+3)\left(p^2+12\right)}{(m+2)(p-3)(p-2)(p-1)p(p+1)(p+2)(p+4)(p+6)}.$$

Overall, the first term is,

$$\mathbb{E}\left[\mathbf{u}_a^\top \mathbf{Q}_m \mathbf{z}_a \cdot \mathbf{u}_a^\top \mathbf{Q}_m \mathbf{v}_a \left(\mathbf{x}_a^\top \mathbf{Q}_m \mathbf{z}_a + \mathbf{x}_b^\top \mathbf{z}_b\right)\left(\mathbf{x}_a^\top \mathbf{Q}_m \mathbf{v}_a + \mathbf{x}_b^\top \mathbf{v}_b\right)\right]$$

$$= \frac{-m^3\left(2p^3 + 11p^2 + p - 30\right) + m^2\left(p^4 + p^3 + 2p^2 + 60p + 72\right) + 2m\left(p^4 + p^3 + 18p^2 + 14p - 60\right) - 8(2p+3)\left(p^2+12\right)}{(m+2)(p-3)(p-2)(p-1)p(p+1)(p+2)(p+4)(p+6)} +$$

$$\frac{(p-m)\left(-2m^2p^2 - 8m^2p + mp^3 + 4mp^2 + 15mp + 18m - 6p^2 - 6p - 12\right)}{(p-3)(p-2)(p-1)p(p+1)(p+2)(p+4)(p+6)}$$

$$= \frac{2m^3p^2 + 8m^3p - 5m^2p^3 - 23m^2p^2 - 16m^2p + 12m^2 + 2mp^4 + 9mp^3 + 45mp^2 + 86mp + 24m - 14p^3 - 18p^2 - 108p - 144}{(p-3)(p-2)(p-1)p(p+1)(p+2)(p+4)(p+6)}.$$

**Term 2.** Notice that $\mathbf{u}, \mathbf{v}, \mathbf{z}, \mathbf{x}$ are exchangeable in the sense that we can swap them freely (see Prop. 9). Therefore,

$$\mathbb{E}\left[\mathbf{u}_b^\top \mathbf{z}_b \cdot \mathbf{u}_b^\top \mathbf{v}_b \left(\mathbf{x}_a^\top \mathbf{Q}_m \mathbf{z}_a + \mathbf{x}_b^\top \mathbf{z}_b\right)\left(\mathbf{x}_a^\top \mathbf{Q}_m \mathbf{v}_a + \mathbf{x}_b^\top \mathbf{v}_b\right)\right]$$

$$\overset{[\text{Cor. 12}]}{=} \mathbb{E}\left[\mathbf{u}_b^\top \mathbf{z}_b \cdot \mathbf{u}_b^\top \mathbf{v}_b \left(\mathbf{x}_a^\top \mathbf{Q}_m \mathbf{z}_a \cdot \mathbf{x}_a^\top \mathbf{Q}_m \mathbf{v}_a + \mathbf{x}_b^\top \mathbf{z}_b \cdot \mathbf{x}_b^\top \mathbf{v}_b\right)\right]$$

$$= \mathbb{E}\left[\mathbf{u}_b^\top \mathbf{z}_b \mathbf{u}_b^\top \mathbf{v}_b \mathbf{x}_a^\top \mathbf{Q}_m \mathbf{z}_a \mathbf{x}_a^\top \mathbf{Q}_m \mathbf{v}_a\right] + \mathbb{E}\left[\mathbf{u}_b^\top \mathbf{z}_b \mathbf{u}_b^\top \mathbf{v}_b \mathbf{x}_b^\top \mathbf{z}_b \mathbf{x}_b^\top \mathbf{v}_b\right]$$

$$\overset{[\text{swap}]}{=} \underbrace{\mathbb{E}\left[\mathbf{u}_a^\top \mathbf{Q}_m \mathbf{z}_a \mathbf{u}_a^\top \mathbf{Q}_m \mathbf{v}_a \mathbf{x}_b^\top \mathbf{z}_b \mathbf{x}_b^\top \mathbf{v}_b\right]}_{\text{swapped } \mathbf{x}, \mathbf{u}; \text{ solved in Eq. (44)}} + \underbrace{\mathbb{E}\left[\mathbf{u}_b^\top \mathbf{v}_b \mathbf{x}_b^\top \mathbf{z}_b \mathbf{u}_b^\top \mathbf{x}_b \mathbf{v}_b^\top \mathbf{z}_b\right]}_{\text{swapped } \mathbf{v}, \mathbf{u}; \text{ solved in Eq. (47)}}$$

$$= \frac{(p-m)\left(-2m^2p^2 - 8m^2p + mp^3 + 4mp^2 + 15mp + 18m - 6p^2 - 6p - 12\right)}{(p-3)(p-2)(p-1)p(p+1)(p+2)(p+4)(p+6)} +$$

$$\frac{m(p-m)\left(5m^2p + 6m^2 - 5mp^2 - 6mp + p^3 + p^2 + 2p\right)}{(p-3)(p-2)(p-1)p(p+1)(p+2)(p+4)(p+6)}$$

$$= (p-m)\left(\frac{5m^3p + 6m^3 - 7m^2p^2 - 14m^2p + 2mp^3 + 5mp^2 + 17mp + 18m - 6p^2 - 6p - 12}{(p-3)(p-2)(p-1)p(p+1)(p+2)(p+4)(p+6)}\right).$$

**Terms 3 and 4.** First, we notice that subterms 3 and 4 are equivalent (we can swap $\mathbf{z}$, $\mathbf{v}$ due to the invariance; Prop. 9), that is

$$\mathbb{E}\left[\mathbf{u}_a^\top \mathbf{Q}_m \mathbf{z}_a \cdot \mathbf{u}_b^\top \mathbf{v}_b \left(\mathbf{x}_a^\top \mathbf{Q}_m \mathbf{z}_a + \mathbf{x}_b^\top \mathbf{z}_b\right)\left(\mathbf{x}_a^\top \mathbf{Q}_m \mathbf{v}_a + \mathbf{x}_b^\top \mathbf{v}_b\right)\right]$$
$$= \mathbb{E}\left[\mathbf{u}_b^\top \mathbf{z}_b \cdot \mathbf{u}_a^\top \mathbf{Q}_m \mathbf{v}_a \left(\mathbf{x}_a^\top \mathbf{Q}_m \mathbf{z}_a + \mathbf{x}_b^\top \mathbf{z}_b\right)\left(\mathbf{x}_a^\top \mathbf{Q}_m \mathbf{v}_a + \mathbf{x}_b^\top \mathbf{v}_b\right)\right]$$
$$[\text{below}] = \frac{(p-m)\left(-2m^2 p^2 - 3m^2 p + 6m^2 + mp^3 - mp^2 + 9mp + 18m + p^3 - 5p^2 - 4p - 12\right)}{(p-3)(p-2)(p-1)p(p+1)(p+2)(p+4)(p+6)},$$

and so we focus on just one of them.

By employing Cor. 12, we see that

$$\mathbb{E}\left[\mathbf{u}_a^\top \mathbf{Q}_m \mathbf{z}_a \cdot \mathbf{u}_b^\top \mathbf{v}_b \left(\mathbf{x}_a^\top \mathbf{Q}_m \mathbf{z}_a + \mathbf{x}_b^\top \mathbf{z}_b\right)\left(\mathbf{x}_a^\top \mathbf{Q}_m \mathbf{v}_a + \mathbf{x}_b^\top \mathbf{v}_b\right)\right]$$
$$= \mathbb{E}\left[\mathbf{u}_a^\top \mathbf{Q}_m \mathbf{z}_a \cdot \mathbf{u}_b^\top \mathbf{v}_b \left(\mathbf{x}_a^\top \mathbf{Q}_m \mathbf{z}_a \cdot \mathbf{x}_b^\top \mathbf{v}_b + \mathbf{x}_b^\top \mathbf{z}_b \cdot \mathbf{x}_a^\top \mathbf{Q}_m \mathbf{v}_a\right)\right]$$
$$= \frac{(p-m)\left(\left(5m^2 p + 6m^2 - 5mp^2 - 6mp + p^3 + p^2 + 2p\right) + \left(-2m^2 p^2 - 8m^2 p + mp^3 + 4mp^2 + 15mp + 18m - 6p^2 - 6p - 12\right)\right)}{(p-3)(p-2)(p-1)p(p+1)(p+2)(p+4)(p+6)}$$
$$= \frac{(p-m)\left(-2m^2 p^2 - 3m^2 p + 6m^2 + mp^3 - mp^2 + 9mp + 18m + p^3 - 5p^2 - 4p - 12\right)}{(p-3)(p-2)(p-1)p(p+1)(p+2)(p+4)(p+6)},$$

where we used the following two derivations, *i.e.,*

$$\mathbb{E}\left[\mathbf{u}_a^\top \mathbf{Q}_m \mathbf{z}_a \cdot \mathbf{u}_b^\top \mathbf{v}_b \cdot \mathbf{x}_b^\top \mathbf{z}_b \cdot \mathbf{x}_a^\top \mathbf{Q}_m \mathbf{v}_a\right] = \mathbb{E}\left[\left(\mathbf{u}_b^\top \mathbf{v}_b \cdot \mathbf{x}_b^\top \mathbf{z}_b\right) \cdot \mathbf{u}_a^\top \mathbf{Q}_m \mathbf{z}_a \cdot \mathbf{x}_a^\top \mathbf{Q}_m \mathbf{v}_a\right]$$
$$= \mathbb{E}\left[\left(\mathbf{u}_b^\top \mathbf{v}_b \cdot \mathbf{x}_b^\top \mathbf{z}_b\right) \cdot \left(\sum_{i=1}^m \sum_{j=1}^m u_i q_{ij} z_j\right) \cdot \left(\sum_{k=1}^m \sum_{\ell=1}^m x_k q_{k\ell} v_\ell\right)\right]$$
$$= \sum_{i,j,k,\ell=1}^m \mathbb{E}\left[\left(\mathbf{u}_b^\top \mathbf{v}_b \cdot \mathbf{x}_b^\top \mathbf{z}_b\right) \cdot u_i v_\ell x_k z_j q_{ij} q_{k\ell}\right]$$
$$= \sum_{i,j,k,\ell=1}^m \mathbb{E}\left[\left(\mathbf{u}_b^\top \mathbf{v}_b \cdot \mathbf{x}_b^\top \mathbf{z}_b\right) \cdot u_i v_\ell x_k z_j \mathbb{E}_{\mathbf{Q}_m}\left[q_{ij} q_{k\ell}\right]\right]$$
$$[\text{Prop. 10}] = \sum_{i,j=1}^m \underbrace{\mathbb{E}_{\mathbf{Q}_m}\left[q_{ij}^2\right]}_{=1/m} \mathbb{E}\left[\left(\mathbf{u}_b^\top \mathbf{v}_b \cdot \mathbf{x}_b^\top \mathbf{z}_b\right) \cdot u_i v_j x_i z_j\right]$$
$$= \frac{1}{m} \underbrace{\mathbb{E}\left[\mathbf{u}_b^\top \mathbf{v}_b \cdot \mathbf{x}_b^\top \mathbf{z}_b \cdot \mathbf{u}_b^\top \mathbf{x}_b \cdot \mathbf{v}_b^\top \mathbf{z}_b\right]}_{\text{solved in Eq. (47)}}$$
$$= \frac{(p-m)\left(5m^2 p + 6m^2 - 5mp^2 - 6mp + p^3 + p^2 + 2p\right)}{(p-3)(p-2)(p-1)p(p+1)(p+2)(p+4)(p+6)},$$

and

$$\mathbb{E}\left[\mathbf{u}_a^\top \mathbf{Q}_m \mathbf{z}_a \cdot \mathbf{u}_b^\top \mathbf{v}_b \mathbf{x}_a^\top \mathbf{Q}_m \mathbf{z}_a \cdot \mathbf{x}_b^\top \mathbf{v}_b\right]$$
$$= \mathbb{E}\left[\|\mathbf{z}_a\|^2 \mathbb{E}_{\mathbf{r}\sim\mathcal{S}^{m-1}}\left(\mathbf{u}_a^\top \mathbf{r}\mathbf{r}^\top \mathbf{x}_a\right) \cdot \mathbf{u}_b^\top \mathbf{v}_b \mathbf{x}_b^\top \mathbf{v}_b\right] = \frac{1}{m}\underbrace{\mathbb{E}\left[\|\mathbf{z}_a\|^2 \mathbf{u}_a^\top \mathbf{x}_a \mathbf{u}_b^\top \mathbf{v}_b \mathbf{x}_b^\top \mathbf{v}_b\right]}_{\text{solved in Eq. (48)}}$$
$$= \frac{(p-m)\left(-2m^2 p^2 - 8m^2 p + mp^3 + 4mp^2 + 15mp + 18m - 6p^2 - 6p - 12\right)}{(p-3)(p-2)(p-1)p(p+1)(p+2)(p+4)(p+6)}.$$

$\square$

**Proposition 26.** *Let* $\mathbf{u} \sim \mathcal{S}^{p-1}$ *and let* $\mathbf{u}_a$ *consists of its first* $m$ *coordinates. The expected* $n^{th}$ *power of the squared norm is,*

$$\mathbb{E}_{\mathbf{u} \sim \mathcal{S}^{p-1}} \|\mathbf{u}_a\|^{2n} = \prod_{r=0}^{n-1} \frac{m+2r}{p+2r} \, .$$

*Specifically, it holds that*

$$\mathbb{E}_{\mathbf{u} \sim \mathcal{S}^{p-1}} \|\mathbf{u}_a\|^2 = \frac{m}{p}$$

$$\mathbb{E}_{\mathbf{u} \sim \mathcal{S}^{p-1}} \|\mathbf{u}_a\|^4 = \frac{m}{p} \cdot \frac{m+2}{p+2}$$

$$\mathbb{E}_{\mathbf{u} \sim \mathcal{S}^{p-1}} \|\mathbf{u}_a\|^6 = \frac{m}{p} \cdot \frac{m+2}{p+2} \cdot \frac{m+4}{p+4}$$

$$\mathbb{E}_{\mathbf{u} \sim \mathcal{S}^{p-1}} \|\mathbf{u}_a\|^8 = \frac{m}{p} \cdot \frac{m+2}{p+2} \cdot \frac{m+4}{p+4} \cdot \frac{m+6}{p+6}$$

$$\implies \mathbb{E}_{\mathbf{u} \sim \mathcal{S}^{p-1}} \|\mathbf{u}_b\|^8 = \frac{p-m}{p} \cdot \frac{p-m+2}{p+2} \cdot \frac{p-m+4}{p+4} \cdot \frac{p-m+6}{p+6}$$

*Proof.* Notice that the squared norm can be parameterized as $\|\mathbf{u}_a\|^2 = \mathbf{u}_a^\top \mathbf{u}_a \triangleq \frac{X}{X+Y}$, where $X \sim \chi_m^2, Y \sim \chi_{p-m}^2$. Consequently, it is distributed as $\|\mathbf{u}_a\|^2 \sim B\left(\frac{m}{2}, \frac{p-m}{2}\right)$. Moreover, given $n \in \mathbb{N}^+$, the $n^{\text{th}}$ raw moment is given by (Chapter 25 in Johnson et al. (1995)),

$$\mathbb{E}_{\mathbf{u} \sim \mathcal{S}^{p-1}} \|\mathbf{u}_a\|^{2n} = \prod_{r=0}^{n-1} \frac{\frac{m}{2}+r}{\frac{p}{2}+r} = \prod_{r=0}^{n-1} \frac{m+2r}{p+2r} \, .$$

$\square$

$$\mathbb{E}\left[\|\mathbf{u}_a\|^4 u_1^2 v_1^2\right] = \mathbb{E}\left[\left(\sum_{i=1}^m u_i^2\right)\left(\sum_{j=1}^m u_j^2\right) u_1^2 v_1^2\right] = \sum_{i=1}^m \sum_{j=1}^m \mathbb{E}\left[u_i^2 u_j^2 u_1^2 v_1^2\right]$$

$$= \underbrace{\mathbb{E}\left[u_1^6 v_1^2\right]}_{i=j=1} + 2(m-1)\underbrace{\mathbb{E}\left[u_1^4 v_1^2 u_2^2\right]}_{i=1\neq j \vee i\neq 1=j} + (m-1)\underbrace{\mathbb{E}\left[u_1^2 u_2^4 v_1^2\right]}_{i=j\geq 2} + (m-1)(m-2)\underbrace{\mathbb{E}\left[u_1^2 u_2^2 u_3^2 v_1^2\right]}_{i\neq j\geq 2}$$

$$= \left\langle \begin{smallmatrix} 6 & 2 \\ 0 & 0 \end{smallmatrix} \right\rangle + 2(m-1)\left\langle \begin{smallmatrix} 4 & 2 \\ 2 & 0 \\ 0 & 0 \end{smallmatrix} \right\rangle + (m-1)\left\langle \begin{smallmatrix} 2 & 2 \\ 4 & 0 \\ 0 & 0 \end{smallmatrix} \right\rangle + (m-1)(m-2)\left\langle \begin{smallmatrix} 2 & 2 \\ 2 & 0 \\ 2 & 0 \\ 0 & 0 \end{smallmatrix} \right\rangle \qquad (11)$$

$$= \frac{15(p-1) + (m-1)(6(p+1) + 3(p+3) + (m-2)(p+3))}{(p-1)p(p+2)(p+4)(p+6)}$$

$$= \frac{m^2(p+3) + m(6p+6) + 8p - 24}{(p-1)p(p+2)(p+4)(p+6)}$$

$$\mathbb{E}\left[\|\mathbf{u}_a\|^4 u_1^2 v_2^2\right] = \mathbb{E}\left[\left(\sum_{i=1}^m u_i^2\right)\left(\sum_{j=1}^m u_j^2\right) u_1^2 v_2^2\right] = \sum_{i=1}^m \sum_{j=1}^m \mathbb{E}\left[u_i^2 u_j^2 u_1^2 v_2^2\right]$$

$$= \underbrace{\mathbb{E}\left[u_1^6 v_2^2\right]}_{i=j=1} + \underbrace{\mathbb{E}\left[u_1^2 u_2^4 v_2^2\right]}_{i=j=2} + 2\underbrace{\mathbb{E}\left[u_1^4 u_2^2 v_2^2\right]}_{i=2,j=1 \vee i=1,j=2} + 2(m-2)\underbrace{\mathbb{E}\left[u_1^4 v_2^2 u_3^2\right]}_{i=1,j\geq 3 \vee j=1,i\geq 3} +$$

$$2(m-2)\underbrace{\mathbb{E}\left[u_1^2 u_2^2 v_2^2 u_3^2\right]}_{i=2,j\geq 3 \vee j=2,i\geq 3} + (m-2)(m-3)\underbrace{\mathbb{E}\left[u_1^2 v_2^2 u_3^2 u_4^2\right]}_{i\neq j\geq 3} + (m-2)\underbrace{\mathbb{E}\left[u_1^2 v_2^2 u_3^4\right]}_{i=j\geq 3}$$

$$= \left\langle \begin{smallmatrix} 6 & 0 \\ 0 & 2 \\ 0 & 0 \end{smallmatrix} \right\rangle + \left\langle \begin{smallmatrix} 2 & 0 \\ 4 & 2 \\ 0 & 0 \end{smallmatrix} \right\rangle + 2\left\langle \begin{smallmatrix} 4 & 0 \\ 2 & 2 \\ 0 & 0 \end{smallmatrix} \right\rangle + 2(m-2)\left\langle \begin{smallmatrix} 4 & 0 \\ 0 & 2 \\ 2 & 0 \\ 0 & 0 \end{smallmatrix} \right\rangle + 2(m-2)\left\langle \begin{smallmatrix} 2 & 0 \\ 2 & 2 \\ 2 & 0 \\ 0 & 0 \end{smallmatrix} \right\rangle +$$

$$(m-2)(m-3)\left\langle \begin{smallmatrix} 2 & 0 \\ 0 & 2 \\ 2 & 0 \\ 2 & 0 \\ 0 & 0 \end{smallmatrix} \right\rangle + (m-2)\left\langle \begin{smallmatrix} 2 & 0 \\ 0 & 2 \\ 4 & 0 \\ 0 & 0 \end{smallmatrix} \right\rangle \tag{12}$$

$$= \left\langle \begin{smallmatrix} 6 & 0 \\ 0 & 2 \\ 0 & 0 \end{smallmatrix} \right\rangle + \left\langle \begin{smallmatrix} 2 & 0 \\ 4 & 2 \\ 0 & 0 \end{smallmatrix} \right\rangle + 2\left\langle \begin{smallmatrix} 4 & 0 \\ 2 & 2 \\ 0 & 0 \end{smallmatrix} \right\rangle + 3(m-2)\left\langle \begin{smallmatrix} 4 & 0 \\ 0 & 2 \\ 2 & 0 \\ 0 & 0 \end{smallmatrix} \right\rangle + 2(m-2)\left\langle \begin{smallmatrix} 2 & 0 \\ 2 & 2 \\ 2 & 0 \\ 0 & 0 \end{smallmatrix} \right\rangle +$$

$$(m-2)(m-3)\left\langle \begin{smallmatrix} 2 & 0 \\ 0 & 2 \\ 2 & 0 \\ 2 & 0 \\ 0 & 0 \end{smallmatrix} \right\rangle$$

$$= \frac{15(p+5) + 3(p+1) + 2(3p+9) + (m-2)(9(p+5) + 2(p+3) + (m-3)(p+5))}{(p-1)p(p+2)(p+4)(p+6)}$$

$$= \frac{m^2(p+5) + 2m(3p+13) + 8(p+3)}{(p-1)p(p+2)(p+4)(p+6)}$$

We notice that the following derivation is symmetric in $i, j$. Thus, we assume $j \geq i$ and multiply everything by 2.

$$\mathbb{E}\left[\|\mathbf{u}_a\|^4 u_1 u_2 v_1 v_2\right] = \mathbb{E}\left[\left(\sum_{i=1}^m u_i^2\right)\left(\sum_{j=1}^m u_j^2\right) u_1 u_2 v_1 v_2\right] = \sum_{i=1}^m \sum_{j=1}^m \mathbb{E}\left[u_i^2 u_j^2 u_1 u_2 v_1 v_2\right]$$

$$= 2\underbrace{\mathbb{E}\left[u_1^5 u_2 v_1 v_2\right]}_{i=j=1 \vee i=j=2} + 2\underbrace{\mathbb{E}\left[u_1^3 u_2^3 v_1 v_2\right]}_{i=1,j=2} + 4(m-2)\underbrace{\mathbb{E}\left[u_1^3 u_2 u_3^2 v_1 v_2\right]}_{i=1,j\geq 3 \vee i=2,j\geq 3} +$$

$$(m-2)(m-3)\underbrace{\mathbb{E}\left[u_1 u_2 u_3^2 u_4^2 v_1 v_2\right]}_{i\neq j\geq 3} + (m-2)\underbrace{\mathbb{E}\left[u_1 u_2 u_3^4 v_1 v_2\right]}_{i=j\geq 3}$$

$$= 2\left\langle \begin{smallmatrix} 5 & 1 \\ 1 & 1 \\ 0 & 0 \end{smallmatrix} \right\rangle + 2\left\langle \begin{smallmatrix} 3 & 1 \\ 3 & 1 \\ 0 & 0 \end{smallmatrix} \right\rangle + 4(m-2)\left\langle \begin{smallmatrix} 3 & 1 \\ 1 & 1 \\ 2 & 0 \\ 0 & 0 \end{smallmatrix} \right\rangle + (m-2)(m-3)\left\langle \begin{smallmatrix} 1 & 1 \\ 1 & 1 \\ 2 & 0 \\ 2 & 0 \\ 0 & 0 \end{smallmatrix} \right\rangle + \tag{13}$$

$$(m-2)\left\langle \begin{smallmatrix} 1 & 1 \\ 1 & 1 \\ 4 & 0 \\ 0 & 0 \end{smallmatrix} \right\rangle$$

$$= \frac{-30 - 18 + (m-2)(-12 - (m-3) - 3)}{(p-1)p(p+2)(p+4)(p+6)}$$

$$= \frac{-(m+6)(m+4)}{(p-1)p(p+2)(p+4)(p+6)}$$

## E.1 Auxiliary derivations with two vectors

Below we attached many auxiliary derivations of simple polynomials that we need in our main propositions and lemmas.

$$
\begin{aligned}
& \mathbb{E}_{\mathbf{u}\perp\mathbf{v},\mathbf{Q}_m} \left(\mathbf{u}_a^\top \mathbf{Q}_m \mathbf{u}_a\right)^2 \left(\mathbf{u}_a^\top \mathbf{Q}_m \mathbf{v}_a\right)^2 \\
& = \mathbb{E}_{\mathbf{u}\perp\mathbf{v}} \left[ \mathbb{E}_{\mathbf{r}\sim\mathcal{S}^{m-1}} \left[ \|\mathbf{u}_a\|^4 \left(\sum_{i=1}^m r_i u_i\right)^2 \left(\sum_{j=1}^m r_j v_j\right)^2 \right]\right] \\
& = \sum_{i=1}^m \sum_{j=1}^m \sum_{k=1}^m \sum_{\ell=1}^m \mathbb{E}_{\mathbf{u}\perp\mathbf{v}} \left[ \mathbb{E}_{\mathbf{r}\sim\mathcal{S}^{m-1}} \left[ \|\mathbf{u}_a\|^4 r_i u_i r_j u_j r_k v_k r_\ell v_\ell \right]\right] \\
& = \sum_{i=1}^m \sum_{j=1}^m \sum_{k=1}^m \sum_{\ell=1}^m \mathbb{E}_{\mathbf{u}\perp\mathbf{v}} \left[ \|\mathbf{u}_a\|^4 u_i u_j v_k v_\ell \mathbb{E}_{\mathbf{r}\sim\mathcal{S}^{m-1}} \left[ r_i r_j r_k r_\ell \right]\right] \\
[\text{Prop. 10}] & = \underbrace{\sum_{i=1}^m \mathbb{E}_{\mathbf{u}\perp\mathbf{v}} \left[ \|\mathbf{u}_a\|^4 u_i^2 v_i^2 \mathbb{E}_{\mathbf{r}} \left[ r_i^4 \right]\right]}_{i=j=k=\ell} + \underbrace{\sum_{i\neq k=1}^m \mathbb{E}_{\mathbf{u}\perp\mathbf{v}} \left[ \|\mathbf{u}_a\|^4 u_i^2 v_k^2 \mathbb{E}_{\mathbf{r}} \left[ r_i^2 r_k^2 \right]\right]}_{i=j\neq k=\ell} + \\
& \underbrace{2 \sum_{i\neq j=1}^m \mathbb{E}_{\mathbf{u}\perp\mathbf{v}} \left[ \|\mathbf{u}_a\|^4 u_i u_j v_i v_j \mathbb{E}_{\mathbf{r}} \left[ r_i^2 r_j^2 \right]\right]}_{i=k\neq j=\ell \,\vee\, i=\ell\neq j=k} \\
& = \left\langle \tfrac{4}{0} \right\rangle_m \sum_{i=1}^m \mathbb{E}\left[ \|\mathbf{u}_a\|^4 u_i^2 v_i^2 \right] + \\
& \left\langle \tfrac{2}{\substack{2 \\ 0}} \right\rangle_m \left( \sum_{i\neq k=1}^m \mathbb{E}\left[ \|\mathbf{u}_a\|^4 u_i^2 v_k^2 \right] + 2\sum_{i\neq j=1}^m \mathbb{E}\left[ \|\mathbf{u}_a\|^4 u_i u_j v_i v_j \right] \right) \\
& = m \left\langle \tfrac{4}{0} \right\rangle_m \mathbb{E}\left[ \|\mathbf{u}_a\|^4 u_1^2 v_1^2 \right] + \\
& m(m-1) \left\langle \tfrac{2}{\substack{2 \\ 0}} \right\rangle_m \left( \mathbb{E}\left[ \|\mathbf{u}_a\|^4 u_1^2 v_2^2 \right] + 2\mathbb{E}\left[ \|\mathbf{u}_a\|^4 u_1 u_2 v_1 v_2 \right] \right) \\
& = \frac{3}{m+2} \underbrace{\mathbb{E}\left[ \|\mathbf{u}_a\|^4 u_1^2 v_1^2 \right]}_{\text{solved in Eq. (11)}} + \frac{m-1}{m+2} \left( \underbrace{\mathbb{E}\left[ \|\mathbf{u}_a\|^4 u_1^2 v_2^2 \right]}_{\text{solved in Eq. (12)}} + 2\underbrace{\mathbb{E}\left[ \|\mathbf{u}_a\|^4 u_1 u_2 v_1 v_2 \right]}_{\text{solved in Eq. (13)}} \right) \\
& = \frac{3\left(m^2(p+3)+m(6p+6)+8p-24\right)}{(m+2)(p-1)p(p+2)(p+4)(p+6)} + \frac{(m-1)\left(m^2(p+5)+2m(3p+13)+8(p+3)\right)}{(m+2)(p-1)p(p+2)(p+4)(p+6)} - \\
& \frac{2(m-1)\cdot(m+6)(m+4)}{(m+2)(p-1)p(p+2)(p+4)(p+6)} \\
& = \frac{(m+2)(m+4)(m(p+3)+2(p-3))}{(m+2)(p-1)p(p+2)(p+4)(p+6)} = \frac{(m+4)(m(p+3)+2(p-3))}{(p-1)p(p+2)(p+4)(p+6)}
\end{aligned}
$$

(14)

$$\mathbb{E}_{\mathbf{u}\perp\mathbf{v}}\left[\|\mathbf{u}_a\|^4\left(\mathbf{u}_b^\top\mathbf{v}_b\right)^2\right] = \mathbb{E}\left[\|\mathbf{u}_a\|^4\left(-\mathbf{u}_a^\top\mathbf{v}_a\right)^2\right] = \sum_{i,j=1}^{m}\mathbb{E}\left[\|\mathbf{u}_a\|^4 u_i u_j v_i v_j\right]$$

$$= \underbrace{\sum_{i=1}^{m}\mathbb{E}\left[\|\mathbf{u}_a\|^4 u_i^2 v_i^2\right]}_{i=j} + \underbrace{\sum_{i\neq j=1}^{m}\mathbb{E}\left[\|\mathbf{u}_a\|^4 u_i u_j v_i v_j\right]}_{i\neq j}$$

$$= m\underbrace{\mathbb{E}\left[\|\mathbf{u}_a\|^4 u_1^2 v_1^2\right]}_{\text{solved in Eq. (11)}} + m(m-1)\underbrace{\mathbb{E}\left[\|\mathbf{u}_a\|^4 u_1 u_2 v_1 v_2\right]}_{\text{solved in Eq. (13)}} \qquad (15)$$

$$= m\left(\frac{m^2(p+3) + m(6p+6) + 8p - 24}{(p-1)p(p+2)(p+4)(p+6)} - \frac{(m-1)\cdot(m+4)(m+6)}{(p-1)p(p+2)(p+4)(p+6)}\right)$$

$$= m\frac{(m+2)(m+4)(p-m)}{(p-1)p(p+2)(p+4)(p+6)}$$

$$\mathbb{E}_{\mathbf{u}\perp\mathbf{v},\mathbf{Q}_m}\left[\left(\mathbf{u}_a^\top\mathbf{Q}_m\mathbf{u}_a\right)^2\left(\mathbf{u}_b^\top\mathbf{v}_b\right)^2\right] = \mathbb{E}_{\mathbf{u}\perp\mathbf{v}}\left[\mathbb{E}_{\mathbf{r}\sim\mathcal{S}^{m-1}}\left(\|\mathbf{u}_a\|\,\mathbf{r}^\top\mathbf{u}_a\right)^2\left(\mathbf{u}_b^\top\mathbf{v}_b\right)^2\right]$$

$$= \mathbb{E}_{\mathbf{u}\perp\mathbf{v}}\left[\|\mathbf{u}_a\|^2\,\mathbb{E}_{\mathbf{r}\sim\mathcal{S}^{m-1}}\mathbf{u}_a^\top\mathbf{r}\mathbf{r}^\top\mathbf{u}_a\left(\mathbf{u}_b^\top\mathbf{v}_b\right)^2\right] = \frac{1}{m}\underbrace{\mathbb{E}_{\mathbf{u}\perp\mathbf{v}}\left[\|\mathbf{u}_a\|^4\left(\mathbf{u}_b^\top\mathbf{v}_b\right)^2\right]}_{\text{solved in Eq. (15)}} \qquad (16)$$

$$= \frac{(m+2)(m+4)(p-m)}{(p-1)p(p+2)(p+4)(p+6)}$$

$$\mathbb{E}\left[\|\mathbf{u}_a\|^2\left(\mathbf{u}_b^\top\mathbf{v}_b\right)^2\right] = \mathbb{E}\left[\sum_{i=1}^{m}u_i^2\left(\sum_{j=m+1}^{p}u_jv_j\right)^2\right] = m\mathbb{E}\left[u_1^2\sum_{j=m+1}^{p}\sum_{k=m+1}^{p}u_jv_ju_kv_k\right]$$

$$= m(p-m)\sum_{k=m+1}^{p}\mathbb{E}\left[u_1^2u_pv_pu_kv_k\right]$$

$$= m(p-m)\left(\underbrace{\mathbb{E}\left[u_1^2u_p^2v_p^2\right]}_{k=p} + \underbrace{(p-m-1)\mathbb{E}\left[u_1^2u_{p-1}v_{p-1}u_pv_p\right]}_{m+1\leq k\leq p-1}\right) \qquad (17)$$

$$= m(p-m)\left(\left\langle\begin{smallmatrix}2&0\\ \frac{2}{0}&\frac{2}{0}\end{smallmatrix}\right\rangle + (p-m-1)\left\langle\begin{smallmatrix}2&0\\ 1&1\\ \frac{1}{0}&\frac{1}{0}\end{smallmatrix}\right\rangle\right)$$

$$= m(p-m)\left(\frac{p+1}{(p-1)p(p+2)(p+4)} + \frac{-(p-m-1)}{(p-1)p(p+2)(p+4)}\right)$$

$$= \frac{(p-m)m(m+2)}{(p-1)p(p+2)(p+4)}$$

Consequently,

$$\mathbb{E}\left[\|\mathbf{u}_b\|^2\left(\mathbf{u}_a^\top\mathbf{v}_a\right)^2\right] = \frac{(p-m)m(p-m+2)}{(p-1)p(p+2)(p+4)} \qquad (18)$$

$$\mathbb{E}\left[\|\mathbf{u}_a\|^2\|\mathbf{v}_a\|^2\|\mathbf{u}_b\|^2\right] = \sum_{i=1}^{m}\sum_{j=1}^{m}\sum_{k=m+1}^{p}\mathbb{E}\left[u_i^2u_k^2v_j^2\right] = (p-m)\sum_{i=1}^{m}\sum_{j=1}^{m}\mathbb{E}\left[u_i^2u_p^2v_j^2\right]$$

$$= m(p-m)\sum_{j=1}^{m}\mathbb{E}\left[u_1^2u_p^2v_j^2\right]$$

$$= m(p-m)\left(\underbrace{\mathbb{E}\left[u_1^2u_p^2v_1^2\right]}_{j=1} + \underbrace{(m-1)\mathbb{E}\left[u_1^2u_p^2v_2^2\right]}_{j\geq 2}\right) \qquad (19)$$

$$= m(p-m)\left(\left\langle\begin{smallmatrix}2&2\\ \frac{2}{0}&\frac{0}{0}\end{smallmatrix}\right\rangle + (m-1)\left\langle\begin{smallmatrix}2&0\\ 0&2\\ 2&0\\ \frac{2}{0}&\frac{0}{0}\end{smallmatrix}\right\rangle\right)$$

$$= m(p-m)\left(\frac{p+1+(m-1)(p+3)}{(p-1)p(p+2)(p+4)}\right)$$

$$= \frac{m(p-m)(m(p+3)-2)}{(p-1)p(p+2)(p+4)}$$

$$\mathbb{E}\left[\left\|\mathbf{u}_a\right\|^4\left\|\mathbf{v}_a\right\|^4\right] = \mathbb{E}\left[\left\|\mathbf{u}_a\right\|^4\left\|\mathbf{v}_a\right\|^4\right] = \mathbb{E}_{\mathbf{u}\perp\mathbf{v}}\left[\left(\sum_{i=1}^m u_i^2\right)^2\left(\sum_{i=1}^m v_i^2\right)^2\right]$$

$$= \mathbb{E}\left[\left(\sum_{i=1}^m u_i^4 + \sum_{i\neq j} u_i^2 u_j^2\right)\left(\sum_{k=1}^m v_k^4 + \sum_{k\neq\ell} v_k^2 v_\ell^2\right)\right]$$

$$= \sum_{i=1}^m\sum_{k=1}^m \mathbb{E}\left[u_i^4 v_k^4\right] + \underbrace{\sum_{i=1}^m\sum_{k\neq\ell}\mathbb{E}\left[u_i^4 v_k^2 v_\ell^2\right] + \sum_{k=1}^m\sum_{i\neq j}\mathbb{E}\left[u_i^2 u_j^2 v_k^4\right]}_{\text{same, due to the identical distributions (see Prop. 9)}} + \sum_{i\neq j}\sum_{k\neq\ell}\mathbb{E}\left[u_i^2 u_j^2 v_k^2 v_\ell^2\right]$$

$$= \sum_{i=1}^m\sum_{k=1}^m \mathbb{E}\left[u_i^4 v_k^4\right] + 2\sum_{i=1}^m\sum_{k\neq\ell}\mathbb{E}_{\mathbf{u}\perp\mathbf{v}}\left[u_i^4 v_k^2 v_\ell^2\right] + \sum_{i\neq j}\sum_{k\neq\ell}\mathbb{E}_{\mathbf{u}\perp\mathbf{v}}\left[u_i^2 u_j^2 v_k^2 v_\ell^2\right]$$

$$= m\sum_{k=1}^m \mathbb{E}\left[u_1^4 v_k^4\right] + 2m\sum_{k\neq\ell}\mathbb{E}\left[u_1^4 v_k^2 v_\ell^2\right] + m(m-1)\sum_{k\neq\ell}\mathbb{E}\left[u_1^2 u_2^2 v_k^2 v_\ell^2\right]$$

$$= m\left(\underbrace{\mathbb{E}\left[u_1^4 v_1^4\right]}_{k=1} + \underbrace{(m-1)\mathbb{E}\left[u_1^4 v_2^4\right]}_{k\geq 2}\right) + 2m\left(\underbrace{2(m-1)\mathbb{E}\left[u_1^4 v_1^2 v_2^2\right]}_{k=1<\ell\,\vee\,\ell=1<k} + \underbrace{(m-1)(m-2)\mathbb{E}\left[u_1^4 v_2^2 v_3^2\right]}_{k\neq\ell\geq 2}\right)$$

$$+ m(m-1)\left(\underbrace{2\mathbb{E}\left[u_1^2 u_2^2 v_1^2 v_2^2\right]}_{k=1,\ell=2\,\vee\,\ell=1,k=2} + \underbrace{4(m-2)\mathbb{E}\left[u_1^2 u_2^2 v_2^2 v_3^2\right]}_{k\leq 2,\ell\geq 3\,\vee\,\ell\leq 2,k\geq 3} + \underbrace{(m-2)(m-3)\mathbb{E}\left[u_1^2 u_2^2 v_3^2 v_4^2\right]}_{k\neq\ell\geq 3}\right)$$

$$= m\left(\left\langle {}^{4}_{0}\,{}^{4}_{0}\right\rangle + (m-1)\left\langle {}^{4}_{0}\,{}^{0}_{4}\right\rangle\right) + 2m\left(2(m-1)\left\langle {}^{4}_{0}\,{}^{2}_{0}\right\rangle + (m-1)(m-2)\left\langle {}^{4}_{0}\,{}^{0}_{2}\,{}^{0}_{2}\right\rangle\right) +$$

$$m(m-1)\left(2\left\langle {}^{2}_{2}\,{}^{2}_{0}\right\rangle + 4(m-2)\left\langle {}^{2}_{0}\,{}^{0}_{2}\right\rangle + (m-2)(m-3)\left\langle {}^{2}_{2}\,{}^{0}_{0}\,{}^{0}_{2}\right\rangle\right)$$

$$= \tfrac{9m}{p(p+2)(p+4)(p+6)} + \tfrac{9m(m-1)(p+3)(p+5)}{(p-1)p(p+1)(p+2)(p+4)(p+6)} + 2m\tfrac{6(m-1)(p+1)(p+3)+3(m-1)(m-2)(p+3)(p+5)}{(p-1)p(p+1)(p+2)(p+4)(p+6)} +$$

$$m(m-1)\left(\tfrac{2(p^2+4p+15)+4(m-2)(p+3)^2}{(p-1)p(p+1)(p+2)(p+4)(p+6)} + \tfrac{(m-2)(m-3)(p+3)(p+5)}{(p-1)p(p+1)(p+2)(p+4)(p+6)}\right)$$

$$= \tfrac{9m(p-1)(p+1)+9m(m-1)(p+3)(p+5)+2m(6(m-1)(p+1)(p+3)+3(m-1)(m-2)(p+3)(p+5))}{(p-1)p(p+1)(p+2)(p+4)(p+6)} +$$

$$\tfrac{m(m-1)\left(2(p^2+4p+15)+4(m-2)(p+3)^2+(m-2)(m-3)(p+3)(p+5)\right)}{(p-1)p(p+1)(p+2)(p+4)(p+6)}$$

$$= \frac{m(m+2)\left(m^2(p+3)(p+5)+2m(p+1)(p+3)-8(2p+3)\right)}{(p-1)p(p+1)(p+2)(p+4)(p+6)}$$

$$\tag{20}$$

$$\mathbb{E}\left[\|\mathbf{u}_a\|^2 \|\mathbf{v}_a\|^2\right] = \sum_{i,j=1}^m \mathbb{E}\left[u_i^2 v_j^2\right] = m \sum_{j=1}^m \mathbb{E}\left[u_1^2 v_j^2\right] = m \underbrace{\mathbb{E}\left[u_1^2 v_1^2\right]}_{j=1} + m \underbrace{(m-1)\,\mathbb{E}\left[u_1^2 v_2^2\right]}_{j\geq 2}$$

$$= m \left\langle \begin{smallmatrix} 2 & 2 \\ 0 & 0 \end{smallmatrix} \right\rangle + m\,(m-1) \left\langle \begin{smallmatrix} 2 & 0 \\ 0 & 2 \end{smallmatrix} \right\rangle = \frac{m}{p\,(p+2)} + \frac{m\,(m-1)\,(p+1)}{(p-1)\,p\,(p+2)} \qquad (21)$$

$$= \frac{m\,(mp + m - 2)}{(p-1)\,p\,(p+2)}$$

$$\mathbb{E}_{\mathbf{u}\perp\mathbf{v},\mathbf{Q}_m}\left(\mathbf{u}_a^\top \mathbf{Q}_m \mathbf{v}_a\right)^2 = \mathbb{E}_{\mathbf{u}\perp\mathbf{v},\mathbf{r}\sim\mathcal{S}^{m-1}}\left(\|\mathbf{u}_a\|\,\mathbf{r}^\top \mathbf{v}_a\right)^2$$

$$= \mathbb{E}_{\mathbf{u}\perp\mathbf{v}}\left[\|\mathbf{u}_a\|^2\,\mathbb{E}_{\mathbf{r}\sim\mathcal{S}^{m-1}}\left(\mathbf{v}_a^\top \mathbf{r}\mathbf{r}^\top \mathbf{v}_a\right)\right] \qquad (22)$$

$$= \frac{1}{m}\underbrace{\mathbb{E}_{\mathbf{u}\perp\mathbf{v}}\left[\|\mathbf{u}_a\|^2 \|\mathbf{v}_a\|^2\right]}_{\text{solved in Eq. (21)}} = \frac{mp + m - 2}{(p-1)\,p\,(p+2)}$$

$$\mathbb{E}\left(\mathbf{u}_a^\top \mathbf{v}_a\right)^2 = \mathbb{E}_{\mathbf{u}\perp\mathbf{v}}\left(-\mathbf{u}_b^\top \mathbf{v}_b\right)^2 = \mathbb{E}_{\mathbf{u}\perp\mathbf{v}}\left(\mathbf{u}_b^\top \mathbf{v}_b\right)^2 = \sum_{i,j=1}^m \mathbb{E}\left[u_i u_j v_i v_j\right]$$

$$= m \sum_{j=1}^m \mathbb{E}\left[u_1 u_j v_1 v_j\right] = m\left(\mathbb{E}\left[u_1^2 v_1^2\right] + (m-1)\,\mathbb{E}\left[u_1 u_2 v_1 v_2\right]\right)$$

$$= m\left(\left\langle \begin{smallmatrix} 2 & 2 \\ 0 & 0 \end{smallmatrix} \right\rangle + (m-1) \left\langle \begin{smallmatrix} 1 & 1 \\ 0 & 0 \end{smallmatrix} \right\rangle\right) = m\left(\frac{1}{p\,(p+2)} + \frac{-1\cdot(m-1)}{(p-1)\,p\,(p+2)}\right) \qquad (23)$$

$$= \frac{m\,(p - 1 + 1 - m)}{(p-1)\,p\,(p+2)} = \frac{m\,(p - m)}{(p-1)\,p\,(p+2)}$$

$$
\mathbb{E}_{\mathbf{u}\perp\mathbf{v}}\left(\mathbf{u}_a^\top\mathbf{v}_a\right)^4 = \mathbb{E}_{\mathbf{u}\perp\mathbf{v}}\left(-\mathbf{u}_b^\top\mathbf{v}_b\right)^4 = \mathbb{E}\left(\mathbf{u}_a^\top\mathbf{v}_a\right)^2\left(\mathbf{u}_b^\top\mathbf{v}_b\right)^2
$$

$$
= \mathbb{E}\left(\sum_{i=1}^m u_i v_i\right)^2\left(\sum_{k=m+1}^p u_k v_k\right)^2
$$

$$
= \sum_{i,j=1}^m \sum_{k,\ell=m+1}^p \mathbb{E}\left[u_i v_i u_j v_j u_k v_k u_\ell v_\ell\right] = m(p-m)\sum_{i=1}^m\sum_{k=m+1}^p \mathbb{E}\left[u_1 v_1 u_i v_i u_k v_k u_p v_p\right]
$$

$$
= m(p-m)\bigg(\underbrace{\mathbb{E}\left[u_1^2 v_1^2 u_p^2 v_p^2\right]}_{i=1,k=p} + \underbrace{(p-m-1)\,\mathbb{E}\left[u_1^2 v_1^2 u_{p-1} v_{p-1} u_p v_p\right]}_{i=1,m+1\le k\le p-1} +
$$

$$
\underbrace{(m-1)\,\mathbb{E}\left[u_1 v_1 u_2 v_2 u_p^2 v_p^2\right]}_{i\ge 2,k=p} + \underbrace{(p-m-1)(m-1)\,\mathbb{E}[u_1 v_1 u_2 v_2 u_{p-1} v_{p-1} u_p v_p]}_{i\ge 2,m+1\le k\le p-1}\bigg) \quad (24)
$$

$$
= m(p-m)\left(\left\langle \begin{smallmatrix}2&2\\2&2\\\vec{0}&\vec{0}\end{smallmatrix}\right\rangle + \big((p-m-1)+(m-1)\big)\left\langle \begin{smallmatrix}2&2\\1&1\\\vec{0}&\vec{0}\end{smallmatrix}\right\rangle + (p-m-1)(m-1)\left\langle \begin{smallmatrix}1&1\\1&1\\\vec{1}&\vec{1}\\\vec{0}&\vec{0}\end{smallmatrix}\right\rangle\right)
$$

$$
= m(p-m)\left(\left\langle \begin{smallmatrix}2&2\\2&2\\\vec{0}&\vec{0}\end{smallmatrix}\right\rangle + (p-2)\left\langle \begin{smallmatrix}2&2\\1&1\\\vec{0}&\vec{0}\end{smallmatrix}\right\rangle + (p-m-1)(m-1)\left\langle \begin{smallmatrix}1&1\\1&1\\1&1\\\vec{0}&\vec{0}\end{smallmatrix}\right\rangle\right)
$$

$$
= m(p-m)\left(\frac{(p^2+4p+15)+(p-2)(-p+3)+3(p-m-1)(m-1)}{(p-1)\,p\,(p+1)\,(p+2)\,(p+4)\,(p+6)}\right)
$$

$$
= \frac{3m^4 - 6m^3 p + 3m^2 p^2 - 6m^2 p - 12m^2 + 6mp^2 + 12mp}{(p-1)\,p\,(p+1)\,(p+2)\,(p+4)\,(p+6)}
$$

$$
\mathbb{E}_{\mathbf{u}\perp\mathbf{v}}\left(\mathbf{u}_a^\top\mathbf{Q}_m\mathbf{v}_a\right)^4 = \mathbb{E}_{\mathbf{u}\perp\mathbf{v},\mathbf{r}\sim\mathcal{S}^{m-1}}\left(\|\mathbf{u}_a\|\,\mathbf{r}^\top\mathbf{v}_a\right)^4 = \mathbb{E}_{\mathbf{u}\perp\mathbf{v}}\left[\|\mathbf{u}_a\|^4\,\mathbb{E}_{\mathbf{r}\sim\mathcal{S}^{m-1}}\left(\mathbf{r}^\top\mathbf{v}_a\right)^4\right]
$$

$$
\left[\begin{smallmatrix}\text{reparameterize } \mathbf{r}\mapsto \mathbf{A}^\top\mathbf{r}\\\text{where } \mathbf{A}\mathbf{v}_a = \mathbf{e}_1\end{smallmatrix}\right] = \mathbb{E}_{\mathbf{u}\perp\mathbf{v}}\left[\|\mathbf{u}_a\|^4\,\mathbb{E}_{\mathbf{r}\sim\mathcal{S}^{m-1}}\left(\|\mathbf{v}_a\|\,\mathbf{r}^\top\mathbf{e}_1\right)^4\right]
$$

$$
= \mathbb{E}_{\mathbf{u}\perp\mathbf{v}}\left[\|\mathbf{u}_a\|^4\,\|\mathbf{v}_a\|^4\,\mathbb{E}_{\mathbf{r}\sim\mathcal{S}^{m-1}} r_1^4\right] = \left\langle\begin{smallmatrix}4\\\vec{0}\end{smallmatrix}\right\rangle_m \mathbb{E}\left[\|\mathbf{u}_a\|^4\,\|\mathbf{v}_a\|^4\right] \quad (25)
$$

$$
= \tfrac{3}{m(m+2)}\cdot\frac{m(m+2)\big(m^2(p+3)(p+5)+2m(p+1)(p+3)-8(2p+3)\big)}{(p-1)p(p+1)(p+2)(p+4)(p+6)}
$$

$$
= \frac{3\left(m^2(p+3)(p+5)+2m(p+1)(p+3)-8(2p+3)\right)}{(p-1)\,p\,(p+1)\,(p+2)\,(p+4)\,(p+6)}\,.
$$

$$\mathbb{E}\left(\mathbf{u}_a^\top \mathbf{Q}_m \mathbf{v}_a\right)^2 \left(\mathbf{u}_b^\top \mathbf{v}_b\right)^2 = \mathbb{E}\left(\mathbf{u}_a^\top \mathbf{Q}_m \mathbf{v}_a\right)^2 \left(\mathbf{u}_a^\top \mathbf{v}_a\right)^2$$

$$= \mathbb{E}_{\mathbf{u}\perp\mathbf{v}}\left[\left(\mathbf{u}_a^\top \mathbf{v}_a\right)^2 \mathbb{E}_{\mathbf{r}\sim\mathcal{S}^{p-1}}\left(\|\mathbf{u}_a\|\,\mathbf{r}^\top \mathbf{v}_a\right)^2\right]$$

$$= \frac{1}{m}\mathbb{E}_{\mathbf{u}\perp\mathbf{v}}\left[\left(\mathbf{u}_a^\top \mathbf{v}_a\right)^2 \|\mathbf{u}_a\|^2 \|\mathbf{v}_a\|^2\right] = \frac{1}{m}\mathbb{E}_{\mathbf{u}\perp\mathbf{v}}\left[\left(-\mathbf{u}_b^\top \mathbf{v}_b\right)^2 \|\mathbf{u}_a\|^2 \|\mathbf{v}_a\|^2\right]$$

$$= \frac{1}{m}\mathbb{E}\left[\|\mathbf{u}_a\|^2 \|\mathbf{v}_a\|^2 \left(\mathbf{u}_b^\top \mathbf{v}_b\right)^2\right] = \frac{1}{m}\sum_{i,j=1}^{m}\sum_{k,\ell=m+1}^{p}\mathbb{E}\left[u_i^2 v_j^2 u_k v_k u_\ell v_\ell\right]$$

$$= \sum_{j=1}^{m}\sum_{k,\ell=m+1}^{p}\mathbb{E}\left[u_1^2 v_j^2 u_k v_k u_\ell v_\ell\right] = (p-m)\sum_{j=1}^{m}\sum_{k=m+1}^{p}\mathbb{E}\left[u_1^2 v_j^2 u_k v_k u_p v_p\right]$$

$$= (p-m)\left(\underbrace{\mathbb{E}\left[u_1^2 v_1^2 u_p^2 v_p^2\right]}_{j=1,k=p} + \underbrace{(p-m-1)\,\mathbb{E}\left[u_1^2 v_1^2 u_{p-1} v_{p-1} u_p v_p\right]}_{j=1,m+1\le k\le p-1} + \right. \tag{26}$$

$$\left. \underbrace{(m-1)\,\mathbb{E}\left[u_1^2 v_2^2 u_p^2 v_p^2\right]}_{j\ge 2,k=p} + \underbrace{(p-m-1)(m-1)\,\mathbb{E}\left[u_1^2 v_2^2 u_{p-1} v_{p-1} u_p v_p\right]}_{j\ge 2,m+1\le k\le p-1}\right)$$

$$= (p-m)\left(\left\langle \begin{smallmatrix} 2 & 2 \\ 2 & 2 \\ 0 & 0 \end{smallmatrix} \right\rangle + (p-m-1)\left\langle \begin{smallmatrix} 2 & 2 \\ 1 & 1 \\ 1 & 1 \\ 0 & 0 \end{smallmatrix} \right\rangle + (m-1)\left\langle \begin{smallmatrix} 2 & 0 \\ 0 & 2 \\ 2 & 2 \\ 0 & 0 \end{smallmatrix} \right\rangle + \right.$$

$$\left. (p-m-1)(m-1)\left\langle \begin{smallmatrix} 2 & 0 \\ 0 & 2 \\ 1 & 1 \\ 1 & 1 \\ 0 & 0 \end{smallmatrix} \right\rangle\right)$$

$$= (p-m)\left(\frac{p^2+4p+15+(p-m-1)(-p+3)+(m-1)(p+3)^2+(m-1)(p-m-1)(-p-3)}{(p-1)p(p+1)(p+2)(p+4)(p+6)}\right)$$

$$= \frac{(m+2)(mp+2p+3m)(p-m)}{(p-1)p(p+1)(p+2)(p+4)(p+6)}$$

$$\mathbb{E}\left[\|\mathbf{u}_b\|^2 \mathbf{v}_a^\top \mathbf{Q}_m \mathbf{u}_a \mathbf{u}_a^\top \mathbf{Q}_m \mathbf{v}_a\right] = \mathbb{E}\left[\|\mathbf{u}_b\|^2 \left(\sum_{i,j=1}^{m} v_i q_{ij} u_j\right)\left(\sum_{k,\ell=1}^{m} u_k q_{k\ell} v_\ell\right)\right]$$

$$= \sum_{i,j=1}^{m}\sum_{k,\ell=1}^{m}\mathbb{E}\left[\|\mathbf{u}_b\|^2 v_i u_j u_k v_\ell\right]\mathbb{E}\left[q_{ij}q_{k\ell}\right]$$

$$\overset{[\text{Prop. }10]}{=} \sum_{i,j=1}^{m}\mathbb{E}\left[\|\mathbf{u}_b\|^2 u_i u_j v_i v_j\right]\mathbb{E}\left[q_{ij}^2\right]$$

$$= \frac{1}{m}\left(\underbrace{m\mathbb{E}\left[\|\mathbf{u}_b\|^2 u_1^2 v_1^2\right]}_{i=j} + \underbrace{m(m-1)\mathbb{E}\left[\|\mathbf{u}_b\|^2 u_1 u_2 v_1 v_2\right]}_{i\ne j}\right) \tag{27}$$

$$= \mathbb{E}\left[\left(\sum_{k=m+1}^{p} u_k^2\right)u_1^2 v_1^2\right] + (m-1)\mathbb{E}\left[\left(\sum_{k=m+1}^{p} u_k^2\right)u_1 u_2 v_1 v_2\right]$$

$$= (p-m)\mathbb{E}\left[u_p^2 u_1^2 v_1^2\right] + (m-1)(p-m)\mathbb{E}\left[u_p^2 u_1 u_2 v_1 v_2\right]$$

$$= (p-m)\left\langle \begin{smallmatrix} 2 & 2 \\ 2 & 0 \\ 0 & 0 \end{smallmatrix} \right\rangle + (m-1)(p-m)\left\langle \begin{smallmatrix} 1 & 1 \\ 1 & 1 \\ 2 & 0 \\ 0 & 0 \\ 0 & 0 \end{smallmatrix} \right\rangle$$

$$= \frac{(p-m)(p+1)-(m-1)(p-m)}{(p-1)p(p+2)(p+4)} = \frac{(p-m)(p-m+2)}{(p-1)p(p+2)(p+4)}$$

$$\mathbb{E}\left[\left\|\mathbf{u}_b\right\|^4 \left(\mathbf{u}_a^\top \mathbf{Q}_m \mathbf{v}_a\right)^2\right] = \mathbb{E}_{\mathbf{u}\perp\mathbf{v}}\left[\left\|\mathbf{u}_b\right\|^4 \mathbf{v}_a^\top \mathbb{E}_{\mathbf{Q}_m}\left(\mathbf{Q}_m \mathbf{u}_a \mathbf{u}_a^\top \mathbf{Q}_m\right)\mathbf{v}_a\right]$$

$$= \mathbb{E}_{\mathbf{u}\perp\mathbf{v}}\left[\left\|\mathbf{u}_b\right\|^4 \mathbf{v}_a^\top \mathbb{E}_{\mathbf{r}\sim\mathcal{S}^{m-1}}\left(\left\|\mathbf{u}_a\right\|^2 \mathbf{r}\mathbf{r}^\top\right)\mathbf{v}_a\right] = \frac{1}{m}\mathbb{E}_{\mathbf{u}\perp\mathbf{v}}\left[\left\|\mathbf{u}_a\right\|^2 \left\|\mathbf{u}_b\right\|^4 \mathbf{v}_a^\top \mathbf{v}_a\right]$$

$$= \frac{1}{m}\mathbb{E}\left[\left\|\mathbf{u}_b\right\|^4 \left\|\mathbf{u}_a\right\|^2 \left\|\mathbf{v}_a\right\|^2\right] = \frac{1}{m}\mathbb{E}\left[\left(\sum_{i=m+1}^p u_i^2\right)\left(\sum_{j=m+1}^p u_j^2\right)\left(\sum_{k=1}^m u_k^2\right)\left(\sum_{\ell=1}^m v_\ell^2\right)\right]$$

$$= \frac{p-m}{m}\mathbb{E}\left[u_p^2\left(\sum_{j=m+1}^p u_j^2\right)\left(\sum_{k=1}^m u_k^2\right)\left(\sum_{\ell=1}^m v_\ell^2\right)\right]$$

$$= \frac{p-m}{m}\sum_{j=m+1}^p \mathbb{E}\left[u_p^2 u_j^2\left(\sum_{k=1}^m u_k^2\right)\left(\sum_{i=1}^m v_\ell^2\right)\right] = (p-m)\sum_{j=m+1}^p \mathbb{E}\left[u_1^2 u_p^2 u_j^2\left(\sum_{i=1}^m v_\ell^2\right)\right]$$

$$= (p-m)\left(\underbrace{\mathbb{E}\left[u_1^2 u_p^4\left(\sum_{i=1}^m v_\ell^2\right)\right]}_{j=p} + (p-m-1)\underbrace{\mathbb{E}\left[u_1^2 u_{p-1}^2 u_p^2\left(\sum_{i=1}^m v_\ell^2\right)\right]}_{m+1\leq j\leq p-1}\right)$$

$$= (p-m)\left(\left(\underbrace{\mathbb{E}\left[u_1^2 u_p^4 v_1^2\right]}_{\ell=1} + (m-1)\underbrace{\mathbb{E}\left[u_1^2 u_p^4 v_2^2\right]}_{2\leq\ell\leq m}\right)+\right. \tag{28}$$

$$(p-m-1)\left(\underbrace{\mathbb{E}\left[u_1^2 u_{p-1}^2 u_p^2 v_1^2\right]}_{\ell=1} + (m-1)\underbrace{\mathbb{E}\left[u_1^2 u_{p-1}^2 u_p^2 v_2^2\right]}_{2\leq\ell\leq m}\right)\Big)$$

$$= (p-m)\left(\mathbb{E}\left[u_1^2 u_2^4 v_1^2\right] + (m-1)\mathbb{E}\left[u_1^2 u_2^4 v_3^2\right] +\right.$$

$$(p-m-1)\left(\mathbb{E}\left[u_1^2 u_2^2 u_3^2 v_1^2\right] + (m-1)\mathbb{E}\left[u_1^2 u_2^2 u_3^2 v_4^2\right]\right)\Big)$$

$$= (p-m)\left(\left\langle \begin{smallmatrix} 2 & 2 \\ 4 & 0 \\ \overrightarrow{0} & \overrightarrow{0} \end{smallmatrix}\right\rangle + (m-1)\left\langle \begin{smallmatrix} 2 & 0 \\ 4 & 0 \\ 0 & 2 \\ \overrightarrow{0} & \overrightarrow{0} \end{smallmatrix}\right\rangle +\right.$$

$$(p-m-1)\left(\left\langle \begin{smallmatrix} 2 & 2 \\ 2 & 0 \\ 2 & 0 \\ \overrightarrow{0} & \overrightarrow{0} \end{smallmatrix}\right\rangle + (m-1)\left\langle \begin{smallmatrix} 2 & 0 \\ 2 & 0 \\ 2 & 0 \\ 0 & 2 \\ \overrightarrow{0} & \overrightarrow{0} \end{smallmatrix}\right\rangle\right)\Big)$$

$$= (p-m)\left(\left(\frac{3(p+3)+3(m-1)(p+5)}{(p-1)p(p+2)(p+4)(p+6)}\right) + (p-m-1)\left(\frac{p+3+(m-1)(p+5)}{(p-1)p(p+2)(p+4)(p+6)}\right)\right)$$

$$= \frac{(p-m)\left(mp^2 - m^2 p - 5m^2 + 7mp + 12m - 2p - 4\right)}{(p-1)\, p\, (p+2)\, (p+4)\, (p+6)}$$

$$\mathbb{E}\left[\left\|\mathbf{u}_a\right\|^2 \mathbf{u}_a^\top \mathbf{v}_a \cdot \mathbf{v}_b^\top \mathbf{u}_b \left\|\mathbf{v}_b\right\|^2\right] = \sum_{i=1}^m \sum_{j=1}^m \sum_{k=m+1}^p \sum_{\ell=m+1}^p \mathbb{E}\left[u_i^2 \cdot u_j v_j \cdot u_k v_k \cdot v_\ell^2\right]$$

$$= m\,(p-m)\sum_{j=1}^m \sum_{k=m+1}^p \mathbb{E}\left[u_1^2 \cdot u_j v_j \cdot u_k v_k \cdot v_p^2\right]$$

$$= m\,(p-m)\left(\sum_{k=m+1}^p \mathbb{E}\left[u_1^3 v_1 \cdot u_k v_k \cdot v_p^2\right] + (m-1)\sum_{k=m+1}^p \mathbb{E}\left[u_1^2 \cdot u_2 v_2 \cdot u_k v_k \cdot v_p^2\right]\right)$$

$$= m\,(p-m)\left(\mathbb{E}\left[u_1^3 v_1 u_p v_p^3\right] + (p-m-1)\,\mathbb{E}\left[u_1^3 v_1 u_{p-1} v_{p-1} v_p^2\right] + \right.$$

$$\left. (m-1)\left(\mathbb{E}\left[u_1^2 u_2 v_2 u_p v_p^3\right] + (p-m-1)\,\mathbb{E}\left[u_1^2 u_2 v_2 u_{p-1} v_{p-1} v_p^2\right]\right)\right)$$

$$= m\,(p-m)\left(\left\langle {\substack{3\\1\\ \overrightarrow{0}}} \; {\substack{1\\3\\ \overrightarrow{0}}} \right\rangle + (p-m-1)\left\langle {\substack{3\\1\\0\\ \overrightarrow{0}}} \; {\substack{1\\1\\2\\ \overrightarrow{0}}} \right\rangle + \right. \tag{29}$$

$$\left. (m-1)\left(\left\langle {\substack{2\\1\\1\\ \overrightarrow{0}}} \; {\substack{0\\1\\3\\ \overrightarrow{0}}} \right\rangle + (p-m-1)\left\langle {\substack{2\\1\\1\\0\\ \overrightarrow{0}}} \; {\substack{0\\1\\1\\2\\ \overrightarrow{0}}} \right\rangle\right)\right)$$

$$= m\,(p-m)\left(\left\langle {\substack{3\\1\\ \overrightarrow{0}}} \; {\substack{1\\3\\ \overrightarrow{0}}} \right\rangle + (p-2)\left\langle {\substack{3\\1\\0\\ \overrightarrow{0}}} \; {\substack{1\\1\\2\\ \overrightarrow{0}}} \right\rangle + (m-1)(p-m-1)\left\langle {\substack{2\\1\\1\\0\\ \overrightarrow{0}}} \; {\substack{0\\1\\1\\2\\ \overrightarrow{0}}} \right\rangle\right)$$

$$= m\,(p-m)\left(\frac{-9(p+3)}{(p-1)p(p+1)(p+2)(p+4)(p+6)}\right.$$

$$\left. + \frac{-3(p-2)(p+3)}{(p-1)p(p+1)(p+2)(p+4)(p+6)} + \frac{(m-1)(p-m-1)(-p-3)}{(p-1)p(p+1)(p+2)(p+4)(p+6)}\right)$$

$$= \frac{m\,(p-m)\,(p+3)\,\left(m^2 - mp - 2\,(p+2)\right)}{(p-1)\,p\,(p+1)\,(p+2)\,(p+4)\,(p+6)}$$

$$\mathbb{E}\left[\mathbf{u}_a^\top \mathbf{Q}_m \mathbf{u}_a \mathbf{v}_a^\top \mathbf{Q}_m \mathbf{v}_a \left(\mathbf{v}_b^\top \mathbf{u}_b\right)^2\right]$$

$$= \mathbb{E}\left[\left(\mathbf{v}_b^\top \mathbf{u}_b\right)^2 \sum_{i,j,k,\ell} u_i q_{ij} u_j v_k q_{k\ell} v_\ell\right] = \sum_{i,j,k,\ell=1}^{m} \mathbb{E}\left[\left(\mathbf{v}_b^\top \mathbf{u}_b\right)^2 u_i u_j v_k v_\ell\right]\mathbb{E}\left[q_{ij}q_{k\ell}\right]$$

[By Prop. 10, most summands are zero]

$$= \sum_{i,j=1}^{m} \mathbb{E}\left[\left(\mathbf{v}_b^\top \mathbf{u}_b\right)^2 u_i u_j v_i v_j\right]\mathbb{E}\left[q_{ij}^2\right] = \frac{1}{m}\sum_{i,j=1}^{m}\mathbb{E}\left[\left(\sum_{k=m+1}^{p} u_k v_k\right)^2 u_i u_j v_i v_j\right]$$

$$= \frac{1}{m}\sum_{i,j=1}^{m}\sum_{k,\ell=m+1}^{p}\mathbb{E}\left[u_i u_j u_k u_\ell v_i v_j v_k v_\ell\right] = \frac{m\,(p-m)}{m}\sum_{j=1}^{m}\sum_{\ell=m+1}^{p}\mathbb{E}\left[u_1 u_j u_p u_\ell v_1 v_j v_p v_\ell\right]$$

$$= (p-m)\sum_{j=1}^{m}\sum_{\ell=m+1}^{p}\mathbb{E}\left[u_1 u_j u_p u_\ell v_1 v_j v_p v_\ell\right]$$

$$= (p-m)\left(\sum_{\ell=m+1}^{p}\mathbb{E}\left[u_1^2 u_p u_\ell v_1^2 v_p v_\ell\right] + (m-1)\sum_{\ell=m+1}^{p}\mathbb{E}\left[u_1 u_2 u_p u_\ell v_1 v_2 v_p v_\ell\right]\right)$$

$$= (p-m)\left(\mathbb{E}\left[u_1^2 u_p^2 v_1^2 v_p^2\right] + (p-m-1)\mathbb{E}\left[u_1^2 u_{p-1} u_p v_1^2 v_{p-1} v_p\right] + \right.$$

$$\left. (m-1)\left(\mathbb{E}\left[u_1 u_2 u_p^2 v_1 v_2 v_p^2\right] + (p-m-1)\mathbb{E}\left[u_1 u_2 u_{p-1} u_p v_1 v_2 v_{p-1} v_p\right]\right)\right)$$

$$= (p-m)\left(\left\langle \begin{smallmatrix} 2 & 2 \\ 2 & 2 \\ \vec{0} & \vec{0} \end{smallmatrix}\right\rangle + (p-m-1)\left\langle \begin{smallmatrix} 2 & 2 \\ 1 & 1 \\ \vec{1} & \vec{1} \\ \vec{0} & \vec{0} \end{smallmatrix}\right\rangle + \right.$$

$$\left. (m-1)\left(\left\langle \begin{smallmatrix} 1 & 1 \\ 1 & 1 \\ 2 & 2 \\ \vec{0} & \vec{0} \end{smallmatrix}\right\rangle + (p-m-1)\left\langle \begin{smallmatrix} 1 & 1 \\ 1 & 1 \\ 1 & 1 \\ \vec{0} & \vec{0} \end{smallmatrix}\right\rangle\right)\right) \tag{30}$$

$$= (p-m)\left(\left\langle \begin{smallmatrix} 2 & 2 \\ 2 & 2 \\ \vec{0} & \vec{0} \end{smallmatrix}\right\rangle + (p-2)\left\langle \begin{smallmatrix} 2 & 2 \\ 1 & 1 \\ 1 & 1 \\ \vec{0} & \vec{0} \end{smallmatrix}\right\rangle + (m-1)(p-m-1)\left\langle \begin{smallmatrix} 1 & 1 \\ 1 & 1 \\ 1 & 1 \\ \vec{0} & \vec{0} \end{smallmatrix}\right\rangle\right)$$

$$= (p-m)\left(\frac{p^2+4p+15}{(p-1)p(p+1)(p+2)(p+4)(p+6)} + \frac{(p-2)(-p+3)+(m-1)(p-m-1)\cdot 3}{(p-1)p(p+1)(p+2)(p+4)(p+6)}\right)$$

$$= \frac{3\,(p-m)\left(-m^2+mp+2p+4\right)}{(p-1)\,p\,(p+1)\,(p+2)\,(p+4)\,(p+6)}$$

$$\mathbb{E}\left[\mathbf{u}_b^\top \mathbf{u}_b \left(\mathbf{v}_a^\top \mathbf{u}_a\right)^2 \mathbf{v}_b^\top \mathbf{v}_b\right] = \mathbb{E}\left[\|\mathbf{u}_b\|^2 \left(\mathbf{v}_a^\top \mathbf{u}_a\right)^2 \|\mathbf{v}_b\|^2\right]$$

$$= \mathbb{E}\left[\left(1-\|\mathbf{u}_a\|^2\right)\left(\mathbf{v}_a^\top \mathbf{u}_a\right)^2 \|\mathbf{v}_b\|^2\right]$$

$$= \mathbb{E}\left[\|\mathbf{v}_b\|^2 \left(\mathbf{v}_a^\top \mathbf{u}_a\right)^2\right] - \mathbb{E}\left[\|\mathbf{u}_a\|^2 \|\mathbf{v}_b\|^2 \left(\mathbf{v}_a^\top \mathbf{u}_a\right)^2\right]$$

$$= \underbrace{\mathbb{E}\left[\|\mathbf{v}_b\|^2 \left(\mathbf{v}_a^\top \mathbf{u}_a\right)^2\right]}_{\text{solved in Eq. (18)}} + \underbrace{\mathbb{E}\left[\|\mathbf{u}_a\|^2 \|\mathbf{v}_b\|^2 \cdot \mathbf{u}_a^\top \mathbf{v}_a \cdot \mathbf{v}_b^\top \mathbf{u}_b\right]}_{\text{solved in Eq. (29)}} \tag{31}$$

$$= \frac{(p-m)m(p-m+2)}{(p-1)p(p+2)(p+4)} + \frac{m(p-m)(p+3)\left(m^2-mp-2(p+2)\right)}{(p-1)p(p+1)(p+2)(p+4)(p+6)}$$

$$= \frac{(p-m)m\left(m^2(p+3)-2m\left(p^2+5p+3\right)+p\left(p^2+7p+10\right)\right)}{(p-1)p(p+1)(p+2)(p+4)(p+6)}$$

$$\mathbb{E}\left[\mathbf{u}_b^\top \mathbf{u}_b \mathbf{u}_a^\top \mathbf{Q}_m \mathbf{v}_a \mathbf{v}_a^\top \mathbf{Q}_m \mathbf{u}_a \mathbf{v}_b^\top \mathbf{v}_b\right] = \sum_{i,j=1}^{m} \sum_{k,\ell=1}^{m} \mathbb{E}\left[\mathbf{u}_b^\top \mathbf{u}_b \mathbf{v}_b^\top \mathbf{v}_b u_i q_{ij} v_j v_k q_{k,\ell} u_\ell\right]$$

$$= \sum_{i,j=1}^{m} \sum_{k,\ell=1}^{m} \mathbb{E}\left[\mathbf{u}_b^\top \mathbf{u}_b \mathbf{v}_b^\top \mathbf{v}_b \cdot u_i v_j v_k u_\ell\right] \mathbb{E}\left[q_{ij} q_{k,\ell}\right]$$

$$\overset{[\text{Prop. 10}]}{=} \sum_{i,j=1}^{m} \mathbb{E}\left[\|\mathbf{u}_b\|^2 \|\mathbf{v}_b\|^2 \cdot u_i u_j v_i v_j\right] \underbrace{\mathbb{E}\left[q_{ij}^2\right]}_{=1/m}$$

$$= \frac{1}{m}\left(\underbrace{m\mathbb{E}\left[\|\mathbf{u}_b\|^2 \|\mathbf{v}_b\|^2 \cdot u_1^2 v_1^2\right]}_{i=j} + \underbrace{m(m-1)\mathbb{E}\left[\|\mathbf{u}_b\|^2 \|\mathbf{v}_b\|^2 \cdot u_1 u_2 v_1 v_2\right]}_{i\neq j}\right)$$

$$= \sum_{i=m+1}^{p} \sum_{j=m+1}^{p} \mathbb{E}\left[u_1^2 v_1^2 \cdot u_i^2 v_j^2\right] + (m-1) \sum_{i=m+1}^{p} \sum_{j=m+1}^{p} \mathbb{E}\left[u_1 u_2 v_1 v_2 \cdot u_i^2 v_j^2\right]$$

$$= (p-m)\left(\sum_{j=m+1}^{p} \mathbb{E}\left[u_1^2 v_1^2 \cdot u_p^2 v_j^2\right] + (m-1) \sum_{j=m+1}^{p} \mathbb{E}\left[u_1 u_2 v_1 v_2 \cdot u_p^2 v_j^2\right]\right) \tag{32}$$

$$= (p-m)\left(\underbrace{\mathbb{E}\left[u_1^2 v_1^2 \cdot u_p^2 v_p^2\right]}_{j=p} + \underbrace{(p-m-1)\mathbb{E}\left[u_1^2 v_1^2 \cdot u_p^2 v_{p-1}^2\right]}_{m+1\leq j\leq p-1} + \right.$$

$$\left. (m-1)\left(\underbrace{\mathbb{E}\left[u_1 u_2 v_1 v_2 \cdot u_p^2 v_p^2\right]}_{j=p} + \underbrace{(p-m-1)\mathbb{E}\left[u_1 u_2 v_1 v_2 \cdot u_p^2 v_{p-1}^2\right]}_{m+1\leq j\leq p-1}\right)\right)$$

$$= (p-m)\left(\left\langle \begin{smallmatrix} 2 & 2 \\ \frac{2}{0} & \overrightarrow{0} \end{smallmatrix} \right\rangle + (p-m-1)\left\langle \begin{smallmatrix} 2 & 2 \\ 0 & 0 \\ \frac{2}{0} & \overrightarrow{0} \end{smallmatrix} \right\rangle + \right.$$

$$\left. (m-1)\left(\left\langle \begin{smallmatrix} 1 & 1 \\ 1 & 1 \\ \frac{2}{0} & \overrightarrow{0} \end{smallmatrix} \right\rangle + (p-m-1)\left\langle \begin{smallmatrix} 1 & 1 \\ 0 & 2 \\ \frac{2}{0} & \overrightarrow{0} \end{smallmatrix} \right\rangle\right)\right)$$

$$= (p-m)\left(\frac{(p^2+4p+15)+(p-m-1)(p+3)^2}{(p-1)p(p+1)(p+2)(p+4)(p+6)} + \frac{(m-1)(-p+3-(p-m-1)(p+3))}{(p-1)p(p+1)(p+2)(p+4)(p+6)}\right)$$

$$= \frac{(p-m)\left(m^2 p + 3m^2 - 2mp^2 - 10mp - 6m + p^3 + 7p^2 + 10p\right)}{(p-1)\,p\,(p+1)\,(p+2)\,(p+4)\,(p+6)}$$

$$\mathbb{E}\left[\mathbf{u}_b^\top \mathbf{u}_b \mathbf{u}_b^\top \mathbf{v}_b \cdot \mathbf{u}_a^\top \mathbf{Q}_m \mathbf{v}_a \mathbf{v}_a^\top \mathbf{Q}_m \mathbf{v}_a\right] = \mathbb{E}\left[\|\mathbf{v}_a\|^2 \mathbf{u}_b^\top \mathbf{u}_b \mathbf{u}_b^\top \mathbf{v}_b \cdot \mathbb{E}_{\mathbf{r}\sim\mathcal{S}^{m-1}}\left(\mathbf{u}_a^\top \mathbf{r} \mathbf{v}_a^\top \mathbf{r}\right)\right]$$

$$= \frac{1}{m}\mathbb{E}\left[\|\mathbf{v}_a\|^2 \mathbf{u}_b^\top \mathbf{u}_b \mathbf{u}_b^\top \mathbf{v}_b \mathbf{u}_a^\top \mathbf{v}_a\right] = -\frac{1}{m}\mathbb{E}\left[\|\mathbf{v}_a\|^2 \|\mathbf{u}_b\|^2 \left(\mathbf{u}_a^\top \mathbf{v}_a\right)^2\right]$$

$$= -\frac{1}{m}\mathbb{E}\left[\left(1 - \|\mathbf{v}_b\|^2\right) \|\mathbf{u}_b\|^2 \left(\mathbf{u}_a^\top \mathbf{v}_a\right)^2\right]$$

$$= -\frac{1}{m}\underbrace{\mathbb{E}\left[\|\mathbf{u}_b\|^2 \left(\mathbf{u}_a^\top \mathbf{v}_a\right)^2\right]}_{\text{solved in Eq. (18)}} + \frac{1}{m}\underbrace{\mathbb{E}\left[\|\mathbf{u}_b\|^2 \|\mathbf{v}_b\|^2 \left(\mathbf{u}_a^\top \mathbf{v}_a\right)^2\right]}_{\text{solved in Eq. (31)}} \tag{33}$$

$$= -\frac{(p-m)(p-m+2)}{(p-1)p(p+2)(p+4)} + \frac{(p-m)\left(m^2(p+3)-2m\left(p^2+5p+3\right)+p\left(p^2+7p+10\right)\right)}{(p-1)p(p+1)(p+2)(p+4)(p+6)}$$

$$= -\frac{(p-m)(p+3)(3(p+1)+(m-1)(p-m-1))}{(p-1)\,p\,(p+1)\,(p+2)\,(p+4)\,(p+6)}$$

## E.2 AUXILIARY DERIVATIONS WITH THREE VECTORS

$$\mathbb{E}_{\mathbf{u}\perp\mathbf{v}\perp\mathbf{z},\mathbf{Q}_m}\left[\left(\mathbf{u}_a^\top\mathbf{Q}_m\mathbf{z}_a\right)^2\left(\mathbf{u}_a^\top\mathbf{Q}_m\mathbf{v}_a\right)^2\right]$$

$$= \mathbb{E}_{\mathbf{u}\perp\mathbf{v}\perp\mathbf{z}}\left[\mathbb{E}_{\mathbf{r}\sim\mathcal{S}^{m-1}}\left[\left(\|\mathbf{u}_a\|\,\mathbf{r}^\top\mathbf{z}_a\right)^2\left(\|\mathbf{u}_a\|\,\mathbf{r}^\top\mathbf{v}_a\right)^2\right]\right]$$

$$= \sum_{i,j=1}^{m}\sum_{k,\ell=1}^{m}\mathbb{E}_{\mathbf{u}\perp\mathbf{v}\perp\mathbf{z}}\left[\|\mathbf{u}_a\|^4\,\mathbb{E}_{\mathbf{r}\sim\mathcal{S}^{m-1}}\left[r_i z_i r_j z_j r_k v_k r_\ell v_\ell\right]\right]$$

$$= \mathbb{E}_{\mathbf{u}\perp\mathbf{v}\perp\mathbf{z}}\left[\|\mathbf{u}_a\|^4\sum_{i,j=1}^{m}\sum_{k,\ell=1}^{m}z_i z_j v_k v_\ell\mathbb{E}_{\mathbf{r}\sim\mathcal{S}^{m-1}}\left[r_i r_j r_k r_\ell\right]\right]$$

[by Prop. 10, most terms become zero]

$$= \left\langle\begin{smallmatrix}4\\0\end{smallmatrix}\right\rangle_m \underbrace{\mathbb{E}\left[\|\mathbf{u}_a\|^4\sum_{i=1}^{m}z_i^2 v_i^2\right]}_{i=j=k=\ell} +$$

$$\left\langle\begin{smallmatrix}2\\2\\0\end{smallmatrix}\right\rangle_m\left(\underbrace{\mathbb{E}\left[\|\mathbf{u}_a\|^4\sum_{i\neq k=1}^{m}z_i^2 v_k^2\right]}_{i=j\neq k=\ell}+2\underbrace{\mathbb{E}\left[\|\mathbf{u}_a\|^4\sum_{i\neq j=1}^{m}z_i z_j v_i v_j\right]}_{i=k\neq j=\ell \,\vee\, i=\ell\neq j=k}\right)\tag{34}$$

$$= \frac{3}{m(m+2)}\sum_{i=1}^{m}\mathbb{E}\left[\|\mathbf{u}_a\|^4 z_i^2 v_i^2\right] +$$

$$\frac{1}{m(m+2)}\left(\sum_{i\neq k=1}^{m}\mathbb{E}\left[\|\mathbf{u}_a\|^4 z_i^2 v_k^2\right]+2\sum_{i\neq j=1}^{m}\mathbb{E}\left[\|\mathbf{u}_a\|^4 z_i z_j v_i v_j\right]\right)$$

$$= \frac{3m}{m(m+2)}\mathbb{E}\left[\|\mathbf{u}_a\|^4 z_1^2 v_1^2\right] + \frac{m(m-1)}{m(m+2)}\left(\mathbb{E}\left[\|\mathbf{u}_a\|^4 z_1^2 v_2^2\right]+2\mathbb{E}\left[\|\mathbf{u}_a\|^4 z_1 z_2 v_1 v_2\right]\right)$$

$$= \frac{3}{m+2}\underbrace{\mathbb{E}\left[\|\mathbf{u}_a\|^4 z_1^2 v_1^2\right]}_{\text{solved in Eq. (36)}}+\frac{m-1}{m+2}\left(\underbrace{\mathbb{E}\left[\|\mathbf{u}_a\|^4 z_1^2 v_2^2\right]}_{\text{solved in Eq. (37)}}+2\underbrace{\mathbb{E}\left[\|\mathbf{u}_a\|^4 z_1 z_2 v_1 v_2\right]}_{\text{solved in Eq. (38)}}\right)$$

$$= \frac{3}{m+2}\frac{(p+3)\left(m^2(p+5)+2m(p+1)\right)-16p-24}{(p-1)p(p+1)(p+2)(p+4)(p+6)} +$$

$$\frac{m-1}{m+2}\frac{m^2\left(p^3+8p^2+13p-2\right)+2m\left(p^3+4p^2-7p-10\right)-8\left(2p^2+9p+6\right)}{(p-2)(p-1)p(p+1)(p+2)(p+4)(p+6)} +$$

$$\frac{2(m-1)}{m+2}\frac{16p-8p^2-(m-2)\left(mp^2+7mp+14m+4p^2+12p+24\right)}{(p-2)(p-1)p(p+1)(p+2)(p+4)(p+6)}$$

$$= \frac{(m+2)(m(p+3)((m+2)p+5m+2)-16p-24)}{(m+2)(p-1)p(p+1)(p+2)(p+4)(p+6)} = \frac{m(p+3)((m+2)p+5m+2)-16p-24}{(p-1)p(p+1)(p+2)(p+4)(p+6)}$$

$$\mathbb{E}_{\mathbf{u}\perp\mathbf{v}\perp\mathbf{z},\mathbf{Q}_m}\left[\left(\mathbf{u}_a^\top\mathbf{Q}_m\mathbf{z}_a\right)^2\left(\mathbf{u}_b^\top\mathbf{v}_b\right)^2\right]=\mathbb{E}_{\mathbf{u}\perp\mathbf{v}\perp\mathbf{z}}\left[\mathbb{E}_{\mathbf{r}\sim\mathcal{S}^{m-1}}\left(\|\mathbf{u}_a\|\,\mathbf{r}^\top\mathbf{z}_a\right)^2\left(\mathbf{u}_b^\top\mathbf{v}_b\right)^2\right]$$

$$=\mathbb{E}_{\mathbf{u}\perp\mathbf{v}\perp\mathbf{z}}\left[\|\mathbf{u}_a\|^2\,\mathbb{E}_{\mathbf{r}\sim\mathcal{S}^{m-1}}\left(\mathbf{z}_a^\top\mathbf{r}\mathbf{r}^\top\mathbf{z}_a\right)\left(\mathbf{u}_b^\top\mathbf{v}_b\right)^2\right]$$

$$=\frac{1}{m}\mathbb{E}_{\mathbf{u}\perp\mathbf{v}\perp\mathbf{z}}\left[\|\mathbf{u}_a\|^2\,\|\mathbf{z}_a\|^2\left(\mathbf{u}_b^\top\mathbf{v}_b\right)^2\right]=\frac{1}{m}\sum_{i,j=1}^{m}\sum_{k,\ell=m+1}^{p}\mathbb{E}\left[u_i^2 z_j^2 u_k u_\ell v_k v_\ell\right]$$

$$=\frac{m\,(p-m)}{m}\sum_{j=1}^{m}\sum_{k=m+1}^{p}\mathbb{E}\left[u_1^2 z_j^2 u_k u_p v_k v_p\right]$$

$$=(p-m)\left(\underbrace{\sum_{k=m+1}^{p}\mathbb{E}\left[u_1^2 z_1^2 u_k u_p v_k v_p\right]}_{j=1}+(m-1)\underbrace{\sum_{k=m+1}^{p}\mathbb{E}\left[u_1^2 z_2^2 u_k u_p v_k v_p\right]}_{j\geq 2}\right)$$

$$=(p-m)\left(\underbrace{\mathbb{E}\left[u_1^2 z_1^2 u_p^2 v_p^2\right]}_{k=p}+\underbrace{(p-m-1)\,\mathbb{E}\left[u_1^2 z_1^2 u_{p-1}u_p v_{p-1}v_p\right]}_{m+1\leq k\leq p-1}+\right.$$

$$\left.(m-1)\left(\underbrace{\mathbb{E}\left[u_1^2 z_2^2 u_p^2 v_p^2\right]}_{k=p}+\underbrace{(p-m-1)\,\mathbb{E}\left[u_1^2 z_2^2 u_{p-1}u_p v_{p-1}v_p\right]}_{m+1\leq k\leq p-1}\right)\right) \tag{35}$$

$$=(p-m)\left(\left\langle\begin{smallmatrix}2&2&0\\2&0&2\\0&0&0\end{smallmatrix}\right\rangle+(p-m-1)\left\langle\begin{smallmatrix}2&0&2\\1&1&0\\0&0&0\end{smallmatrix}\right\rangle+\right.$$

$$\left.(m-1)\left(\left\langle\begin{smallmatrix}2&0&0\\0&2&0\\2&0&2\\0&0&0\end{smallmatrix}\right\rangle+(p-m-1)\left\langle\begin{smallmatrix}2&0&0\\0&0&2\\1&1&0\\0&0&0\end{smallmatrix}\right\rangle\right)\right)$$

$$=(p-m)\left(\frac{(p-2)(p+3)^2-(p-m-1)\left(p^2+3p+6\right)}{(p-2)(p-1)p(p+1)(p+2)(p+4)(p+6)}+\frac{(m-1)\left(p^3+6p^2+3p-6-(p-m-1)\left(p^2+5p+2\right)\right)}{(p-2)(p-1)p(p+1)(p+2)(p+4)(p+6)}\right)$$

$$=(p-m)\left(\frac{mp^2+3mp+6m+2p^2-6p-12+(m-1)\left(mp^2+5mp+2m+2p^2+6p-4\right)}{(p-2)(p-1)p(p+1)(p+2)(p+4)(p+6)}\right)$$

$$=\frac{(p-m)\,(m+2)\,\left(m\left(p^2+5p+2\right)-6p-4\right)}{(p-2)\,(p-1)\,p\,(p+1)\,(p+2)\,(p+4)\,(p+6)}$$

$$\mathbb{E}\left[\|\mathbf{u}_a\|^4 z_1^2 v_1^2\right] = \sum_{i=1}^{m}\sum_{j=1}^{m}\mathbb{E}\left[u_i^2 u_j^2 z_1^2 v_1^2\right]$$

$$= \underbrace{\mathbb{E}\left[u_1^4 z_1^2 v_1^2\right]}_{i=j=1} + \underbrace{2\left(m-1\right)\mathbb{E}\left[u_1^2 u_2^2 z_1^2 v_1^2\right]}_{i=1\neq j\,\vee\,i\neq 1=j} + \underbrace{\left(m-1\right)\mathbb{E}\left[u_2^4 z_1^2 v_1^2\right]}_{i=j\geq 2} +$$

$$\underbrace{\left(m-1\right)\left(m-2\right)\mathbb{E}\left[u_2^2 u_3^2 z_1^2 v_1^2\right]}_{i\neq j\geq 2}$$

$$= \left\langle \begin{smallmatrix} 4 & 2 & 2 \\ \vec{0} & \vec{0} & \vec{0} \end{smallmatrix} \right\rangle + 2\left(m-1\right)\left\langle \begin{smallmatrix} 2 & 2 & 2 \\ 2 & 0 & 0 \\ \vec{0} & \vec{0} & \vec{0} \end{smallmatrix} \right\rangle + \left(m-1\right)\left\langle \begin{smallmatrix} 0 & 2 & 2 \\ 4 & 0 & 0 \\ \vec{0} & \vec{0} & \vec{0} \end{smallmatrix} \right\rangle + \tag{36}$$

$$\left(m-1\right)\left(m-2\right)\left\langle \begin{smallmatrix} 0 & 2 & 2 \\ 2 & 0 & 0 \\ 2 & 0 & 0 \\ \vec{0} & \vec{0} & \vec{0} \end{smallmatrix} \right\rangle$$

$$= \frac{3}{p(p+2)(p+4)(p+6)} + \frac{2(m-1)(p+3)}{(p-1)p(p+2)(p+4)(p+6)} + \frac{3(m-1)(p+3)(p+5)+(m-1)(m-2)(p+3)(p+5)}{(p-1)p(p+1)(p+2)(p+4)(p+6)}$$

$$= \frac{3\left(p-1\right)\left(p+1\right) + 2\left(m-1\right)\left(p+1\right)\left(p+3\right) + \left(m^2-1\right)\left(p+3\right)\left(p+5\right)}{\left(p-1\right)p\left(p+1\right)\left(p+2\right)\left(p+4\right)\left(p+6\right)}$$

$$= \frac{\left(p+3\right)\left(m^2\left(p+5\right)+2m\left(p+1\right)\right) - 16p - 24}{\left(p-1\right)p\left(p+1\right)\left(p+2\right)\left(p+4\right)\left(p+6\right)}$$

$$\mathbb{E}\left[\|\mathbf{u}_a\|^4 z_1^2 v_2^2\right] = \sum_{i=1}^{m}\sum_{j=1}^{m}\mathbb{E}\left[u_i^2 u_j^2 z_1^2 v_2^2\right]$$

$$= 2\underbrace{\mathbb{E}\left[u_1^4 z_1^2 v_2^2\right]}_{i=j\in\{1,2\}} + 2\underbrace{\mathbb{E}\left[u_1^2 u_2^2 z_1^2 v_2^2\right]}_{i=1,j=2\,\vee\,i=2,j=1} + 2\left(m-2\right)\underbrace{\mathbb{E}\left[u_1^2 u_3^2 z_1^2 v_2^2\right]}_{i=1,j\geq 3\,\vee\,j=1,i\geq 3} + 2\left(m-2\right)\underbrace{\mathbb{E}\left[u_2^2 u_3^2 z_1^2 v_2^2\right]}_{i=2<j\,\vee\,i>2=j} +$$

$$\underbrace{\phantom{X}}_{\text{equal}}$$

$$\left(m-2\right)\underbrace{\mathbb{E}\left[u_3^4 z_1^2 v_2^2\right]}_{i=j\geq 3} + \left(m-2\right)\left(m-3\right)\underbrace{\mathbb{E}\left[u_3^2 u_4^2 z_1^2 v_2^2\right]}_{i\neq j\geq 3}$$

$$= 2\left\langle \begin{smallmatrix} 4 & 2 & 0 \\ 0 & 0 & 2 \\ \vec{0} & \vec{0} & \vec{0} \end{smallmatrix} \right\rangle + 2\left\langle \begin{smallmatrix} 2 & 2 & 0 \\ 2 & 0 & 2 \\ \vec{0} & \vec{0} & \vec{0} \end{smallmatrix} \right\rangle + \tag{37}$$

$$\left(m-2\right)\left(4\left\langle \begin{smallmatrix} 2 & 2 & 0 \\ 0 & 0 & 2 \\ 2 & 0 & 0 \\ \vec{0} & \vec{0} & \vec{0} \end{smallmatrix} \right\rangle + \left\langle \begin{smallmatrix} 0 & 2 & 0 \\ 0 & 0 & 2 \\ 4 & 0 & 0 \\ \vec{0} & \vec{0} & \vec{0} \end{smallmatrix} \right\rangle + \left(m-3\right)\left\langle \begin{smallmatrix} 0 & 2 & 0 \\ 0 & 0 & 2 \\ 2 & 0 & 0 \\ 2 & 0 & 0 \\ \vec{0} & \vec{0} & \vec{0} \end{smallmatrix} \right\rangle\right)$$

$$= \frac{6(p+1)(p+5)+2(p+3)^2}{(p-1)p(p+1)(p+2)(p+4)(p+6)} + \frac{4(m-2)\left(p^3+6p^2+3p-6\right)+3(m-2)\left(p^3+8p^2+13p-2\right)}{(p-2)(p-1)p(p+1)(p+2)(p+4)(p+6)} +$$

$$\frac{(m-2)(m-3)\left(p^3+8p^2+13p-2\right)}{(p-2)(p-1)p(p+1)(p+2)(p+4)(p+6)}$$

$$= \frac{m^2\left(p^3+8p^2+13p-2\right) + 2m\left(p^3+4p^2-7p-10\right) - 8\left(2p^2+9p+6\right)}{\left(p-2\right)\left(p-1\right)p\left(p+1\right)\left(p+2\right)\left(p+4\right)\left(p+6\right)}$$

$$\mathbb{E}\left[\|\mathbf{u}_a\|^4 z_1 z_2 v_1 v_2\right] = \sum_{i=1}^{m}\sum_{j=1}^{m}\mathbb{E}\left[u_i^2 u_j^2 z_1 z_2 v_1 v_2\right]$$

$$= 2\underbrace{\mathbb{E}\left[u_1^4 z_1 z_2 v_1 v_2\right]}_{i=j\in\{1,2\}} + 2\underbrace{\mathbb{E}\left[u_1^2 u_2^2 z_1 z_2 v_1 v_2\right]}_{i=1,j=2\,\vee\,i=2,j=1} +$$

$$\underbrace{2\,(m-2)\underbrace{\mathbb{E}\left[u_1^2 u_3^2 z_1 z_2 v_1 v_2\right]}_{i=1,j\geq 3\,\vee\,j=1,i\geq 3} + 2\,(m-2)\underbrace{\mathbb{E}\left[u_2^2 u_3^2 z_1 z_2 v_1 v_2\right]}_{i=2<j\,\vee\,i>2=j}}_{\text{equal}} +$$

$$(m-2)\underbrace{\mathbb{E}\left[u_3^4 z_1 z_2 v_1 v_2\right]}_{i=j\geq 3} + (m-2)\,(m-3)\underbrace{\mathbb{E}\left[u_3^2 u_4^2 z_1 z_2 v_1 v_2\right]}_{i\neq j\geq 3}$$

$$= 2\left\langle\begin{smallmatrix}4&1&1\\0&1&1\\0&0&0\\\rightarrow&\rightarrow&\rightarrow\end{smallmatrix}\right\rangle + 2\left\langle\begin{smallmatrix}2&1&1\\2&1&1\\0&0&0\\\rightarrow&\rightarrow&\rightarrow\end{smallmatrix}\right\rangle + \tag{38}$$

$$(m-2)\left(4\left\langle\begin{smallmatrix}2&1&1\\0&1&1\\2&0&0\\\rightarrow&\rightarrow&\rightarrow\end{smallmatrix}\right\rangle + \left\langle\begin{smallmatrix}0&1&1\\0&1&1\\4&0&0\\\rightarrow&\rightarrow&\rightarrow\end{smallmatrix}\right\rangle + (m-3)\left\langle\begin{smallmatrix}0&1&1\\0&1&1\\2&0&0\\2&0&0\\\rightarrow&\rightarrow&\rightarrow\end{smallmatrix}\right\rangle\right)$$

$$= \frac{-6}{(p-1)p(p+2)(p+4)(p+6)} + \frac{-2(p-3)}{(p-1)p(p+1)(p+2)(p+4)(p+6)} +$$

$$\frac{-4(m-2)\left(p^2+3p+6\right)-3(m-2)\left(p^2+7p+14\right)-(m-2)(m-3)\left(p^2+7p+14\right)}{(p-2)(p-1)p(p+1)(p+2)(p+4)(p+6)}$$

$$= \frac{-6(p-2)(p+1)-2(p-3)(p-2)-4(m-2)\left(p^2+3p+6\right)-3(m-2)\left(p^2+7p+14\right)-(m-2)(m-3)\left(p^2+7p+14\right)}{(p-2)(p-1)p(p+1)(p+2)(p+4)(p+6)}$$

$$= \frac{16p - 8p^2 - (m-2)\left(mp^2 + 7mp + 14m + 4p^2 + 12p + 24\right)}{(p-2)\,(p-1)\,p\,(p+1)\,(p+2)\,(p+4)\,(p+6)}$$

$$\mathbb{E}_{\mathbf{u} \perp \mathbf{v} \perp \mathbf{z}} \left[ \|\mathbf{u}_a\|^2 \, \mathbf{v}_a^\top \mathbf{z}_a \mathbf{u}_b^\top \mathbf{z}_b \mathbf{u}_b^\top \mathbf{v}_b \right] = \sum_{i,j=1}^{m} \sum_{k,\ell=m+1}^{p} \mathbb{E} \left[ u_i^2 v_j z_j u_k z_k u_\ell v_\ell \right]$$

$$= m \, (p-m) \sum_{j=1}^{m} \sum_{k=m+1}^{p} \mathbb{E} \left[ u_1^2 v_j z_j u_k z_k u_p v_p \right]$$

$$= m \, (p-m) \left( \underbrace{\sum_{k=m+1}^{p} \mathbb{E} \left[ u_1^2 v_1 z_1 u_k z_k u_p v_p \right]}_{j=1} + (m-1) \underbrace{\sum_{k=m+1}^{p} \mathbb{E} \left[ u_1^2 v_2 z_2 u_k z_k u_p v_p \right]}_{j \geq 2} \right)$$

$$= m \, (p-m) \left( \underbrace{\mathbb{E} \left[ u_1^2 v_1 z_1 u_p^2 v_p z_p \right]}_{k=p} + (p-m-1) \underbrace{\mathbb{E} \left[ u_1^2 v_1 z_1 u_{p-1} z_{p-1} u_p v_p \right]}_{m+1 \leq k \leq p-1} + \right.$$

$$\left. (m-1) \left( \underbrace{\mathbb{E} \left[ u_1^2 v_2 z_2 u_p^2 v_p z_p \right]}_{k=p} + (p-m-1) \underbrace{\mathbb{E} \left[ u_1^2 v_2 z_2 u_{p-1} z_{p-1} u_p v_p \right]}_{m+1 \leq k \leq p-1} \right) \right) \tag{39}$$

$$= m \, (p-m) \left( \left\langle \begin{smallmatrix} 2 & 1 & 1 \\ \frac{2}{0} & \frac{1}{0} & \frac{1}{0} \end{smallmatrix} \right\rangle + (p-m-1) \left\langle \begin{smallmatrix} 2 & 1 & 1 \\ 1 & 0 & 1 \\ \frac{1}{0} & \frac{0}{0} & \frac{1}{0} \end{smallmatrix} \right\rangle + \right.$$

$$\left. (m-1) \left( \left\langle \begin{smallmatrix} 2 & 0 & 0 \\ 0 & 1 & 1 \\ \frac{2}{0} & \frac{1}{0} & \frac{1}{0} \end{smallmatrix} \right\rangle + (p-m-1) \left\langle \begin{smallmatrix} 2 & 0 & 0 \\ 1 & 1 & 1 \\ \frac{1}{0} & \frac{0}{0} & \frac{1}{0} \end{smallmatrix} \right\rangle \right) \right)$$

$$= m \, (p-m) \left( \frac{-(p-3)(p-2)+4p(p-m-1)}{(p-2)(p-1)p(p+1)(p+2)(p+4)(p+6)} + \frac{(m-1)\left(-\left(p^2+3p+6\right)+2(p-m-1)(p+2)\right)}{(p-2)(p-1)p(p+1)(p+2)(p+4)(p+6)} \right)$$

$$= \frac{m \, (p-m) \left( (m+2) \, p^2 + (2-3m) \, p - 2m^2 \, (p+2) - 6m + 4 \right)}{(p-2) \, (p-1) \, p \, (p+1) \, (p+2) \, (p+4) \, (p+6)}$$

$$
\mathbb{E}\left[\left(\mathbf{u}_b^\top \mathbf{z}_b\right)^2 \left(\mathbf{u}_b^\top \mathbf{v}_b\right)^2\right] = \mathbb{E}\left[\left(-\mathbf{u}_a^\top \mathbf{z}_a\right)^2 \left(\mathbf{u}_b^\top \mathbf{v}_b\right)^2\right] = \mathbb{E}\left[\left(\mathbf{u}_a^\top \mathbf{z}_a\right)^2 \left(\mathbf{u}_b^\top \mathbf{v}_b\right)^2\right]
$$

$$
= \sum_{i,j=1}^{m} \sum_{k,\ell=m+1}^{p-m} \mathbb{E}\left[u_i u_j u_k u_\ell z_i z_j v_k v_\ell\right] = m\,(p-m) \sum_{j=1}^{m} \sum_{k=m+1}^{p-m} \mathbb{E}\left[u_1 u_j u_k u_p z_1 z_j v_k v_p\right]
$$

$$
= m\,(p-m) \Bigg( \underbrace{\mathbb{E}\left[u_1^2 u_p^2 z_1^2 v_p^2\right]}_{j=1,\,k=p} + \underbrace{(p-m-1)\,\mathbb{E}\left[u_1^2 u_{p-1} u_p z_1^2 v_{p-1} v_p\right]}_{j=1,\,m+1\le k\le p-1} +
$$

$$
\underbrace{(m-1)\,\mathbb{E}\left[u_1 u_2 u_p^2 z_1 z_2 v_p^2\right]}_{2\le j\le m,\,k=p} +
$$

$$
\underbrace{(m-1)\,(p-m-1)\,\mathbb{E}\left[u_1 u_2 u_{p-1} u_p z_1 z_2 v_{p-1} v_p\right]}_{2\le j\le m,\,m+1\le k\le p-1} \Bigg)
$$

$$
= m\,(p-m) \left( \left\langle \begin{smallmatrix} 2 & 0 & 2 \\ 2 & 2 & 0 \\ \vec{0} & \vec{0} & \vec{0} \end{smallmatrix} \right\rangle + (p-m-1) \underbrace{\left\langle \begin{smallmatrix} 2 & 0 & 2 \\ 1 & 1 & 0 \\ 1 & 1 & 0 \\ \vec{0} & \vec{0} & \vec{0} \end{smallmatrix} \right\rangle + (m-1) \left\langle \begin{smallmatrix} 1 & 0 & 1 \\ 1 & 0 & 1 \\ 2 & 2 & 0 \\ \vec{0} & \vec{0} & \vec{0} \end{smallmatrix} \right\rangle}_{\text{equal due to invariance (Prop. 9)}} + \right.
$$

$$
\left. (m-1)\,(p-m-1) \left\langle \begin{smallmatrix} 1 & 0 & 1 \\ 1 & 0 & 1 \\ 1 & 1 & 0 \\ 1 & 1 & 0 \\ \vec{0} & \vec{0} & \vec{0} \end{smallmatrix} \right\rangle \right) \tag{40}
$$

$$
= m\,(p-m) \left( \left\langle \begin{smallmatrix} 2 & 0 & 2 \\ 2 & 2 & 0 \\ \vec{0} & \vec{0} & \vec{0} \end{smallmatrix} \right\rangle + (p-2) \left\langle \begin{smallmatrix} 2 & 0 & 2 \\ 1 & 1 & 0 \\ 1 & 1 & 0 \\ \vec{0} & \vec{0} & \vec{0} \end{smallmatrix} \right\rangle + (m-1)\,(p-m-1) \left\langle \begin{smallmatrix} 1 & 0 & 1 \\ 1 & 0 & 1 \\ 1 & 1 & 0 \\ 1 & 1 & 0 \\ \vec{0} & \vec{0} & \vec{0} \end{smallmatrix} \right\rangle \right)
$$

$$
= m\,(p-m) \left( \frac{(p+3)^2}{(p-1)p(p+1)(p+2)(p+4)(p+6)} + \frac{-(p-2)\left(p^2+3p+6\right)}{(p-2)(p-1)p(p+1)(p+2)(p+4)(p+6)} + \right.
$$

$$
\left. \frac{(m-1)(p-m-1)}{(p-1)p(p+1)(p+2)(p+4)(p+6)} \right)
$$

$$
= m\,(p-m) \left( \frac{(p+3)^2 - \left(p^2+3p+6\right) + (m-1)\,(p-m-1)}{(p-1)\,p\,(p+1)\,(p+2)\,(p+4)\,(p+6)} \right)
$$

$$
= \frac{m\,(p-m)\left(-m^2 + mp + 2p + 4\right)}{(p-1)\,p\,(p+1)\,(p+2)\,(p+4)\,(p+6)}
$$

$$
\mathbb{E}_{\mathbf{u}\perp\mathbf{v}\perp\mathbf{z}}\left[\|\mathbf{z}_a\|^2 \left(\mathbf{v}_b^\top \mathbf{u}_b\right)^2\right] = \mathbb{E}\left[\|\mathbf{u}_a\|^2 \left(\mathbf{v}_b^\top \mathbf{z}_b\right)^2\right] = \mathbb{E}\left[\sum_{i=1}^{m} u_i^2 \left(\sum_{j=m+1}^{p} v_j z_j\right)^2\right]
$$

$$
= \sum_{i=1}^{m} \sum_{j,k=m+1}^{p} \mathbb{E}\left[u_i^2 v_j z_j v_k z_k\right] = m \sum_{j,k=m+1}^{p} \mathbb{E}\left[u_1^2 v_j z_j v_k z_k\right]
$$

$$
= m \underbrace{(p-m)\,\mathbb{E}\left[u_1^2 v_p^2 z_p^2\right]}_{j=k} + m \underbrace{(p-m)\,(p-m-1)\,\mathbb{E}\left[u_1^2 v_{p-1} z_{p-1} v_p z_p\right]}_{j\neq k} \tag{41}
$$

$$
= m\,(p-m) \left( \left\langle \begin{smallmatrix} 2 & 2 & 0 \\ 0 & 0 & 2 \\ \vec{0} & \vec{0} & \vec{0} \end{smallmatrix} \right\rangle + (p-m-1) \left\langle \begin{smallmatrix} 2 & 0 & 0 \\ 0 & 1 & 1 \\ 0 & 1 & 1 \\ \vec{0} & \vec{0} & \vec{0} \end{smallmatrix} \right\rangle \right)
$$

$$
= m\,(p-m) \left( \frac{(p+3)}{(p-1)p(p+2)(p+4)} - \frac{(p-m-1)(p+2)}{(p-2)(p-1)p(p+2)(p+4)} \right)
$$

$$
= \frac{m\,(p-m)\,(mp + 2m - 4)}{(p-2)\,(p-1)\,p\,(p+2)\,(p+4)}
$$

$$\mathbb{E}\left[\|\mathbf{u}_b\|^2\,\mathbf{v}_a^\top\mathbf{z}_a\mathbf{u}_a^\top\mathbf{z}_a\mathbf{u}_b^\top\mathbf{v}_b\right]=\mathbb{E}\left[\mathbf{u}_b^\top\mathbf{u}_b\mathbf{u}_b^\top\mathbf{v}_b\mathbf{z}_a^\top\mathbf{u}_a\mathbf{z}_a^\top\mathbf{v}_a\right]$$

$$=\sum_{i,j=1}^m\sum_{k,\ell=m+1}^p\mathbb{E}\left[u_\ell^2u_kv_ku_iz_iz_jv_j\right]=(p-m)\,m\sum_{j=1}^m\sum_{k=m+1}^p\mathbb{E}\left[u_1u_p^2u_kv_kz_1z_jv_j\right]$$

$$=(p-m)\,m\sum_{k=m+1}^p\left(\mathbb{E}\left[u_1u_p^2v_1z_1^2\left(u_kv_k\right)\right]+(m-1)\,\mathbb{E}\left[u_1u_p^2v_2z_1z_2\left(u_kv_k\right)\right]\right)$$

$$=(p-m)\,m\left(\mathbb{E}\left[u_1u_p^3v_1v_pz_1^2\right]+(p-m-1)\,\mathbb{E}\left[u_1u_{p-1}u_p^2v_1v_{p-1}z_1^2\right]\right)+$$

$$(p-m)\,m\,(m-1)\left(\mathbb{E}\left[u_1u_p^3v_2v_pz_1z_2\right]+(p-m-1)\,\mathbb{E}\left[u_1u_{p-1}u_p^2v_2v_{p-1}z_1z_2\right]\right)\tag{42}$$

$$=(p-m)\,m\left(\left\langle\begin{smallmatrix}1&1&2\\3&1&0\\0&0&0\\\rightarrow&\rightarrow&\rightarrow\end{smallmatrix}\right\rangle+(p-m-1)\left\langle\begin{smallmatrix}1&1&2\\2&0&0\\0&0&0\\\rightarrow&\rightarrow&\rightarrow\end{smallmatrix}\right\rangle\right)+$$

$$(p-m)\,m\,(m-1)\left(\left\langle\begin{smallmatrix}1&0&1\\0&1&1\\3&1&0\\\rightarrow&\rightarrow&\rightarrow\end{smallmatrix}\right\rangle+(p-m-1)\left\langle\begin{smallmatrix}1&0&1\\0&1&1\\2&0&0\\\rightarrow&\rightarrow&\rightarrow\end{smallmatrix}\right\rangle\right)$$

$$=(p-m)\,m\left(\tfrac{-3(p+3)-(p+3)(p-m-1)}{(p-1)p(p+1)(p+2)(p+4)(p+6)}+(m-1)\left(\tfrac{6(p+2)+2(p+2)(p-m-1)}{(p-2)(p-1)p(p+1)(p+2)(p+4)(p+6)}\right)\right)$$

$$=\frac{(p-m)\,m\,(p-m+2)\left(2mp+4m-p^2-3p+2\right)}{(p-2)\,(p-1)\,p\,(p+1)\,(p+2)\,(p+4)\,(p+6)}$$

$$\mathbb{E}\left[\|\mathbf{u}_a\|^2\,\mathbf{v}_a^\top\mathbf{z}_a\cdot\mathbf{v}_b^\top\mathbf{x}_b\cdot\mathbf{x}_b^\top\mathbf{z}_b\right]=\sum_{i,j=1}^m\sum_{k,\ell=m+1}^p\mathbb{E}\left[u_i^2v_jz_jv_kx_kx_\ell z_\ell\right]$$

$$=m\,(p-m)\sum_{j=1}^m\sum_{k=m+1}^p\mathbb{E}\left[u_1^2v_jz_jv_kx_kx_pz_p\right]$$

$$=m\,(p-m)\left(\underbrace{\mathbb{E}\left[u_1^2v_1z_1v_px_p^2z_p\right]}_{j=1,\,k=p}+\underbrace{(m-1)\,\mathbb{E}\left[u_1^2v_2z_2v_px_p^2z_p\right]}_{2\le j\le m,\,k=p}\right)+$$

$$m\,(p-m)\,(p-m-1)\left(\underbrace{\mathbb{E}\left[u_1^2v_1z_1v_{p-1}x_{p-1}x_pz_p\right]}_{j=1,\,m+1\le k\le p-1}+\underbrace{(m-1)\,\mathbb{E}\left[u_1^2v_2z_2v_{p-1}x_{p-1}x_pz_p\right]}_{2\le j\le m,\,m+1\le k\le p-1}\right)\tag{43}$$

$$=m\,(p-m)\left(\left\langle\begin{smallmatrix}2&1&1&0\\0&1&1&2\\0&0&0&0\\\rightarrow&\rightarrow&\rightarrow&\rightarrow\end{smallmatrix}\right\rangle+(m-1)\left\langle\begin{smallmatrix}2&0&0&0\\0&1&1&0\\0&1&1&2\\\rightarrow&\rightarrow&\rightarrow&\rightarrow\end{smallmatrix}\right\rangle\right)+$$

$$m\,(p-m)\left((p-m-1)\left\langle\begin{smallmatrix}2&1&1&0\\0&1&0&1\\0&0&1&1\\\rightarrow&\rightarrow&\rightarrow&\rightarrow\end{smallmatrix}\right\rangle+(m-1)\,(p-m-1)\left\langle\begin{smallmatrix}2&0&0&0\\0&1&1&0\\0&1&0&1\\0&0&1&1\\\rightarrow&\rightarrow&\rightarrow&\rightarrow\end{smallmatrix}\right\rangle\right)$$

$$=m\,(p-m)\left(\tfrac{-(p-2)(p+3)+2(p-m-1)(p+2)}{(p-2)(p-1)p(p+1)(p+2)(p+4)(p+6)}+\tfrac{(m-1)\left(2(p-m-1)p(p+4)-\left(p^2+5p+2\right)(p-3)\right)}{(p-3)(p-2)(p-1)p(p+1)(p+2)(p+4)(p+6)}\right)$$

$$=\frac{m(p-m)\left(-2m^2p^2-8m^2p+mp^3+4mp^2+15mp+18m-6p^2-6p-12\right)}{(p-3)(p-2)(p-1)p(p+1)(p+2)(p+4)(p+6)}$$

$$\mathbb{E}\left[\mathbf{u}_a^\top\mathbf{Q}_m\mathbf{z}_a\mathbf{u}_a^\top\mathbf{Q}_m\mathbf{v}_a\mathbf{x}_b^\top\mathbf{z}_b\mathbf{x}_b^\top\mathbf{v}_b\right]$$

$$=\mathbb{E}_{\mathbf{r}\sim\mathcal{S}^{m-1}}\left[\|\mathbf{u}_a\|^2\left(\mathbf{z}_a^\top\mathbf{r}\mathbf{r}^\top\mathbf{v}_a\right)\left(\mathbf{x}_b^\top\mathbf{z}_b\cdot\mathbf{x}_b^\top\mathbf{v}_b\right)\right]=\frac{1}{m}\underbrace{\mathbb{E}\left[\|\mathbf{u}_a\|^2\,\mathbf{z}_a^\top\mathbf{v}_a\cdot\mathbf{x}_b^\top\mathbf{z}_b\cdot\mathbf{x}_b^\top\mathbf{v}_b\right]}_{\text{solved in Eq. (43)}}\tag{44}$$

$$=\frac{(p-m)\left(-2m^2p^2-8m^2p+mp^3+4mp^2+15mp+18m-6p^2-6p-12\right)}{(p-3)\,(p-2)\,(p-1)\,p\,(p+1)\,(p+2)\,(p+4)\,(p+6)}$$

$$\mathbb{E}\left[\mathbf{u}_b^\top\mathbf{u}_b\mathbf{z}_b^\top\mathbf{v}_b\cdot\mathbf{u}_a^\top\mathbf{Q}_m\mathbf{v}_a\mathbf{z}_a^\top\mathbf{Q}_m\mathbf{u}_a\right]=\sum_{i,j=1}^m\sum_{k,\ell=1}^m\mathbb{E}\left[\mathbf{u}_b^\top\mathbf{u}_b\mathbf{z}_b^\top\mathbf{v}_b\cdot u_iq_{ij}v_jz_kq_{k\ell}u_\ell\right]$$

$$=\sum_{i,j=1}^m\sum_{k,\ell=1}^m\mathbb{E}\left[\mathbf{u}_b^\top\mathbf{u}_b\mathbf{z}_b^\top\mathbf{v}_b\cdot u_iv_jz_ku_\ell\right]\mathbb{E}\left[q_{ij}q_{k\ell}\right]$$

$$[\text{Prop. 10}]=\frac{1}{m}\sum_{i,j=1}^m\mathbb{E}\left[\mathbf{u}_b^\top\mathbf{u}_b\mathbf{z}_b^\top\mathbf{v}_b\cdot u_iv_jz_iu_j\right]\underbrace{\mathbb{E}\left[q_{ij}^2\right]}_{=1/m}$$

$$=\frac{1}{m}\mathbb{E}\left[\mathbf{u}_b^\top\mathbf{u}_b\mathbf{z}_b^\top\mathbf{v}_b\mathbf{u}_a^\top\mathbf{v}_a\mathbf{u}_a^\top\mathbf{z}_a\right]=\frac{1}{m}\mathbb{E}\left[\|\mathbf{u}_b\|^2\left(-\mathbf{z}_a^\top\mathbf{v}_a\right)\left(-\mathbf{u}_b^\top\mathbf{v}_b\right)\mathbf{u}_a^\top\mathbf{z}_a\right]$$

$$=\frac{1}{m}\underbrace{\mathbb{E}\left[\|\mathbf{u}_b\|^2\mathbf{v}_a^\top\mathbf{z}_a\mathbf{u}_a^\top\mathbf{z}_a\mathbf{u}_b^\top\mathbf{v}_b\right]}_{\text{solved in Eq. (42)}}$$

$$=\frac{(p-m)(p-m+2)\left(2mp+4m-p^2-3p+2\right)}{(p-2)(p-1)p(p+1)(p+2)(p+4)(p+6)} \tag{45}$$

$$\mathbb{E}\left[\|\mathbf{u}_b\|^2\mathbf{u}_b^\top\mathbf{v}_b\cdot\mathbf{z}_a^\top\mathbf{Q}_m\mathbf{u}_a\mathbf{z}_a^\top\mathbf{Q}_m\mathbf{v}_a\right]$$

$$=\mathbb{E}\left[\|\mathbf{z}_a\|^2\|\mathbf{u}_b\|^2\mathbf{u}_b^\top\mathbf{v}_b\mathbf{u}_a^\top\mathbb{E}_{\mathbf{r}\sim\mathcal{S}^{m-1}}\left(\mathbf{r}^\top\mathbf{r}\right)\mathbf{v}_a\right]=\frac{1}{m}\mathbb{E}\left[\|\mathbf{z}_a\|^2\left(\sum_{\ell=m+1}^pu_\ell^2\right)\mathbf{u}_b^\top\mathbf{v}_b\mathbf{u}_a^\top\mathbf{v}_a\right]$$

$$=\frac{(p-m)}{m}\mathbb{E}\left[u_p^2\|\mathbf{z}_a\|^2\left(\sum_{i=m+1}^pu_iv_i\right)\cdot\left(\sum_{j=1}^mu_jv_j\right)\right]$$

$$=(p-m)\mathbb{E}\left[u_1v_1u_p^2\|\mathbf{z}_a\|^2\left(\sum_{i=m+1}^pu_iv_i\right)\right]$$

$$=(p-m)\left(\mathbb{E}\left[u_1u_p^3v_1v_p\|\mathbf{z}_a\|^2\right]+(p-m-1)\mathbb{E}\left[u_1u_{p-1}u_p^2v_1v_{p-1}\|\mathbf{z}_a\|^2\right]\right)$$

$$=(p-m)\left(\mathbb{E}\left[u_1u_p^3v_1v_p\left(\sum_{i=1}^mz_i^2\right)\right]+(p-m-1)\mathbb{E}\left[u_1u_{p-1}u_p^2v_1v_{p-1}\left(\sum_{i=1}^mz_i^2\right)\right]\right)$$

$$=(p-m)\left(\mathbb{E}\left[u_1u_p^3v_1v_pz_1^2\right]+(m-1)\mathbb{E}\left[u_1u_p^3v_1v_pz_2^2\right]\right)+$$
$$(p-m)(p-m-1)\left(\mathbb{E}\left[u_1u_{p-1}u_p^2v_1v_{p-1}z_1^2\right]+(m-1)\mathbb{E}\left[u_1u_{p-1}u_p^2v_1v_{p-1}z_2^2\right]\right)$$

$$=(p-m)\left(\left\langle\begin{smallmatrix}1&1&2\\3&1&0\\0&0&0\end{smallmatrix}\right\rangle+(m-1)\left\langle\begin{smallmatrix}1&1&0\\0&0&2\\3&1&0\\0&0&0\end{smallmatrix}\right\rangle\right)+$$
$$(p-m)(p-m-1)\left(\left\langle\begin{smallmatrix}1&1&2\\1&1&0\\2&0&0\\0&0&0\end{smallmatrix}\right\rangle+(m-1)\left\langle\begin{smallmatrix}1&1&0\\0&0&2\\1&1&0\\2&0&0\\0&0&0\end{smallmatrix}\right\rangle\right)$$

$$=(p-m)\frac{-3(p+3)}{(p-1)p(p+1)(p+2)(p+4)(p+6)}+$$
$$(p-m)\frac{-3(m-1)\left(p^2+5p+2\right)+(p-m-1)\left(-(p-2)(p+3)-(m-1)\left(p^2+5p+2\right)\right)}{(p-2)(p-1)p(p+1)(p+2)(p+4)(p+6)}$$

$$=(p-m)\left(\frac{(p-2)(-3(p+3))+m^2p^2+5m^2p+2m^2-mp^3-7mp^2-16mp-12m+7p^2+19p-2}{(p-2)(p-1)p(p+1)(p+2)(p+4)(p+6)}\right)$$

$$=\frac{(p-m)\left(m^2p^2+5m^2p+2m^2-mp^3-7mp^2-16mp-12m+4p^2+16p+16\right)}{(p-2)(p-1)p(p+1)(p+2)(p+4)(p+6)} \tag{46}$$

### E.3 Auxiliary derivations with four vectors

$$\mathbb{E}\left[\mathbf{u}_b^\top \mathbf{v}_b \cdot \mathbf{x}_b^\top \mathbf{z}_b \cdot \mathbf{u}_b^\top \mathbf{x}_b \cdot \mathbf{v}_b^\top \mathbf{z}_b\right] = \mathbb{E}\left[\mathbf{u}_a^\top \mathbf{v}_a \cdot \mathbf{x}_a^\top \mathbf{z}_a \cdot \mathbf{u}_b^\top \mathbf{x}_b \cdot \mathbf{v}_b^\top \mathbf{z}_b\right]$$

$$= \sum_{i,j=1}^{m} \sum_{k,\ell=m+1}^{p} \mathbb{E}\left[u_i v_i x_j z_j u_k x_k v_\ell z_\ell\right] = m\,(p-m) \sum_{j=1}^{m} \sum_{k=m+1}^{p} \mathbb{E}\left[u_1 v_1 x_j z_j u_k x_k v_p z_p\right]$$

$$= m\,(p-m)\left(\underbrace{\mathbb{E}\left[u_1 v_1 x_1 z_1 u_p x_p v_p z_p\right]}_{j=1,\,k=p} + \underbrace{(m-1)\,\mathbb{E}\left[u_1 v_1 x_2 z_2 u_p x_p v_p z_p\right]}_{2\le j\le m,\,k=p}\right) +$$

$$m\,(p-m)\left(\underbrace{(p-m-1)\,\mathbb{E}\left[u_1 v_1 x_1 z_1 u_{p-1} x_{p-1} v_p z_p\right]}_{j=1,\,m+1\le k\le p-1} +\right.$$

$$\left.\underbrace{(m-1)\,(p-m-1)\,\mathbb{E}\left[u_1 v_1 x_2 z_2 u_{p-1} x_{p-1} v_p z_p\right]}_{2\le j\le m,\,m+1\le k\le p-1}\right)$$

$$= m\,(p-m)\left(\left\langle \begin{smallmatrix} 1 & 1 & 1 & 1 \\ 1 & 1 & 1 & 1 \\ 0 & 0 & 0 & 0 \\ \vec{} & \vec{} & \vec{} & \vec{} \end{smallmatrix}\right\rangle + (m-1)\left\langle \begin{smallmatrix} 1 & 1 & 1 & 1 \\ 0 & 1 & 1 & 0 \\ 1 & 0 & 0 & 1 \\ \vec{} & \vec{} & \vec{} & \vec{} \end{smallmatrix}\right\rangle + \right.$$

$$\left.(p-m-1)\left\langle \begin{smallmatrix} 1 & 1 & 1 & 1 \\ 1 & 1 & 0 & 0 \\ 0 & 0 & 1 & 1 \\ \vec{} & \vec{} & \vec{} & \vec{} \end{smallmatrix}\right\rangle + (m-1)\,(p-m-1)\left\langle \begin{smallmatrix} 1 & 0 & 1 & 0 \\ 0 & 1 & 0 & 1 \\ 1 & 1 & 0 & 0 \\ 0 & 0 & 1 & 1 \\ \vec{} & \vec{} & \vec{} & \vec{} \end{smallmatrix}\right\rangle\right) \qquad (47)$$

$$= m\,(p-m)\left(\left\langle \begin{smallmatrix} 1 & 1 & 1 & 1 \\ 1 & 1 & 1 & 1 \\ \vec{} & \vec{} & \vec{} & \vec{} \end{smallmatrix}\right\rangle + (p-2)\left\langle \begin{smallmatrix} 1 & 1 & 1 & 1 \\ 1 & 1 & 0 & 0 \\ 0 & 0 & 1 & 1 \\ \vec{} & \vec{} & \vec{} & \vec{} \end{smallmatrix}\right\rangle + \right.$$

$$\left.(m-1)\,(p-m-1)\left\langle \begin{smallmatrix} 1 & 0 & 1 & 0 \\ 0 & 1 & 0 & 1 \\ 1 & 1 & 0 & 0 \\ 0 & 0 & 1 & 1 \\ \vec{} & \vec{} & \vec{} & \vec{} \end{smallmatrix}\right\rangle\right)$$

$$= m\,(p-m)\left(\frac{3(p-3)}{(p-3)(p-1)p(p+1)(p+2)(p+4)(p+6)} + \frac{(p-2)}{(p-1)p(p+1)(p+2)(p+4)(p+6)}\right) +$$

$$m\,(p-m)\left(\frac{(m-1)(p-m-1)(-5p-6)}{(p-3)(p-2)(p-1)p(p+1)(p+2)(p+4)(p+6)}\right)$$

$$= \frac{m\,(p-m)\left(5m^2 p + 6m^2 - 5mp^2 - 6mp + p^3 + p^2 + 2p\right)}{(p-3)\,(p-2)\,(p-1)\,p\,(p+1)\,(p+2)\,(p+4)\,(p+6)}$$

$$\mathbb{E}\left[\left\|\mathbf{z}_a\right\|^2 \mathbf{u}_a^\top \mathbf{x}_a \mathbf{u}_b^\top \mathbf{v}_b \mathbf{x}_b^\top \mathbf{v}_b\right]$$

$$= \sum_{i,j=1}^{m} \sum_{k,\ell=m+1}^{p} \mathbb{E}\left[z_i^2 u_j x_j u_k v_k x_\ell v_\ell\right] = m\left(p-m\right) \sum_{j=1}^{m} \sum_{k=m+1}^{p} \mathbb{E}\left[z_1^2 u_j x_j u_k v_k x_p v_p\right]$$

$$= m\left(p-m\right) \left( \underbrace{\mathbb{E}\left[u_1 x_1 z_1^2 u_p v_p^2 x_p\right]}_{j=1,\, k=p} + \underbrace{\left(m-1\right) \mathbb{E}\left[u_2 x_2 z_1^2 u_p v_p^2 x_p\right]}_{2 \leq j \leq m,\, k=p} \right) +$$

$$m\left(p-m\right)\left(p-m-1\right)\left( \underbrace{\mathbb{E}\left[u_1 x_1 z_1^2 u_{p-1} v_{p-1} x_p v_p\right]}_{j=1,\, m+1 \leq k \leq p-1} + \underbrace{\left(m-1\right) \mathbb{E}\left[u_2 x_2 z_1^2 u_{p-1} v_{p-1} x_p v_p\right]}_{2 \leq j \leq m,\, m+1 \leq k \leq p-1} \right)$$

$$= m\left(p-m\right)\left( \left\langle \begin{smallmatrix} 1 & 0 & 1 & 2 \\ 1 & 2 & 1 & 0 \\ \vec{0} & \vec{0} & \vec{0} & \vec{0} \end{smallmatrix} \right\rangle + \left(m-1\right) \left\langle \begin{smallmatrix} 0 & 0 & 0 & 2 \\ 1 & 0 & 1 & 0 \\ 1 & 2 & 1 & 0 \\ \vec{0} & \vec{0} & \vec{0} & \vec{0} \end{smallmatrix} \right\rangle + \right. \tag{48}$$

$$\left. \left(p-m-1\right) \left\langle \begin{smallmatrix} 1 & 0 & 1 & 2 \\ 1 & 1 & 0 & 0 \\ 0 & 1 & 1 & 0 \\ \vec{0} & \vec{0} & \vec{0} & \vec{0} \end{smallmatrix} \right\rangle + \left(m-1\right)\left(p-m-1\right) \left\langle \begin{smallmatrix} 0 & 0 & 0 & 2 \\ 1 & 0 & 1 & 0 \\ 1 & 1 & 0 & 0 \\ 0 & 1 & 1 & 0 \\ \vec{0} & \vec{0} & \vec{0} & \vec{0} \end{smallmatrix} \right\rangle \right)$$

$$= m\left(p-m\right)\left( \frac{-\left(p-2\right)\left(p+3\right)+2\left(p-m-1\right)\left(p+2\right)}{\left(p-2\right)\left(p-1\right)p\left(p+1\right)\left(p+2\right)\left(p+4\right)\left(p+6\right)} + \right.$$

$$\left. \frac{\left(m-1\right)\left(-\left(p-3\right)\left(p^2+5p+2\right)+2\left(p-m-1\right)p\left(p+4\right)\right)}{\left(p-3\right)\left(p-2\right)\left(p-1\right)p\left(p+1\right)\left(p+2\right)\left(p+4\right)\left(p+6\right)} \right)$$

$$= \frac{m\left(p-m\right)\left(-2m^2p^2-8m^2p+mp^3+4mp^2+15mp+18m-6p^2-6p-12\right)}{\left(p-3\right)\left(p-2\right)\left(p-1\right)p\left(p+1\right)\left(p+2\right)\left(p+4\right)\left(p+6\right)}$$

