# OpenReview forum: "The Joint Effect of Task Similarity and Overparameterization on Catastrophic Forgetting — An Analytical Model"
_ICLR.cc/2024/Conference — ICLR 2024 poster_

### Official Review · Reviewer_SXpA · 2023-10-29

**Soundness:** 2 fair
**Presentation:** 1 poor
**Contribution:** 2 fair
**Rating:** 3
**Confidence:** 4

**Summary:**

The paper studies catastrophic forgetting in an analytical way.
Specifically, they focus on two-task continual linear regression and uncover a pattern in overparameterized models
where intermediate task similarity leads to the most forgetting.

**Strengths:**

1. The results under overparameterized models are interesting.

**Weaknesses:**

1. The writing needs improvement as it is difficult to grasp the central logical structure.
2. Format in assumption 1 seems not correct since there is a long blank.
3. The proof is difficult to follow, I can tell why the three terms in Lemma 6 can be replaced by the long formulas.
4. I think the paper 'Analysis of catastrophic forgetting for random orthogonal transformation tasks in the overparameterized regime' discusses a similar task. It is difficult to tell what the difference and improvement compared to that.

**Questions:**

1. I have tried to read all proof, but it is difficult to follow and check all the long expressions.
I think it will be better to make the proof clearer to understand.

---

> ### Author Response · Authors · 2023-11-17
>
> We thank the reviewer for their comments and we are glad that the reviewer generally found our results interesting.
>
> However, there seems to be a misunderstanding regarding the scope of the paper. The reviewer wrote in their summary that we
> *“uncover a pattern in overparameterized models where intermediate task similarity leads to the most forgetting”*. This is true but it is only a part of our paper.
>
> Generally, our paper is the first to reveal a nuanced analytical interplay between overparameterization and task similarity, and show that overparameterization alone is not enough to understand catastrophic forgetting (in contrast to a current line of continual learning empirical papers which only focuses on the benefits of overparameterization). Our Theorem 3 and Figure 3 clearly demonstrate this.
>
> Moreover, in Weakness 4 the reviewer wrote that they believe that Goldfarb and Hand (2023) already addressed a similar research question.
> We have to respectfully disagree, since there are important differences between our work and theirs, which we have already thoroughly discussed in multiple places in our paper (e.g., in the Introduction, Learning Dynamics of Section 2.2, and in the Simulations of Section 2.4). Specifically, Goldfarb and Hand (2023) only study the effect of overparameterization without considering any notion of task similarity. They analyze the regime of *low* task similarity, i.e., applying completely random transformations to the first data matrix ($\alpha=1$), while our main result analyzes the *full* spectrum of task similarity from $\alpha=0$ to $1$.
> Additionally, that paper uses restrictive data assumptions and provides an upper bound on performance, while our work does not assume any particular data model and our main result is an *exact* expression for expected forgetting.
>
> ---
>
> As to the more minor comments,
>
> - **Weakness 3 & Question 1:**
> > The proof is difficult to follow.
>
>   **Response:** Deriving exact expressions for our analysis was indeed difficult and required 60+ pages of complicated proofs. We originally tried to make our proof’s idea accessible using the Proof Sketch that appears right after Theorem 3.
>   Following the reviewer’s comment, we will further revise the manuscript and add an entire page presenting a Proof Outline in a new appendix for the final version (this requires a lot of time and we want the reviewers-authors discussion to start in the meanwhile).
>
> ---
>
> - **Weakness 2:**
> > Format in Assumption 1 seems incorrect since there is a long blank.
>
>   **Response:** The long blank was not a mistake but a styling choice (using \hfill). We now removed this blank to avoid confusion.
>
> ---
> If we have adequately addressed the reviewer's concerns, we kindly ask the reviewer to consider raising their score. If there are any remaining concerns, please let us know.

---

### Official Review · Reviewer_6udf · 2023-10-30

**Soundness:** 3 good
**Presentation:** 3 good
**Contribution:** 3 good
**Rating:** 8
**Confidence:** 5

**Summary:**

In order to explore the common impact of task similarity and overparameterization on forgotten, the paper obtains a conclusion through formula derivation. The conclusion shows a non-monotonic behavior in task similarity when the model is suitably overparameterized, and a monotonic behavior when it is critically overparameterized. In addition, the paper verified the conclusion through a large number of experiments.

**Strengths:**

This paper obtained a precise and effective conclusion through a large number of formula derivations and experiments. This conclusion can help us better understand the forgetting problems of the model.

**Weaknesses:**

The experimental setting of the paper is very limited, and the scope of application of the conclusions is questionable.

**Questions:**

1.	Is the conclusion proved to be universal? If in a non-linear complex neural network, is the model forgetting the same as the conclusion?
2.	We can only mess up pixels as a standard of dissimilarity, which does not match the real scene. So it's hard to verify the conclusions on other data.
3.	Is overparameterization equivalent to the learning capacity of the model? Do the conclusions of the paper apply to other data and models?
4.	The formulas derived in this paper are rather complicated. Can you please explain Pseudoinverse Properties and Operator Norm Properties in detail?
5.	This paper only conducted experiments on the mnist dataset. Can the same conclusion be reached for other complex datasets?

---

> ### Author Response · Authors · 2023-11-17
>
> We appreciate the reviewer's comments and are pleased that they recognize the value and contribution of our findings to the continual learning literature.
>
> Below we address their questions:
>
> - **Question 1:**
> > Is the conclusion proved to be universal (for non-linear complex neural networks)?
>
>   **Response:** Our rigorous analysis directly applies to linear regression models. Empirically, we verified our analysis’ validity on linear models and showed evidence that our findings can explain some behaviors observed in MLPs trained on continual permutation benchmarks.
>   Extending our analysis to similar continual settings in more complex networks should be very interesting, but will probably require more intricate analytical tools.
>   Existing theoretical results in CL mostly deal with simple models, and there are still very few analytical results on more complex models. We believe that in order to derive results for more complex models, the community must first thoroughly understand the simpler ones (e.g., linear models). Now that we’ve established an interplay between task similarity and overparameterization in linear models, the community can concentrate on extending it to more complex models (e.g., MLPs or CNNs).
>   We now added this as a future direction in the “Future work” section of the revised manuscript.
>
> - **Questions 2&5:**
> > Pixel shuffling is the only current knob for dissimilarity in the experiments, which does not match real world scenarios. Can the same experimental conclusions be reached for other complex datasets?
>
>   **Response:** We hypothesize that the true effect of task similarity comes from similarity in the NTK feature regime as presented in Appendix A. Pixel shuffling is just one proxy for which to generate the desired effect in the NTK feature regime but the relationship between experimental setup and the NTK features is a complex one. We leave the full uncovering of this effect for future work (we mention this in the “Future work” section of the revised manuscript).
>
> -  **Question 3:**
> > Is overparameterization equivalent to learning capacity?
>
>    **Response:** Overparameterization is related to learning capacity but is not equivalent. Any model that surpasses the complexity of the interpolation threshold has an equal capacity to fully fit to the training set. However, more overparameterized models enjoy the beneficial properties of smoother optimization landscapes and sometimes more favorable generalization properties.
>
> - **Question 4:**
> > Can we explain Pseudoinverse Properties and Operator Norm Properties in detail?
>
>   **Response:** The pseudoinverse property that we used in Equation (4) is that for any matrix $\mathbf{X}$ and an orthogonal matrix $\mathbf{O}$, it holds that $\mathbf{X}=\mathbf{X}\mathbf{X}^{+}\mathbf{X}$ and $(\mathbf{X}\mathbf{O})^{+}=\mathbf{O}^{+}\mathbf{X}^{+}=\mathbf{O}^{\top}\mathbf{X}^{+}$. The operator norm property refers to $\Vert\mathbf{X}\mathbf{v}\Vert\le\Vert\mathbf{X}\Vert\Vert\mathbf{v}\Vert$.
>    To improve readability, we now added these explicitly within Equation (4) and underneath it.
>
> ---
>
> We again thank the reviewer for the feedback.
> If we have adequately addressed the reviewer's concerns, we kindly ask the reviewer to continue supporting our submission.
> If there are any remaining concerns, please let us know.

---

### Official Review · Reviewer_ckB1 · 2023-11-01

**Soundness:** 4 excellent
**Presentation:** 4 excellent
**Contribution:** 3 good
**Rating:** 8
**Confidence:** 5

**Summary:**

The paper considers the continual learning problem in the case of two linear regression tasks. The first task is arbitrary, while the second is assumed to be a random orthogonal transformation of the first one.

The main result of the paper is a precise expression of the normalized, expected forgetting (Theorem 3). The expression is related to the level of task similarity and overparameterization.

The paper proceeds to specialize the main theorem to the highly overparameterized regime and at the interpolation threshold, visualizing the amount of forgetting on synthetic data or the MNIST dataset.

**Strengths:**

The paper has several strengths:
- First, while the proof idea is straightforward (e.g., as shown in the proof sketch), it involves lengthy algebraic manipulations that the paper manages to excel in; this is commendable.
- Related to the first strength, even though the proof can be lengthy, the main paper is clearly written and easy to understand.
- The experiments numerically verify the correctness of the theorem and provide interesting insights into catastrophic forgetting. While some of the observations have been made in prior works, the paper suitably discussed that, and moreover provided different results in the highly overparameterized regime.

**Weaknesses:**

While I tend to vote for acceptance, I do have several questions or comments:

- At first glance it seems a little bit weird to have the same $y$ for two different tasks. I think it might be motivated by the case of permuted MNIST where permutations do not change the label. It would be great if the authors could comment on that in their revision.
- Certainly there is some gap between the theory and MNIST experiments. The authors are encouraged to discuss that gap. For example, the training loss in the theory vs test error in the experiments. The least-squares loss in the theory vs classification loss in the experiments (I suppose).
- Is there any deeper connection between the bound in the highly overparameterized regime, $\alpha^2(1-\alpha)^2$, and the corresponding bound in Lemma 9 of Evron et al. (2022) for the case $k=2$?
- While the main theorem is accomplished by algebraic calculations, I wonder whether there is a geometric proof that would achieve the same. For example, the two steps can be thought of as two projections (onto certain subspaces), and the forgetting can be bounded deterministically by some quantity related to the principal angles. With a random orthogonal transformation, the principal angles between the two subspaces would be in some sense random. Would it be easier if one analyzes the randomness of the principal angles, and could the algebraic proofs be replaced by geometric reasoning?

**Questions:**

See above.

---

> ### Author Response · Authors · 2023-11-17
>
> We thank the reviewer for their valuable feedback, and we are glad that they appreciate our work.
> Below we address the reviewer's concerns
>
> ---
> - **Question/Weakness 1:**
> > It seems weird to have the same y for two different tasks.
>
>   **Response:** This is due to our analysis being analogous to the data permutation task benchmarks of continual learning. These tasks are especially amenable to analysis as they are each equally difficult for an MLP to solve. Thus we can fairly compare their accuracies and forgetting between tasks.
>   More generally, many practical continual learning scenarios are subject to domain shifts (e.g., sees another “region” in $\mathcal{X}$), without changing the output space $\mathcal{Y}$.
>
> ---
> - **Question/Weakness 2:**
> > There is some gap between the theory and MNIST experiments (training loss vs. test error; regression vs. classification).
>
>   **Response:** We view the present work as an initial result in understanding the relationship between task similarity and forgetting. Nonetheless there are still clear analogies between the analysis and experiments of our work. The permutation task setting is a variant of the DOTS model where random permutation matrices are used instead of random orthogonal matrices. In continual learning practice, training error and test error are strongly correlated: when forgetting is observed in the test error then it is also observed in the training error. Additionally, studying training error allows us to loosen the data assumptions of prior work (Goldfarb, Lin) to give better insight into the problem’s worst-case performance. Regression and classification problems are also closely related. One can turn the MNIST experiments into a one-hot regression problem and expect to observe the same effects as in the classification problem.
>
> ---
> - **Question 3:** Is there any deeper connection between the bound in the highly overparameterized regime, $\alpha^2 (1-\alpha)^2$, and the corresponding bound in Lemma 9 of Evron et al. (2022) for the case ?
>
>   **Response:** There is indeed a deeper connection! Given no task-repetitions ($k=2$), the result from Evron et al. (2022) yields a forgetting upper bound of $\frac{1}{2} \max_{i} \{ \cos^2 (\theta_i) (1-\cos^2 (\theta_i)) \}$, where $\theta_i$ are the nonzero principal angles between the solution subspaces of the two tasks (here, they are equivalent to the principal angles between the data itself; you can see Claim 19 in their paper).
>   This expression is of course very similar to our overparameterized results, i.e., $\alpha^{2}(1-\alpha)^{2}=\left(\frac{m}{p}\right)^{2}\left(1-\frac{m}{p}\right)^{2}$.
>
>   Trying to explain this very briefly and intuitively: When $m=0$, our random transformation $\textbf{O}$ (applied to $\textbf{X}_1$ to create $\textbf{X}_2$) is a deterministic identity operator. This, of course, yields principal angles of $0$ between tasks, incurring no forgetting.
>   When $m=p$, the random transformation becomes completely random, and since we assume high overparameterization, it means that the two subspaces are going to be almost orthogonal w.h.p. (i.e., have angles of $90^{\circ}$, again incurring no forgetting.
>   To understand $m=\frac{p}{2}$, it is perhaps easy to imagine two random lines in $2$ dimensions. In this case it is known that the expected angle between them should be $\approx 45^{\circ}$, which corresponds to the highest forgetting in Evron et al. (2022).
>
>   Following this review, we have now incorporated a similar discussion (on a slightly different aspect) in our “Geometric interpretation and Comparison to Evron et al. (2022)” paragraph in the discussion of the revised manuscript. We believe that it helps see deeper connections like the one that the reviewer noticed.
>
> ---
> - **Question 4:**
> > Is there a geometric proof that achieves the same result as our algebraic calculations (randomness of principle angles)?
>
>   **Response:** This is a good question. Following the discussion here and with reviewer [TGNt](https://openreview.net/forum?id=u3dHl287oB&noteId=TRtKnkPHKr), and the discussion we’ve added to the manuscript, it is apparent that geometric reasoning is indeed partially possible here. Specifically, analyzing worst-case expected forgetting should be related to the largest principal angle between 2 random subspaces (see, e.g., “On the largest principal angle between random subspaces”, 2006).
>   However, there should be important differences, requiring different proof techniques. Specifically, our analysis allows for the entire range of $m$ (number of directions/pixels that we rotate), while the aforementioned other paper requires the two subspaces to be completely random. Thus, it seems like they can only help analyze the $m=p$ case.
>
> ---
> We again thank the reviewer for the feedback.
> If we have adequately addressed the reviewer's concerns, we kindly ask the reviewer to continue supporting our submission.
> If there are any remaining concerns, please let us know.

---

> > ### Comment · Reviewer_ckB1 · 2023-11-21
> > **Reply**
> >
> > Dear Authors, thank you for the rebuttal and your patience in the discussion phase. I apologize for the late reply.
> >
> > I read your rebuttal and the corresponding modifications in the revised paper. I think my comments have been addressed.
> >
> > I think the paper makes interesting, non-trivial, and to my knowledge, novel theoretical contributions to continual learning (linear regression case), and I'd like to see the paper get accepted. I increased my confidence from 3 to 5. I intended to increase my score from 6 to 7, while there is no such choice.
> >
> > At the same time, I read comments from other reviews and the corresponding rebuttals from the authors.
> >
> > I do agree with Reviewer SXpA that the proofs can be hard to follow (I didn't read them). For me, this at most precludes higher scores (e.g., 8), but it does not form a ground for rejection.
> >
> > I understood the paper received divergent scores. I don't entirely agree with some of them. I, therefore, increased my score from 6 (or one could understand it as 7) to 8 to counteract them.

---

> > > ### Author Response · Authors · 2023-11-21
> > >
> > > We thank the reviewer for acknowledging our paper's contributions and for increasing their review score.
> > >
> > > As mentioned in our response to Reviewer SXpA, we acknowledge that the lengthy proofs may be hard to follow.
> > > We will make effort to make them more accessible and provide a detailed proof outline in the appendix.

---

### Official Review · Reviewer_TGNt · 2023-11-03

**Soundness:** 3 good
**Presentation:** 3 good
**Contribution:** 3 good
**Rating:** 6
**Confidence:** 3

**Summary:**

This paper provides a mathematical analysis of how task similarity and overparameterization jointly affect forgetting in continual learning. By proposing a pair of orthogonally transformed datasets, authors define their similarity measurement DOTS $\alpha$ and the level of overparameterization $\beta$. Theorem 3 provides the worst-case expected forgetting w.r.t $\alpha$ and $\beta$. Furthermore, synthetic and empirical experiments provide more support for the proposed analysis.

**Strengths:**

1. Writing is clear and easy to follow.
2. Their novel result provides a new perspective for forgetting analysis in continual learning.
3. The theoretical result is solid and consistent with empirical results.

**Weaknesses:**

1. The dataset assumption is strong, which only focuses on 2 datasets, and can be converted by orthogonal transformation.
2. It is unclear how DOTS can compare with the notion of similarity in related works.

**Questions:**

1. Is the theory able to provide worst-case forgetting analysis with more general dataset assumption, or T>2 datasets?
2. It is still unclear that how DOTS relates to other notions of similarity, e.g. the principle angles in Evron et al. (2022). Is that possible to plot a figure that shows the relation between DOTS and $\theta$?
3. In Evron et al.(2022), Figure 3 shows the worst-case forgetting on T=2 tasks, where p=d-1(rank(X1)=rank(X2)=d-1), which in this paper, should corresponds to $\beta=1-\frac{d-1}{d}\approx 0$, which should fall into the very less overparameterization regime. However, there results still show the descent with a large angle. Can authors provide some explanation? (And this is also one of the reasons I hope to know how DOTS relates to $\theta$)

---

> ### Author Response · Authors · 2023-11-17
>
> We thank the reviewer for their valuable feedback, and we are glad that they found our paper novel and theoretically sound.
>
> Below we address some of the reviewer's concerns,
>
> - **Weakness:**
> > It is unclear how DOTS can compare with the notion of similarity in related works.
>
>   **Response:** In our linear regression setting, the existing similarity notion most comparable to our DOTS should be the principal angles between the solution subspaces of the two data matrices ($\mathbf{X}_1,\mathbf{X}_2$)}, which were used to quantify task similarity in Doan et al. (2021) and Evron et al. (2022). We elaborate on this further here below, and in the discussion in our revised manuscript.
>
> ---
>
> - **Question 1:**
> > Is the theory able to provide worst-case forgetting analysis with more general dataset assumption, or T>2 datasets?
>
>   **Response:** These are important questions and we added the following discussion to a new section on “Limitations and Future work” in the revised manuscript.
>   Our analysis has primarily examined settings with $T=2$ tasks. Extending these analytical results to $T\ge 3$ tasks poses an immediate challenge. The complexity of our analysis, which already required intricate techniques and proofs, suggests that tackling this extension may be considerably difficult. Moreover, the convergence analysis presented in a previous paper (Evron et al. 2022) for learning $T\ge 3$ tasks cyclically has proven to be notably more challenging than that for $T=2$ tasks, and was further improved in a follow-up paper (Swartworth et al. 2023).
> ---
>
> - **Questions 2&3:**
> > How does DOTS relate to other notions of similarity in prior work? Is there a contradiction to Figure 3 in Evron et al. (2022)?
>
>   **Response:** We thank the reviewer for this interesting question!
>   Evron et al. (2022) showed analytically that intermediate task similarity (a principle angle of $45^{\circ}$) is most difficult in two-task linear regression.
>   Their analysis applies to *any* two arbitrary tasks, and thus seemingly contradicts the behavior observed, e.g., in our Figure 1(b), where maximal dissimilarity is most difficult.
>   The key to settling this apparent disagreement is the *randomness* of our transformations.
>   Their analysis focuses on any two *deterministic* tasks, while our second task is given by a *random* transformation of the first, as done in many popular continual learning benchmarks (e.g., permutation and rotation tasks).
>
>   For simplicity, instead of the case that the reviewer mentioned, where $d=p-1$, let us focus on $d=1$ (in two tasks these are almost equivalent; see Claim 19 in Evron et al. (2022)).
>
>   To gain a geometric intuition, consider two tasks of rank $d=1$ ($\mathbf{x}_1, \mathbf{x}_2$). Consider also a *maximal* DOTS proxy for task dissimilarity ($\alpha=\frac{m}{p}=1$), i.e., $\mathbf{x}_2 = \mathbf{O}\mathbf{x}_1$ is simply a random rotation of $\mathbf{x}_1$ in $p$ dimensions. It is known that $\mathbb{E} \big| \big\langle \frac{\mathbf{x}_1}{\Vert{\mathbf{x}_1}\Vert}, \frac{\mathbf{x}_2}{\Vert{\mathbf{x}_2}\Vert} \big\rangle \big| \approx \frac{1}{\sqrt{p}}$.
>   Near the interpolation threshold, \eg when $p= 2$ (recall that $d=1$), we get that ${\mathbb{E} \big| \big\langle \frac{\mathbf{x}_1}{\Vert{\mathbf{x}_1}\Vert}, \frac{\mathbf{x}_2}{\Vert{\mathbf{x}_2}\Vert} \big\rangle \big| \approx \frac{1}{\sqrt{2}}} \Longrightarrow {\mathbb{E}\angle(\mathbf{x}_1,\mathbf{x}_2) \approx 45^{\circ}}$,
>   corresponding to the *intermediate* task dissimilarity in Evron et al. (2022), where forgetting is *maximal*.
>   On the other hand, given high overparameterization levels ($p\to\infty$), we get that ${\mathbb{E} \big| \big\langle \frac{\mathbf{x}_1}{\Vert{\mathbf{x}_1}\Vert}, \frac{\mathbf{x}_2}{\Vert{\mathbf{x}_2}\Vert} \big\rangle \big| \approx \frac{1}{\sqrt{p}} \to 0} \Longrightarrow {\mathbb{E} \angle(\mathbf{x}_1,\mathbf{x}_2)\to 90^{\circ}}$, corresponding to the *maximal* task dissimilarity in Evron et al. (2022), where forgetting is *minimal*.
>
>
>   We originally compared our results to Evron et al. (2022) in the discussion section. Following the reviewer’s question, we included the discussion above in the revised manuscript to further elaborate on these aspects.
>
>
> ---
> We again thank the reviewer for the feedback.
> If we have adequately addressed the reviewer's concerns, we kindly ask the reviewer to continue supporting our submission.
> If there are any remaining concerns, please let us know.

---

### Official Review · Reviewer_yV41 · 2023-11-05

**Soundness:** 3 good
**Presentation:** 3 good
**Contribution:** 1 poor
**Rating:** 3
**Confidence:** 4

**Summary:**

This paper aims to reveal the connection between the parametrization regime and forgetting. The authors start by analyzing the forgetting in a linear regression model and define their definition of their specific distribution shift for the second task. Moreover, the upper bound of the forgetting (derived as the loss of task 1 after learning task 2), is calculated and later used for different parameterization scenarios. The authors conclude that in an overparametrized regime, forgetting decreases as the task dissimilarity increases, however, in an underparametrized scenario, the trend is reversed (more forgetting for more dissimilar tasks). Finally, they evaluated the soundness of their derivations in a custom version of the permuted-NIST tasks.

**Strengths:**

I think the presentation in the paper is concise and to the point. The mathematical derivations in the given context are sound and clear. Maybe it would be better to show some of the derivations in eq 4 in the appendix but overall it is fine.

The results within the linear regression and simple nist-type experiments look interesting and worth pursuing in future works.

**Weaknesses:**

**Limited scope:** The main issue that I see in the paper is the limited scope of the provided analysis. I am fine with simple experiments in a theoretical paper but the upper bound of forgetting is very specific to the linear regression task.

**Overparametrization proxy:** The second issue is tightly related to the above, in theorem 3,  the proxy for overparametrization is defined as $1 - \frac{d}{p}$, where $d$ is the rank of the input data and $p$ is the data dimensionality. It makes sense in linear regression since the number of parameters is the same as the data dimension, i.e., $p$. This is true to some extent in the MLP layers (at least they are correlated). However, none of the derivations hold in the case of the convolutional layers where there are shared parameters and $p$ cannot be a surrogate for the number of parameters. I encourage the authors to evaluate their upper bound in CNNs.

**Task similarity definition:** Also, the definition of task similarity is very limited to the rotation of a subset of the data dimensions. All of the derivations are based on this initial assumption. I believe a good theoretical paper on this subject should expand this definition so that more real-world cases can be included in the analysis. My comment is the same for the permuted-NIST scenario. A simple permutation of a subset of pixels is not enough to back the main message in this more challenging scenario.

**Questions:**

My question goes back to the previous comments:

**Q1:** Is the overparametrization proxy enough to provide a similar analysis in the CNN case? i.e., can we just substitute the $p$ with the number of parameters in a CNN layer?

**Q2:** Have the authors tried to evaluate their upper bound in more challenging scenarios?

**Q3:** Have the authors tried to test their upper bounds when the definition of the task similarity is different? i.e., more subtle semantic distribution shifts in the data. e.g. CIFAR-100 Superclass.

---

> ### Author Response · Authors · 2023-11-17
>
> We thank the reviewer for their comments on our paper, and we are glad that they found our results interesting.
>
> Unfortunately, there seem to be several misunderstandings that perhaps interfered with the reviewer’s assessment of our paper. For instance, the reviewer wrote that *“in an overparametrized regime, forgetting decreases as the task dissimilarity increases, however, in an underparametrized scenario, the trend is reversed”*. However,
>
> - We only analyzed overparameterized models (from the interpolation threshold to a highly overparameterized regime), as seen from Theorem 3 (the rank $d$ is upper bounded by the dimensionality $p$). Underparameterized regimes are practically less common today, and cannot be very interesting under our realizability assumption (since there would only be a unique solution and there would be no place to discuss the bias incurred by continual learning).
>
> - Even in the overparameterized regime, we did not conclude that forgetting decreases as task dissimilarity increases, but rather showed a nuanced non-monotonic behavior (kindly see Section 2.3.1 on the Extremal Cases and Figures 2 and 3).
>
>
> Importantly, we believe that our paper serves as an analytical counterexample to the findings of several previous empirical papers, which concluded that overparameterization always mitigates forgetting.
> Specifically, we use our model to demonstrate that, and even if we take overparameterization to the extreme, we can still have a considerable amount of forgetting (and it does not necessarily decay to zero), e.g. when $\alpha=1-\frac{m}{p} = 0.5$. We believe that it is more convincing that we were able to find this counterexample even in the simplest model of linear regression and in a simple-yet-standard data model.  This suggests that more complex models probably also have a nuanced behavior with regard to overparameterization, though not necessarily exactly the same behavior.
>
>
> To enhance clarity, we revised the manuscript and incorporated a brief discussion in Section 2.3.2, just before Figure 3. Additionally, we explicitly highlighted this aspect in the contribution list within the introduction.
>
> The reviewer also asked (Q2,Q3) whether we tried to evaluate our upper bound *“in more challenging scenarios”* (e.g., CNNs) or with different notions of task similarity. The answer is that we did not, but neither did we claim that our analysis simply extends to such cases. For example, permutation benchmarks like the one we aim to characterize, are not very suitable for CNNs, since the permutations significantly interfere with the spatial assumptions convolutional models implicitly make (shift-equivariance of representations and shift-invariance of the classification).
> Again, our aim is that our analysis can be used as a counterexample to the practical common belief that overparameterization always mitigates forgetting and as a demonstration of a more nuanced interplay between overparameterization and task similarity. More generally, when doing theoretical research, we believe that one must start from the simplest models as we did, before addressing more complex scenarios as the reviewer rightly suggested.
>
> Following this discussion, we now include these suggestions in the "Future work" section at the end of our revised manuscript.
>
> ---
>
> If we have adequately addressed the reviewer's concerns, we kindly ask the reviewer to consider raising their score. If there are any remaining concerns, please let us know.

---

> > ### Comment · Reviewer_yV41 · 2023-11-22
> > **Response to Authors**
> >
> > I have read the authors' responses and appreciate their efforts in addressing my concerns and clarifying the scope and claims of their papers.
> >
> > While I find the findings interesting, they hinge on a particularly narrow definition of task similarity. This definition isn't widely accepted or realistic. I acknowledge that the authors present a counterexample to previous findings, but it's a specific instance with limited relevance in real-world scenarios. Previous research does not suggest that overparametrization always mitigates forgetting; rather, it appears to be a common occurrence in more realistic situations. The paper could gain significantly if it included some theoretical insights related to CNNs or alternative definitions of task similarity. However, as it stands, the paper’s focus is too narrow for an ICLR audience.
> >
> > For these reasons, I will maintain my initial evaluation score.

---

> > > ### Author Response · Authors · 2023-11-22
> > >
> > > We thank the reviewer for their time and for engaging with us in this discussion.
> > >
> > > -  **Regarding the simplicity of the analyzed model**.
> > > Respectfully, we must disagree that the simple linear regression scenario that we analyze makes our paper *"too narrow for an ICLR audience"*.
> > > Previous theoretical CL papers, published at top venues, often analyze linear models (some even make additional data assumptions, e.g., random isotropic features, which we do not).
> > > For instance, see
> > > [Doan et al. (AISTATS 21)](https://proceedings.mlr.press/v130/doan21a.html),
> > > [Evron et al. (COLT 22)](https://proceedings.mlr.press/v178/evron22a.html),
> > > [Goldfarb & Hand (AISTATS 23)](https://proceedings.mlr.press/v206/goldfarb23a.html),
> > > [Evron et al. (ICML 23)](https://proceedings.mlr.press/v202/evron23a.html),
> > > [Lin et al. (ICML 23)](https://proceedings.mlr.press/v202/lin23f.html),
> > > [Swartworth et al. (NeurIPS 23)](https://openreview.net/forum?id=X25L5AjHig).
> > >   We are not aware of *any* existing theoretical CL result on CNNs, likely due to the hardness of such analysis. We thus believe that our linear model is a more suitable *starting point* for theoretical analysis on the interplay between overparameterization and task similarity in CL.
> > >
> > > - **Regarding our notion of task similarity.** As our paper explains, the notion we analyzed (i.e., Dim. of the Transformed Subspace) *analytically* generalizes the completely random rotation model of [Goldfarb & Hand (2023)](https://proceedings.mlr.press/v206/goldfarb23a.html) (on one extreme); and relates to the principal angles notion from [Doan et al. (2021)](https://proceedings.mlr.press/v130/doan21a.html) and [Evron et al. (2022)](https://proceedings.mlr.press/v178/evron22a.html) (see our revised discussion). *Practically*, it closely characterizes the number of permuted pixels in randomly-permuted benchmarks (see Figure 2 in the seminal paper of [Kirkpatrick et al. (2017)](https://arxiv.org/abs/1612.00796)).
> > > We agree that it should be interesting to investigate related phenomena with other task similarity notions, and hope that our paper will pave the way for such future work.

---

### Official Review · Reviewer_vLwn · 2023-11-09

**Soundness:** 3 good
**Presentation:** 2 fair
**Contribution:** 2 fair
**Rating:** 6
**Confidence:** 4

**Summary:**

The authors present an analytical upper bound for the expected amount of catastrophic forgetting during linear regression with gradient descent as a function of overparameterisation and overlap between tasks. In particular, they show that there exist two phenomenologically different learning regimes: In the overparameterised regime (i.e. $rank({\bf X}) / rank({\bf w})$ small) intermediate similarity between tasks leads to largest amount of forgetting (inverted U-shape) and monotonic behaviour is observed near the interpolation threshold ($rank({\bf X}) / rank({\bf w}) \approx 1$). To this end, the authors make the following assumptions: (1) The target values for both tasks $({\bf y})$ are identical, (2) both tasks are realisable (i.e. a zero-loss solution exists) (3) the second task is an orthogonal projection of the first task (c.f. permutation) and (4) the first task has successfully converged to the global, minimum-norm solution of task one before training on task two. By changing the subspace which the orthogonal operator transforms, the overlap between the tasks can be exactly controlled and its effect analytically studied.


**--------------------------------------**

**Unfortunately, I can not add official comments (anymore), thus I will add my comments to the author's comments here**

**Reply 1**
Weaknesses 1&2 and Questions 1,2,3: Thank you for the clarifications. As it is not obvious from section 2.1 why equation (3) is the closest point on the solution manifold when starting to train from the convergence point of task 1, I would like to suggest to include this derivation in the appendix.

Weakness 3: Thank you for the clarification.

Weakness 6: As there is no direct mapping from permuting MNIST to DOTS, I have mild concerns that the observed phenomena could be explained away by the fact that most information is in the center of the image and thus increasing the permuted area from the center to the outside may result in a non-linear transformation of the dimensional of the data-manifold. If the effect is non-linear, comparisons to phenomena observed when manipulating DOTS would be erroneous. However, I appreciate that the authors try to test their hypotheses on more complex data.

Weakness 4: The minimum-norm solutions studied by the authors are part of the "rich" learning regime. The lazy regime does exist in (multi-layer) linear networks. For example, in the regression model, weight values that lay in the null-space of the training data of the first task could be unequal zero. For example, when initialising the weight matrix from large random values, GD would still converge to a gloabl optimum, however, the convergence points would be unequal to equation (2) and (3). Thus, I would argue that the presented analytical results only apply to the rich learning regime.

Further, in an over-parametrized linear two-layer network, the first layer can be initialised, large and arbitrarily (and in theory even kept fixed). GD would still converge to a global optimum, however, as a consequence, the learned representations in the hidden layer would be "lazy". Again, I don't think that the analytical result apply to this regime and thus simulations that compare these two regimes (i.e. training from (very) small initial weights and training from large initial weights) would be useful.

See also question: For Figures 6 & 7, how are the neural networks initialized? Do they operate in the rich or lazy regime?


**Reply 2**
I would like to thank the authors for addressing my questions.

Weakness 5: Thank you for adding these additional simulations.

Question: Can you give an intuitive explanation for why the expected forgetting is non-asymptotic for overparameterized regime and near interpolation threshold

Sorry, that question was not very well formulated. What I tried to ask is: Do you have an intuitive explanation for why in the overparametrized regime, the expected forgetting for increasing task dissimilarity if falling off, whereas near the interpolation threshold, it keeps increasing?

**The following points remain unaddressed:**

- **Weakness 7: Simulation details and / or code are not provided which makes interpretation of simulation results difficult and reproduction impossible.**
- **Weakness 10: The introduction to continual learning / catastrophic forgetting is, in my opinion, not referenced well enough**


As a result of the changes made by the authors I have increased my rating to a 6.

**--------------------------------------**

**Strengths:**

The authors tackle an important research question: What are the underlying mechanisms that govern catastrophic forgetting during the continuos acquisition of knowledge using gradient descent. Their analytical result fully describes phenomenologically distinct operating regimes during the continuous optimisation of a linear regression model using gradient descent. The authors validate their analytical result using simulation studies and test whether their results generalise to more complicated two-layer networks. Notation is consistent throughout the paper and explanations for variables, equations and derivations are provided. The paper and plots are generally well structured and accessible.

**Weaknesses:**

1. The model (i.e. single linear projection) and algorithm (gradient descent, full-batch gradient descent?!) are not stated in the introduction. Information about the regime to which the analytical result applies is scattered throughout the paper, which makes it difficult to contextualise the results.
2. It would be really helpful if all assumptions of the analytical result would be stated explicitly and collected within one section of the paper. For example, the ${\bf 0}$ initialisation and resulting convergence to the global, minimum-norm solution of the first task is hidden in Scheme 1 but crucial to understand the extent of the analytical result.
3. I think the paper could benefit from making it more explicit that the analysis focuses on the dimensionality and overlap with respect to the manifold of the data distribution, which is often called overparametereised, but should not be confused with the overparameterisation deep networks (i.e. wide hidden layers).
(4. Resulting from 3., the model can not be used to make predictions about the underparameterised regime, e.g. bottlenecked networks.)
5. I think the paper should state explicitly that the analysis is limited to the rich learning regime. It does not provide any insights into the lazy learning regime. Comparisons and references from the paper’s “overparameterised regime” to e.g. the NTK regime are confusing and incorrect.
6. Simulation results in Figure 2. are limited to $d = 2$. There are no simulation results to validate the analytical result for edge-cases like p = d and large(r) d etc.
7. I am not entirely sure if the author’s version of permuted MNIST is suitable to study changes in task similarity that are comparable to DOTS, as vanilla MNIST data is on a very low dimensional data manifold and large parts of the information is centred on the middle of the picture. I think I would prefer a simulation study that uses artificial data as in Figure 4., with well controlled DOTS in two layer networks.
8. Simulation details (initialisation scheme, learning rates etc.) and / or code are not provided.

Minor:
9. The notation in equations (2) and (3) is confusing. There are too many equal signs
10. The introduction to continual learning / catastrophic forgetting is, in my opinion, not referenced well enough
11. Using X_1 and X interchangeably is confusing and seems unnecessary

**Questions:**

- To what optimisation algorithm does the analytical result exactly apply?
- Could you please hint me at the Theorem in Gunasekar et al. (2018) from which equation (3) is derived?
- What assumptions are made on the size of batches and the size of the learning rate?
- Why does Theorem 3 have the assumption of p >= 5?
- Why can you assume that a randomly sampled data matrix X is identical to the worst case in Figure 2?
- The smallest expected forgetting in the interpolation regime (except for values close to 0 DOTS) is larger than the largest expected forgetting in the overparametereised regime. Why is that the case and how is it relevant to the analysis?
- Is there a benefit of using an informal illustration instead of slices from figure 3 in figure 1? The inverted U-shape seems to be slightly exaggerated as expected forgetting for large DOTS don’t fall off nearly as strongly as suggested by the illustration.
- Can you give an intuitive explanation of why the expected forgetting are (non-)asymptotic for overparameterised regime and near the interpolation threshold?
- Do the authors think that their results do generalise to learning more than two tasks? I think that would clearly contradict some of the assumptions and thus maybe should be listed as a limitation of the work?
- Is it maybe possible to use a log-scaled axis in Figure 3 instead of using subplots to represent the results?
- Is it possible that the x-axis of Figure 3, bottom right is labelled incorrectly? Values from 0 to 0.1 seem to be missing
- For figures 6. and 7., how are neural networks initialised? Do they operate in the rich or lazy regime?

---

> ### Author Response · Authors · 2023-11-17
> **Part 1 of the Author Response**
>
> We thank the reviewer for the thorough and helpful review.
> Below we address the weaknesses the reviewer has pointed out and also answer all the questions.
> We believe that the changes made following the discussion below, helped to improve our paper significantly.
>
>
>
> - **Weaknesses 1&2 and Questions 1,2,3:**
> > What is the exact optimization scheme to which our results apply (algorithm, batch size, learning rate)? The model and algorithm are not stated in the introduction. Which theorem from Gunasekar et al. is used.
>
>   **Response:** We apologize about the lack of clarity on the aspects mentioned by the reviewer. Below we address these questions and specify how the revised manuscript clarifies them.
>
>   We study the simplest continual learning procedure - starting from a zero initialization $\textbf{0}_p$ and iteratively minimizing the current task’s loss using (S)GD initialized with the previous task’s solution. In each iteration, we assume that learning “converges” to the limit solution, obtained by (S)GD with *any* batch size (as long as the learning rate is small enough; see Section 2.1 in Gunasekar et al. (2018)). This is a similar procedure to the one studied in several analytical papers (e.g., Evron et al. (2022); Goldfarb and Hand (2023)).
>
>   As can be seen in our revised manuscript, we now (1) specifically mention that we study an (S)GD scheme in the introduction; (2) changed the title of Section 2.2 to “The Analyzed Learning Scheme and its Learning Dynamics”; (3) specifically write before Scheme 1 that this is the scheme that we analyze; (4) added a footnote to the analytical iterates, explaining that any batch size can be use as long as the learning rate is small enough; (5) highlight right after the scheme that we do not actually compute pseudoinverses; (6) and finally, specifically refer to Scheme 1 in our main Theorem 3.
>
>
> ---
>
> - **Weakness 3:** The analysis focuses on the dimensionality and overlap with respect to the manifold of the data distribution, which should not be confused with the overparameterization of deep networks.
>
>   **Response:** We agree with the reviewer and we added this to the “Limitations and Future work” section in our revised manuscript.
>
> ---
>
> - **Weakness 6:**
> > Unsure if permuted MNIST is suitable to study changes in task similarity that are comparable to DOTS. Vanilla MNIST is on a very low dimensional manifold. I would prefer a simulation study that uses artificial data.
>
>   **Response:** The permuted MNIST experiments are intended to illustrate the ideas of the analysis in a more realistic image classification scenario. Note that the permuted image setting is an analogous version of the DOTS simulation where random permutation matrices are used instead of random orthogonal matrices. We recognize that MNIST images lie on a low dimensional manifold and this notion is what motivated our decision in the analysis to have data of low rank d; in this case, low effective dimensionality is a similarity between MNIST and the simulated data.
>
> ---
>
> - **Weakness 4:** The paper should state explicitly that the analysis is limited to the rich learning scheme. It does not provide any insights into the lazy learning regime.
>
>   **Response:** We see two possible interpretations of this statement, and we are slightly confused by both.
>   In case the reviewer was referring to our theoretical analysis, we note that we work in the setting of linear regression, in which there is no rich/lazy regime distinction. As far as we are aware, the lazy vs rich distinction applies only to non-linear parameterizations, where the initialization scale *qualitatively* affects the final solution (e.g., minimizing the L2 norm in "lazy", vs L1 norm in "rich", in the squared linear regression model of Woodworth et al. (COLT 2020)). In contrast, as can be deduced from our Eq. 3, in linear regression, changing the initialization just shifts the final solution location (along a direction orthogonal to the data manifold) — which is not a qualitative change.
>   In case the reviewer meant to the application of our linear regression results to neural networks, then, as far as we are aware, the only formal way to prove that linear regression results directly apply to neural networks is to be in a regime where the neural network is approximately linear, such as the NTK regime. Therefore, we only claim our results can be directly applied to neural networks in this case. Perhaps some of our conclusions also apply to the rich regime, but we do not see how this could be theoretically proven.

---

> > ### Author Response · Authors · 2023-11-17
> > **Part 2 of the Author Response**
> >
> > - **Weakness 5:**
> > > Simulation results in Figure 2 are limited to $d=2$. No simulations for other edge-cases
> >
> >   **Response:** We updated the figure. Instead of plotting only $(p,d)=(30,2),(200,2)$, we now show $(p,d)=(100, 1),(100,10),(100,100)$ in Figure 2, as well as $(10,1),(10,10),(1000,1),(1000,10),(1000,100)$ in the new Appendix B.3, to demonstrate the validity of our analysis more broadly. In all cases, analytical bounds match empirical means.
> >
> > ---
> > - **Question:**
> > > Why does Theorem 3 have the assumption of p>=5?
> >
> >   **Response:** Theorem 3 assumes that the dimension holds $p \ge 5$ (which can actually be improved to $p \ge 4$, we fixed it now). This is because many of our proofs (e.g., in Lemma 8) yield expressions involving a $(p-3)$ factor in their denominators. However, this is a mild restriction when testing for overparameterization effects like we do.
> >
> > ---
> > - **Question:**
> > > In Figure 2, why can you assume that a randomly sampled data matrix X is identical to the worst case?
> >
> >   **Response:** Please notice that the data matrix $\mathbf{X}$ in Figure 2 is indeed random, but we constrain its nonzero singular values to be identical. In Eq. (4), we present an upper bound on the forgetting, and explain that it is *sharp* when the nonzero singular values are identical. This exemplifies that our analysis is tight, and that worst-case matrices are easy to generate.
> >   We added an explicit reference to Eq. (4) in the caption of Figure 2.
> >
> > ---
> > - **Question:**
> > > Why is the smallest expected forgetting in the interpolation regime larger than the largest expected forgetting in the overparameterized regime?
> >
> >   **Response:** This is an interesting question. We'll try to answer it briefly and intuitively. When we are highly overparameterized, the directions randomly rotated in the 2nd task are going to be mostly orthogonal to their corresponding directions in the original first task (thus incurring little forgetting). Near the interpolation regime, randomly rotated directions will have more interference with the original directions, since they will not become orthogonal w.h.p., and forgetting would be higher.
> >   We elaborated on that in the discussion section of our revised manuscript.
> >
> > ---
> > - **Question:**
> > > Is there a benefit of using an informal illustration instead of slices from figure 3 in figure 1? The inverted U-shape seems to be slightly exaggerated as expected forgetting for large DOTS don’t fall off nearly as strongly as suggested by the illustration
> >
> >   **Response:** We believe that at such an early stage of the paper, the informal Figure 1 is clearer than Figure 3. We also believe the inverted U-shape of the highly overparameterized regime’s subfigure is not exaggerated, since it is in fact analytically *exact*. We kindly refer the reviewer to Section 2.3.1 (Extremal Cases), where we show that at the highly overparameterized regime (i.e., when $\beta=1-\frac{d}{p}\to 1$), worst-case forgetting is $\alpha^2 \left(1-\alpha\right)^2$, which forms the inverted U-shape in Figure 1.
> >
> > ---
> > - **Question:**
> > > Can you give an intuitive explanation for why the expected forgetting is non-asymptotic for overparameterized regime and near interpolation threshold
> >
> >   **Response:** Unfortunately, we could not understand the question. Could the reviewer clarify?
> >
> > ---
> > - **Question:**
> > > Do our results generalize to more than 2 tasks? If so, how does that not contradict some of the assumptions?
> >
> >   **Response:** This is an important question and we added the following discussion to a new section on “Future work” in the revised manuscript.
> >   Our analysis has primarily examined settings with $T=2$ tasks. Extending these analytical results to $T\ge 3$ tasks poses an immediate challenge. The complexity of our analysis, which already required intricate proofs, suggests that tackling this extension may be considerably difficult. Moreover, the convergence analysis presented in Evron et al. (2022) for learning $T\ge 3$ tasks cyclically has proven to be notably more challenging than that for $T=2$ tasks, and was further improved in a followup paper (Swartworth et al. 2023).
> >
> > ---
> > - **Question:**
> > > Is it possible to use a log-scaled axis in Figure 3 instead of using subplots?
> >
> >   **Response:** We tried that, but the result was not as good as the current scaling. However, we slightly edited the figure and we believe it looks better now.
> >
> > ---
> > - **Question:**
> > > Values 0 to 0.1 seem to be missing from the x-axis of Figure 3.
> >
> >   **Response:** We fixed it, thank you.
> >
> > ---
> > - **Question:**
> > > For Figures 6 & 7, how are the neural networks initialized? Do they operate in the rich or lazy regime?
> >
> >   **Response:** The networks are initialized with centered Gaussian weights. The MLP’s in Figure 6 and 7 are sufficiently wide to enjoy generalization benefits of lazy training.
> >
> > ---
> > If we have adequately addressed the reviewer's concerns, we kindly ask them to consider raising their score.
> > If there are any remaining concerns, please let us know.

---

### Author Response · Authors · 2023-11-17
**A general comment to all reviewers**

We express our gratitude to all the reviewers for their time and thorough evaluations. While the average score appears to be on the borderline, we are confident that we have effectively addressed the majority of questions and concerns, leveraging the reviews to significantly improve the paper across various aspects.

Specifically, we utilized the feedback to:
- Improve clarity and refine the message throughout the paper.
- Expand Figure 2 to present a broader range of overparameterization and task similarity combinations.
- Elaborate on Figure 3 so as to sharpen the message of the paper.
- Introduce a comprehensive discussion on the geometric interpretation of our results and the connection to Evron et al. (2022).
- Include a detailed section on “Limitations and Future work”.

We have submitted a revised manuscript with changes highlighted in red for the convenience of the reviewers.

We kindly request the reviewers to review our responses and consider adjusting their scores accordingly.
We thank you once again for your time and valuable feedback.

---

### Author Response · Authors · 2023-11-21

Dear reviewers,
We again thank you for your time and valuable feedback.
Having just two days left for the discussion period, we would love to address any issues remaining following our author responses.
Best wishes

---

### Meta-Review · Area_Chair_6DUv · 2023-12-10

**Metareview:**

The submission presents a novel theoretical analysis of sequential learning of two tasks, providing insight into how the interaction of task similarity and overparameterization impacts catastrophic forgetting. The work is precise and provides accompanying intuition, and though it characterizes analytically only a linear setting of two tasks, the phenomenon if not the analysis technique may be foundational for understanding continual learning in more general settings such as the MNIST experiments briefly explored here.

**Justification For Why Not Higher Score:**

scope limited by the linear setting and count (2) of tasks

**Justification For Why Not Lower Score:**

novel theoretical contribution to continual learning

---

### Decision · Program_Chairs · 2024-01-16

Accept (poster)